# CAN WE GENERATE PORTABLE REPRESENTATIONS FOR CLINICAL TIME SERIES DATA USING LLMS?

**Zongliang Ji**[*,1,3]**, Yifei Sun**[*,1,3]**, Andre Amaral**[1,2]**, Anna Goldenberg**[1,3]**, Rahul G. Krishnan**[1,3]
[1]University of Toronto, Canada, [2]Sunnybrook Health Sciences Centre, Canada,
[3]Vector Institute, Canada
`{jerryji, rahulgk}@cs.toronto.edu`

## ABSTRACT

Deploying clinical ML is slow and brittle: models that work at one hospital often degrade under distribution shifts at the next. In this work, we study a simple question – can large language models (LLMs) create *portable* patient embeddings i.e. representations of patients enable a downstream predictor built on one hospital to be used elsewhere with minimal-to-no retraining and fine-tuning. To do so, we map from irregular ICU time series onto concise natural language summaries using a frozen LLM, then embed each summary with a frozen text embedding model to obtain a fixed length vector capable of serving as input to a variety of downstream predictors. Across three cohorts (MIMIC-IV, HIRID, PPICU), on multiple clinically grounded forecasting and classification tasks, we find that our approach is simple, easy to use and competitive with in-distribution with grid imputation, self-supervised representation learning, and time series foundation models, while exhibiting smaller relative performance drops when transferring to new hospitals. We study the variation in performance across prompt design, with structured prompts being crucial to reducing the variance of the predictive models without altering mean accuracy. We find that using these portable representations improves few-shot learning and does not increase demographic recoverability of age or sex relative to baselines, suggesting little additional privacy risk. Our work points to the potential that LLMs hold as tools to enable the scalable deployment of production grade predictive models by reducing the engineering overhead[1].

## 1 INTRODUCTION

Clinical machine learning tools continue to be deployed one site at a time. Teams build a model at Hospital A, run a silent trial, tune thresholds and features, and only then attempt deployment. The process is repeated at Hospital B. Each iteration introduces distribution, population, and incidence shift as laboratory measurement policies, case mix, and disease prevalence change that can degrade model performance. These changes inflate time to deployment, lengthen validation time and calibration cycles, and slow the delivery of benefit to patients.

The status quo in healthcare treats the *model* as the object that is transferrable across institutions. Interoperability standards such as OMOP and FHIR reduce data wrangling but do not guarantee an input representation that will remain predictive when moved to a new site. Domain adaptation and invariant risk minimization typically adapt the model to the target hospital, which preserves accuracy at the cost of additional data, labels, and calibration. Foundation models trained on electronic health records improve in-distribution performance and reduce task-specific engineering, yet they are often coupled to a particular sensor schema or sampling profile, and model transfer across sites still requires tuning. In short, current solutions standardize data formats, sometimes stripping clinically salient context, or recalibrate models.

We adopt a complementary thesis: *portable input representations enable portable models*. i.e. if heterogeneous electronic medical records can be mapped into a semantically aligned, site-agnostic interface, then conventional predictors will require substantially less site-specific adaptation. Rather

---

[1]Code is available at `https://github.com/Jerryji007/Record2Vec-ICLR2026`

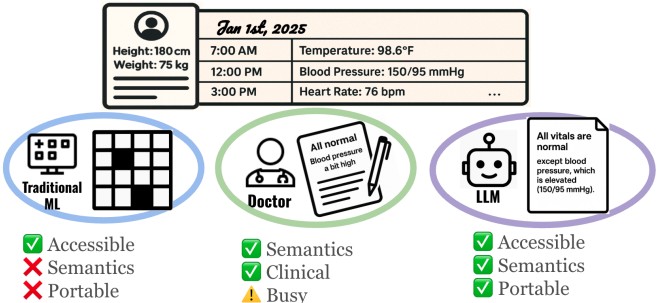

Figure 1: Motivation for Record2Vec. Numeric imputation loses clinical semantics and limits portability; human handoffs preserve meaning but are costly and variable. LLMs can create handoff-style summaries that retain semantics and provide portable inputs for forecasting and classification.

than forcing each hospital's raw schema into a shared syntactic format, we aim to produce a shared semantic representation.

Clinicians already solve an analogous problem. When assuming care for a new patient or starting a new shift, physicians interpret heterogeneous measurements through structured narrative handoffs that foreground clinically salient context while abstracting away measurement idiosyncrasies. We investigate whether large language models (LLMs) can analogously transform irregular multivariate patient histories into consistent, handoff-style summaries that serve as a portable intermediate representation.

Concretely, we instantiate this idea with a summarize-then-embed pipeline. For each patient window, a frozen LLM converts the irregular time-series record into a concise natural-language summary. A frozen text embedding model then maps that summary into a fixed-length vector consumed by standard forecasting and classification models without architectural modification.

This design leverages several properties of language models. First, summarization operates in a semantic space rather than a schema-bound numeric space, allowing normalization of units, resolution of synonyms, and abstraction over site-specific coding artifacts. Second, natural language provides a common representational substrate that aligns heterogeneous sampling policies and missingness patterns into comparable clinical concepts. Third, fixed-length embeddings standardize the downstream interface, simplifying training budgets and enabling zero- or few-shot transfer. Freezing both the summarizer and embedder further constrains overfitting to site-specific artifacts and improves reproducibility by decoupling representation learning from downstream optimization.

Our contributions are threefold:

- We rethink how machine learning in healthcare should operate via a deployment-first framing of prediction that focuses on creating portable input representations for healthcare, aiming to reduce per-site engineering and calibration.

- We present Record2Vec, a practical summarize-then-embed pipeline using frozen language models to transform irregular ICU histories into fixed-length vectors consumed by standard predictors without architectural changes.

- We conduct a multi-site evaluation across three cohorts and multiple tasks showing that the learned representations are competitive in-distribution, more portable across hospitals, more data-efficient, and match or improve on privacy preservation relative to established pipelines.

## 2  RELATED WORK

**Clinical schema and harmonization.** Common data models and interoperability standards such as OMOP and HL7 FHIR have substantially reduced extract–transform–load burden and enabled multi–institutional reuse of EHRs through shared schemas, vocabularies, and APIs (Stang et al., 2010; Bender & Sartipi, 2013; Reich et al., 2024; Vorisek et al., 2022; Tabari et al., 2024). While effective for data sharing and governance, these frameworks standardize *format* rather than learn task–optimized, site–invariant representations for predictive modeling.

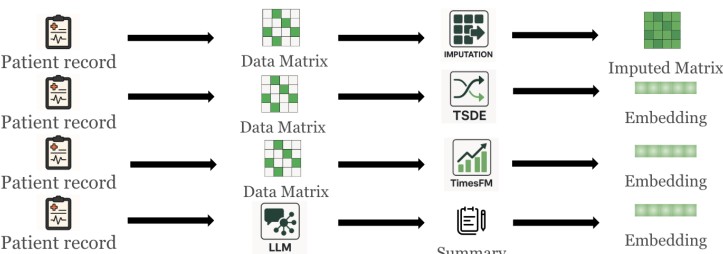

Figure 2: Methods to generate medical-record representations. Top to bottom: imputation pipeline; self-supervised TS representation (TSDE); TS foundation model (TimesFM); and Record2Vec: LLM summary followed by text embedding.

**Domain shift, generalization, and adaptation.** A large literature addresses distribution shift by adapting model parameters to target environments. Domain–adversarial training encourages site–invariant features via a domain classifier (Ganin & Lempitsky, 2016); correlation alignment matches second–order statistics across domains (Sun & Saenko, 2016). Domain generalization methods such as invariant risk minimization seek predictors stable across environments (Arjovsky et al., 2019; Subbaswamy et al., 2022), and distributionally robust optimization emphasizes worst–case groups to improve reliability under shift (Sagawa et al., 2020). These approaches are model–centric, often requiring environment labels, target–site data, or careful tuning. In contrast, we study *input level* portability by learning a site–agnostic representation that standard predictors can consume without architectural changes.

**EHR and self–supervised representation learning.** Representation learning on structured EHRs and clinical time series has advanced rapidly. BEHRT pretrains Transformers over longitudinal code sequences to learn patient timelines (Li et al., 2020b; Steinberg et al., 2020). Recent surveys review concept and patient–level embeddings for EHRs (Kauffman et al., 2025). Time–series foundation models pretrained at scale demonstrate strong zero–/few–shot forecasting, notably TimesFM (Das et al., 2024). Diffusion–style self–supervision for general time–series representation learning, such as TSDE, learns versatile embeddings via imputation–interpolation–forecasting masking with dual–encoder Transformers (Senane et al., 2024). These are strong baselines; however, portability and deployment cost across hospitals are rarely treated as first–class evaluation endpoints.

**LLMs for clinical summarization and structuring.** Large language models have been applied to clinical summarization and structuring tasks. Adapted LLMs can outperform experts on multiple clinical text summarization tasks (Van Veen et al., 2024), and there is growing guidance on prompt design for medical use (Zaghir et al., 2024). Evaluations of medical evidence summarization further characterize strengths and limitations (Tang et al., 2023). Beyond notes, scoping reviews document integrating standardized terminologies like SNOMED CT with LLMs for normalization and coding (Chang & Sung, 2024; Luo et al., 2020). While prior work uses LLMs for text summarization or coding, we repurpose them as an *information transformation layer* for numeric ICU time series, bridging structured and unstructured modalities.

Recent research investigates serializing structured EHR data for LLM processing. Approaches like TabLLM (Hegselmann et al., 2023) and DeLLiriuM (Contreras et al., 2025) fine-tune LLMs on text to perform direct prediction. Other studies (Lee et al., 2025; Gao et al., 2024; Hegselmann et al., 2025) use frozen models to embed raw data serializations. A critical distinction lies in our deployment philosophy. These prior methods optimize the model pipeline and often require the transfer of site-tuned LLMs. In contrast, our approach focuses on preparing a transferable input representation $X$. We use a standard and locally deployed frozen LLM to summarize dense time-series data. This step normalizes site-specific artifacts at the data level. Consequently, hospitals can generate portable representations that allow any lightweight downstream classifier to function effectively without porting entire end-to-end model pipelines.

## 3 METHODOLOGY AND SETUP

We design a controlled study to test whether natural language can serve as an *information transformation layer* that yields *portable inputs* for ICU prediction, formalizing sites, inputs, and tasks

across four method families, three ICU cohorts, and seven prediction tasks, and presenting our Record2Vec pipeline with baselines, data, and evaluation.

**Problem formalization.** Let sites be $S \in \{\text{MIMIC-IV}, \text{HiRID}, \text{PPICU}\}$. For a stay $i$ at site $s$, the irregular ICU record over a 48-hour window is $\mathcal{R}_i^{(s)} = \{(c, \{(t_k, v_k)\}_{k=1}^{K_c}) : c \in \mathcal{C}^{(s)}\}$, a dictionary from clinical concepts $c$ to time–value pairs $(t_k, v_k)$ observed in the window. We consider methods that transform $\mathcal{R}_i^{(s)}$ into either (i) a regular *grid* $X_i \in \mathbb{R}^{D^{(s)} \times T}$ with $T = 48$ hours, or (ii) a fixed-length *vector* $z_i \in \mathbb{R}^d$. A downstream predictor $f_\theta$ maps inputs to task labels $y_i$ across standard ICU outcomes spanning different label-prevalence regimes.

## 3.1 RECORD2VEC

We treat language as an information transformation layer. Given the 48h irregular record $\mathcal{R}_i^{(s)}$ for stay $i$ at site $s$, Record2Vec first produces a concise clinical summary and then embeds that summary into a fixed-length vector used by shared decoders.

**Summarization.** A frozen LLM $g_\phi$ maps each irregular window and a prompt $\pi$ to a concise summary, i.e., $\text{text}_i = g_\phi(\mathcal{R}_i^{(s)}, \pi)$. We compare *structured* slot-based prompts that elicit vitals, labs, therapies, trajectories, salient events, and gaps with *free-form* prompts that request a short narrative of states and trends. We evaluate three summarizers representing different deployment regimes: a large general model (Gemini-2.0 Flash Comanici et al. (2025)), a clinically tuned model (MedGemma Sellergren et al. (2025)), and a small open model (Llama-3.1 Dubey et al. (2024)). All LLMs remain frozen without finetuning.

**Embedding.** A frozen text encoder $h_\psi$ maps the summary to a fixed-length vector used by downstream predictors, $z_i = h_\psi(\text{text}_i) \in \mathbb{R}^d$. Unless noted, we use Qwen3 text-embedding as $h_\psi$ [2] As a no-summarization control, we also embed a canonical serialization of the record directly $z_i^{\text{direct}} = h_\psi(\text{serialize}(\mathcal{R}_i^{(s)}))$.

## 3.2 BASELINES

We compare Record2Vec to three strong families used for ICU time series, plus the direct-embed control.

- **Grid imputation pipelines.** Irregular records are discretized into hourly grids $X \in \mathbb{R}^{D^{(s)} \times 48}$, then completed via (a) mean fill, (b) right-shift carry-forward, or (c) linear interpolation. Completed grids feed the shared decoders for forecasting, regression, and classification. Per-cohort normalization uses training-split statistics and is applied only to grid methods.

- **Self-supervised representation learning (TSDE).** Time-Series Diffusion Embedding (TSDE) (Senane et al., 2024) learns general-purpose embeddings via masked imputation/interpolation/forecasting objectives. We train TSDE on each cohort's training split to obtain one vector per example; vectors are converted to grids by the shared projection in the downstream model and decoded with the same heads as other methods. Model selection uses only training/validation within the source cohort.

- **Time-series foundation model (TimesFM).** TimesFM (Das et al., 2024) is a decoder-only attention model pretrained on diverse time series for strong zero-shot forecasting. We use TimesFM as a *frozen encoder*: after mean imputing each feature to form regular per-feature series, we extract hidden representations feature-by-feature and average across features to obtain a window-level embedding, which is then mapped to a grid by the shared projection and decoded. No TimesFM finetuning is performed.

- **General healthcare predictive framework (GenHPF-variant).** GenHPF (Hur et al., 2024) divides patient features hierarchically, encodes each medical event into an embedding followed by a step to aggregate them. We create a variant of this method with minor modifications to their textual templates and follow the same downstream learning pipeline. [3]

## 3.3 DATASETS AND PREPROCESSING

We use three ICU cohorts: MIMIC-IV (Johnson et al., 2023a), HiRID (Yèche et al., 2021), and PPICU. PPICU is an external dataset from a distinct hospital system. In total, the cohorts include

---

[2] We perform embedding model ablations; the results are recorded in Appendix L.

[3] Details on how mapping was done are shown in Appendix N.

| Dataset → | HiRID | | | | | MIMIC | | | | | PPICU | | | | | Wins |
|---|---|---|---|---|---|---|---|---|---|---|---|---|---|---|---|---|
| Tasks → Method ↓ | Forecast MSE | LoS MAE | Mort AUROC | Drug Recall | Lab Recall | Forecast MSE | LoS MAE | Mort AUROC | Drug Recall | Lab Recall | Forecast MSE | LoS MAE | Mort AUROC | Drug Recall | Lab Recall | |
| Mean | 0.040 | 0.378 | 0.914 | 0.878 | 0.857 | 0.035 | 0.447 | 0.847 | 0.838 | 0.886 | 0.028 | 0.528 | 0.842 | 0.834 | 0.847 | 0 |
| Right shift | 0.041 | **0.342** | 0.923 | 0.884 | 0.858 | 0.037 | 0.409 | 0.886 | 0.841 | 0.884 | 0.031 | 0.490 | 0.868 | 0.831 | 0.846 | 1 |
| Interpolation | 0.435 | 0.370 | 0.918 | 0.879 | 0.853 | 0.103 | 0.430 | 0.873 | 0.844 | 0.893 | 0.133 | 0.493 | 0.857 | 0.837 | 0.847 | 0 |
| TSDE | 0.029 | 0.411 | 0.923 | 0.901 | 0.902 | 0.030 | 0.406 | **0.915** | 0.888 | 0.899 | 0.053 | 0.485 | **0.890** | 0.899 | 0.870 | 1 |
| TimesFM | 0.028 | 0.440 | 0.826 | 0.850 | 0.925 | 0.030 | 0.413 | 0.791 | 0.806 | **0.940** | 0.036 | 0.494 | 0.658 | 0.923 | 0.925 | 0 |
| GenHPF | – | – | 0.836 | 0.770 | 0.741 | – | – | 0.776 | 0.720 | 0.713 | – | – | 0.780 | 0.835 | 0.6773 | 0 |
| Record2Vec | **0.021** | 0.347 | **0.930** | **0.911** | **0.931** | **0.027** | **0.328** | 0.888 | **0.903** | **0.947** | **0.017** | **0.358** | **0.890** | **0.937** | **0.936** | 13 |

Table 1: In-distribution Result (RQ1). Best (per column) in **bold**; second-best underlined. *Wins* counts the number of bests per method across all 15 downstream tasks.

57,212 stays with 60 variables for MIMIC-IV, 32,216 stays with 64 variables for HiRID, and 39,000 stays with 75 variables for PPICU.[4] We segment each stay into non-overlapping windows of 48 hours. The variables cover laboratory tests, vital signs, and clinical interventions.

For the grid baselines, we normalize values within each cohort using statistics computed on the corresponding training split, consistent with prior benchmarks where normalization is crucial for numeric model stability. For text-based transformations, we keep raw magnitudes and the native clinical units to preserve meaning in the summaries i.e we do not perform further modification of the raw data. For language inputs, we create a canonical serialization of the window. For each variable present, the serialization lists the name and the full sequence of timestamps and raw values observed during the window. This serialization is the input to the summarizers and also serves as the no-summarization control described in Section 3.1. Additional details on concept mappings, unit handling, and filtering rules, which were curated with practicing ICU clinicians to ensure correct harmonization of variables, appear in Appendix C.2.

## 3.4 Tasks and Evaluation

We study five predictive tasks and two privacy probes. The tasks are: multivariate forecasting of all variables over the next 24 hours; remaining length of stay at the window end; in-hospital mortality; two treatment indicators for vasopressor and antibiotic use in the next 24 hours; and a multi-label outcome for whether ten common blood tests will be ordered in the next 24 hours. The privacy probes test demographic recoverability by predicting age (clipped to 18–90) and sex.

We use three settings. In-distribution trains and validates within one cohort and tests on its holdout split. Cross-site trains on a source cohort and tests on a distinct target without target labels; we report target accuracy and the drop relative to in-distribution. Few-shot starts from a source-trained model and fine-tunes on 16, 64, or 512 labeled target examples, then tests on the target split.

We report MAE for length of stay and age, masked MSE for forecasting, and micro-averaged recall (with precision and F1) for classification. Results average four seeds with mean and standard deviation. Budgets, early stopping, capacity, and regularization are matched within input type. TSDE is trained on the training split with validation selection. TimesFM, $g_\phi$, and $h_\psi$ remain frozen. Full hyperparameters appear in Appendix D.

## 4 Results

We organize the evaluation around seven research questions probing in-distribution performance (RQ1), cross-cohort transfer (RQ2), the value of summarization and model choice (RQ3), prompt sensitivity (RQ4), few-shot adaptation (RQ5), privacy (RQ6), and information analysis (RQ7). Unless noted, results tables report means over seeds with the top performer highlighted. For consistency, the results sections report numbers from a single downstream model (PatchTSMixer), except for mortality, where we report the best performance across all models tested in the mortality benchmark (Yèche et al., 2022); we verified the same trends hold with neural network architectures including multi-layer perceptron (MLP), LSTM (Hochreiter & Schmidhuber, 1997), TimeMixer (Wang et al., 2024b) in Appendix O.

### 4.1 How do the four methods compare in distribution? (RQ1)

**Record2Vec achieves the strongest in-distribution results overall, winning 13 of 15 tasks and ranking second on the remaining two.**

---

[4]Differences in EHR systems lead to different variable sets across sites.

| Dataset → | HiRID → PPICU | | | | | MIMIC → PPICU | | | | | Wins |
|---|---|---|---|---|---|---|---|---|---|---|---|
| Tasks → 
 Method ↓ | Forecast 
 MSE | LoS 
 MAE | Mort 
 AUROC | Drug 
 Recall | Lab 
 Recall | Forecast 
 MSE | LoS 
 MAE | Mort 
 AUROC | Drug 
 Recall | Lab 
 Recall | |
| Mean | 0.306 | 1.09 | 0.50 | 0.42 | 0.77 | 4.839 | 0.85 | 0.50 | 0.30 | 0.81 | 0 |
| Right shift | 0.417 | 1.03 | 0.50 | 0.53 | 0.88 | 6.344 | 0.76 | 0.50 | 0.23 | 0.82 | 0 |
| Interpolation | 24.80 | 0.91 | 0.50 | 0.34 | 0.81 | 13.41 | 0.71 | 0.50 | 0.21 | 0.80 | 0 |
| TSDE | 0.209 | 0.73 | 0.49 | 0.91 | 0.91 | 0.269 | 0.76 | 0.51 | 0.90 | 0.90 | 0 |
| TimesFM | 0.217 | 0.80 | 0.64 | 0.93 | 0.95 | 0.284 | 0.93 | 0.69 | 0.95 | 0.95 | 2 |
| GenHPF | - | - | 0.582 | 0.699 | 0.316 | - | - | 0.425 | 0.442 | 0.428 | 2 |
| Record2Vec Template | 0.195 | 0.98 | 0.72 | 0.96 | 0.96 | 0.263 | 0.77 | 0.71 | 0.95 | 0.95 | 2 |
| Record2Vec | 0.183 | 0.69 | 0.72 | 0.97 | 0.97 | 0.242 | 0.49 | 0.72 | 0.95 | 0.95 | 10 |

Table 2: Transfer Learning Result (RQ2). Best per column in **bold**; second-best underlined. Wins counts tied bests.

Table 1 summarizes performance within each cohort. A clear pattern emerges: Record2Vec leads on most outcomes in HiRID, sweeps all tasks in MIMIC, and again dominates in PPICU with one exception. The two columns not led by Record2Vec are PPICU mortality, where TSDE is best, and HiRID length of stay, where a simple right-shift imputation edges ahead. Representation-learning baselines (TSDE, TimesFM) are consistently competitive and often place second, while classic grid imputations rarely win and generally trail across cohorts and endpoints. These results indicate that a language-mediated input yields robust utility across forecasting, regression, and classification without tailoring model architectures to a site.

We hypothesize three drivers. First, summarization captures clinically salient semantics—states, trajectories, interventions, salient events—while aligning heterogeneous names, units, and sampling policies into a shared space. This reduces reliance on ad hoc imputations and preserves meaning when measurements are sparse or irregular. Second, the fixed-length embedding offered by Record2Vec stabilizes the downstream interface, limiting sensitivity to missingness patterns and local measurement habits that degrade grid-based inputs. Third, compared with TSDE and TimesFM, which emphasize correlations and trends in numeric streams, Record2Vec adds a layer of clinical context that appears to aid discrimination on classification endpoints while remaining competitive on regression. Overall, the findings support the view that LLM-driven transformation produces more informative and portable inputs than either imputation or purely numeric representation learning.

## 4.2 DOES THE LLM BASED REPRESENTATION IMPROVE TRANSFERABILITY ACROSS ICUS? (RQ2)

**Record2Vec transfers best across ICUs, winning 10 of 10 columns in Table 2 with two ties by TimesFM.**

When models are trained on HiRID or MIMIC and evaluated on PPICU, Record2Vec consistently ranks first across forecasting, length of stay, mortality, treatment, and lab prediction. In contrast, grid imputations degrade sharply under shift, and several classification scores collapse toward chance. TSDE and TimesFM are stronger than imputation, yet they still trail Record2Vec in most settings. TimesFM ties for the top position in two columns and, together with Record2Vec, benefits from the larger MIMIC source when predicting common interventions and labs at PPICU. The pattern is clear: methods that rely on site-specific numeric regularization or self-supervision on the source cohort do not maintain accuracy when confronted with new variable sets, sampling habits, and missingness regimes.

We attribute these results to three factors. First, natural-language summaries align heterogeneous coding choices, units, and documentation styles into a shared clinical space before embedding. This reduces the burden on downstream decoders to re-learn site-specific conventions and helps preserve signal for tasks like mortality and treatment prediction. Second, the fixed-length embedding produced by Record2Vec offers a stable input interface that is less sensitive to irregular sampling and missingness patterns than grids, which explains the large gap from imputation under distribution shift. Third, compared with TSDE and TimesFM, Record2Vec injects semantic context about states, trends, and salient events extracted by the summarizer. To better understand the gains from summarization, we compared against the strategy used by related work (Hur et al., 2024; Gao et al., 2024; Hegselmann et al., 2025) to map patient data onto a fixed template. When comparing Record2Vec with TSDE, and TFM, against fixed-template baselines, we observe additional performance gains that can be attributed to summarization. We hypothesize these gains primarily come due to the standardization of heterogeneous patient profiles across sites while preserving biologically relevant in-

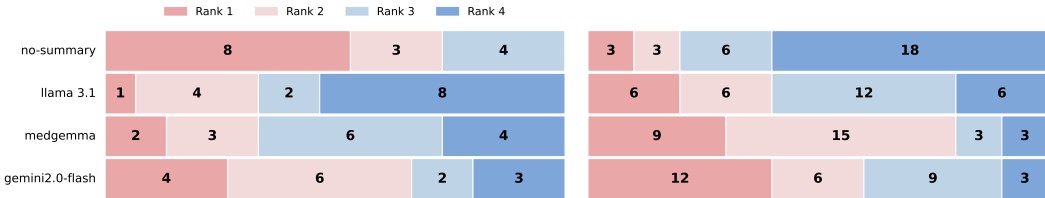

Figure 3: Rank distributions for No-summary vs. three LLM variants across 15 in-distribution tasks (left) and 30 cross-site transfer tasks (right). Methods are ranked based on performance across five downstream tasks: Forecast (MSE), LoS (MAE), Mortality (AUROC), Drug (Recall), and Lab (Recall). See the Appendix M for the detailed values.

formation for making predictions.Foundation pretraining helps TimesFM transfer better than TSDE, but it remains tied to numeric trends alone, whereas Record2Vec retains both numerical and clinical meaning. Together, these properties yield robust portability across hospitals for a broad set of ICU prediction tasks.

### 4.3 DO SUMMARIES HELP, AND WHICH SUMMARIZER WORKS BEST? (RQ3)

**Summarize-then-embed improves cross-site transfer while staying competitive in-distribution, and among summarizers Gemini-2.0 Flash and MedGemma perform best, with Llama-3.1 worse than both.**

Figure 3 shows rank distributions for four variants: *no-summary* (directly embed the raw serialization), and three LLM summarizers (Llama-3.1, MedGemma, Gemini-2.0 Flash). On the left (in-distribution; 15 tasks), *no-summary* often attains the top rank, with the LLM-based variants close behind. Within a single site, feeding the embedding model the full, unsummarized content appears advantageous and does not require normalization into prose. On the right (cross-site transfer; 30 tasks), the pattern reverses. *No-summary* concentrates in the worst ranks overall, while the three summarize-then-embed variants dominate first and second ranks. Comparing the LLMs, Gemini-2.0 Flash and MedGemma account for most of the top positions in both settings; their gap narrows under transfer. Llama-3.1 accumulates the most last-place ranks among summarizers in both settings.

We think there are three reasons for these outcomes. First, summarization imposes a shared clinical language over heterogeneous variable names, units, and sampling practices, which removes site-specific idiosyncrasies before embedding. This harmonization matters most under shift and explains why Record2Vec transfers better than direct embedding of raw, site-specific streams. Second, within-site, direct embeddings preserve all numeric detail and local conventions, which can make representations highly distinctive and strong on held-out splits from the same cohort; however, precisely those site-specific details impair portability to other hospitals. Third, the choice of summarizer influences both fidelity and standardization. MedGemma's medical pretraining promotes clinically faithful phrasing that is stable across institutions, supporting transfer. Gemini-2.0 Flash's instruction following and planning yield concise, information-dense summaries that are marginally stronger in-distribution and remain competitive under transfer, leading to a small gap between the two. Llama-3.1, with smaller capacity and less domain specialization, tends to produce shorter or less standardized summaries, which hurts both within-site utility and cross-site robustness. **A major practical gain is efficiency: without summarization the raw serialization passes, on average, $\sim 25\times$ more tokens to the embedder than the summarized version, cutting inference cost proportionally while improving transfer.**

### 4.4 HOW SENSITIVE IS PERFORMANCE TO PROMPT DESIGN? (RQ4)

**Across four prompting strategies, performance is broadly similar in-distribution and under transfer, with a slight edge for ICD-style prompts in transfer.** Figure 4 plots rank distributions for zero-shot, chain-of-thought, trend-focused, and ICD-focused prompts using Gemini-2.0 Flash. The bars span ranks fairly evenly on both the in-distribution side and the cross-site side, indicating limited sensitivity to prompt choice at the aggregate level. We observe a small shift in transfer where ICD-style prompts collect more top ranks, while the other three strategies remain closely clustered. Within cohorts, no single prompt dominates consistently across forecasting, regression, or classification.

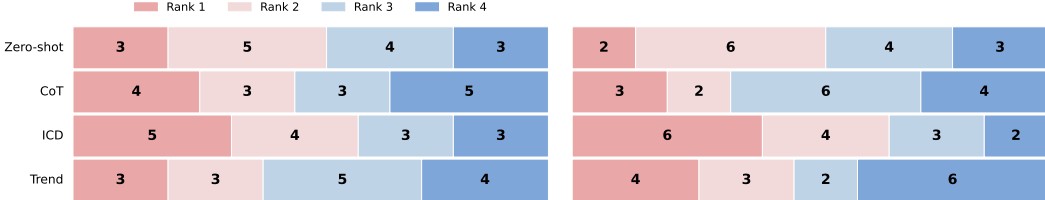

Figure 4: Rank distributions for four prompt variants with Gemini 2.0-flash across 15 in-distribution tasks (left) and 30 cross-site transfer tasks (right). Lower ranks indicate better performance. Methods are ranked based on performance across five downstream tasks: Forecast (MSE), LoS (MAE), Mortality (AUROC), Drug (Recall), and Lab (Recall). See the Appendix M for the detailed values.

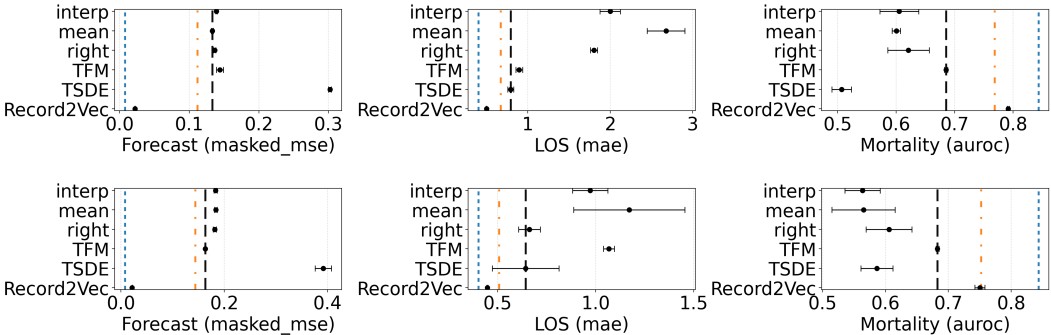

Figure 5: Few-shot finetuning with 16 labeled target samples for mortality prediction (RQ5) shown across six transfer settings. All tasks are reported the same metric as previous section: Forecast: masked mse, LOS: mae, Mortality: AUROC. The first row is the result of Hirid → Ppicu and the second row is Mimic → Ppicu. Reference lines: blue = best in-distribution upper bound, orange = best pre-finetune result, black = best finetuned baseline. Record2Vec surpassed baselines with a large gap, reaching comparable performance to in-distribution.

We think two factors explain this pattern. First, the summarizer is strong enough that high-level clinical content is retained under all four prompts, producing broadly comparable embeddings. Second, the ICD framing may aid transfer by nudging summaries toward standardized terminology that travels better across sites. Overall, these results suggest that prompt choice is not the primary driver of utility in our setup, and that more targeted prompt engineering could be a promising avenue for future improvement. Detailed prompt templates appear in Appendix E.

## 4.5 DOES THE LLM BASED REPRESENTATION IMPROVE FEW-SHOT DOWNSTREAM LEARNING? (RQ5)

**Record2Vec needs fewer samples to achieve generalization on small clinical datasets.** Hospitals that are not part of large medical centers often lack sufficient patient data to train high-performing models from scratch. In Appendix I, we compare models trained on 1,000 PPICU samples with Record2Vec models that are pre-trained on HiRID or MIMIC and then fine-tuned on only 16 randomly selected PPICU samples. Beyond the gains from in-distribution training, Record2Vec delivers substantial improvements over its transfer variants and outperforms all competing methods under the same supervision budget. In multiple settings, the adapted Record2Vec models approach the in-distribution reference model trained on 36,019 samples, indicating that very limited supervision can recover most of the performance lost under distribution shift.

These gains likely arise because Record2Vec produces a compact representation that is already semantically aligned across hospitals, reducing the burden on the downstream head to learn site-specific conventions from scratch. Finetuning can therefore act mainly as a light calibration of task boundaries rather than a wholesale re-learning of measurement scales, missingness patterns, and naming differences. In effect, the summary abstracts away local differences while preserving clinically salient signals—states, trends, and recent interventions—that are informative for mortality, length of stay, and near-term actions. This combination of cross-site alignment and retained clinical

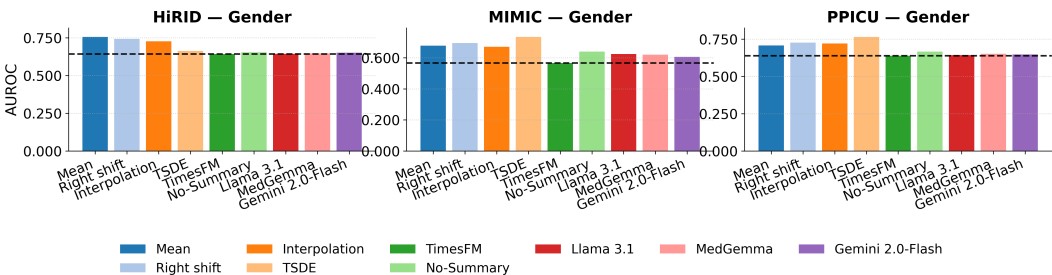

Figure 6: In-distribution privacy prediction on **Gender** across three ICU datasets. Each bar is a different method. The dashed horizontal line in each panel marks the least leaked classical baseline among {TimesFM, TSDE, Mean, Interpolation, Right shift} for that metric/dataset. Our method reached comparable and mostly reduced results in terms of privacy leakage. **Age** results have less than 0.5% gap and can be inferred from Table 4

content makes Record2Vec especially data-efficient in shifted settings, enabling effective few-shot learning when labeled target data are scarce.

## 4.6    Does the portable representation increase privacy risks? (RQ6)

**We find no evidence that Record2Vec increases demographic leakage risk: gender prediction collapses to a constant baseline for all methods, and Record2Vec's age error is similar to or higher than baselines.** Record2Vec's MAE is generally on par with competing approaches across cohorts, suggesting that the portable representation does not make age easier to infer. Figure 6 visualizes demographic recoverability from learned representations. For *gender* (binary), models trained on embeddings from grids, TSDE, TimesFM, and Record2Vec all degenerate to predicting a single class on held-out data, indicating near-chance information about gender regardless of method. We additionally examined performance across minority diagnostic subgroups and found no disparate impact, with Record2Vec tracking overall population trends without exhibiting worst-group performance degradation relative to baselines. Detailed numbers appear in Appendix O.

We think this outcome reflects how the pipeline shapes information. Record2Vec emphasizes clinical states, trends, and recent interventions needed for downstream tasks, rather than demographic markers. Because the summarizer and text encoder are frozen and not trained to predict demographics, they do not amplify demographic signal beyond what is already present in the record windows. In practice, the summaries in our setup rarely include explicit age or sex mentions, and the embedding stage is not optimized to capture them, which helps keep demographic recoverability low while preserving clinical utility. However, we emphasize that these results are specific to demographic leakage and do not preclude other privacy risks such as membership inference or embedding inversion.

## 4.7    What information is obtained or lost in each embedding? (RQ7)

**Record2Vec captures semantically meaningful structure that improves in-distribution performance and, importantly, encodes cross-site invariant patterns, while attenuating demographic attributes.** We compare the Record2Vec embedding—MedGemma summaries prompted with ICD codes—against imputation-based vectors to quantify task-specific information gain or loss (Figure 7). Unless stated otherwise, "gain" denotes the relative improvement ($\Delta\%$) of a metric aligned with task direction (e.g., lower for error; higher for AUROC/recall) with respect to the imputation baseline.

For in-distribution, the largest gains are observed for *forecasting* and *mortality* prediction, indicating that LLM-derived semantic abstractions, such as ICD-anchored summarization, improve outcome modeling and strengthen associations with significant temporal features. For cross-site transfer, We observe consistent improvements for *lab*, *drug*, *forecast*, *LOS*, and *mortality*, with the most pronounced effect on future drug-use prediction. This implies that *Record2Vec* captures patterns that generalize across institutions and reduces sensitivity to site-specific artifacts. For demographic attenuation, signals tied to *age* and *gender* are generally weakened or unchanged. This suggests

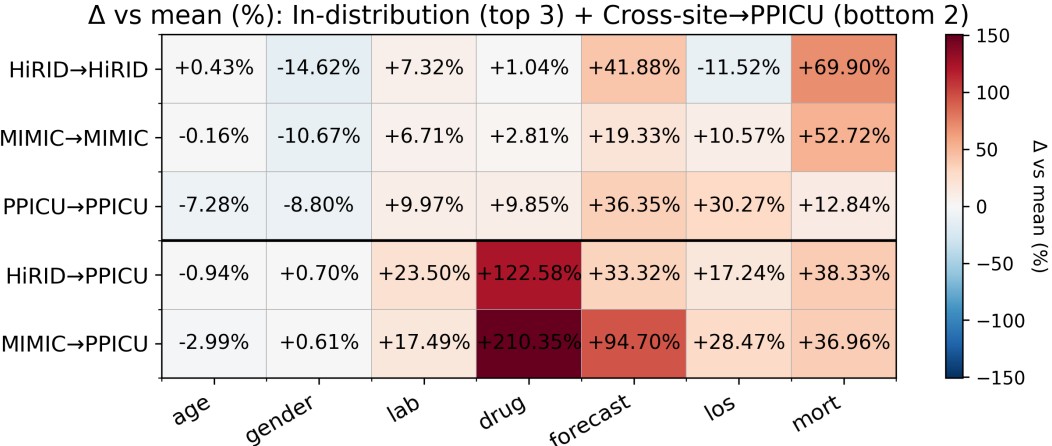

Figure 7: **Information gained vs. lost with Record2Vec (heatmaps; in-distribution and cross-site transfer).** Each cell shows the *task-aligned* relative change (%) of *Record2Vec* over representation vectors. Overall, *Record2Vec* yields the largest gains for *drug*, *forecast*, and *mortality*, including cross-site transfer, while demographic signals (*age*, *gender*) are attenuated or unchanged.

*Record2Vec* does not amplify, and mostly reduces, demographic attribute leakage relative to raw time series, which is another significant advantage under healthcare settings.

## 5  LIMITATIONS

Our framework requires sending patient records to LLM services. Although all experiments were conducted in a secure, IRB-compliant setting with strict controls, some patients and families may be uncomfortable with external processing. At present, this confines the approach to research use unless on-prem or fully compliant deployments are available. Generating LLM-based representations can be resource-intensive. Many hospitals may lack the compute or budget to support large-scale or continuous inference without additional investment (see Appendix H for a detailed analysis of token counts, latency, and costs). Finally, API-based or large local LLMs introduce non-trivial latency compared to lighter models, which can limit real-time applicability even if offline use remains practical. We acknowledge that our study is primarily empirical; we believe developing the theory for when and where this methodology works would significantly improve the reach of our work. We rely on downstream task performance as a proxy for representation quality and do not currently utilize theoretical tools (such as causal diagrams) to formally derive the conditions for site-invariance. Furthermore, we do not directly quantify end-to-end information loss (e.g., via mutual information or per-feature fidelity checks). While our downstream accuracy suggests that our method preserves important signals, a limitation of current work, it is challenging to characterize exactly what information is retained or lost during summarization.

## 6  CONCLUSION

Transforming irregular ICU records by first summarizing with an LLM and then embedding offers a promising route to portable, task-ready representations. Our results indicate competitive in-distribution performance, strong transferability, and improved few-shot learning without increased demographic leakage. We hypothesize that LLMs can leverage their learned priors to produce summaries that are standardized across sites, rich in biological information relevant to downstream decision making and close (in a distributional sense) across sites, enabling the portability of downstream predictive models. Nevertheless, deployment risks remain, particularly around privacy, cost, and latency. Future work should focus on creating compliant on-prem solutions, and prospective evaluations within clinical workflows. Assessing feature-level fidelity and developing theory to better understand and control the properties of LLM-based standardization is an important direction to understand how far this approach can be extended.

REPRODUCIBILITY STATEMENT

We take several steps to enable independent verification of our results. Data preprocessing (filtering, resampling, normalization, masking, and label construction) is documented in Appendix C.2. HiRID and MIMIC-IV are publicly accessible via PhysioNet; PPICU is a privately maintained dataset that will be shared upon reasonable request to the authors once the paper is public and de-anonymized, in accordance with institutional and data-use policies. Training details, including hardware requirements, software environment with pinned versions, all hyperparameters, and early-stopping criteria, are provided in Appendix D. To assess robustness, we tune key hyperparameters and repeat training/evaluation across multiple random seeds; the resulting analyses indicate our conclusions are stable to such variations. The code to reproduce our results is available at https://github.com/Jerryji007/Record2Vec-ICLR2026.

We take several steps to enable independent verification of our results. Data preprocessing (filtering, resampling, normalization, masking, and label construction) is documented in Appendix C.2. HiRID and MIMIC-IV are publicly accessible via PhysioNet; PPICU is a privately maintained dataset that will be shared upon reasonable request to the authors once the paper is public and de-anonymized, in accordance with institutional and data-use policies. Training details—including hardware requirements, software environment with pinned versions, all hyperparameters, and early-stopping criteria—are provided in Appendix D. To assess robustness, we tune key hyperparameters and repeat training/evaluation across multiple random seeds; the resulting analyses indicate our conclusions are stable to such variations.

ETHICS STATEMENT

**Scope and oversight.** Our work is in machine learning for healthcare and involves processing de-identified patient records under institutional oversight. All activities complied with applicable regulations and institutional policies (including HIPAA in the United States and the policies of our institutional review/ethics board). Data use agreements were honored throughout.

**Training and authorization.** All authors completed required human-subjects and privacy training and are certified/authorized to handle clinical data in secure research environments.

**Data protection and privacy.** We followed data-minimization and least-privilege principles; access to protected health information (PHI) was restricted to approved personnel and audit-logged. Data were stored on access-controlled servers with encryption in transit and at rest; exports outside secure infrastructure were prohibited. When sharing intermediate artifacts (e.g., for internal review), we ensured de-identification and suppression of quasi-identifiers.

**Risk assessment and mitigation.** We considered potential harms (privacy leakage, representational bias, inequitable performance) and mitigated them via dataset curation checks, stratified evaluation where applicable, and manual review of outputs used in the paper. No clinical decisions were made using the research system.

**LLM usage.** We employed LLMs in strictly limited ways (see Section A): (i) locally hosted models on institution-managed GPUs within secure, access-controlled servers; and (ii) third-party services (Gemini and ChatGPT families) *only* for grammar/wording polish of author-written text and for generating small iconography within figures (not full figures and not scientific content). No methods, ideas, analyses, code, or experiments were generated by LLMs, and all suggestions were manually reviewed.

**Gemini 2.0 Flash configuration.** Under a dedicated research agreement, we used a sandboxed deployment of the Gemini 2.0 Flash family strictly for research purposes. For any interactions related to healthcare data:

- prompts and inputs excluded PHI whenever possible; when unavoidable for evaluation, only de-identified data were used;

- inference occurred in an access-controlled environment with logging;

- data-retention was disabled per our agreement; the model does not retain or use our prompts or outputs for provider-side training or service improvement.

**Author responsibility.** Beyond copy-editing and minor figure iconography, LLMs were not used. All ideas, study design, algorithm development, experiments, analyses, and substantive writing were performed by the authors, who take full responsibility for the work and its ethical conduct.

ACKNOWLEDGMENTS

RGK is supported by a Canada CIFAR AI Chair and a Canada Research Chair Tier II in Computational Medicine (CRC-2022-00049). This research was supported by an NFRF Special Call NFRFR2022-00526. Resources used in preparing this research were provided, in part, by the Province of Ontario, the Government of Canada through CIFAR, and companies sponsoring the Vector Institute.

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

## A    USAGE OF LARGE LANGUAGE MODELS

We used large language models (LLMs) in two strictly limited ways:

1. **Locally hosted models.**  For development convenience, we ran LLMs on institution-managed GPUs within access-controlled, offline (or firewall-restricted) servers.  These models were used only for debugging assistance and rapid prototyping under our secure computing environment; no research data were exported outside these servers.

2. **Third-party services for copy-editing and minor figure elements.** We used the Gemini and ChatGPT families solely to polish grammar and wording of author-written text and to generate small iconography embedded within figures (e.g., simple symbols), *not* full figures and not scientific content. We did not use these services to generate methods, ideas, analyses, code, or experiments. No sensitive or identifying data were included in prompts. All suggested edits were manually reviewed by the authors.

Beyond the above, LLMs were *not* involved in the conception of ideas, study design, algorithm development, experiments, analyses, or writing of substantive content. All contributions—conceptual, methodological, experimental, and explanatory—are entirely by the authors.

## B    EXTENDED RELATED WORK

**Privacy and attribute inference in embeddings.**  Learned representations may inadvertently reveal sensitive attributes. Recent analyses show that embeddings can leak information via inversion, membership, or attribute–inference attacks, even when protected features are not explicit (Song & Raghunathan, 2020; Zhu et al., 2024).  Emerging auditing frameworks evaluate demographic attribute inference from model outputs and intermediate representations (Panda et al., 2025).  We therefore adopt demographic recoverability (age and sex) from learned embeddings as a practical proxy for privacy risk, situating our evaluation alongside this auditing literature.

**ICU Time Series and Clinical Benchmarks** Public ICU datasets drove clinical ML, notably MIMIC and eICU (Johnson et al., 2016; Pollard et al., 2018; Johnson et al., 2023b). Early pipelines were bespoke (Harutyunyan et al., 2019; McDermott et al., 2021), prompting standardized benchmarks: canonical MIMIC tasks (Harutyunyan et al., 2019), the HiRID ICU benchmark with high frequency signals and evaluation code (Yèche et al., 2021), and YAIB harmonizing MIMIC, eICU, HiRID, and AmsterdamUMCdb (van de Water et al., 2024; Thoral et al., 2021). Aggregations such as BlendedICU broaden multi center evaluation (Oliver et al., 2023), and foundation pretraining over heterogeneous ICU series is emerging (Burger et al., 2024). Deep sequential models improve early warning and trajectory modeling (Hyland et al., 2020; Tomasev et al., 2019; Ji et al., 2024; 2025), and EHR sequence models reach expert level outcomes (Rajkomar et al., 2018). Persistent challenges include explainability, irregular sampling, and missingness (Morid et al., 2021), addressed by interpretable modeling and transformer forecasting in clinical settings (Tan et al., 2021; Hartvigsen et al., 2022; Zhang et al., 2024a). Surveys stress the influence of curation, task design, and preprocessing (Núñez Reiz et al., 2019; Shickel et al., 2018; Suresh et al., 2017). Recent pipelines underscore standardization needs (Gupta et al., 2022; He & Chiang, 2025).

**Imputation and Representation Learning for Time Series.** Irregular sampling and missingness are central in clinical time series. Model based imputers encode absence patterns or learn imputation with prediction (Che et al., 2018; Cao et al., 2018), while generative methods improve accuracy through diffusion and realistic dynamics (Tashiro et al., 2021; Yoon et al., 2019). Continuous time and structure aware encoders address nonuniform sampling and uncertainty (Fortuin et al., 2020; Rubanova et al., 2019; De Brouwer et al., 2019; Kidger et al., 2020; Shukla & Marlin, 2021). Self supervised objectives and transformer encoders learn general purpose representations (Eldele et al., 2021; Yue et al., 2022; Woo et al., 2022; Zerveas et al., 2021; Franceschi et al., 2019). Diffusion style self supervision couples imputation, interpolation, and forecasting masks for transferable embeddings (Senane et al., 2024). Decoder only foundation pretraining yields strong forecasting across domains (Das et al., 2024). Earlier work spans dimensionality reduction and deep recurrent networks (Li, 2019; Balamurali, 2023; Siami-Namini et al., 2019). Recent ideas explore discrete tokenization and approximating LLM embedding spaces (Talukder et al., 2025; Sun et al., 2024). In healthcare, pattern based embeddings and semantic grouping improve interpretability (Feremans et al., 2022;

Kuznetsova et al., 2023). Surveys advocate universal representations robust to noise, sparsity, and shift (Trirat et al., 2024).

**Large Language Models in Healthcare and Time Series.** Large language models have advanced clinical NLP and representation learning. Domain pretrained transformers on coded EHRs improve disease prediction (Li et al., 2020a; Rasmy et al., 2021), clinical scale models achieve strong extraction and inference (Yang et al., 2022; Peng et al., 2023), and general medical systems perform well on exams and challenges (Singhal et al., 2023; 2024; Nori et al., 2023). Early adaptations demonstrated clinical question answering and discharge summarization (Agrawal et al., 2022; Lang et al., 2022; Van Veen et al., 2023). For patient level prediction, audits report underperformance and safety concerns (Brown et al., 2025; Wu et al., 2025; Griot et al., 2025; Moor et al., 2023). Bridging modalities includes in modality pretraining, cross modality transfer and reprogramming, prompt tuning, cross modal fine tuning, autoregressive prediction, and alignment with language space (Kambale et al., 2023; Zhou et al., 2023; Chang et al., 2025; Jin et al., 2024; Cao et al., 2024; Liu et al., 2024a;b; 2025). Textualization strategies often trail specialized forecasters (Ansari et al., 2024; Gruver et al., 2023; Tan et al., 2024). Few shot models show gains, and surveys emphasize formatting, scaling, and evaluation (Liu et al., 2023; Zhang et al., 2024b; Liventsev & Fritz, 2024). Advances in embeddings and domain specific language models motivate LLM derived representations for ICU data (Pennington et al., 2014; Mikolov et al., 2013; Reimers & Gurevych, 2019; Su et al., 2022; Muennighoff et al., 2022; Lee et al., 2024; BehnamGhader et al., 2024; Wang et al., 2024a; Luo et al., 2022; Alsentzer et al., 2019).

Recent studies have also investigated serializing structured EHR data for LLM processing. Approaches like TabLLM (Hegselmann et al., 2023) and DeLLiriuM (Contreras et al., 2025) fine-tune LLMs on serialized records for tabular classification or specific risk predictions. Others, such as Lee et al. (2025), Gao et al. (2024), and Hegselmann et al. (2025), explore using frozen LLMs to embed raw data serializations directly. While these methods demonstrate the utility of LLMs for medical data, they primarily focus on sparse longitudinal records or single-site benchmarks. Our work distinguishes itself by addressing dense, irregular ICU time-series, where we find that direct serialization yields excessive sequence lengths and brittleness to distribution shifts, necessitating a summarization step for effective cross-site portability.

Additionally, domain-specific foundation models like Med-BERT (Rasmy et al., 2021) and GenHPF (Hur et al., 2024) apply Transformer-based pretraining to structured EHR codes. While effective for longitudinal disease prediction within a specific health system, these models rely on fixed vocabularies that can be sensitive to schema variations and distribution shifts across institutions. Furthermore, they often struggle to capture the high-frequency numeric dynamics of intensive care data compared to generalist time-series models or semantic LLM representations, which offer greater flexibility and transferability in diverse deployment settings.

## C  DATASET AND PROCESSING

### C.1  GENERAL INFORMATION

**MIMIC (80,749 samples).** The processed ICU time series comprises 60 features: `ALP, ALT, Bicarbonate, Bilirubin, BloodUreaNitrogen, Calcium, Creatinine, CreatinineKinase, Hemoglobin, INR, Lactate, Magnesium, PaCO2, PaO2, Phosphate, Platelets, Potassium, Sodium, Troponin, WBC, ph, AirwayPressure, DiastolicBloodPressure, FiO2, GCS, HeartRate, ICDSC, MeanBloodPressure, MinuteVentilation, PEEP, RespiratoryRate, SAS, SystolicBloodPressure, Temperature, TidalVolume, UrineOutput, Analgesia, Antiarrhythmics, Antibiotics, Anticoagulants, Antiepileptics, Antihypertensives, Antipsychotics, CTScan, CaReplacement, Dialysis, Diuretics, EnteralNutrition, ICPMonitor, KReplacement, MRI, MgReplacement, PPI, Paralysis, TPN, Transfusions, UltraSound, Vasopressors, Ventilation, Xray.`

**HiRID (36,019 samples).** The processed ICU time series comprises 64 features: `ALP, ALT, AST, Bicarbonate, Bilirubin, BloodUreaNitrogen, Calcium, Chloride, CreatineKinase, Creatinine, Glucose, Hemoglobin, INR, Lactate, Magnesium, PaCO2, PaO2, Phosphate, Platelets, Potassium, Sodium, Troponin, WBC, ph,`

```
AirwayPressure, AirwayPressurePeak, DiastolicBloodPressure, FiO2,
FluidBalance, GCS, HeartRate, MeanBloodPressure, PEEP, RespiratoryRate,
SAS, Saturation, SystolicBloodPressure, Temperature, TidalVolume,
UrineOutput, Analgesia, Antiarrhythmics, Antibiotics, Anticoagulants,
Antiepileptics, Antihypertensives, Aspirin, CaReplacement, Dialysis,
Diuretics, ICPMonitor, Insulin, KReplacement, LiverToxicDrug,
MgReplacement, Neuroleptics, Paralysis, Saline, Sedation, Steroids,
TPN, Transfusions, Vasopressors, Ventilation.
```

**PPICU (47,119 samples).** The processed ICU time series comprises 75 features:
```
ALP, ALT, AST, Bicarbonate, Bilirubin, BloodUreaNitrogen, Calcium,
Chloride, Creatinine, CreatinineKinase, GGT, Glucose, Hemoglobin,
INR, Lactate, Magnesium, PaCO2, PaO2, Phosphate, Platelets, Potassium,
Sodium, Troponin, WBC, ph, AirwayPressure, AirwayPressureIP,
AirwayPressurePeak, DiastolicBloodPressure, FiO2, FluidInput,
FluidOutput, GCS, HeartRate, ICDSC, MeanBloodPressure, MinuteVentilation,
PAVSupport, PC, PEEP, PlateauPressure, RespiratoryRate, SAS,
Saturation, SystolicBloodPressure, Temperature, TidalVolume,
UrineOutput, Analgesia, Antiarrhythmic, Antiarrhythmics, Antibiotics,
Anticoagulants, Antiepileptics, Antihypertensives, Aspirin, Dialysis,
Diuretics, EKG, EVD, ICPMonitor, InhaledVasodilator, Insulin,
Isoproterenol, MgReplacement, Neuroleptics, OsmoticTherapy, PPI, PS,
Paralysis, Sedation, Steroids, TPN, Transfusions, Vasopressors.
```

C.2 DATA PREPROCESSING DETAILS

We provide reproducible preprocessing pipelines for all ICU datasets considered in this work. This appendix summarizes the steps at a high level, without referencing specific files or directories.

**Common setup.** We organize raw and processed data under a central data directory. Access to each dataset follows the relevant data use agreements. Unless stated otherwise, per-stay sequences are constructed at hourly resolution, and features are normalized using statistics computed on the training split. All clinical features are selected in discussion with experienced practicing ICU clinicians.

HIRID

We first obtain authorized access and download the raw HiRID release. Because HiRID associates each patient with a single ICU stay, we group all clinical and medication records by patient identifier and construct a sparse, event-centric representation per patient. We then curate clinically meaningful features and extend the feature dictionary to include start/end times and summary statistics for quality control and normalization. Using this curated set, we produce patient-level targets and serialize time indices, value indices, and per-event counts for downstream modeling. Finally, we standardize signals and combine them on a per-feature basis to ensure consistent scaling across patients and over time.

MIMIC-IV

We begin by obtaining credentialed access and downloading the raw MIMIC-IV data, then convert the raw tables into a columnar layout to improve I/O efficiency. Next, we filter to ICU stays that exceed a minimum duration threshold, retaining this subset for further processing. In collaboration with clinicians, we develop a mapping from measurement identifiers to a study feature set; this mapping is intentionally decoupled so it can be revised without reprocessing earlier stages. We restrict event tables to the curated features and carry forward only necessary metadata into a project-specific processed area. Core administrative and ICU stay tables are merged to derive age, gender, mortality, and length-of-stay labels, excluding pediatric subjects for adult-only cohorts. We then materialize a subject-centric layout (diagnoses, events, procedures, stays), preferring numeric event values where available, and write feature-separated storage to enable efficient access and validation across tasks.

PPICU

For PPICU, we convert all raw sources into a uniform, analysis-friendly tabular format and separate multi-signal streams into per-feature tables for efficient access; laboratory measurements are processed analogously. We merge demographic and outcome information to construct a source stay table with identifiers and key labels (age, gender, length of stay, mortality). The feature space is restricted to the analysis cohort, with per-feature summary statistics computed for clinician review; clinician-verified filters are then applied to remove implausible or out-of-range values. From the merged feature data, we generate per-stay tables sorted by measurement time and align feature names to a harmonized schema. A global time index keyed by stay identifier is built with arrays of time points at the chosen resolution. Each stay is finally converted into both a sparse dictionary-backed representation and a dense, zero-filled discrete representation, and we emit the resulting artifacts alongside train/validation/test splits.

# D  TRAINING DETAILS

## D.1  SYSTEM CONFIGURATION

All experiments were run on NVIDIA L40S GPUs. Unless noted, training and inference can be paused and fully completed on a single L40S (24 GB); large-scale reference runs used 4×L40S for throughput.

## D.2  TSDE EMBEDDING EXTRACTION

The TSDE model is trained in a self-supervised manner on the same training split as downstream tasks. We mask the final 24 h of each sequence and train the model to predict the masked segment. After convergence, we extract the hidden representation as the TSDE embedding for each record.

## D.3  TFM EMBEDDING EXTRACTION

We use the time-series foundation model `google/timesfm-1.0-200m`. We run inference on the full HiRID/MIMIC/PPICU datasets and extract representations from the last two transformer blocks. The penultimate layer yields consistently better downstream performance and is used in all main results.

## D.4  HYPERPARAMETER DETAILS

For baselines, we tune batch size and learning rate with early stopping and report the best configuration per method:

- **right_shift:** batch size 512, learning rate $1 \times 10^{-5}$
- **mean:** batch size 512, learning rate $4 \times 10^{-5}$
- **interpolation:** batch size 256, learning rate $5 \times 10^{-5}$

*Record2Vec* models are trained under a common setting: batch size 512, learning rate $1 \times 10^{-5}$. All methods are trained with seeds $\{42, 84, 1005, 2025\}$; we report the mean and standard deviation across seeds.

## D.5  FEW-SHOT FINETUNING (16-SAMPLE SETTING)

We study an extreme low-data regime with only 16 labeled target samples and no held-out validation set. Finetuning is performed with AdamW and a warmup–cosine schedule; the learning rate scales with the effective batch size:

$$\text{LR}_{\text{scaled}} = \text{BASE\_LR} \times \frac{\text{batch\_size}}{\text{REF\_FINETUNE\_BS}}, \qquad \text{BASE\_LR} = 1 \times 10^{-6}, \ \text{REF\_FINETUNE\_BS} = 16.$$

Weight decay is 0.01 with $\beta_1 = 0.9$, $\beta_2 = 0.999$. Let $S$ be the total number of optimization steps ($S = \text{num\_epochs} \times \text{steps\_per\_epoch}$) and $S_{\text{w}} = \max\{10, \lfloor 0.03\,S \rfloor\}$ the warmup steps. In the final

experiment, $S = 100 \times 16$. The per-step multiplier is

$$\lambda(t) = \begin{cases} \dfrac{t+1}{S_{\mathrm{w}}}, & t < S_{\mathrm{w}}, \\ \mathrm{MIN\_LR\_RATIO} + (1 - \mathrm{MIN\_LR\_RATIO}) \dfrac{1 + \cos\left(\pi \frac{t - S_{\mathrm{w}}}{S - S_{\mathrm{w}}}\right)}{2}, & t \geq S_{\mathrm{w}}, \end{cases}$$

with $\mathrm{MIN\_LR\_RATIO} = 0.10$. Unless otherwise specified, weight updates apply to both the backbone and the task decoder.

**Concrete finetuning setup (Shared across all methods).**

- **Batch size:** batch_size $= 16$, $\mathrm{LR}_{\mathrm{scaled}} = 1 \times 10^{-6}$.
- **Optimizer:** AdamW$(\mathrm{lr} = \mathrm{LR}_{\mathrm{scaled}}, \ \mathrm{weight\_decay} = 0.01, \ \beta = (0.9, 0.999))$.
- **Scheduler:** Warmup (first $\max\{10, 0.03S\}$ steps) then cosine decay to $0.1\times$ the peak LR.

## E  PROMPTS USED FOR LLM SUMMARIZATION AND A CASE STUDY

In RQ5, we discussed the effect of prompts on the quality of embedding. The patient data is summarized on a split-by-feature manner, raw text as following: `ALP has value 66 in hour 31. ALT has value 22 in hour 31. AST has value 17 in hour 31. Bilirubin has value 4 in hour 31. Blood Urea Nitrogen has value 5.6 in hour 31. Calcium has value 1.99 in hour 31. Chloride has value 115 in hour 31. Creatinine has value 103 in hour 31. Creatinine Kinase has value 316 in hour 31. Hemoglobin has value 134 in hour 31. INR has value 0.99 in hour 31. Lactate has value 1.4 in hour 31. Magnesium has value 1.03 in hour 31. Phosphate has value 0.78 in hour 31. Platelets has value 303 in hour 31. Potassium has value 3.4 in hour 31. Sodium has value 150 in hour 31. Troponin has value 8 in hour 31. WBC has value 8.4 in hour 31. Airway Pressure has value 7.36 in hour 31, value 7.21 in hour 32, value 7.13 in hour 33, value 7.1 in hour 34, value 7.36 in hour 35, value 6.29 in hour 36, value 5.08 in hour 37, value 5.3 in hour 38. Diastolic Blood Pressure has value 59 in hour 31, value 56 in hour 32, value 77 in hour 33, value 80 in hour 34, value 75 in hour 35, value 82 in hour 36, value 75 in hour 37, value 78 in hour 38, value 69 in hour 39, value 68 in hour 40, value 59 in hour 41, value 67 in hour 42, value 65 in hour 44. FiO2 has value 30 in hour 31, value 30 in hour 32, value 30 in hour 33, value 30 in hour 34, value 30 in hour 35, value 30 in hour 36, value 30 in hour 37, value 30 in hour 38, value 21 in hour 39, value 21 in hour 40, value 21 in hour 41, value 21 in hour 42, value 21 in hour 43, value 21 in hour 44. FluidInput has value 80 in hour 34, value 160 in hour 39. FluidOutput has value 100 in hour 39. GCS has value 8 in hour 32, value 8 in hour 33, value 8 in hour 34, value 8 in hour 35, value 7 in hour 36, value 8 in hour 37, value 14 in hour 39, value 14 in hour 40, value 14 in hour 44. Heart Rate has value 97 in hour 31, value 101 in hour 32, value 94 in hour 33, value 93 in hour 34, value 99 in hour 35, value 104.5 in hour 36, value 105 in hour 37, value 102 in hour 38, value 109 in hour 39, value 107 in hour 40, value 106 in hour 41, value 102 in hour 42, value 105 in hour 43. Mean Blood Pressure has value 66 in hour 31, value 62 in hour 32, value 83 in hour 33, value 85 in hour 34, value 81 in hour 35, value 87 in hour 36, value 82 in hour 37, value 85 in hour 38, value 80 in hour 39, value 75 in hour 40, value 68 in hour 41, value 74 in hour 42, value 72 in hour 44. Minute Ventilation has value 4 in hour 31, value 3.7 in hour 32, value 3.84 in hour 33, value 4.05`

in hour 34, value 6.54 in hour 35, value 5.51 in hour 36, value 4.56 in hour 37, value 4.4 in hour 38. PC has value 0 in hour 31, value 0 in hour 32, value 0 in hour 33, value 0 in hour 34, value 0 in hour 35, value 0 in hour 36, value 0 in hour 37, value 0 in hour 38. PEEP has value 5.88 in hour 31, value 5.88 in hour 32, value 5.88 in hour 33, value 5.88 in hour 34, value 5.88 in hour 35, value 4.78 in hour 36, value 3.68 in hour 37, value 3.68 in hour 38. Respiratory Rate has value 12.5 in hour 31, value 7 in hour 32, value 7 in hour 33, value 6.5 in hour 34, value 8 in hour 35, value 8.38 in hour 36, value 7.5 in hour 37, value 4 in hour 38, value 20 in hour 39, value 21 in hour 40, value 16 in hour 41, value 15 in hour 42, value 22 in hour 43, value 26 in hour 44, value 20 in hour 47. SAS has value 5 in hour 32, value 5 in hour 33, value 5 in hour 34, value 5 in hour 35, value 5 in hour 36, value 4 in hour 37, value 4 in hour 39, value 4 in hour 40, value 4 in hour 44. Saturation has value 96 in hour 31, value 95 in hour 32, value 99 in hour 33, value 100 in hour 34, value 99 in hour 35, value 99.5 in hour 36, value 100 in hour 37, value 100 in hour 38, value 96 in hour 39, value 93 in hour 40, value 94 in hour 41, value 97 in hour 42, value 96 in hour 43, value 96 in hour 44, value 95 in hour 45, value 94 in hour 46, value 94 in hour 47. Systolic Blood Pressure has value 94 in hour 31, value 84 in hour 32, value 102 in hour 33, value 106 in hour 34, value 106 in hour 35, value 105 in hour 36, value 108 in hour 37, value 109 in hour 38, value 110 in hour 39, value 102 in hour 40, value 101 in hour 41, value 97 in hour 42, value 98 in hour 44. Temperature has value 36.4 in hour 33, value 35.9 in hour 37. Tidal Volume has value 230 in hour 31, value 215 in hour 32, value 235 in hour 33, value 267.5 in hour 34, value 115 in hour 35, value 300 in hour 36, value 255 in hour 37, value 480 in hour 38. Urine Output has value 350 in hour 31, value 100 in hour 32, value 35 in hour 33, value 60 in hour 34, value 55 in hour 35, value 60 in hour 36, value 60 in hour 37, value 300 in hour 38, value 100 in hour 39, value 100 in hour 40, value 100 in hour 44. Patient receives Analgesia at hour 32, 33, 34, 35, 36, 37, 38, 39. Patient receives Anticoagulants at hour 31, 32, 33, 34, 35, 36, 37, 38, 39, 40, 41, 42, 43, 44, 45, 46, 47. Patient receives Neuroleptics at hour 38, 39, 40, 41, 42, 43, 44, 45, 46, 47. Patient receives PS at hour 31, 32, 33, 34, 35, 36, 37, 38. Patient receives Sedation at hour 32, 33, 34, 35, 36, 37, 38, 39. Patient receives TPN at hour 32.

The following is the detailed prompts and samples of summarization

### ZERO-SHOT

> You are a clinical agent that analyze and then provide the most concise summarization on ICU time series data for forecasting.

'**Summary:**\n\nPatient exhibits fluctuating hemodynamics and respiratory status. Key observations:\n* **Hemodynamics:** Initial hypotension (MBP 66, SBP 94) improved, then showed variability (MBP 62-87, SBP 84-110). DBP increased significantly from hour 32-36 (56-82) then decreased. Heart rate is consistently elevated (97-109).\n* **Respiratory:** Airway pressure decreased (7.36 -> 5.08) then increased. PEEP decreased (5.88 -> 3.68). Respiratory rate varied significantly (7-26). FiO2 decreased from 30 to 21. Saturation generally stable (95-100), with a dip to 93 at hour 40.\n* **Renal:** Creatinine is high (103). Urine output is variable (35-350). BUN is slightly elevated (5.6).\n* **Neurologic:** GCS initially low (8), improved to 14, then decreased.\n* **Labs:** Sodium is high (150). Other labs (ALP, ALT, AST, Bilirubin, CK, Troponin, WBC) are relatively stable or within normal limits.\n* **Interventions:** Continuous anticoagulation, analgesia, sedation, and neuroleptics administered. TPN given at hour 32.\n\n**Overall:** Patient shows signs of initial instability with subsequent fluctuations in hemodynamics, respiratory parameters, and neurological status. High sodium and creatinine are notable.'

ICD

You are a clinical analysis agent. Summarize ICU time-series patient data for forecasting using this structure:

- Trend — overall direction of vitals, labs, therapies, and organ support.
- Seasonality — repeating cycles (e.g., circadian).
- Irregularities — acute deviations or events.

Map each diagnosis to its affected organ system (cardiac, respiratory, hepatic, renal, neurologic, etc.). For every system, assign a severity score from 1 (least affected) to 10 (most severe) based on data patterns and level of support required. Output only the summary in clear clinical prose, concluding with a semicolon-separated list of organ systems and scores (e.g., "Cardiovascular 7/10; Respiratory 8/10; Hepatic 3/10"). Do not explain your reasoning.

```
'Overall, the patient exhibits a complex trajectory with periods of relative stability
punctuated by acute deteriorations and improvements. Trends show initial hemodynamic
instability with low blood pressure, followed by a period of hypertension and tachycardia,
then a return towards lower pressures. Respiratory status is marked by a significant decrease
in airway pressure and PEEP, alongside variable respiratory rates and tidal volumes,
suggesting evolving ventilation needs. Neurologic status initially shows depressed GCS, which
improves significantly before a potential decline. Renal function shows initial oliguria
followed by periods of increased output, but creatinine remains elevated. Liver function tests
are relatively stable. There is no clear circadian pattern evident in the provided data.
Irregularities include a sharp drop in respiratory rate and tidal volume around hour 35, a
subsequent increase in tidal volume at hour 38, a significant improvement in GCS around hour
39, and fluctuations in fluid balance. The patient received multiple therapies including
analgesia, anticoagulants, neuroleptics, pressors (PS), and sedation, alongside
TPN.\n\nCardiovascular 7/10; Respiratory 8/10; Hepatic 3/10; Renal 6/10; Neurologic 6/10'
```

TREND

Examine the data closely and describe the trend changes step by step over time. For example: from [start] to [midpoint], what happened? Then from [midpoint] to [end], what happened? After describing each phase, conclude with an overall summary in natural language. Summarize as many feature as possible starting from the most significant ones in concise words. Only include your description and summarization.

```
"**Phase 1: Hour 31 to Hour 38**\n\nFrom hour 31 to hour 38, the patient showed signs of
initial instability followed by some improvement. Airway Pressure decreased steadily from 7.36
to 5.08, then slightly increased to 5.3. Diastolic Blood Pressure initially decreased from 59
to 56, then significantly increased to 80, fluctuating between 75 and 78 before rising again
to 82. FiO2 remained stable at 30 until hour 39. GCS remained low at 8, then dropped to 7
before improving to 14 by hour 39. Heart Rate increased from 97 to 105, then slightly
decreased to 102. Mean Blood Pressure decreased from 66 to 62, then increased significantly to
87, fluctuating between 82 and 85. Minute Ventilation increased from 4 to 6.54, then decreased
to 4.56 and 4.4. PEEP decreased from 5.88 to 3.68. Respiratory Rate decreased significantly
from 12.5 to 4, then increased sharply to 20. Saturation improved from 96 to 100, then
decreased to 96. Systolic Blood Pressure decreased from 94 to 84, then increased significantly
to 109. Temperature decreased from 36.4 to 35.9. Tidal Volume decreased from 230 to 115, then
increased dramatically to 480. Urine Output decreased significantly from 350 to 35, then
increased to 300. The patient received analgesia, anticoagulants, PS, and sedation during this
period.\n\n**Phase 2: Hour 39 to Hour 47**\n\nFrom hour 39 to hour 47, the patient's condition
showed further fluctuations and potential deterioration. Airway Pressure decreased from 5.3 to
6.29, then decreased to 5.08. Diastolic Blood Pressure decreased from 82 to 69, then 68, 59,
67, 65. FiO2 decreased from 21 to 21. GCS remained at 14. Heart Rate increased from 102."
```

CoT

You are a healthcare agent that summarizes ICU patients' time series status for future time series forecasting. Analyze this step by step.

1. Analyze the time series data to identify key trends.
2. Based on the identified trends, determine potential clinical implications.
3. Summarize the findings and suggest possible interventions.

Summarize as many feature as possible starting from the most significant ones in concise words and only respond with your summarization.

| Dataset | Embedding | Silhouette |
|---------|-----------|------------|
| HiRID | Raw time-series | 0.372 |
| HiRID | CoT embedding | 0.452 |
| HiRID | ICD embedding | 0.408 |
| HiRID | zero_shot embedding | 0.638 |
| HiRID | Trend embedding | 0.540 |
| MIMIC | Raw time-series | 0.332 |
| MIMIC | CoT embedding | 0.618 |
| MIMIC | ICD embedding | 0.322 |
| MIMIC | zero_shot embedding | 0.472 |
| MIMIC | Trend embedding | 0.681 |
| PPICU | Raw time-series | 0.486 |
| PPICU | CoT embedding | 0.573 |
| PPICU | ICD embedding | 0.385 |
| PPICU | zero_shot embedding | 0.375 |
| PPICU | Trend embedding | 0.485 |

Table 3: Silhouette scores for different embeddings across datasets (higher is better).

```
'**Summary of ICU Patient Status (Hour 31-47):**\n\n*   **Hemodynamics:** Initial hypotension
(SBP 94, MBP 66) improved significantly (SBP 110, MBP 87) but shows recent instability (DBP
fluctuations, MBP decrease in hour 40-41). Tachycardia (HR 97-109) persists. Airway pressure
decreased significantly (7.36 to 5.08) then increased slightly, suggesting potential
improvement in lung mechanics followed by possible deterioration or change in ventilation
strategy.\n*   **Respiratory Status:** Respiratory rate is variable (initially low 7-6.5, then
increased to 20-26). FiO2 decreased from 30 to 21, indicating potential improvement in
oxygenation. Saturation generally stable (96-100) but decreased recently (93-94). Tidal volume
shows significant fluctuations (230-480).\n*   **Renal Function:** Creatinine is elevated
(103), BUN is borderline high (5.6). Urine output is highly variable (350-35, then
100-300).\n*   **Metabolic/Electrolytes:** Hypernatremia (Na 150). Hypokalemia (K 3.4).
Hypocalcemia (Ca 1.99). Hypophosphatemia (Phos 0.78). Lactate is borderline elevated (1.4).\n*
**Neurological Status:** GCS initially low (8), improved to 14, then decreased again.\n*
**Liver Function:** Liver enzymes (ALP, ALT, AST) are mildly elevated. Bilirubin is borderline
high (4).\n*   **Coagulation:** INR is normal (0.99).\n*   **Other:** Platelets are normal
(303). WBC is normal (8.4). Troponin is elevated (8).\n\n**Clinical Implications:**\n\n*
Hemodynamic instability requires close monitoring and potential vasopressor adjustment.\n*
Variable respiratory status suggests ongoing lung injury or changing ventilation needs.\n*
Renal dysfunction (elevated creatinine, variable urine output) warrants monitoring and
potential intervention.\n*   Electrolyte imbalances (hypernatremia, hypokalemia, hypocalcemia,
hypophosphatemia) need correction.\n*'
```

# F  EMBEDDING QUALITY ASSESSMENT

We assess representation quality by applying the unsupervised dimensionality-reduction method UMAP to the raw time series and four *gemini-2.0-flash*–based summaries (*CoT*, *ICD*, *zero_shot*, *Trend*). Although UMAP is fit without labels, we color points by mortality status (0/1) and quantify cluster separation using the Silhouette score (higher is better), as a proxy for discriminability on the downstream *mortality classification* task. Silhouette scores are reported in Table 3. Results are averaged over target embedding dimensions of 128, 256, and 512.

# G  PRIVACY RESULT TABLE

Table referenced in RQ6 is shown in Table 4.

# H  TOKEN COUNT COMPARISON (QWEN3-EMBEDDING-8B)

We report tokenization statistics for two PPICU text corpora using the `Qwen3-Embedding-8B` tokenizer (special tokens excluded) across raw text and summarization text. Table 5 summarizes per-string token counts. Average tokens for raw text is 6,106.5 and average tokens (Summarization from ICD) is 234.0. Token reduction: 5,872.48  (2,509.3% vs. B).

| Dataset → | HiRID | | MIMIC | | PPICU | |
|---|---|---|---|---|---|---|
| Tasks → | Age | Gender | Age | Gender | Age | Gender |
| Method ↓ | MAE | AUROC | MAE | AUROC | MAE | AUROC |
| Mean | 0.805017 | 0.754615 | 0.803168 | 0.678225 | 0.76247 | 0.707425 |
| Right shift | 0.803392 | 0.742055 | 0.798042 | 0.694878 | 0.767785 | 0.726552 |
| Interpolation | 0.800455 | 0.726648 | 0.799755 | 0.670795 | 0.76095 | 0.720293 |
| TSDE | 0.798725 | 0.662075 | 0.797682 | 0.733638 | 0.76362 | 0.764377 |
| TimesFM | 0.798495 | 0.64247 | 0.795795 | 0.566465 | 0.759935 | 0.63862 |
| No-Summary | 0.801658 | 0.653393 | 0.803828 | 0.640195 | 0.812675 | 0.666798 |
| Llama 3.1 | 0.801658 | 0.64383 | 0.805292 | 0.623775 | 0.81619 | 0.643475 |
| MedGemma | 0.801207 | 0.645748 | 0.803142 | 0.61944 | 0.814385 | 0.652317 |
| Gemini 2.0-Flash | 0.801828 | 0.650965 | 0.805535 | 0.605605 | 0.813318 | 0.646813 |
| Dataset → | HiRID → PPICU | | MIMIC → PPICU | | PPICU → PPICU | |
| Tasks → | Age | Gender | Age | Gender | Age | Gender |
| Method ↓ | MAE | AUROC | MAE | AUROC | MAE | AUROC |
| Mean | 0.805353 | 0.63862 | 0.78551 | 0.48239 | 0.00 | 0.000 |
| Right shift | 0.791562 | 0.63101 | 0.77174 | 0.63016 | 0.00 | 0.000 |
| Interpolation | 0.78547 | 0.6317 | 0.778125 | 0.63008 | 0.00 | 0.000 |
| TSDE | 0.7615 | 0.452603 | 0.755202 | 0.457027 | 0.00 | 0.000 |
| TimesFM | 0.760583 | 0.63862 | 0.75289 | 0.6113 | 0.00 | 0.000 |
| No-Summary | 0.811862 | 0.643015 | 0.80723 | 0.635738 | 0.00 | 0.000 |
| Llama 3.1 | 0.81223 | 0.63101 | 0.80953 | 0.63016 | 0.00 | 0.000 |
| MedGemma | 0.812695 | 0.64306 | 0.805072 | 0.633265 | 0.00 | 0.000 |
| Gemini 2.0-Flash | 0.813743 | 0.642868 | 0.809338 | 0.63083 | 0.00 | 0.000 |

Table 4: RQ6 (Privacy): Age and gender prediction. Age is evaluated with MAE; Gender with AUROC.

Table 5: Per-string tokenization summary (Qwen3-Embedding-8B; no special tokens).

| | count | total | mean | std | max |
|---|---|---|---|---|---|
| File A (Raw Text) | 46,818 | 285,894,402 | 6,106.51 | 3,328.77 | 13,233.00 |
| File B (ICD Summarization) | 47,119 | 11,027,274 | 234.03 | 60.52 | 567.00 |

We add a more detailed report of latency, GPU time, token usage and approximate costs across all of methods in Record2Vec in Table 6. We report inference + embedding latency, GPU time per batch, token counts and costs per patient. Specifically, the language models are served on 4 NVIDIA L40S GPUs using vLLM inference; the embedding model is served on 1 L40S for the summarization-based methods, while the no-summarization baseline uses 2 L40S. Additionally, we compare the inference time using a deterministic template (no summarization). Balancing the results and costs, we find Gemini-2.0-flash results the least latency/costs while maintaining high performance, with a cost of 0.7 dollar per 1000 samples and 0.26s latency per patient.

# I  16-SAMPLE FEW-SHOT FINETUNE RESULTS

In this section, we report the detailed mean±std results under our 16-sample few-shot finetuning scenario as Table 7. This is complementary to Figure 5.

# J  1000 SAMPLES TRAINING RESULTS

In the section, we record the comparison between results of pre-training using 1000 in-distribution samples and 16-sample few-shot finetuning result of Record2Vec. This is a simulation in real world settings for smaller local hospitals, where they have less sample to train a robust model. The 1000 samples are chosen at random and shared across methods to ensure fair comparison.

| Model | Latency (per sample) | GPU time (per batch) | Token counts | Cost (per 1k samples) |
|---|---|---|---|---|
| MedGemma + Qwen | $0.784 + 0.026\,\mathrm{s}$ | $3959.41 + 119.61\,\mathrm{ms}$ | 234.03 tks/sample | – |
| Llama + Qwen | $0.25406 + 0.023\,\mathrm{s}$ | $1299.4 + 117.24\,\mathrm{ms}$ | 254.17 tks/sample | – |
| Gemini + Qwen | $0.28 + 0.036\,\mathrm{s}$ | – | 293.44 tks/sample | 0.7 \$/1k samples |
| Qwen (no summarization) | $2.9675\,\mathrm{s}$ | $23608.14\,\mathrm{ms}$ | 6106.56 tks/sample | – |

Table 6: Latency, GPU time, token usage, and approximate cost of different summarization/inference pipelines.

| Dataset → | HiRID → PPICU | | | MIMIC → PPICU | | | Wins |
|---|---|---|---|---|---|---|---|
| Task → | Forecast | LoS | Mortality | Forecast | LoS | Mortality | |
| Method ↓ | MSE↓ | MAE↓ | AUROC↑ | MSE↓ | MAE↓ | AUROC↑ | |
| Mean | $\underline{0.133 \pm 0.001}$ | $2.678 \pm 0.229$ | $0.600 \pm 0.007$ | $0.183 \pm 0.003$ | $1.171 \pm 0.284$ | $0.565 \pm 0.050$ | 0 |
| Right shift | $0.136 \pm 0.002$ | $1.803 \pm 0.041$ | $0.621 \pm 0.036$ | $0.181 \pm 0.003$ | $0.662 \pm 0.056$ | $0.606 \pm 0.036$ | 0 |
| Interpolation | $0.139 \pm 0.001$ | $1.999 \pm 0.123$ | $0.605 \pm 0.033$ | $0.183 \pm 0.003$ | $0.972 \pm 0.090$ | $0.563 \pm 0.028$ | 0 |
| TFM | $0.144 \pm 0.005$ | $0.896 \pm 0.042$ | $\underline{0.686 \pm 0.003}$ | $\underline{0.163 \pm 0.001}$ | $1.067 \pm 0.028$ | $\underline{0.683 \pm 0.002}$ | 0 |
| TSDE | $0.302 \pm 0.002$ | $\underline{0.795 \pm 0.036}$ | $0.507 \pm 0.017$ | $0.392 \pm 0.016$ | $\underline{0.643 \pm 0.170}$ | $0.586 \pm 0.025$ | 0 |
| Record2Vec | $\mathbf{0.022 \pm 0.000}$ | $\mathbf{0.500 \pm 0.005}$ | $\mathbf{0.792 \pm 0.002}$ | $\mathbf{0.0215 \pm 0.000}$ | $\mathbf{0.448 \pm 0.003}$ | $\mathbf{0.750 \pm 0.008}$ | 6 |

Table 7: Transfer performance from HiRID/MIMIC to PPICU. Values are mean±std. Best per column in **bold**; second-best underlined. Wins count the number of best results per method.

## K    Strong mortality transfer results

In this section, we report transfer learning results for strong mortality baselines: LSTM, GRU, TCN, Transformer as mentioned in Yèche et al. (2022). We observe that none surpass our simple downstream classifier, PatchTSMixer, which achieves an AUROC of 0.72. (Table 9). This suggests that there is still a need to develop models better suited to transfer-learning settings.

## L    Embedder Ablations

We performed additional ablations on the choice of embedding model and pooling / normalization strategy. Overall, we observe three main trends: (i) stronger embedding models generally yield better performance, although the gains are modest across current SOTA models; (ii) a weaker, non-SOTA embedder degrades performance but still remains above our non-summarization baseline; and (iii) our method is largely robust to changes in pooling and normalization, with the recommended configuration from the model documentation giving the most consistent performance across benchmarks.

On our base embedder Qwen3-Embedding-8B, we tested the following pooling + normalization pairs: *mean + L2* (our default), *mean + none*, *CLS + L2*, *last + L2*, and *max + L2*. In addition, we replaced Qwen3 with the current MTEB leader `nvidia/llama-embed-nemotron-8b` and the prior SOTA `gte-Qwen2-7B-instruct`. The in-distribution ablation results on PPICU are reported in Table 10, and the HiRID→PPICU transfer ablations are shown in Table 11.

## M    Detailed results for comparing prompting strategies and summarization methods

In this section, we provide the detailed performance values used to generate the rank distribution plots in Figure 3 and Figure 4. These figures provide an intuitive visualization of how different summarization and prompting strategies affect relative performance across our benchmark.

The rankings are determined by comparing the performance of the competing methods on five specific downstream tasks, using the same metrics reported in the main results (Tables 1 and 2). We utilize Mean Squared Error (MSE) for forecasting, Mean Absolute Error (MAE) for length-of-stay prediction, AUROC for mortality prediction, and Recall for both treatment planning (Drug) and measurement ordering (Lab). For the summarization analysis in Figure 3, ranks are computed across four methods: No-summary, Llama 3.1, MedGemma, and Gemini 2.0 Flash. For the prompting analysis in Figure 4, we compare four variants: Zero-shot, In-Context Learning (ICD), Chain-of-Thought (CoT), and Trend.

| Task | Small PPICU | Few-shot Finetune (Record2Vec) |
|------|-------------|-------------------------------|
| Mortality | 0.6843 | 0.72 |
| LoS | 1.03 | 0.49 |
| Forecast | 0.62 | 0.183 |
| Feature | 0.901 | 0.97 |
| Lab | 0.94 | 0.97 |

Table 8: Performance comparison between models trained on a small PPICU subset vs. few-shot finetuned models.

| | LSTM | GRU | Transformer | TCN |
|---|------|-----|-------------|-----|
| H→P Baseline | 0.640 | 0.687 | 0.579 | 0.719 |
| H→P Record2Vec | 0.69788 | 0.6838 | 0.679 | 0.684 |
| M→P Baseline | 0.5235 | 0.5476 | 0.564 | 0.719 |
| M→P Record2Vec | 0.722 | 0.685 | 0.682 | 0.695 |

Table 9: Model performance for HiRID→PPICU (H→P) and MIMIC→PPICU (M→P), comparing baseline vs. Record2Vec representations.

The rank counts are aggregated across two distinct experimental settings. The "In-Distribution" results (shown in the left subfigures) aggregate rankings over 15 total tasks, corresponding to five tasks evaluated across three source datasets. The "Transfer Learning" results (shown in the right subfigures) aggregate rankings over 30 total tasks, covering five tasks across six distinct transfer directions. For the Figure 4, we only considered three sets of transfers (total 15 numbers). Table 12, Table 13 & 14, Table 15, and Table 16 present the exact performance numbers used to construct these rank distributions, allowing for direct inspection of the underlying values.

## N    GENHPF MODIFICATION AND REPLICATION DETAILS

ICU data contain a far greater number of features observed over a much shorter time horizon, with multiple events often occurring simultaneously. When converting ICU data into a textual representation following GenHPF (Hur et al., 2024), we adopt a simple hierarchy: features are first divided into laboratory, vital, and binary (therapy / intervention) groups, and each feature is concatenated with its group prefix. For example, the feature `Hemoglobin` is encoded as `lab_Hemoglobin`. The feature groupings used for the three datasets are summarized in Table 17.

## O    DETAILED RESULTS

We provide detailed results and tables in this section for a comprehensive comparison between LLM-based summarize-then-embed pipeline with the three baseline methods.

| Ablations | Forecast | LoS | Mort | Drug | Lab |
|---|---|---|---|---|---|
| Qwen3-mean-l2 | 0.021 | 0.347 | 0.90 | 0.911 | 0.931 |
| Qwen3-mean-none | 0.027 | 0.371 | 0.88 | 0.871 | 0.919 |
| Qwen3-cls-l2 | 0.024 | 0.530 | 0.89 | 0.900 | 0.940 |
| Qwen3-last-l2 | 0.022 | 0.423 | 0.89 | 0.920 | 0.923 |
| gte-Qwen2-instruct | 0.028 | 0.480 | 0.88 | 0.886 | 0.918 |
| llama-embed-nemotron | 0.021 | 0.379 | 0.89 | 0.9097 | 0.929 |
| Baseline | 0.040 | 0.378 | 0.52 | 0.878 | 0.857 |

Table 10: Ablations on PPICU (in-distribution).

| Ablations | Forecast | LoS | Mort | Drug | Lab |
|---|---|---|---|---|---|
| Qwen3-mean-l2 | 0.183 | 0.69 | 0.72 | 0.97 | 0.97 |
| Qwen3-mean-none | 0.209 | 0.66 | 0.71 | 0.94 | 0.96 |
| Qwen3-cls-l2 | 0.230 | 0.78 | 0.50 | 0.995 | 0.98 |
| Qwen3-last-l2 | 0.190 | 0.78 | 0.73 | 0.94 | 0.96 |
| gte-Qwen2-instruct | 0.250 | 0.82 | 0.50 | 0.93 | 0.96 |
| llama-embed-nemotron | 0.190 | 0.72 | 0.713 | 0.96 | 0.998 |
| Baseline | 0.306 | 1.09 | 0.50 | 0.42 | 0.77 |

Table 11: Ablations for HiRID→PPICU transfer.

Table 12: Summarization methods performance comparison across 15 in-distribution tasks (Figure 3 left subfigure)

| | HIRID | | | | | MIMIC | | | | | PPICU | | | | |
|---|---|---|---|---|---|---|---|---|---|---|---|---|---|---|---|
| Model | Forecast | LOS | Mort | Drug | Lab | Forecast | LOS | Mort | Drug | Lab | Forecast | LOS | Mort | Drug | Lab |
| no-summary | 0.021 | 0.3538 | 0.9 | 0.899 | 0.931 | 0.027 | 0.328 | 0.82 | 0.886 | 0.947 | 0.0217 | 0.376 | 0.64 | 0.9308 | 0.923 |
| llama 3.1 | 0.0214 | 0.381 | 0.8797 | 0.899 | 0.925 | 0.0288 | 0.406 | 0.8131 | 0.886 | 0.938 | 0.029 | 0.3752 | 0.63 | 0.937 | 0.9271 |
| medgemma | 0.0237 | 0.3682 | 0.83 | 0.911 | 0.925 | 0.0299 | 0.4005 | 0.7788 | 0.8946 | 0.9432 | 0.029 | 0.3626 | 0.6346 | 0.925 | 0.936 |
| gemini | 0.028 | 0.347 | 0.8302 | 0.9064 | 0.9307 | 0.03 | 0.3328 | 0.77 | 0.903 | 0.942 | 0.017 | 0.358 | 0.6377 | 0.9308 | 0.9328 |

Table 13: Summarization methods performance comparison across 30 transfer learning tasks (Part 1: Transfer Pairs 1-3)

| | H → P | | | | | M → P | | | | | H → M | | | | |
|---|---|---|---|---|---|---|---|---|---|---|---|---|---|---|---|
| Model | Forecast | LOS | Mort | Drug | Lab | Forecast | LOS | Mort | Drug | Lab | Forecast | LOS | Mort | Drug | Lab |
| no-summary | 0.21 | 0.98 | 0.66 | 0.92 | 0.96 | 0.263 | 0.77 | 0.7 | 0.92 | 0.92 | 0.1284 | 0.693 | 0.73 | 0.881 | 0.832 |
| llama 3.1 | 0.1908 | 0.71 | 0.735 | 0.9243 | 0.9718 | 0.249 | 0.5522 | 0.7097 | 0.9415 | 0.9285 | 0.134 | 0.6464 | 0.82 | 0.878 | 0.851 |
| medgemma | 0.21 | 0.6981 | 0.74 | 0.97 | 0.98 | 0.239 | 0.5627 | 0.7145 | 0.9428 | 0.95 | 0.0922 | 0.537 | 0.81 | 0.891 | 0.83 |
| gemini | 0.183 | 0.7017 | 0.7001 | 0.92 | 0.9737 | 0.258 | 0.49 | 0.73 | 0.95 | 0.9372 | 0.089 | 0.6661 | 0.8044 | 0.852 | 0.842 |

Table 14: Summarization methods performance comparison across 30 transfer learning tasks (Part 2: Transfer Pairs 4-6)

| | P → M | | | | | M → H | | | | | P → H | | | | |
|---|---|---|---|---|---|---|---|---|---|---|---|---|---|---|---|
| Model | Forecast | LOS | Mort | Drug | Lab | Forecast | LOS | Mort | Drug | Lab | Forecast | LOS | Mort | Drug | Lab |
| no-summary | 0.15971 | 0.518 | 0.75 | 0.913 | 0.8669 | 0.135 | 0.5805 | 0.8277 | 0.816 | 0.8857 | 0.0954 | 0.494 | 0.76 | 0.892 | 0.7895 |
| llama 3.1 | 0.141 | 0.443 | 0.752 | 0.877 | 0.8669 | 0.1741 | 0.572 | 0.83 | 0.903 | 0.9085 | 0.096 | 0.4393 | 0.7657 | 0.835 | 0.734 |
| medgemma | 0.1281 | 0.4493 | 0.81 | 0.88 | 0.9054 | 0.135 | 0.5805 | 0.8277 | 0.8653 | 0.877 | 0.089 | 0.4682 | 0.7725 | 0.8637 | 0.8042 |
| gemini | 0.127 | 0.4886 | 0.7942 | 0.901 | 0.919 | 0.167 | 0.5864 | 0.8037 | 0.8239 | 0.916 | 0.083 | 0.429 | 0.81 | 0.8627 | 0.812 |

Table 15: Prompting methods performance comparison across 15 in-distribution tasks (Figure 4 left subfigure)

| | HIRID | | | | | MIMIC | | | | | PPICU | | | | |
|---|---|---|---|---|---|---|---|---|---|---|---|---|---|---|---|
| Model | Forecast | LOS | Mort | Drug | Lab | Forecast | LOS | Mort | Drug | Lab | Forecast | LOS | Mort | Drug | Lab |
| zero-shot | 0.0244 | 0.354 | 0.8501 | 0.911 | 0.931 | 0.0282 | 0.3951 | 0.77 | 0.8894 | 0.9431 | 0.017 | 0.376 | 0.6367 | 0.9346 | 0.923 |
| CoT | 0.0268 | 0.381 | 0.8311 | 0.9078 | 0.925 | 0.0294 | 0.328 | 0.8101 | 0.886 | 0.938 | 0.029 | 0.358 | 0.6387 | 0.937 | 0.936 |
| ICD | 0.021 | 0.3517 | 0.83 | 0.899 | 0.9305 | 0.027 | 0.3844 | 0.8018 | 0.903 | 0.947 | 0.0284 | 0.3645 | 0.64 | 0.925 | 0.9234 |
| Trend | 0.028 | 0.347 | 0.9 | 0.9042 | 0.9263 | 0.03 | 0.406 | 0.82 | 0.8881 | 0.9469 | 0.0249 | 0.3713 | 0.63 | 0.9324 | 0.9308 |

Table 16: Prompting methods performance comparison across 15 transfer learning tasks (Figure 4 right subfigure)

| | H → P | | | | | M → P | | | | | H → M | | | | |
|---|---|---|---|---|---|---|---|---|---|---|---|---|---|---|---|
| Model | Forecast | LOS | Mort | Drug | Lab | Forecast | LOS | Mort | Drug | Lab | Forecast | LOS | Mort | Drug | Lab |
| zero-shot | 0.183 | 0.6994 | 0.6888 | 0.9359 | 0.9758 | 0.2536 | 0.57 | 0.7045 | 0.949 | 0.95 | 0.1082 | 0.6467 | 0.73 | 0.8878 | 0.83 |
| CoT | 0.1996 | 0.69 | 0.66 | 0.92 | 0.9667 | 0.2461 | 0.49 | 0.7106 | 0.92 | 0.9375 | 0.089 | 0.693 | 0.7328 | 0.868 | 0.832 |
| ICD | 0.21 | 0.6989 | 0.6916 | 0.933 | 0.98 | 0.239 | 0.5199 | 0.73 | 0.926 | 0.9498 | 0.134 | 0.6564 | 0.82 | 0.891 | 0.851 |
| Trend | 0.1927 | 0.71 | 0.74 | 0.97 | 0.96 | 0.258 | 0.5595 | 0.7 | 0.95 | 0.92 — 0.1248 | 0.537 | 0.7629 | 0.852 | 0.847 | |

| Dataset | Lab features | Vital features | Binary features |
|---|---|---|---|
| HiRID | ALP, ALT, AST, Bicarbonate, Bilirubin, BloodUreaNitrogen, Calcium, Chloride, CreatineKinase, Creatinine, Glucose, Hemoglobin, INR, Lactate, Magnesium, PaCO2, PaO2, Phosphate, Platelets, Potassium, Sodium, Troponin, WBC, ph | AirwayPressure, AirwayPressurePeak, DiastolicBloodPressure, FiO2, FluidBalance, GCS, HeartRate, MeanBloodPressure, PEEP, RespiratoryRate, SAS, Saturation, SystolicBloodPressure, Temperature, TidalVolume, UrineOutput, ICPMonitor, Ventilation | Analgesia, Antiarrhythmics, Antibiotics, Anticoagulants, Antiepileptics, Antihypertensives, Aspirin, CaReplacement, Dialysis, Diuretics, ICPMonitor, Insulin, KReplacement, LiverToxicDrug, MgReplacement, Neuroleptics, Paralysis, Saline, Sedation, Steroids, TPN, Transfusions, Vasopressors, Ventilation |
| MIMIC | ALP, ALT, Bicarbonate, Bilirubin, BloodUreaNitrogen, Calcium, Creatinine, CreatinineKinase, Hemoglobin, INR, Lactate, Magnesium, PaCO2, PaO2, Phosphate, Platelets, Potassium, Sodium, Troponin, WBC, ph | AirwayPressure, DiastolicBloodPressure, FiO2, GCS, HeartRate, ICDSC, MeanBloodPressure, MinuteVentilation, PEEP, RespiratoryRate, SAS, SystolicBloodPressure, Temperature, TidalVolume, UrineOutput | Analgesia, Antiarrhythmics, Antibiotics, Anticoagulants, Antiepileptics, Antihypertensives, Antipsychotics, CTScan, CaReplacement, Dialysis, Diuretics, EnteralNutrition, ICPMonitor, KReplacement, MRI, MgReplacement, PPI, Paralysis, TPN, Transfusions, UltraSound, Vasopressors, Ventilation, Xray |
| PPICU | ALP, ALT, AST, Bicarbonate, Bilirubin, BloodUreaNitrogen, Calcium, Chloride, Creatinine, CreatinineKinase, GGT, Glucose, Hemoglobin, INR, Lactate, Magnesium, PaCO2, PaO2, Phosphate, Platelets, Potassium, Sodium, Troponin, WBC, ph | AirwayPressure, AirwayPressureIP, AirwayPressurePeak, DiastolicBloodPressure, FiO2, FluidInput, FluidOutput, GCS, HeartRate, ICDSC, MeanBloodPressure, MinuteVentilation, PAVSupport, PC, PEEP, PlateauPressure, RespiratoryRate, SAS, Saturation, SystolicBloodPressure, Temperature, TidalVolume, UrineOutput | Analgesia, Antiarrhythmic, Antiarrhythmics, Antibiotics, Anticoagulants, Antiepileptics, Antihypertensives, Aspirin, Dialysis, Diuretics, EKG, EVD, ICPMonitor, InhaledVasodilator, Insulin, Isoproterenol, MgReplacement, Neuroleptics, OsmoticTherapy, PPI, PS, Paralysis, Sedation, Steroids, TPN, Transfusions, Vasopressors |

Table 17: Feature groupings for converting ICU data into hierarchical textual representations. Each feature is prefixed with its group label (e.g., `lab_Hemoglobin`, `vital_HeartRate`, `bin_Vasopressors`) before tokenization.

| Method | hirid → hirid | | mimic → mimic | | ppicu → ppicu | |
|---|---|---|---|---|---|---|
| | mae | mse | mae | mse | mae | mse |
| Llama-3.1 CoT +TFM lstm | $0.799 \pm 4.12e-4$ | $1 \pm 1.893e-4$ | $0.802 \pm 2.517e-5$ | $1 \pm 1.732e-5$ | $0.808 \pm 1.323e-3$ | $1 \pm 3.87e-4$ |
| Llama-3.1 CoT +TFM mlp | $0.799 \pm 5.4e-4$ | $1 \pm 1.888e-4$ | $0.802 \pm 6.557e-5$ | $1 \pm 7.572e-5$ | $0.811 \pm 1.514e-3$ | $1.001 \pm 1.177e-3$ |
| Llama-3.1 CoT +TFM patchtsmixer | $0.808 \pm 1.848e-3$ | $1.018 \pm 3.714e-3$ | $0.806 \pm 1.544e-3$ | $1.01 \pm 4.083e-3$ | $0.818 \pm 5.124e-3$ | $1.015 \pm 7.344e-3$ |
| Llama-3.1 CoT +TFM timemixer | $0.8 \pm 1.198e-3$ | $1 \pm 2.955e-4$ | $0.802 \pm 1.229e-4$ | $1 \pm 1.528e-5$ | $0.809 \pm 2.309e-5$ | $1 \pm 1e-5$ |
| Llama-3.1 CoT lstm | $0.799 \pm 4.594e-4$ | $1 \pm 8.139e-5$ | $0.802 \pm 8.139e-5$ | $1 \pm 3.948e-5$ | $0.808 \pm 7.551e-4$ | $1 \pm 1.603e-4$ |
| Llama-3.1 CoT mlp | $0.801 \pm 1.066e-3$ | $1.001 \pm 4.061e-4$ | $0.802 \pm 8.124e-5$ | $1 \pm 3.742e-5$ | $0.811 \pm 8.279e-4$ | $1.001 \pm 3.531e-4$ |
| Llama-3.1 CoT patchtsmixer | $0.802 \pm 4.342e-3$ | $1.009 \pm 5.105e-3$ | $0.805 \pm 1.877e-3$ | $1.004 \pm 2.605e-3$ | $0.814 \pm 3.97e-3$ | $1.007 \pm 2.545e-3$ |
| Llama-3.1 CoT timemixer | $0.8 \pm 1.173e-4$ | $1 \pm 2.363e-5$ | $0.802 \pm 6.351e-5$ | $1 \pm 5.774e-6$ | $0.81 \pm 1.245e-3$ | $1.001 \pm 4.287e-4$ |
| Llama-3.1 ICD +TFM lstm | $0.799 \pm 2.33e-3$ | $1 \pm 7.342e-4$ | $0.802 \pm 1.25e-4$ | $1 \pm 5.292e-5$ | $0.808 \pm 4.464e-4$ | $1 \pm 2.193e-4$ |
| Llama-3.1 ICD +TFM mlp | $0.8 \pm 8.113e-4$ | $1 \pm 6.769e-4$ | $0.802 \pm 1.704e-4$ | $1 \pm 2.887e-5$ | $0.811 \pm 6.393e-4$ | $1.001 \pm 4.518e-4$ |
| Llama-3.1 ICD +TFM patchtsmixer | $0.803 \pm 2.735e-3$ | $1.007 \pm 2.488e-3$ | $0.807 \pm 2.349e-3$ | $1.01 \pm 3.026e-3$ | $0.818 \pm 1.976e-3$ | $1.013 \pm 1.36e-3$ |
| Llama-3.1 ICD +TFM timemixer | $0.799 \pm 5.401e-4$ | $1 \pm 9.165e-5$ | $0.802 \pm 1.528e-5$ | $1 \pm 0e0$ | $0.809 \pm 1.202e-3$ | $1 \pm 2.977e-4$ |
| Llama-3.1 ICD lstm | $0.8 \pm 1.023e-3$ | $1 \pm 2.974e-4$ | $0.802 \pm 1.299e-4$ | $1 \pm 1.291e-5$ | $0.808 \pm 3.73e-4$ | $1 \pm 2.109e-4$ |
| Llama-3.1 ICD mlp | $0.801 \pm 6.392e-4$ | $1.001 \pm 7.034e-4$ | $0.802 \pm 1.688e-4$ | $1 \pm 6.85e-5$ | $0.811 \pm 1.794e-3$ | $1.001 \pm 7.748e-4$ |
| Llama-3.1 ICD patchtsmixer | $0.8 \pm 4.883e-3$ | $1.007 \pm 2.973e-3$ | $0.803 \pm 9.965e-4$ | $1.003 \pm 1.069e-3$ | $0.811 \pm 3.361e-3$ | $1.005 \pm 1.447e-3$ |
| Llama-3.1 ICD timemixer | $0.799 \pm 1.554e-3$ | $1 \pm 1.954e-4$ | $0.802 \pm 1.087e-4$ | $1 \pm 1.155e-5$ | $0.809 \pm 2.757e-4$ | $1 \pm 7.632e-5$ |
| Llama-3.1 Trend +TFM lstm | $0.799 \pm 1.36e-3$ | $1 \pm 1.986e-4$ | $0.802 \pm 1.007e-4$ | $1 \pm 9.815e-5$ | $0.808 \pm 1.002e-3$ | $1 \pm 1.05e-4$ |
| Llama-3.1 Trend +TFM mlp | $0.8 \pm 7.948e-4$ | $1.001 \pm 5.086e-4$ | $0.802 \pm 2.108e-4$ | $1 \pm 6.506e-5$ | $0.81 \pm 9.158e-4$ | $1.001 \pm 7e-5$ |
| Llama-3.1 Trend +TFM patchtsmixer | $0.805 \pm 1.46e-3$ | $1.013 \pm 1.356e-3$ | $0.806 \pm 3.252e-4$ | $1.008 \pm 2.559e-3$ | $0.816 \pm 2.077e-3$ | $1.013 \pm 1.48e-3$ |
| Llama-3.1 Trend +TFM timemixer | $0.799 \pm 1.756e-3$ | $1 \pm 3.102e-4$ | $0.802 \pm 9.165e-5$ | $1 \pm 5.774e-6$ | $0.809 \pm 3.219e-4$ | $1 \pm 1.026e-4$ |
| Llama-3.1 Trend lstm | $0.799 \pm 7.871e-4$ | $1 \pm 1.794e-4$ | $0.802 \pm 2.5e-5$ | $1 \pm 4.573e-5$ | $0.809 \pm 1.105e-3$ | $1 \pm 3.561e-4$ |
| Llama-3.1 Trend mlp | $0.801 \pm 1.622e-3$ | $1.001 \pm 8.542e-4$ | $0.802 \pm 1.209e-4$ | $1 \pm 1.414e-5$ | $0.81 \pm 1.283e-3$ | $1.001 \pm 2.9e-4$ |
| Llama-3.1 Trend patchtsmixer | $0.804 \pm 4.987e-3$ | $1.007 \pm 2.488e-3$ | $0.804 \pm 1.247e-3$ | $1.004 \pm 1.947e-3$ | $0.815 \pm 2.49e-3$ | $1.007 \pm 1.752e-3$ |
| Llama-3.1 Trend timemixer | $0.8 \pm 1.757e-3$ | $1.001 \pm 5.354e-4$ | $0.802 \pm 5.058e-5$ | $1 \pm 5e-6$ | $0.809 \pm 1.221e-3$ | $1.001 \pm 3.344e-4$ |
| Llama-3.1 zero_shot +TFM lstm | $0.799 \pm 7.994e-4$ | $1 \pm 3.576e-4$ | $0.802 \pm 1.904e-4$ | $1 \pm 6.11e-5$ | $0.808 \pm 7.767e-4$ | $1 \pm 4.842e-4$ |
| Llama-3.1 zero_shot +TFM mlp | $0.799 \pm 1.048e-3$ | $1 \pm 8.607e-4$ | $0.802 \pm 7.767e-5$ | $1 \pm 1.914e-4$ | $0.809 \pm 6.928e-5$ | $1.001 \pm 8.336e-4$ |
| Llama-3.1 zero_shot +TFM patchtsmixer | $0.806 \pm 4.216e-3$ | $1.018 \pm 6.421e-3$ | $0.807 \pm 2.386e-3$ | $1.011 \pm 3.676e-3$ | $0.82 \pm 5.632e-3$ | $1.02 \pm 1.084e-2$ |
| Llama-3.1 zero_shot +TFM timemixer | $0.8 \pm 2.454e-4$ | $1 \pm 5.568e-5$ | $0.802 \pm 3.464e-5$ | $1 \pm 5.774e-6$ | $0.809 \pm 9.822e-4$ | $1 \pm 2.2e-4$ |
| Llama-3.1 zero_shot lstm | $0.799 \pm 6.685e-4$ | $1 \pm 2.419e-4$ | $0.802 \pm 1.103e-4$ | $1 \pm 3.594e-5$ | $0.809 \pm 9.524e-4$ | $1 \pm 2.787e-4$ |
| Llama-3.1 zero_shot mlp | $0.801 \pm 5.996e-4$ | $1.001 \pm 3.377e-4$ | $0.802 \pm 1.721e-4$ | $1 \pm 6.272e-5$ | $0.812 \pm 2.202e-4$ | $1.002 \pm 3.597e-4$ |
| Llama-3.1 zero_shot patchtsmixer | $0.802 \pm 3.087e-3$ | $1.005 \pm 1.034e-3$ | $0.805 \pm 3.432e-3$ | $1.005 \pm 3.432e-3$ | $0.816 \pm 4.356e-3$ | $1.009 \pm 6.16e-3$ |
| Llama-3.1 zero_shot timemixer | $0.8 \pm 1.042e-3$ | $1 \pm 2.07e-4$ | $0.802 \pm 2.217e-5$ | $1 \pm 5e-6$ | $0.809 \pm 8.542e-4$ | $1 \pm 2.232e-4$ |
| TFM lstm | $0.794 \pm 1.222e-3$ | $0.995 \pm 2.307e-4$ | $0.793 \pm 1.426e-4$ | $0.978 \pm 1.36e-4$ | $0.752 \pm 2.269e-4$ | $0.934 \pm 7.789e-5$ |
| TFM mlp | $0.795 \pm 1.084e-3$ | $0.995 \pm 2.241e-4$ | $0.794 \pm 1.422e-4$ | $0.978 \pm 2.623e-4$ | $0.753 \pm 8.388e-4$ | $0.935 \pm 1.005e-3$ |
| TFM patchtsmixer | $0.798 \pm 2.747e-3$ | $1.007 \pm 1.228e-3$ | $0.796 \pm 9.829e-4$ | $0.985 \pm 1.854e-3$ | $0.76 \pm 2.015e-3$ | $0.947 \pm 2.213e-3$ |
| TFM timemixer | $0.795 \pm 1.268e-3$ | $0.994 \pm 1.484e-4$ | $0.794 \pm 1.97e-4$ | $0.978 \pm 7.767e-5$ | $0.753 \pm 8.918e-4$ | $0.934 \pm 1.767e-4$ |
| TSDE lstm | $0.794 \pm 1.283e-3$ | $0.994 \pm 2.651e-4$ | $0.793 \pm 2.316e-4$ | $0.977 \pm 3.348e-4$ | $0.752 \pm 1.179e-4$ | $0.934 \pm 3.616e-4$ |
| TSDE mlp | $0.795 \pm 1.53e-3$ | $0.995 \pm 8.894e-4$ | $0.793 \pm 1.951e-4$ | $0.977 \pm 2.168e-4$ | $0.753 \pm 5.773e-4$ | $0.934 \pm 1.016e-3$ |
| TSDE patchtsmixer | $0.799 \pm 4.027e-3$ | $1.012 \pm 3.068e-3$ | $0.798 \pm 1.979e-3$ | $0.987 \pm 3.758e-3$ | $0.764 \pm 5.447e-3$ | $0.951 \pm 5.592e-3$ |
| TSDE timemixer | $0.795 \pm 1.15e-3$ | $0.994 \pm 1.347e-4$ | $0.793 \pm 1.949e-4$ | $0.978 \pm 6.702e-5$ | $0.753 \pm 9.068e-4$ | $0.934 \pm 1.663e-4$ |
| gemini-2.0-flash CoT +TFM lstm | $0.798 \pm 7.966e-4$ | $1 \pm 1.955e-4$ | $0.802 \pm 1.419e-4$ | $1 \pm 1.25e-4$ | $0.807 \pm 2.479e-4$ | $0.999 \pm 3.291e-4$ |
| gemini-2.0-flash CoT +TFM mlp | $0.8 \pm 8.967e-4$ | $1 \pm 2.458e-4$ | $0.802 \pm 8.021e-5$ | $1 \pm 1.656e-4$ | $0.81 \pm 1.556e-3$ | $1.001 \pm 1.334e-3$ |
| gemini-2.0-flash CoT +TFM patchtsmixer | $0.802 \pm 3.295e-3$ | $1.015 \pm 5.567e-3$ | $0.806 \pm 2.782e-3$ | $1.011 \pm 4.901e-3$ | $0.818 \pm 8.316e-3$ | $1.017 \pm 1.24e-2$ |
| gemini-2.0-flash CoT +TFM timemixer | $0.8 \pm 1.567e-3$ | $1 \pm 3.384e-4$ | $0.802 \pm 1.253e-4$ | $1 \pm 1.155e-5$ | $0.809 \pm 9.109e-4$ | $1 \pm 2.371e-4$ |
| gemini-2.0-flash CoT lstm | $0.8 \pm 7.464e-4$ | $1 \pm 2.128e-4$ | $0.802 \pm 1.795e-4$ | $1 \pm 7.719e-5$ | $0.809 \pm 1.255e-3$ | $1 \pm 2.233e-4$ |
| gemini-2.0-flash CoT mlp | $0.801 \pm 8.308e-4$ | $1.001 \pm 4.742e-4$ | $0.802 \pm 6.928e-5$ | $1 \pm 9.309e-5$ | $0.812 \pm 1.04e-3$ | $1.001 \pm 4.701e-4$ |
| gemini-2.0-flash CoT patchtsmixer | $0.803 \pm 5.182e-3$ | $1.007 \pm 3.614e-3$ | $0.804 \pm 1.278e-3$ | $1.003 \pm 8.689e-4$ | $0.814 \pm 5.678e-3$ | $1.006 \pm 3.776e-3$ |
| gemini-2.0-flash CoT timemixer | $0.799 \pm 4.21e-4$ | $1 \pm 6.683e-5$ | $0.802 \pm 1.17e-4$ | $1 \pm 9.574e-6$ | $0.809 \pm 1.109e-4$ | $1 \pm 2.646e-5$ |
| gemini-2.0-flash ICD +TFM lstm | $0.798 \pm 6.035e-4$ | $1 \pm 3.291e-4$ | $0.802 \pm 1.286e-4$ | $0.999 \pm 1.311e-4$ | $0.808 \pm 7.289e-4$ | $1 \pm 3.786e-5$ |
| gemini-2.0-flash ICD +TFM mlp | $0.801 \pm 1.713e-3$ | $1.001 \pm 9.697e-4$ | $0.802 \pm 1.443e-4$ | $1 \pm 1.752e-5$ | $0.81 \pm 9.042e-4$ | $1 \pm 4.102e-4$ |
| gemini-2.0-flash ICD +TFM patchtsmixer | $0.806 \pm 6.972e-3$ | $1.019 \pm 9.015e-3$ | $0.805 \pm 1.407e-3$ | $1.009 \pm 3.682e-3$ | $0.819 \pm 6.351e-3$ | $1.016 \pm 5.938e-3$ |
| gemini-2.0-flash ICD +TFM timemixer | $0.799 \pm 5.112e-4$ | $1 \pm 9.713e-5$ | $0.802 \pm 7.234e-5$ | $1 \pm 1.155e-5$ | $0.81 \pm 2.857e-4$ | $1.001 \pm 8.718e-5$ |
| gemini-2.0-flash ICD lstm | $0.799 \pm 1.504e-3$ | $1 \pm 3.638e-4$ | $0.802 \pm 5.909e-5$ | $1 \pm 2.38e-5$ | $0.809 \pm 1.431e-3$ | $1 \pm 3.902e-4$ |
| gemini-2.0-flash ICD mlp | $0.802 \pm 7.823e-4$ | $1.001 \pm 3.651e-4$ | $0.802 \pm 1.741e-4$ | $1 \pm 5.377e-5$ | $0.811 \pm 9.929e-4$ | $1.001 \pm 4.109e-4$ |
| gemini-2.0-flash ICD patchtsmixer | $0.8 \pm 2.129e-3$ | $1.005 \pm 1.943e-3$ | $0.805 \pm 2.47e-3$ | $1.005 \pm 7.609e-4$ | $0.815 \pm 4.102e-3$ | $1.007 \pm 1.912e-3$ |
| gemini-2.0-flash ICD timemixer | $0.8 \pm 1.003e-3$ | $1 \pm 2.123e-4$ | $0.802 \pm 2.708e-5$ | $1 \pm 5e-6$ | $0.809 \pm 6.5e-4$ | $1 \pm 1.513e-4$ |
| gemini-2.0-flash Trend +TFM lstm | $0.798 \pm 6.421e-4$ | $1 \pm 9.019e-5$ | $0.802 \pm 1.212e-4$ | $1 \pm 6.083e-5$ | $0.807 \pm 4.572e-4$ | $1 \pm 1.253e-4$ |
| gemini-2.0-flash Trend +TFM mlp | $0.8 \pm 1.213e-3$ | $1 \pm 9.142e-4$ | $0.802 \pm 5.292e-5$ | $1 \pm 1.389e-4$ | $0.81 \pm 9.521e-4$ | $1 \pm 1.044e-3$ |
| gemini-2.0-flash Trend +TFM patchtsmixer | $0.809 \pm 2.167e-3$ | $1.021 \pm 4.501e-3$ | $0.806 \pm 1.91e-3$ | $1.011 \pm 5.336e-3$ | $0.816 \pm 2.824e-3$ | $1.017 \pm 4.701e-3$ |
| gemini-2.0-flash Trend +TFM timemixer | $0.801 \pm 7.143e-4$ | $1.001 \pm 2.248e-4$ | $0.802 \pm 1.039e-4$ | $1 \pm 5.774e-6$ | $0.809 \pm 4.07e-4$ | $1 \pm 8.888e-5$ |
| gemini-2.0-flash Trend lstm | $0.8 \pm 2.157e-3$ | $1.001 \pm 5.701e-4$ | $0.802 \pm 7.089e-5$ | $1 \pm 5.477e-5$ | $0.809 \pm 1.029e-3$ | $1 \pm 3.321e-4$ |
| gemini-2.0-flash Trend mlp | $0.8 \pm 7.542e-4$ | $1.001 \pm 5.427e-4$ | $0.802 \pm 1.263e-4$ | $1 \pm 3.317e-5$ | $0.811 \pm 9.28e-4$ | $1.001 \pm 4.237e-4$ |
| gemini-2.0-flash Trend patchtsmixer | $0.804 \pm 5.877e-3$ | $1.009 \pm 8.576e-3$ | $0.804 \pm 2.883e-3$ | $1.006 \pm 2.217e-3$ | $0.815 \pm 3.544e-3$ | $1.007 \pm 2.735e-3$ |
| gemini-2.0-flash Trend timemixer | $0.8 \pm 1.186e-3$ | $1 \pm 2.575e-4$ | $0.802 \pm 9.274e-5$ | $1 \pm 5e-6$ | $0.81 \pm 8.4e-4$ | $1.001 \pm 2.402e-4$ |
| gemini-2.0-flash zero_shot +TFM lstm | $0.798 \pm 4.85e-4$ | $1 \pm 3.037e-4$ | $0.802 \pm 6.245e-5$ | $1 \pm 5.508e-5$ | $0.808 \pm 6.14e-4$ | $0.999 \pm 2.887e-5$ |
| gemini-2.0-flash zero_shot +TFM mlp | $0.8 \pm 1.932e-3$ | $1.001 \pm 1.177e-3$ | $0.803 \pm 1.717e-4$ | $1 \pm 1.801e-4$ | $0.81 \pm 1.093e-3$ | $1.001 \pm 3.439e-4$ |
| gemini-2.0-flash zero_shot +TFM patchtsmixer | $0.803 \pm 2.508e-3$ | $1.014 \pm 4.794e-3$ | $0.806 \pm 3.949e-3$ | $1.009 \pm 5.323e-3$ | $0.817 \pm 2.7e-3$ | $1.013 \pm 1.66e-3$ |
| gemini-2.0-flash zero_shot +TFM timemixer | $0.8 \pm 1.125e-3$ | $1 \pm 2.359e-4$ | $0.802 \pm 3.205e-4$ | $1 \pm 5.774e-6$ | $0.81 \pm 8.261e-4$ | $1.001 \pm 2.601e-4$ |
| gemini-2.0-flash zero_shot lstm | $0.799 \pm 1.286e-3$ | $1 \pm 2.53e-4$ | $0.802 \pm 8.583e-5$ | $1 \pm 3.594e-5$ | $0.809 \pm 8.283e-4$ | $1 \pm 1.135e-4$ |
| gemini-2.0-flash zero_shot mlp | $0.801 \pm 7.43e-4$ | $1.001 \pm 3.448e-4$ | $0.802 \pm 5.944e-5$ | $1 \pm 5.354e-5$ | $0.811 \pm 1.49e-3$ | $1.001 \pm 8.31e-4$ |
| gemini-2.0-flash zero_shot patchtsmixer | $0.802 \pm 2.607e-3$ | $1.006 \pm 3.524e-3$ | $0.806 \pm 4.285e-3$ | $1.005 \pm 3.005e-3$ | $0.813 \pm 1.74e-3$ | $1.006 \pm 1.151e-3$ |
| gemini-2.0-flash zero_shot timemixer | $0.8 \pm 7.074e-4$ | $1 \pm 1.916e-4$ | $0.802 \pm 1.711e-4$ | $1 \pm 8.165e-6$ | $0.809 \pm 1.455e-3$ | $1 \pm 3.885e-4$ |
| interp lstm | $0.795 \pm 7.933e-4$ | $0.994 \pm 4.948e-4$ | $0.793 \pm 2.49e-4$ | $0.977 \pm 2.41e-4$ | $0.754 \pm 9.792e-4$ | $0.934 \pm 4.055e-4$ |
| interp mlp | $0.796 \pm 7.269e-4$ | $0.996 \pm 6.276e-4$ | $0.793 \pm 3.807e-4$ | $0.978 \pm 7.938e-4$ | $0.754 \pm 9.573e-4$ | $0.935 \pm 3.97e-4$ |
| interp patchtsmixer | $0.8 \pm 2.155e-3$ | $1.013 \pm 3.985e-3$ | $0.8 \pm 3.223e-3$ | $0.989 \pm 3.73e-3$ | $0.761 \pm 7.619e-3$ | $0.947 \pm 8.651e-3$ |
| interp timemixer | $0.795 \pm 1.088e-3$ | $0.995 \pm 1.42e-4$ | $0.793 \pm 8.104e-5$ | $0.978 \pm 3.109e-5$ | $0.753 \pm 8.277e-4$ | $0.935 \pm 1.857e-4$ |
| mean lstm | $0.795 \pm 6.998e-4$ | $0.994 \pm 7.05e-4$ | $0.793 \pm 2.029e-4$ | $0.977 \pm 4.399e-4$ | $0.754 \pm 7.075e-4$ | $0.935 \pm 2.757e-4$ |
| mean mlp | $0.796 \pm 5.046e-4$ | $0.994 \pm 7.89e-5$ | $0.793 \pm 3.061e-4$ | $0.978 \pm 3.035e-4$ | $0.754 \pm 1.199e-3$ | $0.934 \pm 1.003e-3$ |
| mean patchtsmixer | $0.805 \pm 2.516e-3$ | $1.014 \pm 5.21e-3$ | $0.803 \pm 4.826e-3$ | $0.997 \pm 5.996e-3$ | $0.762 \pm 8.643e-3$ | $0.956 \pm 8.724e-3$ |
| mean timemixer | $0.795 \pm 1.078e-3$ | $0.995 \pm 1.274e-4$ | $0.793 \pm 1.079e-4$ | $0.978 \pm 4.193e-5$ | $0.753 \pm 8.874e-4$ | $0.935 \pm 1.971e-4$ |
| No_sum_CoT CoT +TFM lstm | $0.8 \pm 4.027e-4$ | $1 \pm 7.5e-4$ | $0.802 \pm 9.292e-5$ | $0.999 \pm 2.042e-4$ | $0.807 \pm 2.136e-4$ | $0.998 \pm 8.445e-4$ |
| No_sum_CoT CoT +TFM mlp | $0.8 \pm 1.12e-3$ | $1.001 \pm 7.881e-4$ | $0.802 \pm 8.888e-5$ | $1 \pm 1.68e-4$ | $0.81 \pm 8.729e-4$ | $1 \pm 1.419e-4$ |
| No_sum_CoT CoT +TFM patchtsmixer | $0.801 \pm 1.353e-3$ | $1.012 \pm 3.439e-3$ | $0.809 \pm 3.78e-3$ | $1.012 \pm 5.47e-3$ | $0.816 \pm 4.633e-3$ | $1.013 \pm 2.616e-3$ |
| No_sum_CoT CoT +TFM timemixer | $0.8 \pm 1.564e-4$ | $1 \pm 4.1e-4$ | $0.802 \pm 1.06e-4$ | $1 \pm 1e-5$ | $0.81 \pm 5.75e-4$ | $1.001 \pm 1.735e-4$ |
| No_sum_CoT lstm | $0.799 \pm 7.968e-4$ | $1 \pm 1.644e-4$ | $0.802 \pm 1.173e-4$ | $1 \pm 4.272e-5$ | $0.808 \pm 4.664e-4$ | $1 \pm 9.815e-5$ |
| No_sum_CoT mlp | $0.802 \pm 1.165e-3$ | $1.001 \pm 5.263e-4$ | $0.802 \pm 1.573e-4$ | $1 \pm 7.659e-5$ | $0.811 \pm 1.329e-3$ | $1.001 \pm 7.081e-4$ |
| No_sum_CoT patchtsmixer | $0.8 \pm 7.784e-3$ | $1.008 \pm 3.88e-3$ | $0.805 \pm 2.973e-3$ | $1.004 \pm 2.265e-3$ | $0.813 \pm 5.557e-3$ | $1.007 \pm 3.969e-3$ |
| No_sum_CoT timemixer | $0.799 \pm 8.839e-4$ | $1 \pm 1.209e-4$ | $0.802 \pm 1.038e-4$ | $1 \pm 5.774e-6$ | $0.81 \pm 1.31e-3$ | $1.001 \pm 3.4e-4$ |
| No_sum_ICD ICD +TFM lstm | $0.797 \pm 4.851e-4$ | $0.999 \pm 2.136e-4$ | $0.802 \pm 1.082e-4$ | $0.999 \pm 1.85e-4$ | $0.808 \pm 8.193e-4$ | $0.999 \pm 9.914e-4$ |
| No_sum_ICD ICD +TFM mlp | $0.8 \pm 5.373e-4$ | $1.001 \pm 5.696e-4$ | $0.802 \pm 1.572e-4$ | $1 \pm 2.651e-4$ | $0.81 \pm 2.277e-3$ | $1 \pm 3.089e-4$ |
| No_sum_ICD ICD +TFM patchtsmixer | $0.808 \pm 1.307e-3$ | $1.021 \pm 3.624e-3$ | $0.805 \pm 7.229e-4$ | $1.007 \pm 2.108e-3$ | $0.821 \pm 4.215e-3$ | $1.018 \pm 5.508e-3$ |
| No_sum_ICD ICD +TFM timemixer | $0.8 \pm 1.246e-3$ | $1 \pm 2.969e-4$ | $0.802 \pm 5.033e-5$ | $1 \pm 0e0$ | $0.809 \pm 3.927e-4$ | $1 \pm 8.387e-5$ |
| No_sum_ICD lstm | $0.8 \pm 1.627e-3$ | $1.001 \pm 4.364e-4$ | $0.802 \pm 1.964e-4$ | $1 \pm 1.056e-4$ | $0.809 \pm 5.778e-4$ | $1 \pm 2.636e-4$ |
| No_sum_ICD mlp | $0.801 \pm 7.965e-4$ | $1.001 \pm 1.358e-4$ | $0.802 \pm 8.578e-5$ | $1 \pm 5.323e-5$ | $0.811 \pm 1.656e-3$ | $1.001 \pm 5.301e-4$ |
| No_sum_ICD patchtsmixer | $0.798 \pm 3.902e-3$ | $1.005 \pm 1.06e-3$ | $0.803 \pm 1.226e-3$ | $1.004 \pm 1.687e-3$ | $0.814 \pm 4.127e-3$ | $1.007 \pm 3.227e-3$ |
| No_sum_ICD timemixer | $0.799 \pm 3.359e-4$ | $1 \pm 6.481e-5$ | $0.802 \pm 4.646e-5$ | $1 \pm 5.774e-6$ | $0.809 \pm 5.38e-4$ | $1 \pm 1.349e-4$ |
| No_sum_Trend Trend +TFM lstm | $0.799 \pm 6.553e-4$ | $0.999 \pm 4.143e-4$ | $0.802 \pm 8.083e-5$ | $0.999 \pm 1.908e-4$ | $0.807 \pm 5.631e-4$ | $0.999 \pm 5.116e-4$ |
| No_sum_Trend Trend +TFM mlp | $0.8 \pm 9.115e-4$ | $1 \pm 6.274e-4$ | $0.802 \pm 2.307e-4$ | $1 \pm 1.861e-4$ | $0.81 \pm 2.249e-3$ | $1.001 \pm 1.942e-3$ |
| No_sum_Trend Trend +TFM patchtsmixer | $0.806 \pm 8.208e-4$ | $1.02 \pm 7.728e-3$ | $0.806 \pm 7.015e-4$ | $1.011 \pm 1.952e-3$ | $0.814 \pm 1.981e-3$ | $1.011 \pm 1.88e-3$ |
| No_sum_Trend Trend +TFM timemixer | $0.8 \pm 1.685e-3$ | $1 \pm 4.328e-4$ | $0.802 \pm 1e-4$ | $1 \pm 5.774e-6$ | $0.809 \pm 6.892e-4$ | $1 \pm 1.815e-4$ |

Table 18: In-distribution results - Age. (part 1/2)

| Method | hirid → hirid | | mimic → mimic | | ppicu → ppicu | |
|---|---|---|---|---|---|---|
| | mae | mse | mae | mse | mae | mse |
| No_sum_Trend lstm | $0.799 \pm 1.177e-3$ | $1 \pm 2.004e-4$ | $0.802 \pm 5.188e-5$ | $1 \pm 1.5e-5$ | $0.808 \pm 6.139e-4$ | $1 \pm 2.061e-4$ |
| No_sum_Trend mlp | $0.801 \pm 1.357e-3$ | $1.001 \pm 6.168e-4$ | $0.802 \pm 2.016e-4$ | $1 \pm 1.005e-4$ | $0.81 \pm 1.434e-3$ | $1 \pm 5.804e-4$ |
| No_sum_Trend patchtsmixer | $0.803 \pm 2.482e-3$ | $1.006 \pm 2.681e-3$ | $0.805 \pm 1.96e-3$ | $1.004 \pm 7.381e-4$ | $0.812 \pm 1.637e-3$ | $1.005 \pm 1.892e-3$ |
| No_sum_Trend timemixer | $0.8 \pm 6.989e-4$ | $1 \pm 1.409e-4$ | $0.802 \pm 4.933e-5$ | $1 \pm 5.774e-6$ | $0.809 \pm 1.931e-4$ | $1 \pm 5.888e-5$ |
| medgemma CoT +TFM lstm | $0.798 \pm 3.301e-4$ | $1 \pm 5.014e-4$ | $0.802 \pm 1.617e-4$ | $0.999 \pm 1.401e-4$ | $0.808 \pm 1.51e-4$ | $1 \pm 2.768e-4$ |
| medgemma CoT +TFM mlp | $0.8 \pm 8.638e-4$ | $1.001 \pm 4.661e-4$ | $0.802 \pm 1.155e-5$ | $1 \pm 1.582e-4$ | $0.81 \pm 1.223e-3$ | $1 \pm 1.119e-3$ |
| medgemma CoT +TFM patchtsmixer | $0.807 \pm 3.471e-3$ | $1.024 \pm 1.068e-2$ | $0.805 \pm 8.303e-4$ | $1.009 \pm 2.689e-3$ | $0.815 \pm 1.743e-3$ | $1.016 \pm 4.846e-3$ |
| medgemma CoT +TFM timemixer | $0.8 \pm 1.343e-3$ | $1.001 \pm 3.816e-4$ | $0.802 \pm 8.021e-5$ | $1 \pm 5.774e-6$ | $0.809 \pm 1.636e-3$ | $1.001 \pm 5.227e-4$ |
| medgemma CoT lstm | $0.799 \pm 1.998e-3$ | $1 \pm 5.651e-4$ | $0.802 \pm 5.62e-5$ | $1 \pm 5.774e-5$ | $0.809 \pm 1.522e-3$ | $1 \pm 3.647e-4$ |
| medgemma CoT mlp | $0.802 \pm 1.233e-3$ | $1.001 \pm 7.105e-4$ | $0.802 \pm 1.242e-4$ | $1 \pm 2.16e-5$ | $0.811 \pm 1.828e-3$ | $1.001 \pm 1.194e-3$ |
| medgemma CoT patchtsmixer | $0.801 \pm 2.015e-3$ | $1.005 \pm 1.861e-3$ | $0.805 \pm 1.934e-3$ | $1.005 \pm 2.257e-3$ | $0.815 \pm 3.297e-3$ | $1.006 \pm 2.075e-3$ |
| medgemma CoT timemixer | $0.8 \pm 5.616e-4$ | $1 \pm 1.266e-4$ | $0.802 \pm 1.491e-4$ | $1 \pm 8.165e-6$ | $0.809 \pm 1.859e-3$ | $1.001 \pm 6.096e-4$ |
| medgemma ICD +TFM lstm | $0.799 \pm 7.548e-4$ | $1 \pm 2.272e-4$ | $0.802 \pm 1.249e-4$ | $1 \pm 2.627e-4$ | $0.808 \pm 7.91e-4$ | $1 \pm 2.363e-4$ |
| medgemma ICD +TFM mlp | $0.8 \pm 1.484e-3$ | $1.001 \pm 1.013e-3$ | $0.802 \pm 2.113e-4$ | $1 \pm 2.715e-4$ | $0.81 \pm 1.608e-3$ | $1.001 \pm 9.855e-4$ |
| medgemma ICD +TFM patchtsmixer | $0.807 \pm 3.378e-3$ | $1.019 \pm 4.825e-3$ | $0.807 \pm 1.582e-3$ | $1.01 \pm 5.048e-4$ | $0.816 \pm 5.003e-3$ | $1.019 \pm 3.843e-3$ |
| medgemma ICD +TFM timemixer | $0.799 \pm 4.107e-4$ | $1 \pm 8.544e-5$ | $0.802 \pm 5.292e-5$ | $1 \pm 5.774e-6$ | $0.809 \pm 1.185e-3$ | $1 \pm 2.358e-4$ |
| medgemma ICD lstm | $0.799 \pm 1.006e-3$ | $1 \pm 2.429e-4$ | $0.802 \pm 1.48e-4$ | $1 \pm 1.291e-5$ | $0.809 \pm 8.411e-4$ | $1 \pm 1.823e-4$ |
| medgemma ICD mlp | $0.802 \pm 4.164e-4$ | $1.001 \pm 3.267e-4$ | $0.802 \pm 2.65e-4$ | $1 \pm 1.431e-4$ | $0.81 \pm 1.05e-3$ | $1.001 \pm 4.768e-4$ |
| medgemma ICD patchtsmixer | $0.802 \pm 1.518e-3$ | $1.007 \pm 2.384e-3$ | $0.804 \pm 1.19e-3$ | $1.003 \pm 9.347e-4$ | $0.818 \pm 4.967e-3$ | $1.01 \pm 4.364e-3$ |
| medgemma ICD timemixer | $0.799 \pm 6.545e-4$ | $1 \pm 1.09e-4$ | $0.802 \pm 5.477e-5$ | $1 \pm 5e-6$ | $0.809 \pm 8.681e-4$ | $1 \pm 1.848e-4$ |
| medgemma Trend +TFM lstm | $0.798 \pm 5.125e-4$ | $1 \pm 1.361e-4$ | $0.802 \pm 1.159e-4$ | $1 \pm 5.859e-5$ | $0.807 \pm 2.721e-4$ | $1 \pm 2.403e-4$ |
| medgemma Trend +TFM mlp | $0.8 \pm 1.693e-3$ | $1.001 \pm 1.002e-3$ | $0.802 \pm 1.531e-4$ | $1 \pm 4.041e-5$ | $0.81 \pm 2.434e-3$ | $1.001 \pm 1.474e-3$ |
| medgemma Trend +TFM patchtsmixer | $0.806 \pm 9.166e-4$ | $1.018 \pm 1.32e-3$ | $0.806 \pm 2.89e-3$ | $1.011 \pm 4.614e-3$ | $0.815 \pm 2.136e-3$ | $1.015 \pm 2.962e-3$ |
| medgemma Trend +TFM timemixer | $0.799 \pm 1.216e-3$ | $1 \pm 2.237e-4$ | $0.802 \pm 9.539e-5$ | $1 \pm 1e-5$ | $0.809 \pm 1.571e-3$ | $1.001 \pm 5.096e-4$ |
| medgemma Trend lstm | $0.799 \pm 5.958e-4$ | $1 \pm 2.047e-4$ | $0.802 \pm 7.32e-5$ | $1 \pm 3.096e-5$ | $0.808 \pm 1.07e-3$ | $1 \pm 3.157e-4$ |
| medgemma Trend mlp | $0.801 \pm 1.054e-3$ | $1.001 \pm 7.023e-4$ | $0.802 \pm 1.398e-4$ | $1 \pm 1.742e-4$ | $0.811 \pm 6.312e-4$ | $1.001 \pm 4.551e-4$ |
| medgemma Trend patchtsmixer | $0.799 \pm 2.469e-3$ | $1.004 \pm 1.798e-3$ | $0.804 \pm 7.189e-4$ | $1.003 \pm 9.537e-4$ | $0.813 \pm 2.898e-3$ | $1.006 \pm 2.387e-3$ |
| medgemma Trend timemixer | $0.799 \pm 8.691e-4$ | $1 \pm 1.797e-4$ | $0.802 \pm 1.252e-4$ | $1 \pm 1.291e-5$ | $0.808 \pm 6.085e-4$ | $1 \pm 8.756e-5$ |
| medgemma zero_shot +TFM lstm | $0.799 \pm 6.902e-4$ | $1 \pm 3.201e-4$ | $0.802 \pm 4.163e-5$ | $1 \pm 1.967e-4$ | $0.807 \pm 1.253e-4$ | $1 \pm 2.721e-4$ |
| medgemma zero_shot +TFM mlp | $0.8 \pm 6.208e-4$ | $1 \pm 6.301e-4$ | $0.802 \pm 1.007e-4$ | $1 \pm 7.572e-5$ | $0.81 \pm 1.547e-3$ | $1.001 \pm 1.437e-4$ |
| medgemma zero_shot +TFM patchtsmixer | $0.806 \pm 3.184e-3$ | $1.019 \pm 6.825e-3$ | $0.805 \pm 1.699e-3$ | $1.008 \pm 6.408e-4$ | $0.818 \pm 3.971e-3$ | $1.016 \pm 6.382e-3$ |
| medgemma zero_shot +TFM timemixer | $0.799 \pm 1.214e-3$ | $1 \pm 2.055e-4$ | $0.802 \pm 2.517e-5$ | $1 \pm 0e0$ | $0.809 \pm 1.219e-3$ | $1 \pm 2.479e-4$ |
| medgemma zero_shot lstm | $0.799 \pm 1.36e-3$ | $1 \pm 3.178e-4$ | $0.802 \pm 1.184e-4$ | $1 \pm 2.944e-5$ | $0.808 \pm 3.326e-4$ | $1 \pm 1.36e-4$ |
| medgemma zero_shot mlp | $0.8 \pm 9.336e-4$ | $1 \pm 5.176e-4$ | $0.802 \pm 1.412e-4$ | $1 \pm 6e-5$ | $0.811 \pm 1.18e-3$ | $1.001 \pm 3.718e-4$ |
| medgemma zero_shot patchtsmixer | $0.801 \pm 1.17e-3$ | $1.006 \pm 1.53e-3$ | $0.803 \pm 2.27e-3$ | $1.005 \pm 1.433e-3$ | $0.814 \pm 1.797e-3$ | $1.005 \pm 1.552e-3$ |
| medgemma zero_shot timemixer | $0.799 \pm 7.274e-4$ | $1 \pm 1.086e-4$ | $0.802 \pm 1.065e-4$ | $1 \pm 1.258e-5$ | $0.808 \pm 3.534e-4$ | $1 \pm 6.397e-5$ |
| No_sum_zero_shot zero_shot +TFM lstm | $0.799 \pm 2.811e-4$ | $1 \pm 2.29e-4$ | $0.802 \pm 8.083e-5$ | $0.999 \pm 6.351e-5$ | $0.807 \pm 1.231e-4$ | $0.999 \pm 6.614e-4$ |
| No_sum_zero_shot zero_shot +TFM mlp | $0.8 \pm 1.34e-3$ | $1 \pm 6.005e-4$ | $0.802 \pm 2.732e-4$ | $1 \pm 3.02e-4$ | $0.81 \pm 4.343e-4$ | $1 \pm 4.859e-4$ |
| No_sum_zero_shot zero_shot +TFM patchtsmixer | $0.803 \pm 1.455e-3$ | $1.016 \pm 4.599e-3$ | $0.806 \pm 1.482e-3$ | $1.011 \pm 1.815e-3$ | $0.82 \pm 3.048e-3$ | $1.016 \pm 3.578e-3$ |
| No_sum_zero_shot zero_shot +TFM timemixer | $0.8 \pm 6.369e-4$ | $1 \pm 1.5e-4$ | $0.802 \pm 1.35e-4$ | $1 \pm 2.082e-5$ | $0.809 \pm 2.229e-3$ | $1.001 \pm 7.192e-4$ |
| No_sum_zero_shot lstm | $0.801 \pm 1.9e-3$ | $1.001 \pm 5.583e-4$ | $0.802 \pm 7.188e-5$ | $1 \pm 4.193e-5$ | $0.809 \pm 4.696e-4$ | $1 \pm 1.339e-4$ |
| No_sum_zero_shot mlp | $0.801 \pm 1.75e-3$ | $1.001 \pm 8.046e-4$ | $0.802 \pm 1.159e-4$ | $1 \pm 3.862e-5$ | $0.811 \pm 1.089e-3$ | $1.001 \pm 3.03e-4$ |
| No_sum_zero_shot patchtsmixer | $0.802 \pm 2.277e-3$ | $1.005 \pm 2.572e-3$ | $0.804 \pm 2.032e-3$ | $1.004 \pm 1.011e-3$ | $0.813 \pm 4.082e-3$ | $1.005 \pm 2.859e-3$ |
| No_sum_zero_shot timemixer | $0.8 \pm 6.189e-4$ | $1 \pm 1.24e-4$ | $0.802 \pm 1.409e-4$ | $1 \pm 1.893e-5$ | $0.81 \pm 5.082e-4$ | $1.001 \pm 1.623e-4$ |
| right lstm | $0.795 \pm 7.442e-4$ | $0.994 \pm 7.231e-4$ | $0.793 \pm 2.285e-4$ | $0.977 \pm 1.96e-4$ | $0.754 \pm 8.797e-4$ | $0.934 \pm 1.571e-4$ |
| right mlp | $0.795 \pm 7.886e-4$ | $0.993 \pm 2.05e-4$ | $0.793 \pm 1.916e-4$ | $0.977 \pm 2.916e-4$ | $0.754 \pm 2.142e-3$ | $0.933 \pm 9.933e-4$ |
| right patchtsmixer | $0.803 \pm 3.212e-3$ | $1.015 \pm 3.019e-3$ | $0.798 \pm 3.152e-3$ | $0.988 \pm 3.915e-3$ | $0.768 \pm 4.237e-3$ | $0.947 \pm 3.971e-3$ |
| right timemixer | $0.795 \pm 1.039e-3$ | $0.995 \pm 1.193e-4$ | $0.793 \pm 8.347e-5$ | $0.978 \pm 2.062e-5$ | $0.753 \pm 1.141e-3$ | $0.935 \pm 2.627e-4$ |

Table 19: In-distribution results - Age. (part 2/2)

| Method | hirid → ppicu | | mimic → ppicu | |
|---|---|---|---|---|
| | mae | mse | mae | mse |
| Llama-3.1 CoT +TFM lstm | $0.808 \pm 1.305e-3$ | $1 \pm 4.309e-4$ | $0.806 \pm 1.069e-4$ | $1 \pm 6.028e-5$ |
| Llama-3.1 CoT +TFM mlp | $0.809 \pm 7.101e-4$ | $1.001 \pm 6.509e-4$ | $0.806 \pm 1.457e-4$ | $1 \pm 3.215e-5$ |
| Llama-3.1 CoT +TFM patchtsmixer | $0.82 \pm 2.642e-3$ | $1.019 \pm 4.362e-3$ | $0.81 \pm 2.535e-3$ | $1.01 \pm 6.56e-3$ |
| Llama-3.1 CoT +TFM timemixer | $0.809 \pm 1.014e-3$ | $1 \pm 2.955e-4$ | $0.806 \pm 1.986e-4$ | $1 \pm 1e-5$ |
| Llama-3.1 CoT lstm | $0.809 \pm 2.399e-4$ | $1 \pm 9.274e-5$ | $0.806 \pm 1.097e-4$ | $1 \pm 2.944e-5$ |
| Llama-3.1 CoT mlp | $0.811 \pm 9.011e-4$ | $1.001 \pm 2.553e-4$ | $0.806 \pm 2.191e-4$ | $1 \pm 4.349e-5$ |
| Llama-3.1 CoT patchtsmixer | $0.813 \pm 3.711e-3$ | $1.008 \pm 3.914e-3$ | $0.809 \pm 3.074e-3$ | $1.004 \pm 3.167e-3$ |
| Llama-3.1 CoT timemixer | $0.809 \pm 9.832e-5$ | $1 \pm 2.363e-5$ | $0.806 \pm 9.743e-5$ | $1 \pm 5.774e-6$ |
| Llama-3.1 ICD +TFM lstm | $0.808 \pm 1.906e-3$ | $1.001 \pm 6.766e-4$ | $0.806 \pm 1.652e-4$ | $1 \pm 1.193e-4$ |
| Llama-3.1 ICD +TFM mlp | $0.81 \pm 7.51e-4$ | $1.001 \pm 6.311e-4$ | $0.806 \pm 1.825e-4$ | $1 \pm 3.786e-5$ |
| Llama-3.1 ICD +TFM patchtsmixer | $0.815 \pm 2.4e-3$ | $1.016 \pm 5.238e-3$ | $0.811 \pm 3.936e-3$ | $1.01 \pm 3.038e-3$ |
| Llama-3.1 ICD +TFM timemixer | $0.809 \pm 4.583e-4$ | $1 \pm 8.544e-5$ | $0.806 \pm 2.517e-5$ | $1 \pm 0e0$ |
| Llama-3.1 ICD lstm | $0.809 \pm 8.811e-4$ | $1 \pm 3.636e-4$ | $0.806 \pm 2.156e-4$ | $1 \pm 1.258e-5$ |
| Llama-3.1 ICD mlp | $0.811 \pm 2.589e-4$ | $1.001 \pm 4.671e-4$ | $0.806 \pm 2.696e-4$ | $1 \pm 9.287e-5$ |
| Llama-3.1 ICD patchtsmixer | $0.812 \pm 3.624e-3$ | $1.007 \pm 1.829e-3$ | $0.806 \pm 2.141e-3$ | $1.003 \pm 5.351e-4$ |
| Llama-3.1 ICD timemixer | $0.808 \pm 1.322e-3$ | $1 \pm 1.954e-4$ | $0.806 \pm 1.735e-4$ | $1 \pm 1.155e-5$ |
| Llama-3.1 Trend +TFM lstm | $0.808 \pm 7.317e-4$ | $1 \pm 1.686e-4$ | $0.806 \pm 1.852e-4$ | $1 \pm 2.122e-4$ |
| Llama-3.1 Trend +TFM mlp | $0.81 \pm 5.956e-4$ | $1.002 \pm 7.123e-4$ | $0.806 \pm 2.358e-4$ | $1 \pm 1.332e-4$ |
| Llama-3.1 Trend +TFM patchtsmixer | $0.817 \pm 1.749e-3$ | $1.015 \pm 1.827e-3$ | $0.81 \pm 1.337e-2$ | $1.008 \pm 2.62e-3$ |
| Llama-3.1 Trend +TFM timemixer | $0.809 \pm 1.489e-3$ | $1 \pm 3.053e-4$ | $0.806 \pm 1.429e-4$ | $1 \pm 5.774e-6$ |
| Llama-3.1 Trend lstm | $0.809 \pm 6.381e-4$ | $1 \pm 1.916e-4$ | $0.806 \pm 6.055e-5$ | $1 \pm 1.204e-4$ |
| Llama-3.1 Trend mlp | $0.81 \pm 1.442e-3$ | $1 \pm 6.115e-4$ | $0.806 \pm 2.001e-4$ | $1 \pm 5.56e-5$ |
| Llama-3.1 Trend patchtsmixer | $0.814 \pm 5.117e-3$ | $1.006 \pm 2.978e-3$ | $0.808 \pm 2.131e-3$ | $1.003 \pm 8.315e-4$ |
| Llama-3.1 Trend timemixer | $0.809 \pm 1.561e-3$ | $1.001 \pm 5.321e-4$ | $0.806 \pm 7.544e-5$ | $1 \pm 5.774e-6$ |
| Llama-3.1 zero_shot +TFM lstm | $0.808 \pm 1.195e-3$ | $1 \pm 2.488e-4$ | $0.806 \pm 2.303e-4$ | $1 \pm 6.245e-5$ |
| Llama-3.1 zero_shot +TFM mlp | $0.809 \pm 1.027e-3$ | $1.001 \pm 5.934e-4$ | $0.806 \pm 6.986e-4$ | $1 \pm 4.151e-4$ |
| Llama-3.1 zero_shot +TFM patchtsmixer | $0.818 \pm 3.996e-3$ | $1.017 \pm 4.479e-3$ | $0.81 \pm 3.067e-3$ | $1.011 \pm 4.664e-3$ |
| Llama-3.1 zero_shot +TFM timemixer | $0.809 \pm 2.103e-4$ | $1 \pm 5.568e-5$ | $0.806 \pm 5.774e-5$ | $1 \pm 5.774e-6$ |
| Llama-3.1 zero_shot lstm | $0.809 \pm 6.559e-4$ | $1 \pm 1.919e-4$ | $0.806 \pm 1.746e-4$ | $1 \pm 3.5e-5$ |
| Llama-3.1 zero_shot mlp | $0.811 \pm 6.466e-4$ | $1.001 \pm 3.994e-4$ | $0.806 \pm 2.325e-4$ | $1 \pm 1.041e-4$ |
| Llama-3.1 zero_shot patchtsmixer | $0.812 \pm 2.08e-3$ | $1.006 \pm 1.313e-3$ | $0.81 \pm 2.757e-3$ | $1.006 \pm 2.814e-3$ |
| Llama-3.1 zero_shot timemixer | $0.809 \pm 8.806e-4$ | $1 \pm 2.117e-4$ | $0.806 \pm 2.986e-5$ | $1 \pm 0e0$ |
| TFM lstm | $0.752 \pm 8.624e-4$ | $0.935 \pm 1.439e-4$ | $0.75 \pm 1.715e-4$ | $0.935 \pm 2.179e-4$ |
| TFM mlp | $0.753 \pm 1.029e-3$ | $0.935 \pm 1.105e-4$ | $0.75 \pm 9.912e-5$ | $0.935 \pm 2.134e-4$ |
| TFM patchtsmixer | $0.761 \pm 3.671e-3$ | $0.947 \pm 1.981e-3$ | $0.753 \pm 1.024e-3$ | $0.943 \pm 1.584e-3$ |
| TFM timemixer | $0.753 \pm 8.952e-4$ | $0.934 \pm 2.106e-4$ | $0.75 \pm 2.007e-4$ | $0.934 \pm 4.435e-5$ |
| TSDE lstm | $0.751 \pm 4.323e-3$ | $0.936 \pm 9.137e-4$ | $0.751 \pm 3.034e-4$ | $0.935 \pm 7.839e-4$ |
| TSDE mlp | $0.748 \pm 1.451e-3$ | $0.939 \pm 5.069e-3$ | $0.752 \pm 8.398e-3$ | $0.936 \pm 5.379e-3$ |
| TSDE patchtsmixer | $0.762 \pm 2.922e-3$ | $0.951 \pm 3.481e-3$ | $0.755 \pm 2.716e-3$ | $0.944 \pm 6.36e-3$ |
| TSDE timemixer | $0.753 \pm 8.294e-4$ | $0.934 \pm 1.949e-4$ | $0.75 \pm 1.949e-4$ | $0.934 \pm 4.646e-5$ |
| gemini-2.0-flash CoT +TFM lstm | $0.807 \pm 1.032e-3$ | $1 \pm 6.643e-4$ | $0.806 \pm 1.493e-4$ | $0.999 \pm 3.544e-4$ |
| gemini-2.0-flash CoT +TFM mlp | $0.81 \pm 1.06e-3$ | $1.001 \pm 5.029e-4$ | $0.806 \pm 2.875e-4$ | $1 \pm 9.074e-5$ |
| gemini-2.0-flash CoT +TFM patchtsmixer | $0.814 \pm 3.898e-3$ | $1.016 \pm 6.893e-3$ | $0.809 \pm 3.754e-3$ | $1.011 \pm 5.063e-3$ |
| gemini-2.0-flash CoT +TFM timemixer | $0.809 \pm 1.33e-3$ | $1 \pm 3.384e-4$ | $0.806 \pm 1.955e-4$ | $1 \pm 1.528e-5$ |
| gemini-2.0-flash CoT lstm | $0.809 \pm 6.393e-4$ | $1 \pm 2.156e-4$ | $0.806 \pm 3.094e-4$ | $1 \pm 4.5e-5$ |
| gemini-2.0-flash CoT mlp | $0.81 \pm 5.832e-4$ | $1.001 \pm 1.909e-4$ | $0.806 \pm 9.032e-5$ | $1 \pm 1.152e-4$ |
| gemini-2.0-flash CoT patchtsmixer | $0.813 \pm 7.459e-3$ | $1.007 \pm 5.039e-3$ | $0.808 \pm 1.732e-3$ | $1.003 \pm 7.908e-4$ |
| gemini-2.0-flash CoT timemixer | $0.808 \pm 3.555e-4$ | $1 \pm 6.397e-5$ | $0.806 \pm 1.782e-4$ | $1 \pm 5.774e-6$ |
| gemini-2.0-flash ICD +TFM lstm | $0.808 \pm 4.688e-4$ | $1 \pm 7.38e-4$ | $0.806 \pm 1.361e-4$ | $1 \pm 1.32e-4$ |
| gemini-2.0-flash ICD +TFM mlp | $0.81 \pm 1.535e-3$ | $1.001 \pm 4.917e-4$ | $0.806 \pm 2.272e-4$ | $1 \pm 4.293e-4$ |
| gemini-2.0-flash ICD +TFM patchtsmixer | $0.818 \pm 8.371e-3$ | $1.02 \pm 1.025e-2$ | $0.809 \pm 1.245e-3$ | $1.009 \pm 2.844e-3$ |
| gemini-2.0-flash ICD +TFM timemixer | $0.809 \pm 4.362e-4$ | $1 \pm 9.539e-5$ | $0.806 \pm 1.127e-4$ | $1 \pm 1.155e-5$ |
| gemini-2.0-flash ICD lstm | $0.809 \pm 1.269e-3$ | $1.001 \pm 3.017e-4$ | $0.806 \pm 1.464e-4$ | $1 \pm 8.098e-5$ |
| gemini-2.0-flash ICD mlp | $0.811 \pm 8.003e-4$ | $1.001 \pm 2.453e-4$ | $0.806 \pm 2.757e-4$ | $1 \pm 4.031e-5$ |
| gemini-2.0-flash ICD patchtsmixer | $0.812 \pm 2.618e-3$ | $1.006 \pm 1.148e-3$ | $0.809 \pm 5.485e-3$ | $1.005 \pm 1.02e-3$ |
| gemini-2.0-flash ICD timemixer | $0.809 \pm 8.427e-4$ | $1 \pm 2.123e-4$ | $0.806 \pm 4.717e-5$ | $1 \pm 8.165e-6$ |
| gemini-2.0-flash Trend +TFM lstm | $0.807 \pm 6.799e-4$ | $1 \pm 4.355e-4$ | $0.806 \pm 2.787e-4$ | $1 \pm 1.25e-4$ |
| gemini-2.0-flash Trend +TFM mlp | $0.81 \pm 8.972e-4$ | $1.001 \pm 3.132e-4$ | $0.806 \pm 4.257e-4$ | $1 \pm 3.781e-4$ |
| gemini-2.0-flash Trend +TFM patchtsmixer | $0.821 \pm 1.446e-3$ | $1.02 \pm 3.373e-3$ | $0.809 \pm 1.505e-3$ | $1.012 \pm 4.289e-3$ |
| gemini-2.0-flash Trend +TFM timemixer | $0.81 \pm 6.006e-4$ | $1.001 \pm 2.248e-4$ | $0.806 \pm 1.589e-4$ | $1 \pm 1.155e-5$ |
| gemini-2.0-flash Trend lstm | $0.809 \pm 1.841e-3$ | $1.001 \pm 5.033e-4$ | $0.806 \pm 2.012e-4$ | $1 \pm 1.688e-4$ |
| gemini-2.0-flash Trend mlp | $0.81 \pm 7.437e-4$ | $1.001 \pm 5.1e-4$ | $0.806 \pm 2.729e-4$ | $1 \pm 4.573e-5$ |
| gemini-2.0-flash Trend patchtsmixer | $0.814 \pm 5.695e-3$ | $1.009 \pm 6.999e-3$ | $0.807 \pm 4.126e-3$ | $1.005 \pm 2.361e-3$ |
| gemini-2.0-flash Trend timemixer | $0.809 \pm 1.008e-3$ | $1 \pm 2.575e-4$ | $0.806 \pm 1.406e-4$ | $1 \pm 5e-6$ |
| gemini-2.0-flash zero_shot +TFM lstm | $0.808 \pm 4.114e-4$ | $1 \pm 3.051e-4$ | $0.806 \pm 4.583e-5$ | $1 \pm 1.825e-4$ |
| gemini-2.0-flash zero_shot +TFM mlp | $0.81 \pm 1.831e-3$ | $1.002 \pm 5.443e-4$ | $0.806 \pm 2.179e-4$ | $1 \pm 2.193e-4$ |
| gemini-2.0-flash zero_shot +TFM patchtsmixer | $0.816 \pm 2.724e-3$ | $1.015 \pm 4.367e-3$ | $0.809 \pm 4.541e-3$ | $1.009 \pm 5.198e-3$ |
| gemini-2.0-flash zero_shot +TFM timemixer | $0.809 \pm 9.551e-4$ | $1 \pm 2.359e-4$ | $0.806 \pm 4.966e-4$ | $1 \pm 5.774e-6$ |
| gemini-2.0-flash zero_shot lstm | $0.808 \pm 9.849e-4$ | $1 \pm 2.015e-4$ | $0.806 \pm 1.696e-4$ | $1 \pm 4.435e-5$ |
| gemini-2.0-flash zero_shot mlp | $0.81 \pm 6.791e-4$ | $1.001 \pm 3.608e-4$ | $0.806 \pm 9.592e-5$ | $1 \pm 7.5e-5$ |
| gemini-2.0-flash zero_shot patchtsmixer | $0.814 \pm 2.721e-3$ | $1.007 \pm 3.521e-3$ | $0.809 \pm 5.301e-3$ | $1.005 \pm 2.145e-3$ |
| gemini-2.0-flash zero_shot timemixer | $0.809 \pm 6.028e-4$ | $1 \pm 1.916e-4$ | $0.806 \pm 2.674e-4$ | $1 \pm 8.165e-6$ |
| interp lstm | $0.751 \pm 8.926e-4$ | $0.933 \pm 1.031e-3$ | $0.751 \pm 1.276e-3$ | $0.935 \pm 7.468e-4$ |
| interp mlp | $0.754 \pm 6.643e-4$ | $0.939 \pm 3.273e-3$ | $0.752 \pm 1.327e-3$ | $0.938 \pm 4.643e-3$ |
| interp patchtsmixer | $0.785 \pm 2.196e-2$ | $0.998 \pm 2.299e-2$ | $0.778 \pm 6.66e-3$ | $0.976 \pm 8.186e-3$ |
| interp timemixer | $0.753 \pm 7.487e-4$ | $0.934 \pm 1.144e-4$ | $0.75 \pm 7.165e-5$ | $0.935 \pm 2.449e-5$ |
| mean lstm | $0.752 \pm 1.66e-3$ | $0.934 \pm 3.463e-4$ | $0.75 \pm 1.994e-3$ | $0.934 \pm 8.96e-4$ |
| mean mlp | $0.752 \pm 1.087e-3$ | $0.933 \pm 1.557e-3$ | $0.751 \pm 1.89e-3$ | $0.936 \pm 1.947e-3$ |
| mean patchtsmixer | $0.805 \pm 7.924e-3$ | $1.022 \pm 1.104e-2$ | $0.786 \pm 1.582e-2$ | $0.991 \pm 2.368e-2$ |
| mean timemixer | $0.753 \pm 7.514e-4$ | $0.934 \pm 1.1e-4$ | $0.75 \pm 1.05e-4$ | $0.934 \pm 2.363e-5$ |
| No_sum_CoT CoT +TFM lstm | $0.809 \pm 2.112e-3$ | $1 \pm 6.208e-4$ | $0.805 \pm 7.478e-4$ | $1 \pm 4.276e-4$ |
| No_sum_CoT CoT +TFM mlp | $0.809 \pm 4.597e-4$ | $1.001 \pm 6.295e-4$ | $0.806 \pm 5.831e-4$ | $1 \pm 4.072e-4$ |
| No_sum_CoT CoT +TFM patchtsmixer | $0.813 \pm 2.364e-4$ | $1.012 \pm 1.163e-3$ | $0.815 \pm 5.372e-3$ | $1.013 \pm 5.639e-3$ |
| No_sum_CoT CoT +TFM timemixer | $0.809 \pm 1.329e-3$ | $1 \pm 4.188e-4$ | $0.806 \pm 1.617e-4$ | $1 \pm 1e-5$ |
| No_sum_CoT lstm | $0.809 \pm 7.066e-4$ | $1 \pm 1.576e-4$ | $0.806 \pm 1.992e-4$ | $1 \pm 1.435e-4$ |
| No_sum_CoT mlp | $0.811 \pm 7.703e-4$ | $1.001 \pm 8.169e-4$ | $0.806 \pm 2.83e-4$ | $1 \pm 7.5e-5$ |
| No_sum_CoT patchtsmixer | $0.811 \pm 6.678e-4$ | $1.007 \pm 3.376e-3$ | $0.809 \pm 4.199e-3$ | $1.003 \pm 2.354e-4$ |
| No_sum_CoT timemixer | $0.808 \pm 7.487e-4$ | $1 \pm 1.182e-4$ | $0.806 \pm 1.556e-4$ | $1 \pm 9.574e-6$ |
| No_sum_ICD ICD +TFM lstm | $0.807 \pm 8.242e-4$ | $1 \pm 6.817e-4$ | $0.806 \pm 1.559e-4$ | $0.999 \pm 1.57e-4$ |
| No_sum_ICD ICD +TFM mlp | $0.81 \pm 1.585e-3$ | $1 \pm 1.733e-3$ | $0.806 \pm 4.96e-4$ | $1.001 \pm 7.355e-4$ |
| No_sum_ICD ICD +TFM patchtsmixer | $0.82 \pm 5.856e-4$ | $1.019 \pm 2.224e-3$ | $0.808 \pm 1.143e-3$ | $1.006 \pm 3.275e-3$ |
| No_sum_ICD ICD +TFM timemixer | $0.809 \pm 1.065e-3$ | $1 \pm 3.055e-4$ | $0.806 \pm 7.506e-5$ | $1 \pm 5.774e-6$ |
| No_sum_ICD lstm | $0.809 \pm 1.448e-3$ | $1.001 \pm 4.847e-4$ | $0.806 \pm 4.021e-4$ | $1 \pm 1.452e-4$ |
| No_sum_ICD mlp | $0.81 \pm 7.661e-4$ | $1.001 \pm 1.434e-4$ | $0.806 \pm 1.575e-4$ | $1 \pm 8.421e-5$ |
| No_sum_ICD patchtsmixer | $0.81 \pm 3.309e-3$ | $1.006 \pm 2.53e-3$ | $0.807 \pm 2.762e-3$ | $1.004 \pm 2.054e-3$ |
| No_sum_ICD timemixer | $0.809 \pm 2.861e-4$ | $1 \pm 6.481e-5$ | $0.806 \pm 7.394e-5$ | $1 \pm 5.774e-6$ |
| No_sum_Trend Trend +TFM lstm | $0.807 \pm 4.004e-4$ | $0.999 \pm 1.106e-3$ | $0.806 \pm 5.069e-4$ | $0.999 \pm 8.869e-4$ |
| No_sum_Trend Trend +TFM mlp | $0.81 \pm 8.718e-5$ | $1.001 \pm 1e-3$ | $0.806 \pm 3.279e-4$ | $1 \pm 3.315e-4$ |
| No_sum_Trend Trend +TFM patchtsmixer | $0.818 \pm 1.8e-3$ | $1.02 \pm 6.814e-3$ | $0.808 \pm 5.538e-4$ | $1.009 \pm 2.116e-3$ |
| No_sum_Trend Trend +TFM timemixer | $0.809 \pm 1.429e-3$ | $1 \pm 4.236e-4$ | $0.806 \pm 1.501e-4$ | $1 \pm 1.155e-5$ |

Table 20: Cross-site transfer results - Age. (part 1/2)

| Method | hirid → ppicu | | mimic → ppicu | |
|---|---|---|---|---|
| | mae | mse | mae | mse |
| No_sum_Trend lstm | $0.808 \pm 1.044e-3$ | $1 \pm 3.807e-4$ | $0.806 \pm 1.065e-4$ | $1 \pm 1.88e-4$ |
| No_sum_Trend mlp | $0.81 \pm 1.264e-3$ | $1 \pm 7.27e-4$ | $0.806 \pm 3.889e-4$ | $1 \pm 2.887e-5$ |
| No_sum_Trend patchtsmixer | $0.814 \pm 3.215e-3$ | $1.007 \pm 3.595e-3$ | $0.808 \pm 4.422e-3$ | $1.005 \pm 3.252e-3$ |
| No_sum_Trend timemixer | $0.809 \pm 5.934e-4$ | $1 \pm 1.409e-4$ | $0.806 \pm 7.5e-5$ | $1 \pm 5e-6$ |
| medgemma CoT +TFM lstm | $0.807 \pm 6.886e-4$ | $1 \pm 3.897e-4$ | $0.806 \pm 2.281e-4$ | $1 \pm 1.756e-4$ |
| medgemma CoT +TFM mlp | $0.81 \pm 9.819e-4$ | $1.002 \pm 4.46e-4$ | $0.806 \pm 7.937e-5$ | $1 \pm 3.215e-5$ |
| medgemma CoT +TFM patchtsmixer | $0.819 \pm 4.652e-3$ | $1.026 \pm 1.106e-2$ | $0.809 \pm 1.123e-3$ | $1.009 \pm 1.528e-3$ |
| medgemma CoT +TFM timemixer | $0.81 \pm 1.138e-3$ | $1.001 \pm 3.758e-4$ | $0.806 \pm 1.301e-4$ | $1 \pm 5.774e-6$ |
| medgemma CoT lstm | $0.808 \pm 1.51e-3$ | $1 \pm 4.267e-4$ | $0.806 \pm 5.745e-5$ | $1 \pm 4.573e-5$ |
| medgemma CoT mlp | $0.811 \pm 1.383e-3$ | $1.001 \pm 6.851e-4$ | $0.806 \pm 1.718e-4$ | $1 \pm 9.5e-5$ |
| medgemma CoT patchtsmixer | $0.812 \pm 2.312e-3$ | $1.006 \pm 1.899e-3$ | $0.811 \pm 2.944e-3$ | $1.006 \pm 2.257e-3$ |
| medgemma CoT timemixer | $0.809 \pm 4.816e-4$ | $1 \pm 1.329e-4$ | $0.806 \pm 2.278e-4$ | $1 \pm 9.574e-6$ |
| medgemma ICD +TFM lstm | $0.809 \pm 5.368e-4$ | $1 \pm 2.483e-4$ | $0.806 \pm 1.159e-4$ | $1 \pm 2.397e-4$ |
| medgemma ICD +TFM mlp | $0.811 \pm 1.223e-3$ | $1.002 \pm 1.109e-3$ | $0.806 \pm 1.345e-4$ | $1 \pm 4.107e-4$ |
| medgemma ICD +TFM patchtsmixer | $0.819 \pm 2.332e-3$ | $1.018 \pm 3.501e-3$ | $0.81 \pm 2.333e-3$ | $1.009 \pm 1.351e-3$ |
| medgemma ICD +TFM timemixer | $0.809 \pm 3.502e-4$ | $1 \pm 7.506e-5$ | $0.806 \pm 7.572e-5$ | $1 \pm 1.155e-5$ |
| medgemma ICD lstm | $0.808 \pm 8.344e-4$ | $1 \pm 1.992e-4$ | $0.806 \pm 1.976e-4$ | $1 \pm 4.349e-5$ |
| medgemma ICD mlp | $0.811 \pm 5.262e-4$ | $1.001 \pm 5.917e-4$ | $0.806 \pm 3.189e-4$ | $1 \pm 1.192e-4$ |
| medgemma ICD patchtsmixer | $0.813 \pm 4.584e-4$ | $1.007 \pm 3.122e-3$ | $0.809 \pm 2.417e-3$ | $1.004 \pm 1.357e-3$ |
| medgemma ICD timemixer | $0.808 \pm 5.545e-4$ | $1 \pm 1.08e-4$ | $0.806 \pm 8.655e-5$ | $1 \pm 0e0$ |
| medgemma Trend +TFM lstm | $0.807 \pm 1.015e-3$ | $1 \pm 1.852e-4$ | $0.806 \pm 1.069e-4$ | $1 \pm 1.701e-4$ |
| medgemma Trend +TFM mlp | $0.81 \pm 1.065e-3$ | $1.001 \pm 9.487e-4$ | $0.806 \pm 1.931e-4$ | $1 \pm 1.222e-4$ |
| medgemma Trend +TFM patchtsmixer | $0.818 \pm 1.52e-3$ | $1.017 \pm 3.255e-3$ | $0.809 \pm 3.577e-3$ | $1.011 \pm 5.994e-3$ |
| medgemma Trend +TFM timemixer | $0.809 \pm 1.036e-3$ | $1 \pm 2.227e-4$ | $0.806 \pm 1.501e-4$ | $1 \pm 1e-5$ |
| medgemma Trend lstm | $0.808 \pm 4.689e-4$ | $1 \pm 1.658e-4$ | $0.806 \pm 1.389e-4$ | $1 \pm 4.031e-5$ |
| medgemma Trend mlp | $0.81 \pm 9.645e-4$ | $1.001 \pm 5.056e-4$ | $0.806 \pm 1.05e-4$ | $1 \pm 1.323e-4$ |
| medgemma Trend patchtsmixer | $0.811 \pm 2.372e-3$ | $1.006 \pm 1.387e-3$ | $0.807 \pm 1.916e-3$ | $1.004 \pm 4.877e-4$ |
| medgemma Trend timemixer | $0.809 \pm 7.393e-4$ | $1 \pm 1.797e-4$ | $0.806 \pm 1.977e-4$ | $1 \pm 9.574e-6$ |
| medgemma zero_shot +TFM lstm | $0.808 \pm 1.404e-3$ | $1 \pm 8.8e-4$ | $0.806 \pm 1.114e-4$ | $1 \pm 1.629e-4$ |
| medgemma zero_shot +TFM mlp | $0.81 \pm 1.093e-3$ | $1.001 \pm 5.905e-4$ | $0.806 \pm 1.206e-4$ | $1 \pm 1.277e-4$ |
| medgemma zero_shot +TFM patchtsmixer | $0.819 \pm 2.865e-3$ | $1.019 \pm 5.564e-3$ | $0.809 \pm 2.487e-3$ | $1.008 \pm 1.207e-3$ |
| medgemma zero_shot +TFM timemixer | $0.809 \pm 1.033e-3$ | $1 \pm 2.088e-4$ | $0.806 \pm 3.606e-5$ | $1 \pm 5.774e-6$ |
| medgemma zero_shot lstm | $0.808 \pm 1.186e-3$ | $1 \pm 3.106e-4$ | $0.806 \pm 2.29e-4$ | $1 \pm 8.347e-5$ |
| medgemma zero_shot mlp | $0.809 \pm 7.985e-4$ | $1 \pm 4.812e-4$ | $0.806 \pm 2.216e-4$ | $1 \pm 4.272e-5$ |
| medgemma zero_shot patchtsmixer | $0.813 \pm 2.217e-3$ | $1.007 \pm 3.561e-3$ | $0.805 \pm 4.02e-3$ | $1.005 \pm 1.791e-3$ |
| medgemma zero_shot timemixer | $0.808 \pm 6.186e-4$ | $1 \pm 1.103e-4$ | $0.806 \pm 1.676e-4$ | $1 \pm 1.291e-5$ |
| No_sum_zero_shot zero_shot +TFM lstm | $0.809 \pm 3.194e-4$ | $1.001 \pm 4.528e-4$ | $0.806 \pm 1.311e-3$ | $1 \pm 1.171e-3$ |
| No_sum_zero_shot zero_shot +TFM mlp | $0.81 \pm 2.697e-3$ | $1 \pm 3.251e-3$ | $0.805 \pm 9.241e-4$ | $1 \pm 3.913e-4$ |
| No_sum_zero_shot zero_shot +TFM patchtsmixer | $0.816 \pm 1.487e-3$ | $1.016 \pm 5.008e-3$ | $0.81 \pm 1.121e-3$ | $1.01 \pm 1.845e-3$ |
| No_sum_zero_shot zero_shot +TFM timemixer | $0.809 \pm 5.515e-4$ | $1 \pm 1.752e-4$ | $0.806 \pm 2.05e-4$ | $1 \pm 1.528e-5$ |
| No_sum_zero_shot lstm | $0.81 \pm 1.741e-3$ | $1.001 \pm 4.948e-4$ | $0.806 \pm 1.261e-4$ | $1 \pm 5.123e-5$ |
| No_sum_zero_shot mlp | $0.81 \pm 1.638e-3$ | $1.001 \pm 9.81e-4$ | $0.806 \pm 1.682e-4$ | $1 \pm 1.037e-4$ |
| No_sum_zero_shot patchtsmixer | $0.812 \pm 1.574e-3$ | $1.007 \pm 2.336e-3$ | $0.807 \pm 4.249e-3$ | $1.004 \pm 9.251e-4$ |
| No_sum_zero_shot timemixer | $0.809 \pm 5.216e-4$ | $1 \pm 1.253e-4$ | $0.806 \pm 2.181e-4$ | $1 \pm 1.732e-5$ |
| right lstm | $0.751 \pm 9.323e-4$ | $0.933 \pm 1.35e-3$ | $0.751 \pm 1.816e-3$ | $0.935 \pm 5.847e-4$ |
| right mlp | $0.753 \pm 1.278e-3$ | $0.933 \pm 1.511e-3$ | $0.75 \pm 6.719e-4$ | $0.934 \pm 5.928e-4$ |
| right patchtsmixer | $0.792 \pm 1.487e-2$ | $0.994 \pm 1.356e-2$ | $0.772 \pm 2.593e-3$ | $0.966 \pm 5.168e-3$ |
| right timemixer | $0.753 \pm 7.362e-4$ | $0.934 \pm 1.1e-4$ | $0.75 \pm 1.352e-4$ | $0.935 \pm 4.655e-5$ |

Table 21: Cross-site transfer results - Age. (part 2/2)

| Method | hirid → hirid | | mimic → mimic | | ppicu → ppicu | |
|---|---|---|---|---|---|---|
| | auprc | auroc | auprc | auroc | auprc | auroc |
| Llama-3.1 CoT +TFM lstm | $0.682 \pm 8.894e-3$ | $0.555 \pm 1.246e-2$ | $0.655 \pm 1.33e-2$ | $0.585 \pm 1.425e-2$ | $0.693 \pm 7.3e-3$ | $0.573 \pm 1.212e-2$ |
| Llama-3.1 CoT +TFM mlp | $0.72 \pm 7.983e-3$ | $0.6 \pm 1.015e-2$ | $0.705 \pm 1.309e-2$ | $0.64 \pm 1.339e-2$ | $0.725 \pm 1.677e-2$ | $0.608 \pm 2.07e-2$ |
| Llama-3.1 CoT +TFM patchtsmixer | $0.704 \pm 3.667e-4$ | $0.58 \pm 1.341e-3$ | $0.702 \pm 8.468e-4$ | $0.63 \pm 2e-4$ | $0.714 \pm 5.538e-3$ | $0.595 \pm 6.329e-3$ |
| Llama-3.1 CoT +TFM timemixer | $0.707 \pm 2.153e-3$ | $0.588 \pm 2.084e-3$ | $0.696 \pm 1.456e-2$ | $0.631 \pm 1.694e-2$ | $0.7 \pm 6.089e-3$ | $0.584 \pm 5.492e-3$ |
| Llama-3.1 CoT lstm | $0.702 \pm 3.827e-3$ | $0.579 \pm 5.909e-3$ | $0.685 \pm 2.532e-2$ | $0.612 \pm 2.763e-2$ | $0.696 \pm 6.709e-3$ | $0.573 \pm 1.042e-2$ |
| Llama-3.1 CoT mlp | $0.704 \pm 3.167e-3$ | $0.581 \pm 4.62e-3$ | $0.702 \pm 8.377e-3$ | $0.634 \pm 9.491e-3$ | $0.707 \pm 1.567e-2$ | $0.586 \pm 2.046e-2$ |
| Llama-3.1 CoT patchtsmixer | $0.697 \pm 3.109e-3$ | $0.572 \pm 3.425e-3$ | $0.698 \pm 1.118e-2$ | $0.62 \pm 1.096e-2$ | $0.713 \pm 1.649e-2$ | $0.593 \pm 2.004e-2$ |
| Llama-3.1 CoT timemixer | $0.7 \pm 8.593e-4$ | $0.574 \pm 9.507e-4$ | $0.682 \pm 6.902e-3$ | $0.608 \pm 6.689e-3$ | $0.698 \pm 1.743e-3$ | $0.573 \pm 2.178e-3$ |
| Llama-3.1 ICD +TFM lstm | $0.681 \pm 1.215e-2$ | $0.555 \pm 1.428e-2$ | $0.624 \pm 1.094e-2$ | $0.554 \pm 9.214e-3$ | $0.692 \pm 4.155e-3$ | $0.567 \pm 5.781e-3$ |
| Llama-3.1 ICD +TFM mlp | $0.721 \pm 1.447e-2$ | $0.603 \pm 1.76e-2$ | $0.658 \pm 1.66e-2$ | $0.591 \pm 1.936e-2$ | $0.717 \pm 8.973e-3$ | $0.596 \pm 1.237e-2$ |
| Llama-3.1 ICD +TFM patchtsmixer | $0.708 \pm 5.577e-4$ | $0.587 \pm 3.686e-4$ | $0.657 \pm 8.486e-3$ | $0.587 \pm 8.952e-3$ | $0.719 \pm 7.324e-3$ | $0.598 \pm 8.555e-3$ |
| Llama-3.1 ICD +TFM timemixer | $0.705 \pm 1.578e-3$ | $0.586 \pm 3.109e-3$ | $0.662 \pm 4.638e-3$ | $0.597 \pm 5.18e-3$ | $0.695 \pm 8.775e-3$ | $0.571 \pm 8.912e-3$ |
| Llama-3.1 ICD lstm | $0.69 \pm 3.973e-3$ | $0.565 \pm 6.009e-3$ | $0.643 \pm 5.652e-3$ | $0.571 \pm 5.622e-3$ | $0.69 \pm 5.427e-3$ | $0.565 \pm 7.829e-3$ |
| Llama-3.1 ICD mlp | $0.698 \pm 7.389e-3$ | $0.575 \pm 9.757e-3$ | $0.645 \pm 3.347e-3$ | $0.575 \pm 4.081e-3$ | $0.699 \pm 3.943e-3$ | $0.574 \pm 4.436e-3$ |
| Llama-3.1 ICD patchtsmixer | $0.692 \pm 1.261e-3$ | $0.569 \pm 2.164e-3$ | $0.648 \pm 9.119e-3$ | $0.577 \pm 8.647e-3$ | $0.689 \pm 2.35e-3$ | $0.561 \pm 3.104e-3$ |
| Llama-3.1 ICD timemixer | $0.695 \pm 1.662e-3$ | $0.571 \pm 1.684e-3$ | $0.65 \pm 2.146e-3$ | $0.577 \pm 2.153e-3$ | $0.688 \pm 1.409e-3$ | $0.558 \pm 2.071e-3$ |
| Llama-3.1 Trend +TFM lstm | $0.683 \pm 8.034e-3$ | $0.556 \pm 9.091e-3$ | $0.62 \pm 2.583e-3$ | $0.553 \pm 9.563e-4$ | $0.697 \pm 3.774e-3$ | $0.575 \pm 4.552e-3$ |
| Llama-3.1 Trend +TFM mlp | $0.713 \pm 1.134e-2$ | $0.593 \pm 1.302e-2$ | $0.669 \pm 1.752e-2$ | $0.599 \pm 2.069e-2$ | $0.711 \pm 5.604e-4$ | $0.587 \pm 7.629e-4$ |
| Llama-3.1 Trend +TFM patchtsmixer | $0.708 \pm 3.699e-3$ | $0.585 \pm 5.518e-3$ | $0.672 \pm 6.595e-3$ | $0.602 \pm 7.253e-3$ | $0.713 \pm 7.321e-3$ | $0.591 \pm 9.864e-3$ |
| Llama-3.1 Trend +TFM timemixer | $0.707 \pm 8.358e-3$ | $0.589 \pm 9.674e-3$ | $0.661 \pm 7.25e-3$ | $0.591 \pm 7.592e-3$ | $0.713 \pm 3.603e-3$ | $0.596 \pm 4.388e-3$ |
| Llama-3.1 Trend lstm | $0.7 \pm 4.246e-3$ | $0.576 \pm 6.409e-3$ | $0.649 \pm 4.433e-3$ | $0.575 \pm 4.399e-3$ | $0.701 \pm 5.662e-3$ | $0.579 \pm 7.719e-3$ |
| Llama-3.1 Trend mlp | $0.699 \pm 5.085e-3$ | $0.574 \pm 8.231e-3$ | $0.663 \pm 8.269e-3$ | $0.592 \pm 9.672e-3$ | $0.7 \pm 5.528e-3$ | $0.576 \pm 6.935e-3$ |
| Llama-3.1 Trend patchtsmixer | $0.7 \pm 9.98e-4$ | $0.575 \pm 1.986e-3$ | $0.667 \pm 1.93e-2$ | $0.592 \pm 1.845e-2$ | $0.701 \pm 3.676e-3$ | $0.577 \pm 4.355e-3$ |
| Llama-3.1 Trend timemixer | $0.701 \pm 5.141e-4$ | $0.575 \pm 8.269e-4$ | $0.664 \pm 9.245e-3$ | $0.589 \pm 1.015e-2$ | $0.7 \pm 1.113e-3$ | $0.574 \pm 1.374e-3$ |
| Llama-3.1 zero_shot +TFM lstm | $0.687 \pm 6.535e-3$ | $0.56 \pm 1.161e-2$ | $0.64 \pm 9.101e-3$ | $0.573 \pm 9.565e-3$ | $0.687 \pm 1.161e-2$ | $0.564 \pm 1.657e-2$ |
| Llama-3.1 zero_shot +TFM mlp | $0.719 \pm 1.746e-2$ | $0.602 \pm 2.091e-2$ | $0.729 \pm 8.446e-3$ | $0.675 \pm 9.738e-3$ | $0.716 \pm 1.302e-2$ | $0.595 \pm 1.367e-2$ |
| Llama-3.1 zero_shot +TFM patchtsmixer | $0.704 \pm 3.635e-4$ | $0.581 \pm 3.557e-4$ | $0.702 \pm 2.06e-2$ | $0.64 \pm 2.288e-2$ | $0.716 \pm 5.97e-3$ | $0.595 \pm 6.605e-3$ |
| Llama-3.1 zero_shot +TFM timemixer | $0.702 \pm 2.551e-3$ | $0.583 \pm 5.376e-3$ | $0.681 \pm 1.312e-2$ | $0.621 \pm 1.569e-2$ | $0.703 \pm 3.164e-3$ | $0.577 \pm 5.926e-3$ |
| Llama-3.1 zero_shot lstm | $0.697 \pm 3.063e-3$ | $0.575 \pm 5.793e-3$ | $0.67 \pm 1.631e-2$ | $0.605 \pm 1.946e-2$ | $0.7 \pm 8.366e-3$ | $0.578 \pm 1.064e-2$ |
| Llama-3.1 zero_shot mlp | $0.7 \pm 2.574e-3$ | $0.577 \pm 2.975e-3$ | $0.693 \pm 9.806e-3$ | $0.635 \pm 1.078e-2$ | $0.698 \pm 5.621e-3$ | $0.573 \pm 5.56e-3$ |
| Llama-3.1 zero_shot patchtsmixer | $0.698 \pm 1.777e-3$ | $0.575 \pm 3.099e-3$ | $0.709 \pm 4.749e-3$ | $0.651 \pm 4.87e-3$ | $0.699 \pm 6.637e-3$ | $0.574 \pm 6.645e-3$ |
| Llama-3.1 zero_shot timemixer | $0.702 \pm 1.709e-3$ | $0.579 \pm 2.678e-3$ | $0.699 \pm 9.912e-3$ | $0.639 \pm 1.057e-2$ | $0.696 \pm 8.549e-3$ | $0.569 \pm 8.527e-3$ |
| TFM lstm | $0.67 \pm 5.001e-3$ | $0.541 \pm 4.894e-3$ | $0.594 \pm 4.994e-3$ | $0.529 \pm 3.431e-3$ | $0.654 \pm 5.393e-3$ | $0.517 \pm 3.594e-3$ |
| TFM mlp | $0.671 \pm 4.555e-3$ | $0.545 \pm 5.417e-3$ | $0.586 \pm 2.113e-3$ | $0.532 \pm 1.159e-3$ | $0.661 \pm 1.712e-3$ | $0.528 \pm 1.28e-3$ |
| TFM patchtsmixer | $0.673 \pm 1.693e-3$ | $0.548 \pm 8.308e-4$ | $0.589 \pm 1.442e-3$ | $0.532 \pm 5.452e-4$ | $0.664 \pm 6.923e-4$ | $0.53 \pm 1.105e-3$ |
| TFM timemixer | $0.671 \pm 3.346e-3$ | $0.549 \pm 3.121e-3$ | $0.601 \pm 3.253e-3$ | $0.536 \pm 1.391e-3$ | $0.661 \pm 2.082e-3$ | $0.526 \pm 1.573e-3$ |
| TSDE lstm | $0.681 \pm 6.848e-3$ | $0.554 \pm 1.058e-2$ | $0.819 \pm 2.1e-3$ | $0.773 \pm 2.974e-3$ | $0.87 \pm 5.957e-3$ | $0.795 \pm 8.274e-3$ |
| TSDE mlp | $0.791 \pm 3.345e-3$ | $0.681 \pm 3.648e-3$ | $0.85 \pm 2.178e-3$ | $0.811 \pm 2.348e-3$ | $0.898 \pm 1.83e-3$ | $0.832 \pm 3.731e-3$ |
| TSDE patchtsmixer | $0.799 \pm 7.006e-4$ | $0.688 \pm 2.859e-3$ | $0.851 \pm 1.679e-3$ | $0.812 \pm 2.108e-3$ | $0.901 \pm 2.816e-3$ | $0.836 \pm 4.382e-3$ |
| TSDE timemixer | $0.767 \pm 1.683e-2$ | $0.65 \pm 2.076e-2$ | $0.846 \pm 3.735e-3$ | $0.804 \pm 4.207e-3$ | $0.885 \pm 1.801e-3$ | $0.815 \pm 1.597e-3$ |
| gemini-2.0-flash CoT +TFM lstm | $0.688 \pm 5.348e-3$ | $0.564 \pm 5.267e-3$ | $0.669 \pm 1.464e-2$ | $0.605 \pm 1.303e-2$ | $0.717 \pm 1.128e-2$ | $0.601 \pm 1.495e-2$ |
| gemini-2.0-flash CoT +TFM mlp | $0.718 \pm 1.129e-3$ | $0.597 \pm 2.09e-3$ | $0.712 \pm 1.346e-2$ | $0.653 \pm 1.543e-2$ | $0.75 \pm 1.31e-2$ | $0.637 \pm 1.75e-2$ |
| gemini-2.0-flash CoT +TFM patchtsmixer | $0.72 \pm 1.202e-3$ | $0.6 \pm 1.908e-4$ | $0.695 \pm 1.63e-3$ | $0.631 \pm 1.855e-3$ | $0.747 \pm 7.968e-4$ | $0.632 \pm 1.418e-3$ |
| gemini-2.0-flash CoT +TFM timemixer | $0.714 \pm 2.798e-3$ | $0.597 \pm 1.736e-3$ | $0.688 \pm 5.022e-3$ | $0.629 \pm 5.829e-3$ | $0.723 \pm 8.138e-3$ | $0.605 \pm 1.228e-2$ |
| gemini-2.0-flash CoT lstm | $0.706 \pm 6.648e-3$ | $0.585 \pm 8.732e-3$ | $0.691 \pm 1.918e-2$ | $0.628 \pm 2.157e-2$ | $0.721 \pm 4.388e-3$ | $0.604 \pm 6.022e-3$ |
| gemini-2.0-flash CoT mlp | $0.71 \pm 4.685e-3$ | $0.588 \pm 5.742e-3$ | $0.685 \pm 8.121e-3$ | $0.624 \pm 8.335e-3$ | $0.723 \pm 5.409e-3$ | $0.603 \pm 6.741e-3$ |
| gemini-2.0-flash CoT patchtsmixer | $0.707 \pm 1.058e-3$ | $0.585 \pm 1.192e-3$ | $0.692 \pm 7.343e-3$ | $0.627 \pm 7.429e-3$ | $0.723 \pm 1.102e-2$ | $0.601 \pm 1.311e-2$ |
| gemini-2.0-flash CoT timemixer | $0.707 \pm 2.137e-3$ | $0.584 \pm 1.325e-3$ | $0.683 \pm 6.923e-3$ | $0.617 \pm 7.006e-3$ | $0.724 \pm 8.265e-3$ | $0.601 \pm 9.491e-3$ |
| gemini-2.0-flash ICD +TFM lstm | $0.688 \pm 8.135e-3$ | $0.566 \pm 1.118e-2$ | $0.667 \pm 1.216e-2$ | $0.607 \pm 1.217e-2$ | $0.704 \pm 1.131e-2$ | $0.589 \pm 1.698e-2$ |
| gemini-2.0-flash ICD +TFM mlp | $0.725 \pm 1.092e-2$ | $0.605 \pm 1.293e-2$ | $0.709 \pm 3.413e-3$ | $0.653 \pm 2.715e-3$ | $0.732 \pm 1.242e-2$ | $0.615 \pm 1.693e-2$ |
| gemini-2.0-flash ICD +TFM patchtsmixer | $0.727 \pm 1.79e-2$ | $0.606 \pm 2.454e-2$ | $0.703 \pm 1.196e-2$ | $0.642 \pm 1.193e-2$ | $0.741 \pm 5.058e-3$ | $0.627 \pm 6.805e-3$ |
| gemini-2.0-flash ICD +TFM timemixer | $0.714 \pm 8.728e-4$ | $0.596 \pm 2.768e-3$ | $0.696 \pm 4.552e-3$ | $0.641 \pm 5.327e-3$ | $0.711 \pm 1.125e-2$ | $0.596 \pm 1.01e-2$ |
| gemini-2.0-flash ICD lstm | $0.706 \pm 4.078e-3$ | $0.584 \pm 5.169e-3$ | $0.681 \pm 1.47e-2$ | $0.621 \pm 1.588e-2$ | $0.719 \pm 5.841e-3$ | $0.601 \pm 9.043e-3$ |
| gemini-2.0-flash ICD mlp | $0.717 \pm 8.876e-3$ | $0.597 \pm 1.284e-2$ | $0.693 \pm 2.445e-3$ | $0.635 \pm 2.166e-3$ | $0.721 \pm 6.321e-3$ | $0.601 \pm 8.13e-3$ |
| gemini-2.0-flash ICD patchtsmixer | $0.707 \pm 1.419e-3$ | $0.583 \pm 1.455e-3$ | $0.696 \pm 1.089e-2$ | $0.637 \pm 1.187e-2$ | $0.718 \pm 8.059e-3$ | $0.597 \pm 9.69e-3$ |
| gemini-2.0-flash ICD timemixer | $0.707 \pm 5.886e-4$ | $0.581 \pm 1.143e-3$ | $0.688 \pm 7.978e-3$ | $0.628 \pm 8.641e-3$ | $0.714 \pm 4.16e-3$ | $0.592 \pm 4.244e-3$ |
| gemini-2.0-flash Trend +TFM lstm | $0.677 \pm 4.555e-3$ | $0.546 \pm 8.996e-3$ | $0.659 \pm 7.126e-3$ | $0.592 \pm 7.249e-3$ | $0.716 \pm 5.095e-3$ | $0.605 \pm 6.79e-3$ |
| gemini-2.0-flash Trend +TFM mlp | $0.719 \pm 1.302e-3$ | $0.596 \pm 1.14e-3$ | $0.701 \pm 1.563e-3$ | $0.638 \pm 1.693e-3$ | $0.719 \pm 1.442e-3$ | $0.599 \pm 1.209e-3$ |
| gemini-2.0-flash Trend +TFM patchtsmixer | $0.707 \pm 7.753e-4$ | $0.581 \pm 1.084e-3$ | $0.692 \pm 1.293e-3$ | $0.625 \pm 1.285e-3$ | $0.721 \pm 7.448e-3$ | $0.603 \pm 9.94e-3$ |
| gemini-2.0-flash Trend +TFM timemixer | $0.707 \pm 6.438e-4$ | $0.588 \pm 3.874e-4$ | $0.685 \pm 1.121e-2$ | $0.623 \pm 1.354e-2$ | $0.715 \pm 1.523e-3$ | $0.601 \pm 3.991e-3$ |
| gemini-2.0-flash Trend lstm | $0.698 \pm 1.22e-4$ | $0.574 \pm 1.025e-3$ | $0.671 \pm 7.364e-3$ | $0.6 \pm 7.048e-3$ | $0.712 \pm 2.628e-3$ | $0.595 \pm 4.96e-3$ |
| gemini-2.0-flash Trend mlp | $0.7 \pm 2.748e-3$ | $0.576 \pm 2.57e-3$ | $0.68 \pm 9.396e-3$ | $0.612 \pm 1.043e-2$ | $0.709 \pm 5.213e-3$ | $0.587 \pm 6.987e-3$ |
| gemini-2.0-flash Trend patchtsmixer | $0.699 \pm 1.321e-3$ | $0.575 \pm 1.197e-3$ | $0.686 \pm 7.111e-3$ | $0.614 \pm 7.194e-3$ | $0.708 \pm 2.187e-3$ | $0.585 \pm 3.249e-3$ |
| gemini-2.0-flash Trend timemixer | $0.7 \pm 1.712e-3$ | $0.575 \pm 1.076e-3$ | $0.678 \pm 3.487e-3$ | $0.608 \pm 3.882e-3$ | $0.707 \pm 3.588e-3$ | $0.583 \pm 4.458e-3$ |
| gemini-2.0-flash zero_shot +TFM lstm | $0.7 \pm 1.392e-2$ | $0.579 \pm 2e-2$ | $0.662 \pm 1.173e-2$ | $0.597 \pm 1.356e-2$ | $0.714 \pm 6.285e-3$ | $0.6 \pm 8.373e-3$ |
| gemini-2.0-flash zero_shot +TFM mlp | $0.742 \pm 1.127e-2$ | $0.627 \pm 1.357e-2$ | $0.694 \pm 2.292e-2$ | $0.634 \pm 2.392e-2$ | $0.749 \pm 1.657e-3$ | $0.637 \pm 2.262e-3$ |
| gemini-2.0-flash zero_shot +TFM patchtsmixer | $0.748 \pm 5.132e-3$ | $0.635 \pm 6.363e-3$ | $0.702 \pm 1.048e-2$ | $0.64 \pm 1.122e-2$ | $0.742 \pm 1.007e-2$ | $0.628 \pm 1.227e-2$ |
| gemini-2.0-flash zero_shot +TFM timemixer | $0.73 \pm 1.432e-2$ | $0.611 \pm 1.471e-2$ | $0.675 \pm 2.83e-3$ | $0.616 \pm 3.266e-3$ | $0.718 \pm 9.66e-3$ | $0.601 \pm 1.524e-2$ |
| gemini-2.0-flash zero_shot lstm | $0.715 \pm 9.405e-3$ | $0.597 \pm 1.684e-2$ | $0.678 \pm 7.654e-3$ | $0.613 \pm 8.143e-3$ | $0.72 \pm 7.176e-3$ | $0.603 \pm 1.028e-2$ |
| gemini-2.0-flash zero_shot mlp | $0.738 \pm 9.791e-3$ | $0.621 \pm 1.109e-2$ | $0.684 \pm 5.914e-3$ | $0.623 \pm 7.038e-3$ | $0.723 \pm 6.817e-3$ | $0.604 \pm 8.556e-3$ |
| gemini-2.0-flash zero_shot patchtsmixer | $0.749 \pm 2.068e-3$ | $0.63 \pm 2.351e-3$ | $0.687 \pm 2.964e-3$ | $0.624 \pm 2.399e-3$ | $0.725 \pm 1.792e-3$ | $0.606 \pm 2.215e-3$ |
| gemini-2.0-flash zero_shot timemixer | $0.733 \pm 1.782e-2$ | $0.612 \pm 2.063e-2$ | $0.673 \pm 7.553e-3$ | $0.608 \pm 7.72e-3$ | $0.721 \pm 6.148e-3$ | $0.601 \pm 7.37e-3$ |
| interp lstm | $0.664 \pm 4.33e-2$ | $0.528 \pm 5.538e-2$ | $0.586 \pm 3.228e-2$ | $0.52 \pm 3.973e-2$ | $0.658 \pm 3.933e-2$ | $0.526 \pm 5.13e-2$ |
| interp mlp | $0.785 \pm 3.045e-2$ | $0.684 \pm 6.478e-2$ | $0.748 \pm 2.634e-3$ | $0.695 \pm 3.103e-3$ | $0.821 \pm 3.047e-3$ | $0.729 \pm 3.135e-3$ |
| interp patchtsmixer | $0.868 \pm 4.851e-4$ | $0.786 \pm 1.611e-2$ | $0.782 \pm 1.957e-3$ | $0.73 \pm 2.016e-3$ | $0.863 \pm 2.761e-3$ | $0.78 \pm 2.455e-3$ |
| interp timemixer | $0.739 \pm 3.808e-3$ | $0.612 \pm 5.098e-3$ | $0.664 \pm 4.762e-4$ | $0.6 \pm 3.306e-3$ | $0.732 \pm 5.991e-3$ | $0.612 \pm 1.174e-2$ |
| mean lstm | $0.684 \pm 8.276e-2$ | $0.555 \pm 1.094e-1$ | $0.61 \pm 7.946e-2$ | $0.545 \pm 8.924e-2$ | $0.639 \pm 0e0$ | $0.5 \pm 0e0$ |
| mean mlp | $0.837 \pm 1.092e-2$ | $0.751 \pm 1.993e-3$ | $0.731 \pm 2.236e-3$ | $0.684 \pm 2.358e-3$ | $0.854 \pm 1.325e-3$ | $0.763 \pm 2.016e-3$ |
| mean patchtsmixer | $0.879 \pm 2.175e-3$ | $0.81 \pm 4.419e-3$ | $0.79 \pm 6.211e-3$ | $0.741 \pm 5.469e-3$ | $0.847 \pm 1.344e-3$ | $0.756 \pm 4.7e-3$ |
| mean timemixer | $0.872 \pm 1.497e-3$ | $0.793 \pm 1.085e-3$ | $0.808 \pm 1.141e-3$ | $0.761 \pm 2.156e-3$ | $0.836 \pm 1.65e-3$ | $0.743 \pm 2.735e-3$ |
| No_sum_CoT CoT +TFM lstm | $0.723 \pm 1.544e-3$ | $0.619 \pm 1.426e-3$ | $0.75 \pm 1.476e-2$ | $0.695 \pm 1.678e-2$ | $0.777 \pm 8.079e-3$ | $0.688 \pm 1.272e-2$ |
| No_sum_CoT CoT +TFM mlp | $0.765 \pm 1.529e-3$ | $0.654 \pm 2.325e-3$ | $0.78 \pm 8.862e-4$ | $0.729 \pm 2.401e-4$ | $0.8 \pm 1.956e-2$ | $0.703 \pm 2.468e-2$ |
| No_sum_CoT CoT +TFM patchtsmixer | $0.762 \pm 3.793e-3$ | $0.649 \pm 5.79e-3$ | $0.774 \pm 3.033e-2$ | $0.711 \pm 3.255e-2$ | $0.814 \pm 1.059e-2$ | $0.716 \pm 1.263e-2$ |
| No_sum_CoT CoT +TFM timemixer | $0.755 \pm 1.472e-2$ | $0.641 \pm 2.036e-2$ | $0.766 \pm 2.158e-2$ | $0.712 \pm 2.287e-2$ | $0.764 \pm 2.202e-2$ | $0.659 \pm 2.382e-2$ |
| No_sum_CoT lstm | $0.735 \pm 3.67e-3$ | $0.625 \pm 3.715e-3$ | $0.716 \pm 1.338e-2$ | $0.656 \pm 1.607e-2$ | $0.75 \pm 8.467e-3$ | $0.64 \pm 1.349e-2$ |
| No_sum_CoT mlp | $0.755 \pm 9.858e-3$ | $0.643 \pm 1.27e-2$ | $0.745 \pm 3.568e-3$ | $0.689 \pm 4.358e-3$ | $0.788 \pm 3.896e-3$ | $0.688 \pm 5.356e-3$ |
| No_sum_CoT patchtsmixer | $0.755 \pm 8.893e-3$ | $0.64 \pm 1.08e-2$ | $0.746 \pm 1.058e-2$ | $0.686 \pm 1.145e-2$ | $0.782 \pm 3.733e-3$ | $0.673 \pm 5.333e-3$ |
| No_sum_CoT timemixer | $0.751 \pm 2.922e-3$ | $0.633 \pm 3.603e-3$ | $0.727 \pm 1.61e-2$ | $0.666 \pm 1.753e-2$ | $0.779 \pm 1.016e-2$ | $0.672 \pm 1.441e-2$ |
| No_sum_ICD ICD +TFM lstm | $0.714 \pm 4.662e-3$ | $0.604 \pm 7.298e-3$ | $0.755 \pm 1.394e-2$ | $0.702 \pm 2.384e-2$ | $0.771 \pm 3.647e-3$ | $0.677 \pm 5.379e-3$ |
| No_sum_ICD ICD +TFM mlp | $0.771 \pm 6.736e-3$ | $0.661 \pm 8.792e-3$ | $0.787 \pm 7.887e-3$ | $0.737 \pm 9.344e-3$ | $0.797 \pm 1.163e-2$ | $0.698 \pm 1.555e-2$ |
| No_sum_ICD ICD +TFM patchtsmixer | $0.762 \pm 4.834e-3$ | $0.649 \pm 6.052e-3$ | $0.798 \pm 9.244e-3$ | $0.74 \pm 9.79e-3$ | $0.814 \pm 1.707e-2$ | $0.714 \pm 2.154e-2$ |
| No_sum_ICD ICD +TFM timemixer | $0.765 \pm 6.781e-3$ | $0.655 \pm 9.045e-3$ | $0.789 \pm 1.89e-3$ | $0.739 \pm 3.565e-3$ | $0.778 \pm 3.037e-3$ | $0.673 \pm 2.51e-3$ |
| No_sum_ICD lstm | $0.733 \pm 1.321e-2$ | $0.621 \pm 2.176e-3$ | $0.735 \pm 7.334e-3$ | $0.681 \pm 9.124e-3$ | $0.75 \pm 3.354e-3$ | $0.641 \pm 4.25e-3$ |
| No_sum_ICD mlp | $0.751 \pm 5.511e-3$ | $0.636 \pm 6.843e-3$ | $0.751 \pm 1.153e-2$ | $0.696 \pm 1.24e-2$ | $0.781 \pm 5.254e-3$ | $0.675 \pm 6.613e-3$ |
| No_sum_ICD patchtsmixer | $0.761 \pm 5.675e-3$ | $0.645 \pm 7.142e-3$ | $0.747 \pm 3.78e-3$ | $0.689 \pm 4.212e-3$ | $0.79 \pm 9.069e-3$ | $0.683 \pm 1.25e-2$ |
| No_sum_ICD timemixer | $0.751 \pm 1.169e-3$ | $0.633 \pm 1.206e-3$ | $0.742 \pm 1.722e-2$ | $0.685 \pm 1.817e-2$ | $0.775 \pm 3.997e-3$ | $0.666 \pm 5.715e-3$ |
| No_sum_Trend Trend +TFM lstm | $0.722 \pm 4.245e-3$ | $0.614 \pm 5.841e-3$ | $0.758 \pm 2.191e-2$ | $0.703 \pm 2.598e-2$ | $0.787 \pm 2.879e-2$ | $0.702 \pm 4.131e-2$ |
| No_sum_Trend Trend +TFM mlp | $0.756 \pm 1.644e-2$ | $0.643 \pm 2.253e-2$ | $0.784 \pm 1.677e-2$ | $0.732 \pm 1.867e-2$ | $0.79 \pm 1.126e-2$ | $0.69 \pm 1.437e-2$ |
| No_sum_Trend Trend +TFM patchtsmixer | $0.754 \pm 4.161e-2$ | $0.639 \pm 5.573e-3$ | $0.797 \pm 1.046e-2$ | $0.735 \pm 1.128e-2$ | $0.808 \pm 1.192e-2$ | $0.707 \pm 1.543e-2$ |
| No_sum_Trend Trend +TFM timemixer | $0.75 \pm 1.1e-2$ | $0.639 \pm 8.155e-3$ | $0.761 \pm 3.013e-3$ | $0.706 \pm 3.665e-3$ | $0.762 \pm 2.516e-2$ | $0.656 \pm 2.332e-2$ |

Table 22: In-distribution results - Gender. (part 1/2)

| Method | hirid → hirid | | mimic → mimic | | ppicu → ppicu | |
| --- | --- | --- | --- | --- | --- | --- |
| | auprc | auroc | auprc | auroc | auprc | auroc |
| No_sum_Trend lstm | $0.74 \pm 2.23e-3$ | $0.63 \pm 3.426e-3$ | $0.722 \pm 1.427e-2$ | $0.663 \pm 1.613e-2$ | $0.778 \pm 2.76e-3$ | $0.681 \pm 3.846e-3$ |
| No_sum_Trend mlp | $0.755 \pm 3.027e-3$ | $0.641 \pm 3.93e-3$ | $0.753 \pm 1.008e-2$ | $0.698 \pm 1.139e-2$ | $0.787 \pm 9.173e-3$ | $0.686 \pm 1.275e-2$ |
| No_sum_Trend patchtsmixer | $0.749 \pm 4.704e-3$ | $0.631 \pm 5.715e-3$ | $0.754 \pm 6.242e-3$ | $0.695 \pm 7.373e-3$ | $0.79 \pm 1.108e-2$ | $0.684 \pm 1.554e-2$ |
| No_sum_Trend timemixer | $0.751 \pm 2.802e-3$ | $0.632 \pm 3.398e-3$ | $0.736 \pm 1.755e-2$ | $0.675 \pm 1.889e-2$ | $0.78 \pm 2.308e-3$ | $0.674 \pm 2.988e-3$ |
| medgemma CoT +TFM lstm | $0.701 \pm 9.768e-3$ | $0.582 \pm 1.444e-2$ | $0.659 \pm 4.775e-3$ | $0.598 \pm 4.392e-3$ | $0.724 \pm 1.238e-2$ | $0.615 \pm 2.059e-2$ |
| medgemma CoT +TFM mlp | $0.73 \pm 1.005e-2$ | $0.614 \pm 1.175e-2$ | $0.716 \pm 1.392e-2$ | $0.657 \pm 1.404e-2$ | $0.762 \pm 1.148e-3$ | $0.657 \pm 2.568e-3$ |
| medgemma CoT +TFM patchtsmixer | $0.74 \pm 8.523e-3$ | $0.626 \pm 9.423e-3$ | $0.714 \pm 2.428e-3$ | $0.651 \pm 1.848e-3$ | $0.741 \pm 8.583e-3$ | $0.627 \pm 1.134e-2$ |
| medgemma CoT +TFM timemixer | $0.718 \pm 9.627e-3$ | $0.598 \pm 9.804e-3$ | $0.71 \pm 8.869e-3$ | $0.653 \pm 9.106e-3$ | $0.726 \pm 3.822e-3$ | $0.612 \pm 3.668e-3$ |
| medgemma CoT lstm | $0.716 \pm 5.031e-4$ | $0.603 \pm 1.864e-3$ | $0.694 \pm 8.079e-3$ | $0.632 \pm 8.779e-3$ | $0.734 \pm 2.271e-2$ | $0.628 \pm 3.664e-2$ |
| medgemma CoT mlp | $0.717 \pm 3.543e-3$ | $0.598 \pm 4.867e-3$ | $0.719 \pm 3.884e-3$ | $0.66 \pm 4.348e-3$ | $0.753 \pm 1.132e-2$ | $0.647 \pm 1.391e-2$ |
| medgemma CoT patchtsmixer | $0.721 \pm 2.206e-3$ | $0.602 \pm 2.562e-3$ | $0.715 \pm 9.317e-3$ | $0.652 \pm 9.335e-3$ | $0.73 \pm 2.966e-2$ | $0.613 \pm 3.604e-2$ |
| medgemma CoT timemixer | $0.716 \pm 3.997e-3$ | $0.595 \pm 4.219e-3$ | $0.705 \pm 5.001e-3$ | $0.642 \pm 5.563e-3$ | $0.711 \pm 2.667e-3$ | $0.59 \pm 3.162e-3$ |
| medgemma ICD +TFM lstm | $0.704 \pm 1.716e-2$ | $0.586 \pm 2.449e-2$ | $0.674 \pm 5.636e-3$ | $0.61 \pm 6.298e-3$ | $0.688 \pm 7.053e-3$ | $0.564 \pm 8.507e-3$ |
| medgemma ICD +TFM mlp | $0.733 \pm 1.209e-2$ | $0.615 \pm 1.689e-2$ | $0.693 \pm 1.062e-2$ | $0.633 \pm 1.192e-2$ | $0.724 \pm 1.222e-2$ | $0.606 \pm 1.465e-2$ |
| medgemma ICD +TFM patchtsmixer | $0.736 \pm 5.126e-3$ | $0.615 \pm 6.441e-3$ | $0.691 \pm 9.462e-3$ | $0.633 \pm 9.006e-3$ | $0.721 \pm 8.924e-3$ | $0.604 \pm 1.059e-2$ |
| medgemma ICD +TFM timemixer | $0.716 \pm 9.304e-3$ | $0.597 \pm 5.295e-3$ | $0.69 \pm 2.087e-3$ | $0.631 \pm 2.762e-3$ | $0.718 \pm 6.332e-3$ | $0.602 \pm 4.055e-3$ |
| medgemma ICD lstm | $0.711 \pm 4.333e-3$ | $0.588 \pm 6.317e-3$ | $0.673 \pm 1.307e-2$ | $0.61 \pm 1.482e-2$ | $0.704 \pm 6.042e-3$ | $0.584 \pm 7.67e-3$ |
| medgemma ICD mlp | $0.717 \pm 1.656e-3$ | $0.593 \pm 2.615e-3$ | $0.693 \pm 9.13e-3$ | $0.634 \pm 9.341e-3$ | $0.713 \pm 4.461e-3$ | $0.595 \pm 4.885e-3$ |
| medgemma ICD patchtsmixer | $0.719 \pm 1.953e-3$ | $0.595 \pm 2.597e-3$ | $0.687 \pm 8.396e-3$ | $0.625 \pm 8.509e-3$ | $0.71 \pm 7.246e-3$ | $0.589 \pm 8.342e-3$ |
| medgemma ICD timemixer | $0.717 \pm 3.023e-3$ | $0.59 \pm 3.521e-3$ | $0.676 \pm 3.96e-3$ | $0.614 \pm 3.539e-3$ | $0.705 \pm 4.061e-3$ | $0.582 \pm 4.864e-3$ |
| medgemma Trend +TFM lstm | $0.69 \pm 1.662e-3$ | $0.568 \pm 5.877e-4$ | $0.666 \pm 4.095e-2$ | $0.599 \pm 4.256e-2$ | $0.69 \pm 1.934e-2$ | $0.567 \pm 2.605e-2$ |
| medgemma Trend +TFM mlp | $0.723 \pm 9.772e-3$ | $0.602 \pm 1.185e-2$ | $0.711 \pm 1.28e-2$ | $0.651 \pm 1.555e-2$ | $0.719 \pm 3.211e-3$ | $0.596 \pm 4.299e-3$ |
| medgemma Trend +TFM patchtsmixer | $0.709 \pm 7.731e-4$ | $0.585 \pm 4.571e-4$ | $0.715 \pm 1.91e-3$ | $0.649 \pm 1.747e-3$ | $0.715 \pm 6.824e-4$ | $0.592 \pm 5.277e-4$ |
| medgemma Trend +TFM timemixer | $0.709 \pm 7.321e-4$ | $0.59 \pm 3.491e-3$ | $0.689 \pm 3.296e-3$ | $0.627 \pm 3.766e-3$ | $0.713 \pm 3.195e-4$ | $0.598 \pm 1.229e-3$ |
| medgemma Trend lstm | $0.7 \pm 6.957e-3$ | $0.576 \pm 7.229e-3$ | $0.686 \pm 7.68e-3$ | $0.619 \pm 8.363e-3$ | $0.701 \pm 9.49e-3$ | $0.574 \pm 1.131e-2$ |
| medgemma Trend mlp | $0.707 \pm 4.45e-3$ | $0.583 \pm 5.281e-3$ | $0.7 \pm 6.592e-3$ | $0.635 \pm 7.411e-3$ | $0.705 \pm 4.399e-3$ | $0.578 \pm 5.072e-3$ |
| medgemma Trend patchtsmixer | $0.705 \pm 2.01e-3$ | $0.581 \pm 2.071e-3$ | $0.706 \pm 9.576e-3$ | $0.635 \pm 9.128e-3$ | $0.704 \pm 1.071e-3$ | $0.576 \pm 1.102e-3$ |
| medgemma Trend timemixer | $0.707 \pm 1.524e-3$ | $0.581 \pm 1.171e-3$ | $0.688 \pm 1.065e-2$ | $0.622 \pm 1.365e-2$ | $0.702 \pm 9.072e-4$ | $0.573 \pm 1.085e-3$ |
| medgemma zero_shot +TFM lstm | $0.696 \pm 2.587e-2$ | $0.578 \pm 3.51e-2$ | $0.675 \pm 1.045e-2$ | $0.612 \pm 1.099e-2$ | $0.731 \pm 1.36e-2$ | $0.624 \pm 2.043e-2$ |
| medgemma zero_shot +TFM mlp | $0.737 \pm 1.751e-2$ | $0.62 \pm 2.358e-2$ | $0.716 \pm 1.13e-2$ | $0.658 \pm 1.286e-2$ | $0.752 \pm 7.858e-3$ | $0.642 \pm 1.103e-2$ |
| medgemma zero_shot +TFM patchtsmixer | $0.732 \pm 2.435e-3$ | $0.612 \pm 4.336e-3$ | $0.716 \pm 2.947e-2$ | $0.654 \pm 3.129e-2$ | $0.741 \pm 7.613e-3$ | $0.627 \pm 8.152e-3$ |
| medgemma zero_shot +TFM timemixer | $0.728 \pm 7.906e-3$ | $0.611 \pm 1.199e-2$ | $0.692 \pm 5.17e-3$ | $0.633 \pm 6.136e-3$ | $0.722 \pm 9.043e-3$ | $0.609 \pm 9.301e-3$ |
| medgemma zero_shot lstm | $0.717 \pm 4.841e-3$ | $0.603 \pm 9.247e-3$ | $0.67 \pm 9.442e-3$ | $0.606 \pm 1.063e-2$ | $0.729 \pm 1.197e-2$ | $0.621 \pm 2.029e-2$ |
| medgemma zero_shot mlp | $0.723 \pm 7.89e-3$ | $0.602 \pm 1.147e-2$ | $0.696 \pm 1.046e-2$ | $0.636 \pm 1.195e-2$ | $0.733 \pm 2.28e-3$ | $0.621 \pm 2.99e-3$ |
| medgemma zero_shot patchtsmixer | $0.722 \pm 7.805e-3$ | $0.602 \pm 1.014e-2$ | $0.709 \pm 8.92e-3$ | $0.647 \pm 9.963e-3$ | $0.735 \pm 9.767e-3$ | $0.621 \pm 1.114e-2$ |
| medgemma zero_shot timemixer | $0.723 \pm 2.405e-3$ | $0.599 \pm 3.898e-3$ | $0.687 \pm 1.206e-2$ | $0.622 \pm 1.365e-2$ | $0.736 \pm 3.054e-3$ | $0.621 \pm 3.758e-3$ |
| No_sum_zero_shot zero_shot +TFM lstm | $0.722 \pm 5.1e-3$ | $0.616 \pm 1.094e-2$ | $0.742 \pm 4.867e-3$ | $0.684 \pm 5.837e-3$ | $0.76 \pm 2.029e-2$ | $0.664 \pm 2.82e-2$ |
| No_sum_zero_shot zero_shot +TFM mlp | $0.751 \pm 9.506e-3$ | $0.634 \pm 1.456e-2$ | $0.752 \pm 9.467e-3$ | $0.696 \pm 1.189e-2$ | $0.806 \pm 6.152e-3$ | $0.71 \pm 8.249e-3$ |
| No_sum_zero_shot zero_shot +TFM patchtsmixer | $0.753 \pm 5.825e-3$ | $0.639 \pm 7.477e-3$ | $0.773 \pm 1.104e-2$ | $0.71 \pm 1.255e-2$ | $0.81 \pm 1.939e-2$ | $0.709 \pm 2.312e-2$ |
| No_sum_zero_shot zero_shot +TFM timemixer | $0.749 \pm 5.594e-3$ | $0.632 \pm 7.303e-3$ | $0.759 \pm 1.388e-2$ | $0.706 \pm 1.614e-2$ | $0.761 \pm 1.982e-2$ | $0.655 \pm 2.072e-2$ |
| No_sum_zero_shot lstm | $0.735 \pm 1.049e-3$ | $0.625 \pm 3.211e-4$ | $0.727 \pm 1.031e-2$ | $0.669 \pm 1.276e-2$ | $0.734 \pm 9.081e-3$ | $0.618 \pm 1.173e-2$ |
| No_sum_zero_shot mlp | $0.751 \pm 6.221e-3$ | $0.637 \pm 8.498e-3$ | $0.75 \pm 8.165e-3$ | $0.695 \pm 9.902e-3$ | $0.782 \pm 6.794e-3$ | $0.678 \pm 9.684e-3$ |
| No_sum_zero_shot patchtsmixer | $0.749 \pm 3.79e-3$ | $0.633 \pm 4.857e-3$ | $0.742 \pm 8.508e-3$ | $0.682 \pm 8.934e-3$ | $0.784 \pm 6.863e-3$ | $0.675 \pm 9.66e-3$ |
| No_sum_zero_shot timemixer | $0.749 \pm 1.463e-3$ | $0.631 \pm 1.507e-3$ | $0.727 \pm 7.659e-3$ | $0.666 \pm 8.418e-3$ | $0.742 \pm 2.761e-2$ | $0.626 \pm 3.416e-2$ |
| right lstm | $0.662 \pm 3.943e-2$ | $0.527 \pm 5.419e-2$ | $0.57 \pm 0e0$ | $0.5 \pm 0e0$ | $0.639 \pm 0e0$ | $0.5 \pm 5e-6$ |
| right mlp | $0.842 \pm 1.711e-2$ | $0.763 \pm 4.486e-3$ | $0.786 \pm 8.476e-4$ | $0.736 \pm 1.542e-3$ | $0.876 \pm 1.087e-3$ | $0.797 \pm 1.703e-3$ |
| right patchtsmixer | $0.875 \pm 8.592e-4$ | $0.802 \pm 1.787e-3$ | $0.809 \pm 2.09e-3$ | $0.764 \pm 2.146e-3$ | $0.876 \pm 2.212e-3$ | $0.795 \pm 3.708e-3$ |
| right timemixer | $0.755 \pm 8.163e-3$ | $0.633 \pm 1.476e-2$ | $0.727 \pm 2.988e-3$ | $0.681 \pm 1.945e-3$ | $0.768 \pm 3.868e-3$ | $0.66 \pm 4.337e-3$ |

Table 23: In-distribution results - Gender. (part 2/2)

| Method | hirid → ppicu auprc | auroc | mimic → ppicu auprc | auroc |
|---|---|---|---|---|
| Llama-3.1 CoT +TFM lstm | $0.664 \pm 6.426e-3$ | $0.528 \pm 5.865e-3$ | $0.673 \pm 2.82e-3$ | $0.539 \pm 4.609e-3$ |
| Llama-3.1 CoT +TFM mlp | $0.673 \pm 3.117e-3$ | $0.535 \pm 3.658e-3$ | $0.674 \pm 2.093e-3$ | $0.54 \pm 4.741e-3$ |
| Llama-3.1 CoT +TFM patchtsmixer | $0.68 \pm 1.007e-3$ | $0.548 \pm 1.27e-3$ | $0.678 \pm 4.151e-4$ | $0.546 \pm 1.174e-3$ |
| Llama-3.1 CoT +TFM timemixer | $0.677 \pm 3.143e-3$ | $0.544 \pm 6.236e-3$ | $0.675 \pm 1.276e-3$ | $0.542 \pm 1.627e-3$ |
| Llama-3.1 CoT lstm | $0.674 \pm 5.913e-4$ | $0.542 \pm 2.421e-3$ | $0.675 \pm 3.617e-4$ | $0.543 \pm 8.516e-4$ |
| Llama-3.1 CoT mlp | $0.676 \pm 8.942e-4$ | $0.543 \pm 1.65e-3$ | $0.677 \pm 1.266e-3$ | $0.543 \pm 2.263e-3$ |
| Llama-3.1 CoT patchtsmixer | $0.677 \pm 1.087e-3$ | $0.544 \pm 1.92e-3$ | $0.676 \pm 1.235e-3$ | $0.545 \pm 1.534e-3$ |
| Llama-3.1 CoT timemixer | $0.677 \pm 1.531e-3$ | $0.545 \pm 6.271e-4$ | $0.674 \pm 1.844e-3$ | $0.543 \pm 1.893e-3$ |
| Llama-3.1 ICD +TFM lstm | $0.657 \pm 2.57e-3$ | $0.52 \pm 1.982e-3$ | $0.664 \pm 2.207e-3$ | $0.53 \pm 3.553e-3$ |
| Llama-3.1 ICD +TFM mlp | $0.664 \pm 3.384e-3$ | $0.526 \pm 3.965e-3$ | $0.667 \pm 1.799e-3$ | $0.535 \pm 3.067e-3$ |
| Llama-3.1 ICD +TFM patchtsmixer | $0.672 \pm 6.484e-4$ | $0.535 \pm 1.28e-3$ | $0.671 \pm 7.286e-4$ | $0.54 \pm 1.842e-3$ |
| Llama-3.1 ICD +TFM timemixer | $0.669 \pm 3.524e-3$ | $0.533 \pm 3.805e-3$ | $0.67 \pm 7.071e-4$ | $0.538 \pm 3.141e-3$ |
| Llama-3.1 ICD lstm | $0.667 \pm 1.733e-3$ | $0.531 \pm 1.912e-3$ | $0.665 \pm 1.754e-3$ | $0.534 \pm 1.14e-3$ |
| Llama-3.1 ICD mlp | $0.667 \pm 1.79e-3$ | $0.532 \pm 2.489e-3$ | $0.668 \pm 1.333e-3$ | $0.536 \pm 1.124e-4$ |
| Llama-3.1 ICD patchtsmixer | $0.668 \pm 2.909e-4$ | $0.533 \pm 1.122e-3$ | $0.667 \pm 1.141e-3$ | $0.536 \pm 6.438e-4$ |
| Llama-3.1 ICD timemixer | $0.669 \pm 7.352e-4$ | $0.535 \pm 7.313e-4$ | $0.667 \pm 1.223e-3$ | $0.536 \pm 4.909e-4$ |
| Llama-3.1 Trend +TFM lstm | $0.662 \pm 5.545e-3$ | $0.527 \pm 4.806e-3$ | $0.663 \pm 3.815e-3$ | $0.528 \pm 3.898e-3$ |
| Llama-3.1 Trend +TFM mlp | $0.668 \pm 2.891e-3$ | $0.533 \pm 6.51e-3$ | $0.667 \pm 4.11e-3$ | $0.53 \pm 3.45e-3$ |
| Llama-3.1 Trend +TFM patchtsmixer | $0.677 \pm 1.747e-3$ | $0.541 \pm 3.033e-3$ | $0.672 \pm 1.755e-3$ | $0.537 \pm 2.944e-3$ |
| Llama-3.1 Trend +TFM timemixer | $0.674 \pm 1.827e-3$ | $0.542 \pm 3.232e-3$ | $0.672 \pm 1.816e-3$ | $0.539 \pm 1.319e-3$ |
| Llama-3.1 Trend lstm | $0.669 \pm 9.636e-4$ | $0.534 \pm 2.643e-3$ | $0.667 \pm 8.415e-4$ | $0.534 \pm 9.089e-4$ |
| Llama-3.1 Trend mlp | $0.67 \pm 2.436e-3$ | $0.537 \pm 4.101e-3$ | $0.667 \pm 3.737e-4$ | $0.534 \pm 8.304e-4$ |
| Llama-3.1 Trend patchtsmixer | $0.672 \pm 7.463e-4$ | $0.537 \pm 8.551e-4$ | $0.667 \pm 8.675e-4$ | $0.533 \pm 1.001e-3$ |
| Llama-3.1 Trend timemixer | $0.673 \pm 8.969e-4$ | $0.54 \pm 6.96e-4$ | $0.667 \pm 9.77e-4$ | $0.533 \pm 1.519e-3$ |
| Llama-3.1 zero_shot +TFM lstm | $0.656 \pm 2.526e-3$ | $0.518 \pm 3.192e-3$ | $0.671 \pm 3.138e-3$ | $0.539 \pm 1.484e-3$ |
| Llama-3.1 zero_shot +TFM mlp | $0.662 \pm 2.076e-3$ | $0.521 \pm 2.602e-3$ | $0.672 \pm 1.873e-3$ | $0.539 \pm 2.119e-3$ |
| Llama-3.1 zero_shot +TFM patchtsmixer | $0.664 \pm 1.153e-3$ | $0.527 \pm 1.649e-3$ | $0.676 \pm 1.949e-3$ | $0.543 \pm 2.982e-3$ |
| Llama-3.1 zero_shot +TFM timemixer | $0.667 \pm 3.793e-3$ | $0.534 \pm 4.487e-3$ | $0.673 \pm 9.395e-4$ | $0.542 \pm 2.297e-3$ |
| Llama-3.1 zero_shot lstm | $0.661 \pm 1.774e-3$ | $0.522 \pm 1.272e-3$ | $0.672 \pm 2.568e-3$ | $0.539 \pm 2.97e-3$ |
| Llama-3.1 zero_shot mlp | $0.662 \pm 2.248e-3$ | $0.523 \pm 2.711e-3$ | $0.675 \pm 1.282e-3$ | $0.545 \pm 2.477e-3$ |
| Llama-3.1 zero_shot patchtsmixer | $0.662 \pm 1.64e-4$ | $0.524 \pm 7.276e-4$ | $0.676 \pm 8.784e-4$ | $0.546 \pm 7.578e-4$ |
| Llama-3.1 zero_shot timemixer | $0.662 \pm 1.446e-3$ | $0.525 \pm 1.079e-3$ | $0.672 \pm 6.774e-4$ | $0.541 \pm 1.045e-3$ |
| TFM lstm | $0.647 \pm 1.332e-3$ | $0.514 \pm 4.081e-3$ | $0.647 \pm 3.477e-3$ | $0.513 \pm 4.214e-3$ |
| TFM mlp | $0.656 \pm 4.364e-3$ | $0.521 \pm 3.65e-3$ | $0.652 \pm 2.906e-3$ | $0.521 \pm 1.976e-3$ |
| TFM patchtsmixer | $0.661 \pm 1.895e-3$ | $0.527 \pm 1.407e-3$ | $0.651 \pm 1.066e-3$ | $0.522 \pm 1.018e-3$ |
| TFM timemixer | $0.664 \pm 2.007e-3$ | $0.529 \pm 2.858e-3$ | $0.658 \pm 1.02e-3$ | $0.526 \pm 1.029e-3$ |
| TSDE lstm | $0.638 \pm 9.617e-4$ | $0.496 \pm 4.173e-3$ | $0.64 \pm 6.798e-3$ | $0.502 \pm 9.967e-3$ |
| TSDE mlp | $0.653 \pm 2.68e-2$ | $0.516 \pm 3.931e-3$ | $0.691 \pm 1.726e-2$ | $0.571 \pm 2.475e-2$ |
| TSDE patchtsmixer | $0.65 \pm 1.516e-2$ | $0.521 \pm 2.29e-2$ | $0.695 \pm 1.768e-2$ | $0.574 \pm 1.8e-2$ |
| TSDE timemixer | $0.64 \pm 2.737e-2$ | $0.498 \pm 3.735e-3$ | $0.66 \pm 3.474e-2$ | $0.528 \pm 4.731e-3$ |
| gemini-2.0-flash CoT +TFM lstm | $0.667 \pm 3.387e-3$ | $0.534 \pm 6.672e-3$ | $0.684 \pm 2.266e-3$ | $0.553 \pm 2.386e-3$ |
| gemini-2.0-flash CoT +TFM mlp | $0.681 \pm 3.592e-3$ | $0.549 \pm 4.444e-3$ | $0.688 \pm 2.068e-3$ | $0.559 \pm 3.566e-3$ |
| gemini-2.0-flash CoT +TFM patchtsmixer | $0.687 \pm 1.273e-3$ | $0.558 \pm 1.278e-3$ | $0.695 \pm 2.92e-3$ | $0.567 \pm 2.923e-3$ |
| gemini-2.0-flash CoT +TFM timemixer | $0.681 \pm 2.181e-3$ | $0.555 \pm 1.612e-3$ | $0.694 \pm 1.233e-3$ | $0.565 \pm 1.432e-3$ |
| gemini-2.0-flash CoT lstm | $0.68 \pm 6.536e-3$ | $0.552 \pm 9.249e-3$ | $0.685 \pm 3.19e-3$ | $0.558 \pm 3.331e-3$ |
| gemini-2.0-flash CoT mlp | $0.683 \pm 4.006e-3$ | $0.556 \pm 4.798e-3$ | $0.688 \pm 1.267e-3$ | $0.561 \pm 1.209e-3$ |
| gemini-2.0-flash CoT patchtsmixer | $0.681 \pm 1.202e-3$ | $0.553 \pm 1.498e-3$ | $0.69 \pm 1.952e-3$ | $0.563 \pm 1.933e-3$ |
| gemini-2.0-flash CoT timemixer | $0.679 \pm 1.521e-3$ | $0.552 \pm 1.252e-3$ | $0.687 \pm 2.086e-3$ | $0.561 \pm 1.86e-3$ |
| gemini-2.0-flash ICD +TFM lstm | $0.671 \pm 6.3e-3$ | $0.54 \pm 1.002e-2$ | $0.688 \pm 1.499e-3$ | $0.56 \pm 3.216e-3$ |
| gemini-2.0-flash ICD +TFM mlp | $0.681 \pm 1.129e-3$ | $0.549 \pm 4.585e-3$ | $0.691 \pm 1.102e-3$ | $0.564 \pm 3.598e-3$ |
| gemini-2.0-flash ICD +TFM patchtsmixer | $0.688 \pm 2.907e-3$ | $0.56 \pm 1.108e-3$ | $0.692 \pm 2.075e-3$ | $0.569 \pm 2.007e-3$ |
| gemini-2.0-flash ICD +TFM timemixer | $0.686 \pm 2.3e-3$ | $0.561 \pm 1.33e-3$ | $0.69 \pm 3.373e-3$ | $0.565 \pm 2.886e-3$ |
| gemini-2.0-flash ICD lstm | $0.679 \pm 3.447e-3$ | $0.552 \pm 5.017e-3$ | $0.689 \pm 1.697e-3$ | $0.568 \pm 1.536e-3$ |
| gemini-2.0-flash ICD mlp | $0.686 \pm 1.743e-3$ | $0.559 \pm 1.456e-3$ | $0.688 \pm 6.494e-4$ | $0.567 \pm 7.23e-4$ |
| gemini-2.0-flash ICD patchtsmixer | $0.683 \pm 6.85e-4$ | $0.556 \pm 6.835e-4$ | $0.688 \pm 2.879e-3$ | $0.568 \pm 3.261e-3$ |
| gemini-2.0-flash ICD timemixer | $0.678 \pm 1.054e-3$ | $0.552 \pm 1.234e-3$ | $0.685 \pm 8.961e-4$ | $0.565 \pm 1.171e-3$ |
| gemini-2.0-flash Trend +TFM lstm | $0.664 \pm 3.821e-3$ | $0.529 \pm 4.386e-3$ | $0.677 \pm 7.19e-4$ | $0.543 \pm 5.579e-4$ |
| gemini-2.0-flash Trend +TFM mlp | $0.678 \pm 6.497e-3$ | $0.547 \pm 6.906e-3$ | $0.682 \pm 4.529e-3$ | $0.551 \pm 5.453e-3$ |
| gemini-2.0-flash Trend +TFM patchtsmixer | $0.681 \pm 2.555e-3$ | $0.552 \pm 3.869e-3$ | $0.685 \pm 2.19e-3$ | $0.554 \pm 3.018e-3$ |
| gemini-2.0-flash Trend +TFM timemixer | $0.682 \pm 6.809e-4$ | $0.558 \pm 2.641e-4$ | $0.685 \pm 5.95e-4$ | $0.554 \pm 5.81e-4$ |
| gemini-2.0-flash Trend lstm | $0.677 \pm 1.297e-3$ | $0.547 \pm 2.634e-3$ | $0.682 \pm 2.711e-3$ | $0.554 \pm 2.045e-3$ |
| gemini-2.0-flash Trend mlp | $0.681 \pm 1.482e-3$ | $0.551 \pm 2.168e-3$ | $0.681 \pm 1.688e-3$ | $0.553 \pm 1.432e-3$ |
| gemini-2.0-flash Trend patchtsmixer | $0.68 \pm 1.276e-3$ | $0.551 \pm 1.95e-3$ | $0.681 \pm 9.38e-4$ | $0.554 \pm 1.776e-3$ |
| gemini-2.0-flash Trend timemixer | $0.68 \pm 1.174e-3$ | $0.551 \pm 7.688e-4$ | $0.682 \pm 6.751e-4$ | $0.555 \pm 1.328e-3$ |
| gemini-2.0-flash zero_shot +TFM lstm | $0.673 \pm 3.99e-3$ | $0.541 \pm 5.391e-3$ | $0.684 \pm 2.167e-3$ | $0.556 \pm 1.379e-3$ |
| gemini-2.0-flash zero_shot +TFM mlp | $0.688 \pm 4.124e-3$ | $0.558 \pm 5.348e-3$ | $0.691 \pm 1.859e-3$ | $0.564 \pm 3.858e-3$ |
| gemini-2.0-flash zero_shot +TFM patchtsmixer | $0.691 \pm 1.151e-3$ | $0.559 \pm 6.519e-4$ | $0.693 \pm 2.076e-3$ | $0.566 \pm 2.471e-3$ |
| gemini-2.0-flash zero_shot +TFM timemixer | $0.689 \pm 2.983e-3$ | $0.56 \pm 1.665e-3$ | $0.692 \pm 1.139e-3$ | $0.566 \pm 1.297e-3$ |
| gemini-2.0-flash zero_shot lstm | $0.682 \pm 4.034e-3$ | $0.553 \pm 3.975e-3$ | $0.692 \pm 7.208e-4$ | $0.565 \pm 1.489e-3$ |
| gemini-2.0-flash zero_shot mlp | $0.686 \pm 2.353e-3$ | $0.558 \pm 2.876e-3$ | $0.691 \pm 1.573e-3$ | $0.565 \pm 1.388e-3$ |
| gemini-2.0-flash zero_shot patchtsmixer | $0.685 \pm 2.944e-3$ | $0.557 \pm 3.471e-3$ | $0.693 \pm 1.073e-3$ | $0.566 \pm 1.022e-3$ |
| gemini-2.0-flash zero_shot timemixer | $0.687 \pm 5.444e-3$ | $0.558 \pm 5.147e-3$ | $0.692 \pm 1.375e-3$ | $0.566 \pm 1.655e-3$ |
| interp lstm | $0.644 \pm 1.164e-2$ | $0.508 \pm 1.574e-2$ | $0.641 \pm 3.775e-3$ | $0.504 \pm 7.525e-3$ |
| interp mlp | $0.657 \pm 1.27e-3$ | $0.52 \pm 9.396e-4$ | $0.637 \pm 4.554e-4$ | $0.499 \pm 6.287e-4$ |
| interp patchtsmixer | $0.681 \pm 3.759e-3$ | $0.546 \pm 5.672e-3$ | $0.655 \pm 2.889e-3$ | $0.534 \pm 4.291e-3$ |
| interp timemixer | $0.659 \pm 4.302e-3$ | $0.524 \pm 7.454e-3$ | $0.662 \pm 7.809e-4$ | $0.526 \pm 1.762e-3$ |
| mean lstm | $0.638 \pm 1.495e-3$ | $0.499 \pm 2.835e-3$ | $0.643 \pm 9.4e-3$ | $0.507 \pm 1.477e-2$ |
| mean mlp | $0.66 \pm 3.843e-3$ | $0.528 \pm 3.351e-3$ | $0.667 \pm 5.15e-3$ | $0.549 \pm 7.58e-3$ |
| mean patchtsmixer | $0.694 \pm 6.732e-3$ | $0.567 \pm 3.063e-3$ | $0.663 \pm 1.742e-2$ | $0.542 \pm 3.163e-2$ |
| mean timemixer | $0.622 \pm 2.893e-3$ | $0.468 \pm 1.83e-3$ | $0.644 \pm 1.102e-2$ | $0.497 \pm 1.143e-2$ |
| No_sum_CoT CoT +TFM lstm | $0.689 \pm 1.166e-3$ | $0.56 \pm 2.305e-3$ | $0.711 \pm 1.29e-3$ | $0.583 \pm 2.859e-3$ |
| No_sum_CoT CoT +TFM mlp | $0.701 \pm 1.357e-3$ | $0.568 \pm 1.455e-3$ | $0.701 \pm 1.873e-3$ | $0.574 \pm 3.119e-3$ |
| No_sum_CoT CoT +TFM patchtsmixer | $0.701 \pm 9.133e-4$ | $0.568 \pm 8.514e-4$ | $0.711 \pm 5.239e-3$ | $0.586 \pm 6.898e-3$ |
| No_sum_CoT CoT +TFM timemixer | $0.702 \pm 1.437e-3$ | $0.572 \pm 3.253e-3$ | $0.709 \pm 3.145e-3$ | $0.582 \pm 2.542e-3$ |
| No_sum_CoT lstm | $0.691 \pm 1.583e-3$ | $0.565 \pm 1.564e-3$ | $0.708 \pm 2.701e-3$ | $0.586 \pm 3.573e-3$ |
| No_sum_CoT mlp | $0.694 \pm 3.843e-3$ | $0.572 \pm 6.092e-3$ | $0.697 \pm 1.963e-3$ | $0.576 \pm 1.669e-3$ |
| No_sum_CoT patchtsmixer | $0.695 \pm 2.291e-3$ | $0.571 \pm 3.742e-3$ | $0.705 \pm 1.04e-3$ | $0.582 \pm 2.895e-3$ |
| No_sum_CoT timemixer | $0.696 \pm 1.185e-3$ | $0.57 \pm 1.322e-3$ | $0.697 \pm 2.566e-3$ | $0.577 \pm 3.58e-3$ |
| No_sum_ICD ICD +TFM lstm | $0.687 \pm 7.433e-4$ | $0.559 \pm 9.199e-4$ | $0.711 \pm 2.782e-3$ | $0.587 \pm 3.324e-3$ |
| No_sum_ICD ICD +TFM mlp | $0.703 \pm 5.818e-3$ | $0.572 \pm 7.284e-3$ | $0.712 \pm 5.727e-3$ | $0.586 \pm 6.684e-3$ |
| No_sum_ICD ICD +TFM patchtsmixer | $0.703 \pm 1.164e-3$ | $0.573 \pm 2.192e-3$ | $0.714 \pm 2.505e-3$ | $0.589 \pm 1.042e-3$ |
| No_sum_ICD ICD +TFM timemixer | $0.702 \pm 1.975e-3$ | $0.57 \pm 4.537e-4$ | $0.709 \pm 7.882e-3$ | $0.583 \pm 8.063e-3$ |
| No_sum_ICD lstm | $0.695 \pm 1.836e-3$ | $0.573 \pm 2.081e-3$ | $0.707 \pm 1.467e-3$ | $0.584 \pm 1.121e-3$ |
| No_sum_ICD mlp | $0.697 \pm 2.346e-3$ | $0.576 \pm 3.113e-3$ | $0.706 \pm 6.114e-3$ | $0.585 \pm 6.285e-3$ |
| No_sum_ICD patchtsmixer | $0.699 \pm 2.085e-3$ | $0.578 \pm 1.83e-3$ | $0.711 \pm 1.838e-3$ | $0.589 \pm 1.995e-3$ |
| No_sum_ICD timemixer | $0.703 \pm 1.428e-3$ | $0.58 \pm 1.191e-3$ | $0.705 \pm 2.969e-3$ | $0.584 \pm 2.066e-3$ |
| No_sum_Trend Trend +TFM lstm | $0.687 \pm 2.651e-3$ | $0.556 \pm 3.955e-3$ | $0.715 \pm 3.465e-3$ | $0.589 \pm 4.91e-3$ |
| No_sum_Trend Trend +TFM mlp | $0.698 \pm 5.702e-3$ | $0.568 \pm 5.788e-3$ | $0.699 \pm 8.263e-3$ | $0.572 \pm 1.13e-2$ |
| No_sum_Trend Trend +TFM patchtsmixer | $0.7 \pm 1.456e-3$ | $0.568 \pm 3.095e-3$ | $0.708 \pm 4.263e-3$ | $0.586 \pm 5.725e-3$ |
| No_sum_Trend Trend +TFM timemixer | $0.701 \pm 1.005e-3$ | $0.569 \pm 3.575e-3$ | $0.714 \pm 1.818e-3$ | $0.588 \pm 1.287e-3$ |

Table 24: Cross-site transfer results - Gender. (part 1/2)

| Method | hirid → ppicu | | mimic → ppicu | |
|---|---|---|---|---|
| | auprc | auroc | auprc | auroc |
| No_sum_Trend lstm | $0.686 \pm 2.715e-3$ | $0.561 \pm 3.173e-3$ | $0.712 \pm 2.536e-3$ | $0.588 \pm 2.901e-3$ |
| No_sum_Trend mlp | $0.687 \pm 1.638e-3$ | $0.563 \pm 1.828e-3$ | $0.707 \pm 3.987e-3$ | $0.588 \pm 3.721e-3$ |
| No_sum_Trend patchtsmixer | $0.691 \pm 1.504e-3$ | $0.566 \pm 1.569e-3$ | $0.702 \pm 2.505e-3$ | $0.584 \pm 2.975e-3$ |
| No_sum_Trend timemixer | $0.694 \pm 1.373e-3$ | $0.568 \pm 1.335e-3$ | $0.702 \pm 1.888e-3$ | $0.582 \pm 3.587e-3$ |
| medgemma CoT +TFM lstm | $0.674 \pm 3.333e-3$ | $0.542 \pm 4.159e-3$ | $0.68 \pm 9.923e-4$ | $0.55 \pm 1.624e-3$ |
| medgemma CoT +TFM mlp | $0.687 \pm 2.608e-4$ | $0.553 \pm 5.26e-4$ | $0.686 \pm 1.44e-3$ | $0.558 \pm 2.216e-3$ |
| medgemma CoT +TFM patchtsmixer | $0.69 \pm 8.329e-4$ | $0.556 \pm 8.536e-4$ | $0.687 \pm 1.499e-3$ | $0.56 \pm 1.612e-3$ |
| medgemma CoT +TFM timemixer | $0.685 \pm 6.618e-3$ | $0.555 \pm 5.886e-3$ | $0.685 \pm 8.743e-4$ | $0.559 \pm 1.573e-3$ |
| medgemma CoT lstm | $0.683 \pm 6.904e-4$ | $0.551 \pm 1.304e-3$ | $0.683 \pm 1.401e-3$ | $0.558 \pm 1.623e-3$ |
| medgemma CoT mlp | $0.685 \pm 1.403e-3$ | $0.555 \pm 2.095e-3$ | $0.685 \pm 9.946e-5$ | $0.561 \pm 6.614e-4$ |
| medgemma CoT patchtsmixer | $0.687 \pm 7.489e-4$ | $0.558 \pm 8.431e-4$ | $0.686 \pm 6.106e-4$ | $0.561 \pm 3.438e-4$ |
| medgemma CoT timemixer | $0.686 \pm 1.968e-3$ | $0.556 \pm 2.005e-3$ | $0.681 \pm 1.419e-3$ | $0.556 \pm 1.152e-3$ |
| medgemma ICD +TFM lstm | $0.67 \pm 2.968e-3$ | $0.539 \pm 6.595e-3$ | $0.675 \pm 1.207e-3$ | $0.543 \pm 1.559e-3$ |
| medgemma ICD +TFM mlp | $0.677 \pm 1.649e-3$ | $0.548 \pm 2.954e-3$ | $0.676 \pm 3.333e-3$ | $0.547 \pm 3.011e-3$ |
| medgemma ICD +TFM patchtsmixer | $0.682 \pm 9.678e-4$ | $0.551 \pm 1.749e-3$ | $0.679 \pm 7.088e-4$ | $0.55 \pm 1.454e-3$ |
| medgemma ICD +TFM timemixer | $0.681 \pm 7.295e-4$ | $0.552 \pm 1.295e-3$ | $0.68 \pm 1.527e-3$ | $0.55 \pm 9.28e-4$ |
| medgemma ICD lstm | $0.673 \pm 2.193e-3$ | $0.546 \pm 1.734e-3$ | $0.676 \pm 1.544e-3$ | $0.55 \pm 9.854e-4$ |
| medgemma ICD mlp | $0.676 \pm 9.677e-4$ | $0.551 \pm 6.531e-4$ | $0.673 \pm 1.987e-3$ | $0.546 \pm 2.311e-3$ |
| medgemma ICD patchtsmixer | $0.679 \pm 2.547e-4$ | $0.554 \pm 2.541e-4$ | $0.674 \pm 1.957e-3$ | $0.548 \pm 2.653e-3$ |
| medgemma ICD timemixer | $0.68 \pm 5.372e-4$ | $0.554 \pm 1.056e-3$ | $0.675 \pm 6.08e-4$ | $0.549 \pm 6.74e-4$ |
| medgemma Trend +TFM lstm | $0.67 \pm 5.701e-4$ | $0.538 \pm 3.443e-4$ | $0.676 \pm 3.496e-3$ | $0.544 \pm 4.863e-3$ |
| medgemma Trend +TFM mlp | $0.682 \pm 3.816e-3$ | $0.546 \pm 4.199e-3$ | $0.683 \pm 3.324e-3$ | $0.554 \pm 5.511e-3$ |
| medgemma Trend +TFM patchtsmixer | $0.685 \pm 1.427e-3$ | $0.552 \pm 1.289e-3$ | $0.683 \pm 2.212e-3$ | $0.554 \pm 1.831e-3$ |
| medgemma Trend +TFM timemixer | $0.683 \pm 3.498e-3$ | $0.553 \pm 4.794e-3$ | $0.684 \pm 1.091e-3$ | $0.555 \pm 1.287e-3$ |
| medgemma Trend lstm | $0.68 \pm 3.709e-3$ | $0.547 \pm 4.37e-3$ | $0.683 \pm 2.223e-3$ | $0.557 \pm 1.936e-3$ |
| medgemma Trend mlp | $0.683 \pm 2.096e-3$ | $0.551 \pm 2.115e-3$ | $0.685 \pm 8.504e-4$ | $0.56 \pm 1.115e-3$ |
| medgemma Trend patchtsmixer | $0.683 \pm 1.228e-3$ | $0.551 \pm 8.638e-4$ | $0.685 \pm 9.167e-4$ | $0.56 \pm 1.063e-3$ |
| medgemma Trend timemixer | $0.684 \pm 1.472e-3$ | $0.553 \pm 6.251e-4$ | $0.68 \pm 1.174e-3$ | $0.557 \pm 1.081e-3$ |
| medgemma zero_shot +TFM lstm | $0.671 \pm 6.166e-3$ | $0.538 \pm 5.835e-3$ | $0.681 \pm 2.502e-3$ | $0.552 \pm 2.85e-3$ |
| medgemma zero_shot +TFM mlp | $0.679 \pm 6.633e-3$ | $0.548 \pm 9.38e-3$ | $0.683 \pm 3.667e-3$ | $0.555 \pm 3.361e-3$ |
| medgemma zero_shot +TFM patchtsmixer | $0.684 \pm 2.966e-3$ | $0.553 \pm 4.421e-3$ | $0.688 \pm 4.088e-3$ | $0.56 \pm 5.639e-3$ |
| medgemma zero_shot +TFM timemixer | $0.681 \pm 1.818e-3$ | $0.55 \pm 2.888e-3$ | $0.684 \pm 7.047e-4$ | $0.558 \pm 9.539e-4$ |
| medgemma zero_shot lstm | $0.68 \pm 1.026e-3$ | $0.553 \pm 1.347e-3$ | $0.682 \pm 8.385e-4$ | $0.558 \pm 1.123e-3$ |
| medgemma zero_shot mlp | $0.68 \pm 1.632e-3$ | $0.555 \pm 2.142e-3$ | $0.685 \pm 1.068e-3$ | $0.56 \pm 2.674e-3$ |
| medgemma zero_shot patchtsmixer | $0.681 \pm 1.601e-3$ | $0.557 \pm 1.468e-3$ | $0.683 \pm 1.665e-3$ | $0.557 \pm 2.348e-3$ |
| medgemma zero_shot timemixer | $0.682 \pm 7.89e-4$ | $0.556 \pm 5.766e-4$ | $0.681 \pm 1.748e-3$ | $0.556 \pm 3.155e-3$ |
| No_sum_zero_shot zero_shot +TFM lstm | $0.687 \pm 2.507e-3$ | $0.555 \pm 2.461e-3$ | $0.709 \pm 1.499e-3$ | $0.582 \pm 2.53e-3$ |
| No_sum_zero_shot zero_shot +TFM mlp | $0.699 \pm 1.001e-3$ | $0.567 \pm 4.949e-4$ | $0.701 \pm 6.731e-3$ | $0.573 \pm 8.181e-3$ |
| No_sum_zero_shot zero_shot +TFM patchtsmixer | $0.7 \pm 5.422e-3$ | $0.567 \pm 6.611e-3$ | $0.709 \pm 7.127e-4$ | $0.58 \pm 1.602e-3$ |
| No_sum_zero_shot zero_shot +TFM timemixer | $0.698 \pm 1.78e-3$ | $0.57 \pm 2.616e-3$ | $0.709 \pm 8.243e-3$ | $0.584 \pm 6.554e-3$ |
| No_sum_zero_shot lstm | $0.689 \pm 8.802e-4$ | $0.562 \pm 8.527e-4$ | $0.699 \pm 2.181e-3$ | $0.573 \pm 3.481e-3$ |
| No_sum_zero_shot mlp | $0.688 \pm 1.266e-3$ | $0.564 \pm 2.873e-3$ | $0.701 \pm 1.758e-3$ | $0.579 \pm 2.061e-3$ |
| No_sum_zero_shot patchtsmixer | $0.689 \pm 1.275e-3$ | $0.564 \pm 1.727e-3$ | $0.698 \pm 1.747e-3$ | $0.575 \pm 1.635e-3$ |
| No_sum_zero_shot timemixer | $0.688 \pm 1.633e-3$ | $0.563 \pm 1.083e-3$ | $0.691 \pm 2.098e-3$ | $0.568 \pm 2.346e-3$ |
| right lstm | $0.642 \pm 6.967e-3$ | $0.506 \pm 1.217e-2$ | $0.639 \pm 0e0$ | $0.5 \pm 0e0$ |
| right mlp | $0.65 \pm 1.583e-3$ | $0.511 \pm 2.671e-3$ | $0.64 \pm 1.61e-3$ | $0.504 \pm 2.429e-3$ |
| right patchtsmixer | $0.682 \pm 2.058e-3$ | $0.55 \pm 1.6e-3$ | $0.649 \pm 5.373e-3$ | $0.519 \pm 2.978e-3$ |
| right timemixer | $0.662 \pm 6.162e-3$ | $0.528 \pm 6.041e-3$ | $0.67 \pm 2.405e-3$ | $0.534 \pm 7.208e-3$ |

Table 25: Cross-site transfer results - Gender. (part 2/2)

| Method | hirid → hirid | | mimic → mimic | | ppicu → ppicu | |
| --- | --- | --- | --- | --- | --- | --- |
| | f1_micro | recall_micro | f1_micro | recall_micro | f1_micro | recall_micro |
| Llama-3.1 CoT +TFM lstm | $0.759 \pm 1.504e-3$ | $0.848 \pm 5.911e-3$ | $0.76 \pm 4.839e-3$ | $0.836 \pm 2.494e-3$ | $0.808 \pm 2.843e-3$ | $0.917 \pm 5.773e-3$ |
| Llama-3.1 CoT +TFM mlp | $0.771 \pm 2.913e-3$ | $0.847 \pm 3.027e-3$ | $0.77 \pm 7.323e-3$ | $0.831 \pm 4.135e-3$ | $0.814 \pm 3.384e-3$ | $0.915 \pm 2.589e-3$ |
| Llama-3.1 CoT +TFM patchtsmixer | $0.736 \pm 1.973e-2$ | $0.859 \pm 1.498e-2$ | $0.753 \pm 7.457e-3$ | $0.82 \pm 6.964e-3$ | $0.798 \pm 8.902e-3$ | $0.921 \pm 4.925e-3$ |
| Llama-3.1 CoT +TFM timemixer | $0.747 \pm 6.439e-3$ | $0.854 \pm 1.833e-3$ | $0.737 \pm 1.381e-2$ | $0.811 \pm 1.056e-2$ | $0.805 \pm 8.986e-3$ | $0.925 \pm 2.113e-3$ |
| Llama-3.1 CoT lstm | $0.733 \pm 9.055e-3$ | $0.855 \pm 3.436e-3$ | $0.692 \pm 6.113e-3$ | $0.815 \pm 4.863e-3$ | $0.793 \pm 3.488e-3$ | $0.935 \pm 4.021e-3$ |
| Llama-3.1 CoT mlp | $0.738 \pm 8.59e-3$ | $0.854 \pm 2.156e-3$ | $0.69 \pm 4.948e-3$ | $0.811 \pm 4.759e-3$ | $0.793 \pm 2.72e-3$ | $0.935 \pm 7.473e-3$ |
| Llama-3.1 CoT patchtsmixer | $0.725 \pm 6.203e-3$ | $0.855 \pm 1.603e-2$ | $0.683 \pm 3.666e-3$ | $0.813 \pm 2.29e-3$ | $0.787 \pm 2.354e-3$ | $0.937 \pm 5.183e-3$ |
| Llama-3.1 CoT timemixer | $0.728 \pm 4.499e-3$ | $0.855 \pm 1.809e-2$ | $0.684 \pm 3.584e-3$ | $0.813 \pm 3.468e-3$ | $0.79 \pm 2.229e-3$ | $0.939 \pm 2.984e-3$ |
| Llama-3.1 ICD +TFM lstm | $0.782 \pm 2.866e-3$ | $0.86 \pm 4.17e-3$ | $0.784 \pm 6.085e-3$ | $0.858 \pm 7.348e-3$ | $0.82 \pm 3.275e-3$ | $0.919 \pm 2.594e-3$ |
| Llama-3.1 ICD +TFM mlp | $0.791 \pm 6.305e-3$ | $0.86 \pm 3.877e-3$ | $0.786 \pm 7.182e-3$ | $0.845 \pm 6.57e-3$ | $0.826 \pm 5.987e-3$ | $0.91 \pm 3.833e-3$ |
| Llama-3.1 ICD +TFM patchtsmixer | $0.77 \pm 1.35e-2$ | $0.854 \pm 1.041e-2$ | $0.767 \pm 4.433e-3$ | $0.84 \pm 4.443e-3$ | $0.805 \pm 9.498e-3$ | $0.917 \pm 5.297e-3$ |
| Llama-3.1 ICD +TFM timemixer | $0.759 \pm 9.084e-3$ | $0.856 \pm 3.575e-3$ | $0.768 \pm 5.454e-3$ | $0.846 \pm 6.041e-3$ | $0.816 \pm 2.132e-3$ | $0.922 \pm 4.818e-3$ |
| Llama-3.1 ICD lstm | $0.752 \pm 8.791e-3$ | $0.857 \pm 5.166e-3$ | $0.728 \pm 4.776e-3$ | $0.834 \pm 6.282e-3$ | $0.804 \pm 2.747e-3$ | $0.932 \pm 7.23e-3$ |
| Llama-3.1 ICD mlp | $0.756 \pm 7.923e-3$ | $0.86 \pm 2.418e-3$ | $0.733 \pm 5.609e-3$ | $0.827 \pm 2.671e-3$ | $0.803 \pm 4.286e-3$ | $0.93 \pm 6.317e-3$ |
| Llama-3.1 ICD patchtsmixer | $0.744 \pm 9.307e-3$ | $0.853 \pm 4.546e-3$ | $0.719 \pm 8.979e-3$ | $0.821 \pm 2.673e-3$ | $0.795 \pm 2.433e-3$ | $0.934 \pm 6.717e-3$ |
| Llama-3.1 ICD timemixer | $0.748 \pm 6.901e-3$ | $0.864 \pm 3.019e-3$ | $0.723 \pm 7.084e-3$ | $0.826 \pm 1.007e-3$ | $0.803 \pm 3.418e-3$ | $0.935 \pm 8.746e-3$ |
| Llama-3.1 Trend +TFM lstm | $0.809 \pm 5.031e-3$ | $0.861 \pm 1.306e-3$ | $0.792 \pm 5.698e-3$ | $0.85 \pm 4.228e-3$ | $0.853 \pm 3.27e-3$ | $0.905 \pm 1.584e-3$ |
| Llama-3.1 Trend +TFM mlp | $0.817 \pm 2.262e-3$ | $0.858 \pm 5.886e-3$ | $0.797 \pm 8.779e-3$ | $0.833 \pm 4.905e-3$ | $0.856 \pm 6.346e-3$ | $0.906 \pm 5.637e-3$ |
| Llama-3.1 Trend +TFM patchtsmixer | $0.789 \pm 1.251e-2$ | $0.847 \pm 1.06e-2$ | $0.781 \pm 6.412e-3$ | $0.828 \pm 5.052e-3$ | $0.831 \pm 1.194e-2$ | $0.896 \pm 6.258e-3$ |
| Llama-3.1 Trend +TFM timemixer | $0.788 \pm 8.871e-3$ | $0.852 \pm 6.465e-4$ | $0.776 \pm 9.525e-4$ | $0.832 \pm 8.903e-4$ | $0.827 \pm 1.403e-2$ | $0.898 \pm 5.407e-3$ |
| Llama-3.1 Trend lstm | $0.777 \pm 1.378e-2$ | $0.837 \pm 1.26e-2$ | $0.744 \pm 5.898e-3$ | $0.816 \pm 4.685e-3$ | $0.835 \pm 6.602e-3$ | $0.901 \pm 4.228e-3$ |
| Llama-3.1 Trend mlp | $0.781 \pm 1.431e-2$ | $0.847 \pm 6.816e-3$ | $0.747 \pm 7.288e-3$ | $0.815 \pm 1.982e-3$ | $0.836 \pm 7.237e-3$ | $0.903 \pm 2.543e-3$ |
| Llama-3.1 Trend patchtsmixer | $0.766 \pm 1.143e-2$ | $0.845 \pm 3.502e-3$ | $0.733 \pm 6.448e-3$ | $0.807 \pm 2.011e-3$ | $0.824 \pm 5.114e-3$ | $0.899 \pm 3.857e-3$ |
| Llama-3.1 Trend timemixer | $0.771 \pm 1.132e-2$ | $0.844 \pm 3.25e-3$ | $0.738 \pm 8.357e-3$ | $0.814 \pm 2.792e-3$ | $0.828 \pm 5.605e-3$ | $0.898 \pm 4.965e-3$ |
| Llama-3.1 zero_shot +TFM lstm | $0.809 \pm 2.576e-3$ | $0.872 \pm 7.616e-3$ | $0.778 \pm 3.649e-3$ | $0.856 \pm 4.224e-3$ | $0.822 \pm 5.651e-3$ | $0.917 \pm 1.668e-3$ |
| Llama-3.1 zero_shot +TFM mlp | $0.819 \pm 2.948e-3$ | $0.872 \pm 5.474e-3$ | $0.782 \pm 5.126e-3$ | $0.844 \pm 3.205e-3$ | $0.824 \pm 3.438e-3$ | $0.914 \pm 1.721e-3$ |
| Llama-3.1 zero_shot +TFM patchtsmixer | $0.791 \pm 5.223e-3$ | $0.865 \pm 3.37e-3$ | $0.76 \pm 5.657e-3$ | $0.836 \pm 3.153e-3$ | $0.804 \pm 3.756e-3$ | $0.915 \pm 4.702e-3$ |
| Llama-3.1 zero_shot +TFM timemixer | $0.793 \pm 8.257e-3$ | $0.865 \pm 4.343e-3$ | $0.77 \pm 5.763e-3$ | $0.845 \pm 9.868e-4$ | $0.815 \pm 3.645e-3$ | $0.923 \pm 4.415e-3$ |
| Llama-3.1 zero_shot lstm | $0.778 \pm 9.139e-3$ | $0.858 \pm 7.707e-3$ | $0.72 \pm 9.989e-3$ | $0.825 \pm 5.409e-3$ | $0.805 \pm 1.598e-3$ | $0.931 \pm 2.449e-3$ |
| Llama-3.1 zero_shot mlp | $0.785 \pm 1.097e-2$ | $0.862 \pm 7.484e-3$ | $0.72 \pm 8.082e-3$ | $0.82 \pm 4.275e-3$ | $0.804 \pm 3.139e-3$ | $0.93 \pm 4.183e-3$ |
| Llama-3.1 zero_shot patchtsmixer | $0.772 \pm 1.008e-2$ | $0.861 \pm 5.025e-3$ | $0.71 \pm 6.547e-3$ | $0.824 \pm 1.434e-3$ | $0.799 \pm 1.577e-3$ | $0.934 \pm 8.589e-4$ |
| Llama-3.1 zero_shot timemixer | $0.773 \pm 1.038e-2$ | $0.853 \pm 1.372e-2$ | $0.708 \pm 5.895e-3$ | $0.823 \pm 5.838e-3$ | $0.8 \pm 4.755e-4$ | $0.934 \pm 1.143e-3$ |
| TFM lstm | $0.741 \pm 9.034e-4$ | $0.848 \pm 4.087e-3$ | $0.722 \pm 1.278e-3$ | $0.817 \pm 5.07e-3$ | $0.801 \pm 1.61e-3$ | $0.924 \pm 2.687e-3$ |
| TFM mlp | $0.742 \pm 4.86e-4$ | $0.85 \pm 7.832e-3$ | $0.726 \pm 1.487e-3$ | $0.813 \pm 3.211e-3$ | $0.801 \pm 1.695e-3$ | $0.922 \pm 2.964e-3$ |
| TFM patchtsmixer | $0.721 \pm 7.536e-3$ | $0.85 \pm 1.464e-2$ | $0.713 \pm 1.868e-3$ | $0.806 \pm 2.089e-3$ | $0.794 \pm 1.514e-3$ | $0.923 \pm 4.105e-3$ |
| TFM timemixer | $0.728 \pm 3.021e-3$ | $0.844 \pm 3.174e-3$ | $0.719 \pm 1.011e-3$ | $0.805 \pm 8.749e-3$ | $0.797 \pm 1.061e-3$ | $0.927 \pm 6.955e-3$ |
| TSDE mlp | $0.882 \pm 5.15e-3$ | $0.9 \pm 6.275e-3$ | $0.858 \pm 3.274e-3$ | $0.889 \pm 2.551e-3$ | $0.914 \pm 3.144e-3$ | $0.897 \pm 4.171e-3$ |
| TSDE lstm | $0.885 \pm 4.684e-3$ | $0.902 \pm 5.807e-3$ | $0.862 \pm 2.948e-3$ | $0.89 \pm 2.344e-3$ | $0.915 \pm 2.896e-3$ | $0.899 \pm 3.746e-3$ |
| TSDE patchtsmixer | $0.882 \pm 2.559e-3$ | $0.901 \pm 3.322e-3$ | $0.864 \pm 1.931e-3$ | $0.888 \pm 1.861e-3$ | $0.917 \pm 5.714e-4$ | $0.899 \pm 1.961e-3$ |
| TSDE timemixer | $0.887 \pm 4.508e-3$ | $0.904 \pm 5.48e-3$ | $0.866 \pm 2.983e-3$ | $0.892 \pm 2.44e-3$ | $0.914 \pm 4.716e-3$ | $0.902 \pm 3.798e-3$ |
| gemini-2.0-flash CoT +TFM lstm | $0.823 \pm 7.215e-3$ | $0.879 \pm 2.338e-3$ | $0.803 \pm 5.517e-3$ | $0.866 \pm 3.885e-3$ | $0.858 \pm 3.412e-3$ | $0.911 \pm 2.789e-3$ |
| gemini-2.0-flash CoT +TFM mlp | $0.826 \pm 6.77e-3$ | $0.874 \pm 2.381e-3$ | $0.801 \pm 5.121e-3$ | $0.854 \pm 2.831e-3$ | $0.863 \pm 8.082e-3$ | $0.907 \pm 1.713e-3$ |
| gemini-2.0-flash CoT +TFM patchtsmixer | $0.799 \pm 6.955e-3$ | $0.872 \pm 2.665e-4$ | $0.774 \pm 3.662e-3$ | $0.837 \pm 2.663e-2$ | $0.846 \pm 9.188e-4$ | $0.9 \pm 3.033e-3$ |
| gemini-2.0-flash CoT +TFM timemixer | $0.816 \pm 5.945e-3$ | $0.875 \pm 8.525e-3$ | $0.79 \pm 3.963e-3$ | $0.855 \pm 1.119e-3$ | $0.839 \pm 2.197e-3$ | $0.897 \pm 3.462e-3$ |
| gemini-2.0-flash CoT lstm | $0.8 \pm 8.11e-3$ | $0.869 \pm 7.762e-3$ | $0.759 \pm 8.907e-3$ | $0.823 \pm 6.927e-3$ | $0.842 \pm 7.014e-3$ | $0.909 \pm 5.895e-3$ |
| gemini-2.0-flash CoT mlp | $0.805 \pm 9.615e-3$ | $0.868 \pm 4.96e-3$ | $0.762 \pm 6.676e-3$ | $0.827 \pm 9.066e-3$ | $0.845 \pm 7.499e-3$ | $0.909 \pm 4.946e-3$ |
| gemini-2.0-flash CoT patchtsmixer | $0.794 \pm 8.547e-3$ | $0.867 \pm 1.346e-3$ | $0.753 \pm 8.305e-3$ | $0.82 \pm 9.924e-3$ | $0.833 \pm 5.914e-3$ | $0.907 \pm 2.657e-3$ |
| gemini-2.0-flash CoT timemixer | $0.794 \pm 8.417e-3$ | $0.868 \pm 2.328e-3$ | $0.752 \pm 6.761e-3$ | $0.821 \pm 7.768e-3$ | $0.839 \pm 5.666e-3$ | $0.914 \pm 4.218e-3$ |
| gemini-2.0-flash ICD +TFM lstm | $0.819 \pm 5.572e-3$ | $0.885 \pm 5.662e-3$ | $0.783 \pm 6.991e-3$ | $0.85 \pm 5.63e-3$ | $0.825 \pm 3.245e-3$ | $0.905 \pm 2.993e-3$ |
| gemini-2.0-flash ICD +TFM mlp | $0.821 \pm 3.211e-3$ | $0.877 \pm 1.819e-3$ | $0.782 \pm 6.43e-3$ | $0.838 \pm 5.876e-3$ | $0.83 \pm 5.002e-4$ | $0.905 \pm 5.133e-3$ |
| gemini-2.0-flash ICD +TFM patchtsmixer | $0.806 \pm 2.056e-3$ | $0.87 \pm 5.716e-3$ | $0.767 \pm 6.095e-3$ | $0.832 \pm 4.4e-3$ | $0.81 \pm 8.236e-3$ | $0.907 \pm 2.886e-3$ |
| gemini-2.0-flash ICD +TFM timemixer | $0.804 \pm 7.385e-3$ | $0.878 \pm 1.637e-3$ | $0.771 \pm 7.967e-3$ | $0.841 \pm 4.318e-3$ | $0.813 \pm 9.3e-3$ | $0.91 \pm 3.109e-3$ |
| gemini-2.0-flash ICD lstm | $0.799 \pm 7.573e-3$ | $0.873 \pm 8.271e-3$ | $0.73 \pm 6.661e-3$ | $0.829 \pm 4.84e-3$ | $0.809 \pm 4.763e-3$ | $0.913 \pm 5.972e-3$ |
| gemini-2.0-flash ICD mlp | $0.803 \pm 9.185e-3$ | $0.875 \pm 6.48e-3$ | $0.733 \pm 6.468e-3$ | $0.822 \pm 3.005e-3$ | $0.809 \pm 5.461e-3$ | $0.912 \pm 6.076e-3$ |
| gemini-2.0-flash ICD patchtsmixer | $0.794 \pm 1.271e-2$ | $0.875 \pm 5.899e-3$ | $0.717 \pm 1.344e-2$ | $0.819 \pm 5.034e-3$ | $0.802 \pm 4.614e-3$ | $0.916 \pm 1.032e-2$ |
| gemini-2.0-flash ICD timemixer | $0.789 \pm 4.491e-3$ | $0.873 \pm 3.424e-3$ | $0.719 \pm 5.836e-3$ | $0.822 \pm 2.4e-3$ | $0.806 \pm 2.704e-3$ | $0.918 \pm 4.693e-3$ |
| gemini-2.0-flash Trend +TFM lstm | $0.805 \pm 2.186e-3$ | $0.869 \pm 3.724e-3$ | $0.804 \pm 7.659e-3$ | $0.857 \pm 7.629e-3$ | $0.851 \pm 7.153e-3$ | $0.906 \pm 5.343e-4$ |
| gemini-2.0-flash Trend +TFM mlp | $0.814 \pm 1.586e-3$ | $0.864 \pm 4.928e-3$ | $0.811 \pm 3.376e-3$ | $0.85 \pm 6.824e-4$ | $0.851 \pm 3.532e-3$ | $0.908 \pm 1.825e-3$ |
| gemini-2.0-flash Trend +TFM patchtsmixer | $0.79 \pm 1.242e-2$ | $0.851 \pm 7.54e-3$ | $0.789 \pm 5.27e-3$ | $0.84 \pm 4.126e-3$ | $0.837 \pm 7.07e-3$ | $0.896 \pm 2.298e-3$ |
| gemini-2.0-flash Trend +TFM timemixer | $0.795 \pm 4.414e-3$ | $0.864 \pm 1.924e-3$ | $0.788 \pm 3.605e-3$ | $0.843 \pm 4.348e-3$ | $0.83 \pm 1.029e-2$ | $0.904 \pm 2.206e-3$ |
| gemini-2.0-flash Trend lstm | $0.786 \pm 1.001e-2$ | $0.854 \pm 5.272e-3$ | $0.76 \pm 9.8e-3$ | $0.824 \pm 2.86e-3$ | $0.838 \pm 7e-3$ | $0.91 \pm 5.622e-3$ |
| gemini-2.0-flash Trend mlp | $0.79 \pm 8.753e-3$ | $0.852 \pm 6.831e-3$ | $0.762 \pm 9.526e-3$ | $0.821 \pm 4.046e-3$ | $0.838 \pm 7.23e-3$ | $0.912 \pm 7.303e-3$ |
| gemini-2.0-flash Trend patchtsmixer | $0.775 \pm 6.171e-3$ | $0.853 \pm 1.358e-2$ | $0.757 \pm 1.251e-2$ | $0.816 \pm 8.842e-3$ | $0.827 \pm 3.24e-3$ | $0.916 \pm 5.829e-3$ |
| gemini-2.0-flash Trend timemixer | $0.776 \pm 1.131e-2$ | $0.849 \pm 1.048e-2$ | $0.746 \pm 4.479e-3$ | $0.817 \pm 7.589e-3$ | $0.825 \pm 4.43e-3$ | $0.914 \pm 7.294e-4$ |
| gemini-2.0-flash zero_shot +TFM lstm | $0.863 \pm 4.987e-3$ | $0.917 \pm 5.998e-3$ | $0.864 \pm 5.265e-3$ | $0.913 \pm 5.542e-3$ | $0.896 \pm 1.43e-3$ | $0.92 \pm 8.127e-4$ |
| gemini-2.0-flash zero_shot +TFM mlp | $0.862 \pm 2.848e-3$ | $0.902 \pm 2.268e-3$ | $0.864 \pm 3.762e-3$ | $0.903 \pm 2.7e-3$ | $0.901 \pm 6.076e-3$ | $0.924 \pm 2.546e-3$ |
| gemini-2.0-flash zero_shot +TFM patchtsmixer | $0.816 \pm 2.215e-2$ | $0.877 \pm 1.103e-2$ | $0.847 \pm 3.211e-3$ | $0.897 \pm 2.887e-3$ | $0.886 \pm 4.837e-3$ | $0.911 \pm 2.735e-3$ |
| gemini-2.0-flash zero_shot +TFM timemixer | $0.849 \pm 2.128e-3$ | $0.895 \pm 3.301e-3$ | $0.848 \pm 4.601e-3$ | $0.902 \pm 1.879e-3$ | $0.876 \pm 1.335e-2$ | $0.904 \pm 6.197e-3$ |
| gemini-2.0-flash zero_shot lstm | $0.843 \pm 9.137e-3$ | $0.897 \pm 5.75e-3$ | $0.852 \pm 8.835e-3$ | $0.903 \pm 3.167e-3$ | $0.889 \pm 9.931e-3$ | $0.918 \pm 7.836e-3$ |
| gemini-2.0-flash zero_shot mlp | $0.846 \pm 7.775e-3$ | $0.894 \pm 6.53e-3$ | $0.851 \pm 9.353e-3$ | $0.9 \pm 1.722e-3$ | $0.889 \pm 1.015e-2$ | $0.914 \pm 5.727e-3$ |
| gemini-2.0-flash zero_shot patchtsmixer | $0.832 \pm 1.033e-2$ | $0.89 \pm 2.778e-3$ | $0.838 \pm 7.206e-3$ | $0.894 \pm 2.895e-3$ | $0.869 \pm 8.608e-3$ | $0.907 \pm 2.16e-3$ |
| gemini-2.0-flash zero_shot timemixer | $0.832 \pm 6.102e-3$ | $0.89 \pm 4.892e-3$ | $0.842 \pm 7.26e-3$ | $0.895 \pm 5.414e-3$ | $0.876 \pm 7.177e-3$ | $0.91 \pm 3.559e-3$ |
| interp lstm | $0.899 \pm 9.323e-4$ | $0.886 \pm 3.813e-3$ | $0.882 \pm 1.06e-3$ | $0.875 \pm 5.809e-3$ | $0.923 \pm 1.772e-3$ | $0.882 \pm 5.66e-3$ |
| interp mlp | $0.898 \pm 9.976e-4$ | $0.889 \pm 2.169e-3$ | $0.869 \pm 2.169e-3$ | $0.84 \pm 4.341e-3$ | $0.925 \pm 6.745e-4$ | $0.887 \pm 4.192e-3$ |
| interp patchtsmixer | $0.887 \pm 1.387e-3$ | $0.879 \pm 6.754e-3$ | $0.848 \pm 2.734e-3$ | $0.844 \pm 1.791e-2$ | $0.848 \pm 4.845e-3$ | $0.837 \pm 1.682e-2$ |
| interp timemixer | $0.889 \pm 3.617e-3$ | $0.89 \pm 6.172e-3$ | $0.857 \pm 2.195e-3$ | $0.843 \pm 7.52e-3$ | $0.851 \pm 3.304e-3$ | $0.846 \pm 1.36e-2$ |
| mean lstm | $0.905 \pm 1.05e-3$ | $0.892 \pm 3.051e-3$ | $0.888 \pm 2.364e-3$ | $0.884 \pm 4.763e-3$ | $0.924 \pm 7.926e-4$ | $0.881 \pm 2.693e-3$ |
| mean mlp | $0.901 \pm 6.388e-4$ | $0.887 \pm 2.212e-3$ | $0.875 \pm 1.632e-3$ | $0.855 \pm 4.365e-3$ | $0.926 \pm 2.396e-4$ | $0.887 \pm 1.678e-3$ |
| mean patchtsmixer | $0.883 \pm 3.368e-3$ | $0.878 \pm 1.275e-2$ | $0.844 \pm 5.126e-3$ | $0.838 \pm 1.595e-2$ | $0.844 \pm 2.806e-3$ | $0.834 \pm 1.359e-2$ |
| mean timemixer | $0.892 \pm 2.522e-3$ | $0.898 \pm 5.943e-3$ | $0.861 \pm 2.061e-3$ | $0.854 \pm 4.307e-3$ | $0.863 \pm 9.474e-4$ | $0.859 \pm 4.567e-3$ |
| No_sum_CoT CoT +TFM lstm | $0.9 \pm 4.496e-3$ | $0.915 \pm 1.419e-2$ | $0.892 \pm 9.205e-4$ | $0.907 \pm 2.807e-3$ | $0.931 \pm 3.132e-3$ | $0.915 \pm 2.932e-3$ |
| No_sum_CoT CoT +TFM patchtsmixer | $0.879 \pm 0e0$ | $0.905 \pm 0e0$ | $0.875 \pm 0e0$ | $0.903 \pm 0e0$ | $0.92 \pm 0e0$ | $0.913 \pm 0e0$ |
| No_sum_CoT lstm | $0.879 \pm 3.153e-3$ | $0.911 \pm 1.34e-3$ | $0.876 \pm 2.63e-3$ | $0.903 \pm 2.297e-3$ | $0.92 \pm 4.208e-3$ | $0.911 \pm 2.396e-3$ |
| No_sum_CoT mlp | $0.88 \pm 5.433e-3$ | $0.911 \pm 3.895e-3$ | $0.875 \pm 2.234e-3$ | $0.901 \pm 8.699e-4$ | $0.922 \pm 4.788e-3$ | $0.913 \pm 4.862e-3$ |
| No_sum_CoT patchtsmixer | $0.868 \pm 1.84e-3$ | $0.91 \pm 2.465e-3$ | $0.867 \pm 1.265e-3$ | $0.9 \pm 2.293e-3$ | $0.915 \pm 6.256e-3$ | $0.914 \pm 5.206e-3$ |
| No_sum_CoT timemixer | $0.866 \pm 4.418e-3$ | $0.912 \pm 3.306e-3$ | $0.864 \pm 2.326e-3$ | $0.902 \pm 2.705e-3$ | $0.91 \pm 1.786e-3$ | $0.913 \pm 1.181e-3$ |
| No_sum_ICD ICD +TFM lstm | $0.9 \pm 1.977e-3$ | $0.915 \pm 1.642e-3$ | $0.895 \pm 2.344e-3$ | $0.909 \pm 1.955e-3$ | $0.932 \pm 2.082e-3$ | $0.916 \pm 2.257e-3$ |
| No_sum_ICD ICD +TFM patchtsmixer | $0.883 \pm 0e0$ | $0.906 \pm 0e0$ | $0.867 \pm 0e0$ | $0.897 \pm 0e0$ | $0.908 \pm 0e0$ | $0.906 \pm 0e0$ |
| No_sum_ICD lstm | $0.876 \pm 2.698e-3$ | $0.91 \pm 2.129e-3$ | $0.873 \pm 1.294e-3$ | $0.902 \pm 1.303e-3$ | $0.92 \pm 3.666e-3$ | $0.91 \pm 1.341e-3$ |
| No_sum_ICD mlp | $0.873 \pm 2.943e-3$ | $0.908 \pm 2.162e-3$ | $0.873 \pm 4.679e-3$ | $0.901 \pm 1.18e-3$ | $0.922 \pm 3.821e-3$ | $0.911 \pm 3.056e-3$ |
| No_sum_ICD patchtsmixer | $0.868 \pm 2.209e-3$ | $0.911 \pm 1.463e-3$ | $0.867 \pm 1.446e-3$ | $0.9 \pm 2.08e-3$ | $0.911 \pm 4.231e-3$ | $0.91 \pm 2.351e-3$ |
| No_sum_ICD timemixer | $0.861 \pm 3.385e-3$ | $0.911 \pm 1.886e-3$ | $0.861 \pm 3.087e-3$ | $0.901 \pm 2.361e-3$ | $0.909 \pm 5.276e-3$ | $0.91 \pm 3.126e-3$ |
| No_sum_Trend Trend +TFM lstm | $0.899 \pm 3.564e-3$ | $0.913 \pm 1.287e-3$ | $0.893 \pm 4.497e-3$ | $0.908 \pm 1.725e-3$ | $0.933 \pm 2.694e-3$ | $0.917 \pm 2.875e-4$ |
| No_sum_Trend Trend +TFM patchtsmixer | $0.891 \pm 0e0$ | $0.91 \pm 0e0$ | $0.88 \pm 0e0$ | $0.904 \pm 0e0$ | $0.927 \pm 0e0$ | $0.921 \pm 0e0$ |
| No_sum_Trend lstm | $0.879 \pm 6.174e-3$ | $0.91 \pm 3.441e-3$ | $0.877 \pm 3.142e-3$ | $0.901 \pm 1.489e-3$ | $0.919 \pm 5.606e-3$ | $0.91 \pm 2.737e-3$ |
| No_sum_Trend mlp | $0.882 \pm 3.976e-3$ | $0.91 \pm 2.915e-3$ | $0.877 \pm 3.551e-3$ | $0.901 \pm 7.657e-4$ | $0.922 \pm 5.061e-3$ | $0.914 \pm 3.682e-3$ |
| No_sum_Trend patchtsmixer | $0.869 \pm 5.659e-3$ | $0.909 \pm 2.656e-3$ | $0.869 \pm 2.656e-3$ | $0.9 \pm 1.744e-3$ | $0.914 \pm 2.523e-3$ | $0.911 \pm 3.485e-3$ |
| No_sum_Trend timemixer | $0.862 \pm 3.996e-3$ | $0.912 \pm 3.809e-3$ | $0.869 \pm 1.694e-3$ | $0.901 \pm 1.501e-3$ | $0.91 \pm 3.811e-3$ | $0.911 \pm 1.347e-3$ |
| medgemma CoT +TFM lstm | $0.845 \pm 3.854e-3$ | $0.893 \pm 3.063e-3$ | $0.822 \pm 7.75e-3$ | $0.886 \pm 5.495e-3$ | $0.853 \pm 2.784e-3$ | $0.914 \pm 9.8e-4$ |
| medgemma CoT +TFM mlp | $0.845 \pm 7.988e-3$ | $0.893 \pm 5.966e-3$ | $0.828 \pm 8.741e-4$ | $0.874 \pm 2.578e-3$ | $0.859 \pm 5.718e-3$ | $0.914 \pm 2.687e-3$ |

Table 26: In-distribution results - Drug. (part 1/2)

| Method | hirid → hirid | | mimic → mimic | | ppicu → ppicu | |
|---|---|---|---|---|---|---|
| | f1_micro | recall_micro | f1_micro | recall_micro | f1_micro | recall_micro |
| medgemma CoT +TFM patchtsmixer | $0.816 \pm 1.606e-2$ | $0.876 \pm 6.866e-3$ | $0.811 \pm 6.325e-3$ | $0.867 \pm 3.918e-3$ | $0.833 \pm 1.357e-2$ | $0.912 \pm 5.332e-3$ |
| medgemma CoT +TFM timemixer | $0.824 \pm 4.624e-3$ | $0.887 \pm 2.053e-3$ | $0.805 \pm 6.915e-3$ | $0.873 \pm 6.623e-3$ | $0.841 \pm 8.596e-3$ | $0.904 \pm 7.508e-3$ |
| medgemma CoT lstm | $0.821 \pm 8.278e-3$ | $0.88 \pm 5.981e-3$ | $0.787 \pm 1.078e-2$ | $0.855 \pm 2.705e-3$ | $0.84 \pm 7.288e-3$ | $0.919 \pm 4.913e-3$ |
| medgemma CoT mlp | $0.824 \pm 7.467e-3$ | $0.884 \pm 2.531e-3$ | $0.788 \pm 1.038e-2$ | $0.849 \pm 4.009e-3$ | $0.841 \pm 6.244e-3$ | $0.918 \pm 3.274e-3$ |
| medgemma CoT patchtsmixer | $0.816 \pm 6.976e-3$ | $0.882 \pm 1.833e-3$ | $0.773 \pm 8.922e-3$ | $0.848 \pm 5.822e-3$ | $0.828 \pm 4.332e-3$ | $0.919 \pm 3.551e-3$ |
| medgemma CoT timemixer | $0.81 \pm 4.429e-3$ | $0.88 \pm 1.775e-3$ | $0.775 \pm 5.807e-3$ | $0.847 \pm 2.43e-3$ | $0.833 \pm 5.616e-3$ | $0.922 \pm 3.817e-3$ |
| medgemma ICD +TFM lstm | $0.833 \pm 5.142e-3$ | $0.893 \pm 6.274e-3$ | $0.832 \pm 5.928e-3$ | $0.891 \pm 4.639e-3$ | $0.863 \pm 3.365e-3$ | $0.916 \pm 3.334e-3$ |
| medgemma ICD +TFM mlp | $0.835 \pm 4.035e-3$ | $0.89 \pm 1.505e-3$ | $0.831 \pm 3.681e-3$ | $0.877 \pm 2.38e-3$ | $0.869 \pm 3.292e-3$ | $0.914 \pm 2.275e-3$ |
| medgemma ICD +TFM patchtsmixer | $0.821 \pm 5.796e-3$ | $0.882 \pm 2.384e-3$ | $0.816 \pm 1.204e-2$ | $0.87 \pm 4.382e-3$ | $0.847 \pm 1.173e-2$ | $0.907 \pm 3.869e-3$ |
| medgemma ICD +TFM timemixer | $0.824 \pm 9.086e-3$ | $0.891 \pm 2.021e-3$ | $0.82 \pm 2.987e-3$ | $0.876 \pm 5.201e-3$ | $0.846 \pm 2.419e-3$ | $0.912 \pm 2.627e-3$ |
| medgemma ICD lstm | $0.814 \pm 8.569e-3$ | $0.888 \pm 3.051e-3$ | $0.79 \pm 6.621e-3$ | $0.869 \pm 2.355e-3$ | $0.854 \pm 9.239e-3$ | $0.916 \pm 3.989e-3$ |
| medgemma ICD mlp | $0.812 \pm 7.204e-3$ | $0.882 \pm 2.433e-3$ | $0.797 \pm 9.16e-3$ | $0.863 \pm 8.231e-4$ | $0.856 \pm 7.955e-3$ | $0.914 \pm 4.12e-3$ |
| medgemma ICD patchtsmixer | $0.801 \pm 5.345e-3$ | $0.887 \pm 5.366e-3$ | $0.782 \pm 5.648e-3$ | $0.861 \pm 5.315e-3$ | $0.843 \pm 7.95e-3$ | $0.916 \pm 2.581e-3$ |
| medgemma ICD timemixer | $0.804 \pm 7.716e-3$ | $0.883 \pm 2.69e-3$ | $0.784 \pm 6.648e-3$ | $0.86 \pm 2.863e-3$ | $0.848 \pm 6.34e-3$ | $0.916 \pm 1.932e-3$ |
| medgemma Trend +TFM lstm | $0.807 \pm 5.339e-3$ | $0.878 \pm 4.003e-3$ | $0.791 \pm 5.387e-3$ | $0.856 \pm 4.259e-3$ | $0.826 \pm 9.506e-4$ | $0.915 \pm 5.121e-3$ |
| medgemma Trend +TFM mlp | $0.825 \pm 5.077e-3$ | $0.875 \pm 3.013e-3$ | $0.792 \pm 4.857e-3$ | $0.842 \pm 1.411e-3$ | $0.83 \pm 7.246e-3$ | $0.911 \pm 2.66e-3$ |
| medgemma Trend +TFM patchtsmixer | $0.791 \pm 3.926e-3$ | $0.871 \pm 3.617e-3$ | $0.769 \pm 6.276e-3$ | $0.829 \pm 6.85e-3$ | $0.812 \pm 1.385e-2$ | $0.912 \pm 6.464e-3$ |
| medgemma Trend +TFM timemixer | $0.797 \pm 1.197e-2$ | $0.872 \pm 3.333e-3$ | $0.776 \pm 3.095e-3$ | $0.838 \pm 2.689e-3$ | $0.818 \pm 9.05e-3$ | $0.915 \pm 3.635e-3$ |
| medgemma Trend lstm | $0.789 \pm 9.638e-3$ | $0.872 \pm 4.757e-3$ | $0.737 \pm 8.961e-3$ | $0.818 \pm 5.73e-3$ | $0.813 \pm 4.827e-3$ | $0.924 \pm 1.156e-2$ |
| medgemma Trend mlp | $0.794 \pm 9.509e-3$ | $0.868 \pm 5.495e-3$ | $0.74 \pm 9.047e-3$ | $0.82 \pm 4.334e-3$ | $0.812 \pm 3.613e-3$ | $0.923 \pm 8.403e-3$ |
| medgemma Trend patchtsmixer | $0.781 \pm 6.313e-3$ | $0.87 \pm 7.499e-3$ | $0.73 \pm 6.975e-3$ | $0.821 \pm 5.311e-3$ | $0.809 \pm 2.999e-3$ | $0.926 \pm 9.635e-3$ |
| medgemma Trend timemixer | $0.781 \pm 5.326e-3$ | $0.87 \pm 9.221e-3$ | $0.728 \pm 6.475e-3$ | $0.822 \pm 2.376e-3$ | $0.81 \pm 3.251e-3$ | $0.927 \pm 1.016e-2$ |
| medgemma zero_shot +TFM lstm | $0.853 \pm 3.201e-3$ | $0.908 \pm 2.792e-3$ | $0.863 \pm 7.753e-4$ | $0.914 \pm 6.976e-4$ | $0.89 \pm 5.206e-3$ | $0.919 \pm 3.282e-3$ |
| medgemma zero_shot +TFM mlp | $0.871 \pm 5.77e-3$ | $0.908 \pm 2.679e-3$ | $0.855 \pm 4.225e-3$ | $0.899 \pm 1.999e-3$ | $0.891 \pm 5.443e-3$ | $0.917 \pm 3.266e-3$ |
| medgemma zero_shot +TFM patchtsmixer | $0.84 \pm 7.043e-3$ | $0.897 \pm 7.143e-4$ | $0.844 \pm 7.993e-3$ | $0.893 \pm 6.471e-3$ | $0.87 \pm 9.705e-3$ | $0.904 \pm 5.752e-3$ |
| medgemma zero_shot +TFM timemixer | $0.837 \pm 5.705e-3$ | $0.892 \pm 4.526e-3$ | $0.842 \pm 2.596e-3$ | $0.896 \pm 3.402e-3$ | $0.869 \pm 7.885e-3$ | $0.907 \pm 6.656e-3$ |
| medgemma zero_shot lstm | $0.838 \pm 7.46e-3$ | $0.901 \pm 3.792e-3$ | $0.838 \pm 9.671e-3$ | $0.895 \pm 9.065e-3$ | $0.874 \pm 9.389e-3$ | $0.915 \pm 3.963e-3$ |
| medgemma zero_shot mlp | $0.84 \pm 8.177e-3$ | $0.899 \pm 1.083e-2$ | $0.839 \pm 9.327e-3$ | $0.893 \pm 6.112e-3$ | $0.878 \pm 8.874e-3$ | $0.91 \pm 3.853e-3$ |
| medgemma zero_shot patchtsmixer | $0.825 \pm 2.188e-3$ | $0.887 \pm 3.195e-3$ | $0.828 \pm 1.111e-2$ | $0.883 \pm 6.946e-3$ | $0.866 \pm 8.997e-3$ | $0.907 \pm 4.078e-3$ |
| medgemma zero_shot timemixer | $0.828 \pm 9.289e-3$ | $0.895 \pm 5.052e-3$ | $0.83 \pm 8.241e-3$ | $0.888 \pm 4.706e-3$ | $0.868 \pm 8.289e-3$ | $0.911 \pm 3.518e-3$ |
| No_sum_zero_shot zero_shot +TFM lstm | $0.901 \pm 6.503e-3$ | $0.912 \pm 1.125e-3$ | $0.891 \pm 4.215e-3$ | $0.906 \pm 1.323e-3$ | $0.932 \pm 3.949e-3$ | $0.916 \pm 3.882e-3$ |
| No_sum_zero_shot zero_shot +TFM patchtsmixer | $0.875 \pm 0e0$ | $0.903 \pm 0e0$ | $0.883 \pm 0e0$ | $0.905 \pm 0e0$ | $0.921 \pm 0e0$ | $0.914 \pm 0e0$ |
| No_sum_zero_shot lstm | $0.884 \pm 3.829e-3$ | $0.908 \pm 2.785e-3$ | $0.878 \pm 5.327e-3$ | $0.902 \pm 2.01e-3$ | $0.921 \pm 6.224e-3$ | $0.911 \pm 2.664e-3$ |
| No_sum_zero_shot mlp | $0.884 \pm 4.159e-3$ | $0.911 \pm 3.471e-3$ | $0.875 \pm 5.187e-3$ | $0.901 \pm 6.915e-4$ | $0.923 \pm 5.792e-3$ | $0.913 \pm 4.059e-3$ |
| No_sum_zero_shot patchtsmixer | $0.872 \pm 4.123e-3$ | $0.907 \pm 3.57e-3$ | $0.867 \pm 1.975e-3$ | $0.903 \pm 3.036e-3$ | $0.913 \pm 3.249e-3$ | $0.914 \pm 1.405e-3$ |
| No_sum_zero_shot timemixer | $0.864 \pm 1.586e-3$ | $0.912 \pm 1.225e-3$ | $0.867 \pm 1.081e-3$ | $0.902 \pm 2.729e-3$ | $0.908 \pm 3.663e-3$ | $0.915 \pm 3.573e-3$ |
| right lstm | $0.903 \pm 9.254e-4$ | $0.891 \pm 2.123e-3$ | $0.886 \pm 2.159e-3$ | $0.882 \pm 5.877e-3$ | $0.924 \pm 1.166e-3$ | $0.883 \pm 3.702e-3$ |
| right mlp | $0.901 \pm 2.061e-4$ | $0.88 \pm 8.09e-4$ | $0.86 \pm 3.343e-3$ | $0.927 \pm 6.405e-4$ | $0.892 \pm 1.716e-3$ |
| right patchtsmixer | $0.888 \pm 2.581e-3$ | $0.884 \pm 6.439e-3$ | $0.849 \pm 3.944e-3$ | $0.841 \pm 2.423e-2$ | $0.848 \pm 6.675e-3$ | $0.831 \pm 2.645e-2$ |
| right timemixer | $0.9 \pm 1.44e-3$ | $0.895 \pm 4.593e-3$ | $0.863 \pm 3.713e-4$ | $0.85 \pm 2.895e-3$ | $0.859 \pm 4.092e-3$ | $0.846 \pm 1.355e-2$ |

Table 27: In-distribution results - Drug. (part 2/2)

| Method | hirid → ppicu | | mimic → ppicu | |
|---|---|---|---|---|
| | f1_micro | recall_micro | f1_micro | recall_micro |
| Llama-3.1 CoT +TFM lstm | $0.769 \pm 1.565e-3$ | $0.931 \pm 1.886e-3$ | $0.777 \pm 2.04e-3$ | $0.931 \pm 1.593e-3$ |
| Llama-3.1 CoT +TFM mlp | $0.763 \pm 1.471e-3$ | $0.933 \pm 1.939e-3$ | $0.774 \pm 1.628e-3$ | $0.938 \pm 4.576e-3$ |
| Llama-3.1 CoT +TFM patchtsmixer | $0.75 \pm 2.781e-3$ | $0.945 \pm 4.113e-3$ | $0.767 \pm 3.535e-3$ | $0.932 \pm 1.907e-3$ |
| Llama-3.1 CoT +TFM timemixer | $0.764 \pm 3.39e-3$ | $0.937 \pm 5.739e-3$ | $0.772 \pm 1.664e-3$ | $0.936 \pm 6.964e-4$ |
| Llama-3.1 CoT lstm | $0.761 \pm 3.771e-3$ | $0.944 \pm 3.279e-3$ | $0.766 \pm 4.793e-4$ | $0.95 \pm 7.115e-3$ |
| Llama-3.1 CoT mlp | $0.762 \pm 2.565e-3$ | $0.944 \pm 7.042e-3$ | $0.765 \pm 1.757e-3$ | $0.95 \pm 6.242e-3$ |
| Llama-3.1 CoT patchtsmixer | $0.757 \pm 2.87e-3$ | $0.953 \pm 8.317e-3$ | $0.764 \pm 1.703e-3$ | $0.949 \pm 6.546e-3$ |
| Llama-3.1 CoT timemixer | $0.76 \pm 3.443e-3$ | $0.951 \pm 6.63e-3$ | $0.764 \pm 9.235e-4$ | $0.952 \pm 8.599e-3$ |
| Llama-3.1 ICD +TFM lstm | $0.769 \pm 1.131e-3$ | $0.927 \pm 4.855e-3$ | $0.78 \pm 2.845e-4$ | $0.934 \pm 3.966e-3$ |
| Llama-3.1 ICD +TFM mlp | $0.772 \pm 3.155e-3$ | $0.937 \pm 1.884e-3$ | $0.78 \pm 1.354e-3$ | $0.937 \pm 3.736e-3$ |
| Llama-3.1 ICD +TFM patchtsmixer | $0.754 \pm 1.392e-3$ | $0.935 \pm 4.183e-3$ | $0.774 \pm 2.245e-3$ | $0.928 \pm 2.228e-3$ |
| Llama-3.1 ICD +TFM timemixer | $0.76 \pm 6.022e-3$ | $0.934 \pm 4.85e-3$ | $0.779 \pm 1.468e-3$ | $0.937 \pm 5.118e-3$ |
| Llama-3.1 ICD lstm | $0.773 \pm 1.46e-3$ | $0.947 \pm 3.975e-3$ | $0.776 \pm 1.883e-3$ | $0.948 \pm 8.712e-3$ |
| Llama-3.1 ICD mlp | $0.774 \pm 1.227e-3$ | $0.942 \pm 6.249e-3$ | $0.775 \pm 2.145e-3$ | $0.947 \pm 3.594e-3$ |
| Llama-3.1 ICD patchtsmixer | $0.771 \pm 2.857e-3$ | $0.941 \pm 4.237e-3$ | $0.774 \pm 2.882e-3$ | $0.952 \pm 3.825e-3$ |
| Llama-3.1 ICD timemixer | $0.774 \pm 2.123e-3$ | $0.946 \pm 2.746e-3$ | $0.776 \pm 2.973e-3$ | $0.95 \pm 2.51e-3$ |
| Llama-3.1 Trend +TFM lstm | $0.789 \pm 1.766e-3$ | $0.925 \pm 2.241e-3$ | $0.789 \pm 1.162e-3$ | $0.922 \pm 3.262e-3$ |
| Llama-3.1 Trend +TFM mlp | $0.786 \pm 3.692e-3$ | $0.931 \pm 1.383e-3$ | $0.79 \pm 2.015e-3$ | $0.93 \pm 3.038e-3$ |
| Llama-3.1 Trend +TFM patchtsmixer | $0.771 \pm 2.673e-3$ | $0.923 \pm 3.235e-3$ | $0.784 \pm 1.017e-3$ | $0.926 \pm 6.444e-3$ |
| Llama-3.1 Trend +TFM timemixer | $0.785 \pm 2.486e-3$ | $0.932 \pm 2.821e-3$ | $0.788 \pm 1.255e-3$ | $0.921 \pm 2.751e-3$ |
| Llama-3.1 Trend lstm | $0.789 \pm 1.779e-3$ | $0.935 \pm 3.835e-3$ | $0.785 \pm 1.248e-3$ | $0.925 \pm 1.794e-3$ |
| Llama-3.1 Trend mlp | $0.789 \pm 1.819e-3$ | $0.937 \pm 2.334e-3$ | $0.784 \pm 2.716e-3$ | $0.929 \pm 5.426e-3$ |
| Llama-3.1 Trend patchtsmixer | $0.786 \pm 2.478e-3$ | $0.939 \pm 8.783e-4$ | $0.779 \pm 4.816e-3$ | $0.934 \pm 4.188e-3$ |
| Llama-3.1 Trend timemixer | $0.789 \pm 2.811e-3$ | $0.939 \pm 3.333e-3$ | $0.782 \pm 2.986e-3$ | $0.932 \pm 2.18e-3$ |
| Llama-3.1 zero_shot +TFM lstm | $0.77 \pm 2.102e-3$ | $0.94 \pm 2.732e-3$ | $0.779 \pm 2.321e-3$ | $0.933 \pm 2.519e-3$ |
| Llama-3.1 zero_shot +TFM mlp | $0.767 \pm 2.487e-3$ | $0.942 \pm 1.719e-3$ | $0.778 \pm 1.441e-3$ | $0.94 \pm 4.035e-3$ |
| Llama-3.1 zero_shot +TFM patchtsmixer | $0.747 \pm 1.155e-3$ | $0.941 \pm 3.095e-3$ | $0.771 \pm 2.038e-3$ | $0.928 \pm 4.455e-3$ |
| Llama-3.1 zero_shot +TFM timemixer | $0.763 \pm 3.452e-3$ | $0.941 \pm 1.541e-3$ | $0.773 \pm 1.115e-3$ | $0.934 \pm 3.409e-3$ |
| Llama-3.1 zero_shot lstm | $0.774 \pm 4.573e-3$ | $0.951 \pm 6.489e-4$ | $0.773 \pm 3.396e-3$ | $0.95 \pm 3.991e-3$ |
| Llama-3.1 zero_shot mlp | $0.773 \pm 3.335e-3$ | $0.95 \pm 4.034e-3$ | $0.774 \pm 2.273e-3$ | $0.95 \pm 2.508e-3$ |
| Llama-3.1 zero_shot patchtsmixer | $0.772 \pm 4.054e-3$ | $0.952 \pm 2.209e-3$ | $0.774 \pm 2.935e-3$ | $0.949 \pm 4.178e-3$ |
| Llama-3.1 zero_shot timemixer | $0.775 \pm 3.184e-3$ | $0.952 \pm 2.072e-3$ | $0.775 \pm 2.595e-3$ | $0.953 \pm 1.233e-3$ |
| TFM lstm | $0.792 \pm 6.625e-4$ | $0.934 \pm 3.326e-3$ | $0.788 \pm 2.603e-4$ | $0.941 \pm 2.725e-3$ |
| TFM mlp | $0.79 \pm 1.096e-3$ | $0.932 \pm 1.591e-3$ | $0.783 \pm 1.592e-3$ | $0.944 \pm 2.947e-3$ |
| TFM patchtsmixer | $0.779 \pm 1.063e-3$ | $0.935 \pm 2.672e-3$ | $0.781 \pm 1.161e-3$ | $0.947 \pm 2.087e-3$ |
| TFM timemixer | $0.785 \pm 1.221e-3$ | $0.943 \pm 3.055e-3$ | $0.787 \pm 9.39e-4$ | $0.941 \pm 1.408e-3$ |
| TSDE mlp | $0.731 \pm 5.055e-4$ | $0.916 \pm 1.303e-3$ | $0.731 \pm 1.478e-3$ | $0.939 \pm 3.459e-3$ |
| TSDE lstm | $0.732 \pm 5.055e-4$ | $0.917 \pm 1.159e-3$ | $0.733 \pm 1.27e-3$ | $0.905 \pm 2.959e-3$ |
| TSDE patchtsmixer | $0.732 \pm 5.084e-4$ | $0.909 \pm 9.785e-4$ | $0.733 \pm 3.643e-4$ | $0.899 \pm 2.846e-4$ |
| TSDE timemixer | $0.732 \pm 5.027e-4$ | $0.918 \pm 1.937e-3$ | $0.734 \pm 1.911e-3$ | $0.904 \pm 3.655e-3$ |
| gemini-2.0-flash CoT +TFM lstm | $0.795 \pm 2.031e-3$ | $0.922 \pm 4.589e-3$ | $0.797 \pm 1.453e-3$ | $0.919 \pm 4.039e-3$ |
| gemini-2.0-flash CoT +TFM mlp | $0.792 \pm 9.546e-4$ | $0.927 \pm 4.131e-3$ | $0.799 \pm 2.066e-3$ | $0.926 \pm 1.863e-3$ |
| gemini-2.0-flash CoT +TFM patchtsmixer | $0.77 \pm 6.553e-4$ | $0.912 \pm 3.054e-3$ | $0.778 \pm 6.061e-3$ | $0.916 \pm 4.519e-3$ |
| gemini-2.0-flash CoT +TFM timemixer | $0.779 \pm 2.736e-3$ | $0.921 \pm 2.092e-3$ | $0.793 \pm 1.69e-3$ | $0.921 \pm 2.353e-3$ |
| gemini-2.0-flash CoT lstm | $0.79 \pm 3.174e-3$ | $0.93 \pm 1.465e-3$ | $0.798 \pm 2.544e-4$ | $0.934 \pm 4.794e-3$ |
| gemini-2.0-flash CoT mlp | $0.791 \pm 2.826e-3$ | $0.93 \pm 2.745e-3$ | $0.798 \pm 1.862e-3$ | $0.931 \pm 6.755e-3$ |
| gemini-2.0-flash CoT patchtsmixer | $0.788 \pm 2.483e-3$ | $0.93 \pm 2.14e-3$ | $0.797 \pm 1.708e-3$ | $0.932 \pm 5.13e-3$ |
| gemini-2.0-flash CoT timemixer | $0.793 \pm 1.82e-3$ | $0.93 \pm 2.094e-3$ | $0.799 \pm 2.242e-3$ | $0.929 \pm 1.58e-3$ |
| gemini-2.0-flash ICD +TFM lstm | $0.76 \pm 1.129e-3$ | $0.929 \pm 1.224e-3$ | $0.774 \pm 1.572e-3$ | $0.936 \pm 2.818e-3$ |
| gemini-2.0-flash ICD +TFM mlp | $0.761 \pm 1.293e-3$ | $0.937 \pm 3.918e-3$ | $0.777 \pm 1.703e-3$ | $0.935 \pm 1.565e-3$ |
| gemini-2.0-flash ICD +TFM patchtsmixer | $0.736 \pm 2.042e-3$ | $0.968 \pm 2.401e-3$ | $0.766 \pm 7.831e-4$ | $0.929 \pm 1.971e-3$ |
| gemini-2.0-flash ICD +TFM timemixer | $0.753 \pm 4.107e-3$ | $0.947 \pm 6.716e-3$ | $0.773 \pm 8.622e-4$ | $0.933 \pm 1.654e-3$ |
| gemini-2.0-flash ICD lstm | $0.749 \pm 2.683e-3$ | $0.954 \pm 5.802e-3$ | $0.781 \pm 2.21e-3$ | $0.938 \pm 7.867e-4$ |
| gemini-2.0-flash ICD mlp | $0.749 \pm 3.572e-3$ | $0.956 \pm 2.767e-3$ | $0.781 \pm 2.93e-3$ | $0.939 \pm 4.633e-3$ |
| gemini-2.0-flash ICD patchtsmixer | $0.749 \pm 5.385e-3$ | $0.959 \pm 4.889e-3$ | $0.779 \pm 2.487e-3$ | $0.94 \pm 4.035e-3$ |
| gemini-2.0-flash ICD timemixer | $0.757 \pm 1.928e-3$ | $0.949 \pm 3.16e-3$ | $0.78 \pm 2.748e-3$ | $0.94 \pm 4.242e-3$ |
| gemini-2.0-flash Trend +TFM lstm | $0.781 \pm 4.03e-3$ | $0.928 \pm 5.388e-3$ | $0.783 \pm 6.712e-4$ | $0.93 \pm 1.714e-3$ |
| gemini-2.0-flash Trend +TFM mlp | $0.777 \pm 2.635e-3$ | $0.931 \pm 4.03e-3$ | $0.784 \pm 4.19e-3$ | $0.933 \pm 2.811e-3$ |
| gemini-2.0-flash Trend +TFM patchtsmixer | $0.759 \pm 5.442e-3$ | $0.925 \pm 6.722e-3$ | $0.774 \pm 2.541e-3$ | $0.923 \pm 2.473e-3$ |
| gemini-2.0-flash Trend +TFM timemixer | $0.774 \pm 3.092e-3$ | $0.934 \pm 3.535e-3$ | $0.783 \pm 9.199e-4$ | $0.928 \pm 6.544e-3$ |
| gemini-2.0-flash Trend lstm | $0.786 \pm 1.8e-3$ | $0.935 \pm 4.027e-3$ | $0.785 \pm 2.572e-3$ | $0.941 \pm 4.679e-3$ |
| gemini-2.0-flash Trend mlp | $0.785 \pm 2.353e-3$ | $0.935 \pm 4.557e-3$ | $0.787 \pm 1.862e-3$ | $0.94 \pm 3.453e-3$ |
| gemini-2.0-flash Trend patchtsmixer | $0.783 \pm 2.399e-3$ | $0.939 \pm 5.041e-3$ | $0.786 \pm 1.849e-3$ | $0.943 \pm 3.641e-3$ |
| gemini-2.0-flash Trend timemixer | $0.788 \pm 1.085e-3$ | $0.941 \pm 2.566e-3$ | $0.787 \pm 2.902e-3$ | $0.941 \pm 1.593e-3$ |
| gemini-2.0-flash zero_shot +TFM lstm | $0.805 \pm 4.417e-4$ | $0.934 \pm 2.364e-3$ | $0.803 \pm 8.592e-4$ | $0.931 \pm 1.903e-3$ |
| gemini-2.0-flash zero_shot +TFM mlp | $0.802 \pm 1.462e-3$ | $0.937 \pm 3.03e-3$ | $0.805 \pm 1.826e-3$ | $0.935 \pm 1.732e-3$ |
| gemini-2.0-flash zero_shot +TFM patchtsmixer | $0.777 \pm 4.705e-3$ | $0.917 \pm 4.439e-3$ | $0.791 \pm 2.184e-3$ | $0.924 \pm 1.232e-3$ |
| gemini-2.0-flash zero_shot +TFM timemixer | $0.792 \pm 3.123e-3$ | $0.927 \pm 2.555e-3$ | $0.801 \pm 1.244e-3$ | $0.929 \pm 1.953e-3$ |
| gemini-2.0-flash zero_shot lstm | $0.808 \pm 1.19e-3$ | $0.946 \pm 1.719e-3$ | $0.808 \pm 1.656e-3$ | $0.936 \pm 2.038e-3$ |
| gemini-2.0-flash zero_shot mlp | $0.807 \pm 8.551e-4$ | $0.941 \pm 2.503e-3$ | $0.808 \pm 1.284e-3$ | $0.935 \pm 3.185e-3$ |
| gemini-2.0-flash zero_shot patchtsmixer | $0.804 \pm 1.035e-3$ | $0.941 \pm 3.829e-3$ | $0.806 \pm 1.31e-3$ | $0.936 \pm 3.405e-3$ |
| gemini-2.0-flash zero_shot timemixer | $0.808 \pm 4.102e-4$ | $0.947 \pm 3.22e-3$ | $0.808 \pm 5.868e-4$ | $0.938 \pm 2.419e-3$ |
| interp lstm | $0.742 \pm 7.07e-3$ | $0.87 \pm 5.803e-2$ | $0.516 \pm 1.005e-2$ | $0.4 \pm 1.348e-2$ |
| interp mlp | $0.723 \pm 5.267e-3$ | $0.834 \pm 3.155e-2$ | $0.621 \pm 1.595e-2$ | $0.554 \pm 2.712e-2$ |
| interp patchtsmixer | $0.461 \pm 3.294e-2$ | $0.345 \pm 3.335e-2$ | $0.325 \pm 3.759e-2$ | $0.213 \pm 3.304e-2$ |
| interp timemixer | $0.646 \pm 1.65e-2$ | $0.609 \pm 3.489e-2$ | $0.343 \pm 1.279e-2$ | $0.227 \pm 1.028e-2$ |
| mean lstm | $0.669 \pm 9.804e-3$ | $0.629 \pm 1.499e-2$ | $0.682 \pm 2.045e-2$ | $0.663 \pm 4.92e-2$ |
| mean mlp | $0.711 \pm 8.323e-3$ | $0.712 \pm 2.228e-2$ | $0.662 \pm 7.924e-4$ | $0.621 \pm 1.92e-2$ |
| mean patchtsmixer | $0.511 \pm 5.235e-2$ | $0.422 \pm 7.283e-2$ | $0.424 \pm 3.362e-2$ | $0.303 \pm 3.908e-2$ |
| mean timemixer | $0.7 \pm 4.65e-3$ | $0.719 \pm 1.265e-2$ | $0.497 \pm 7.052e-3$ | $0.384 \pm 1.168e-2$ |
| No_sum_CoT CoT +TFM lstm | $0.831 \pm 8.552e-4$ | $0.953 \pm 2.666e-3$ | $0.8 \pm 4.439e-3$ | $0.91 \pm 8.198e-3$ |
| No_sum_CoT CoT +TFM patchtsmixer | $0.808 \pm 0e0$ | $0.93 \pm 0e0$ | $0.779 \pm 0e0$ | $0.922 \pm 0e0$ |
| No_sum_CoT lstm | $0.839 \pm 1.693e-3$ | $0.961 \pm 2.933e-3$ | $0.819 \pm 1.368e-3$ | $0.946 \pm 2.732e-3$ |
| No_sum_CoT mlp | $0.837 \pm 3.725e-4$ | $0.957 \pm 4.288e-3$ | $0.819 \pm 2.082e-3$ | $0.946 \pm 4.004e-3$ |
| No_sum_CoT patchtsmixer | $0.836 \pm 1.488e-3$ | $0.963 \pm 1.443e-3$ | $0.818 \pm 6.36e-4$ | $0.944 \pm 3.774e-3$ |
| No_sum_CoT timemixer | $0.837 \pm 1.145e-3$ | $0.966 \pm 2.013e-3$ | $0.824 \pm 3.503e-3$ | $0.949 \pm 2.709e-3$ |
| No_sum_ICD ICD +TFM lstm | $0.828 \pm 1.955e-3$ | $0.947 \pm 3.566e-3$ | $0.801 \pm 4.128e-3$ | $0.916 \pm 1.229e-2$ |
| No_sum_ICD ICD +TFM patchtsmixer | $0.81 \pm 0e0$ | $0.931 \pm 0e0$ | $0.781 \pm 0e0$ | $0.924 \pm 0e0$ |
| No_sum_ICD lstm | $0.836 \pm 5.28e-4$ | $0.963 \pm 4.649e-3$ | $0.82 \pm 1.416e-3$ | $0.95 \pm 2.91e-3$ |
| No_sum_ICD mlp | $0.836 \pm 9.8e-4$ | $0.963 \pm 2.615e-3$ | $0.817 \pm 1.021e-3$ | $0.944 \pm 2.966e-3$ |
| No_sum_ICD patchtsmixer | $0.836 \pm 1.277e-3$ | $0.965 \pm 1.212e-3$ | $0.818 \pm 1.81e-3$ | $0.946 \pm 2.933e-3$ |
| No_sum_ICD timemixer | $0.836 \pm 1.503e-3$ | $0.971 \pm 9.228e-4$ | $0.825 \pm 2.262e-3$ | $0.954 \pm 2.281e-3$ |
| No_sum_Trend Trend +TFM lstm | $0.83 \pm 3.097e-3$ | $0.952 \pm 5.162e-3$ | $0.805 \pm 6.704e-3$ | $0.929 \pm 2.766e-3$ |
| No_sum_Trend Trend +TFM patchtsmixer | $0.806 \pm 0e0$ | $0.925 \pm 0e0$ | $0.781 \pm 0e0$ | $0.925 \pm 0e0$ |
| No_sum_Trend lstm | $0.836 \pm 2e-3$ | $0.961 \pm 4.458e-3$ | $0.823 \pm 1.1e-3$ | $0.948 \pm 2.113e-3$ |
| No_sum_Trend mlp | $0.833 \pm 8.873e-4$ | $0.953 \pm 3.464e-3$ | $0.821 \pm 3.409e-3$ | $0.945 \pm 2.64e-3$ |
| No_sum_Trend patchtsmixer | $0.836 \pm 1.59e-3$ | $0.962 \pm 2.871e-3$ | $0.82 \pm 1.466e-3$ | $0.944 \pm 3.025e-3$ |
| No_sum_Trend timemixer | $0.836 \pm 1.247e-3$ | $0.97 \pm 1.808e-3$ | $0.826 \pm 1.335e-3$ | $0.949 \pm 2.029e-3$ |
| medgemma CoT +TFM lstm | $0.791 \pm 4.491e-4$ | $0.912 \pm 1.675e-3$ | $0.796 \pm 1.672e-3$ | $0.908 \pm 9.96e-4$ |
| medgemma CoT +TFM mlp | $0.79 \pm 1.493e-3$ | $0.909 \pm 5.324e-3$ | $0.798 \pm 2.988e-3$ | $0.911 \pm 7.002e-3$ |

Table 28: Cross-site transfer results - Drug. (part 1/2)

| Method | hirid → ppicu | | mimic → ppicu | |
| --- | --- | --- | --- | --- |
| | f1_micro | recall_micro | f1_micro | recall_micro |
| medgemma CoT +TFM patchtsmixer | $0.772 \pm 2.573e-3$ | $0.907 \pm 3.85e-3$ | $0.784 \pm 9.672e-4$ | $0.916 \pm 9.366e-4$ |
| medgemma CoT +TFM timemixer | $0.783 \pm 1.245e-3$ | $0.912 \pm 4.037e-3$ | $0.793 \pm 2.516e-4$ | $0.91 \pm 2.613e-3$ |
| medgemma CoT lstm | $0.788 \pm 2.26e-3$ | $0.924 \pm 1.781e-3$ | $0.797 \pm 6.506e-4$ | $0.923 \pm 5.205e-3$ |
| medgemma CoT mlp | $0.79 \pm 6.558e-4$ | $0.924 \pm 3.799e-3$ | $0.795 \pm 6.553e-4$ | $0.926 \pm 7.525e-4$ |
| medgemma CoT patchtsmixer | $0.791 \pm 1.586e-3$ | $0.922 \pm 8.402e-4$ | $0.793 \pm 1.714e-3$ | $0.924 \pm 3.783e-3$ |
| medgemma CoT timemixer | $0.796 \pm 1.224e-3$ | $0.926 \pm 6.978e-4$ | $0.795 \pm 1.366e-3$ | $0.927 \pm 5.454e-3$ |
| medgemma ICD +TFM lstm | $0.803 \pm 1.08e-3$ | $0.936 \pm 1.261e-3$ | $0.805 \pm 1.541e-3$ | $0.927 \pm 2.397e-3$ |
| medgemma ICD +TFM mlp | $0.803 \pm 2.028e-3$ | $0.94 \pm 3.247e-3$ | $0.809 \pm 6.264e-4$ | $0.937 \pm 3.902e-3$ |
| medgemma ICD +TFM patchtsmixer | $0.781 \pm 6.616e-4$ | $0.919 \pm 1.42e-3$ | $0.794 \pm 2.607e-3$ | $0.919 \pm 5.06e-3$ |
| medgemma ICD +TFM timemixer | $0.792 \pm 6.278e-3$ | $0.927 \pm 7.214e-3$ | $0.803 \pm 1.484e-3$ | $0.928 \pm 2.035e-3$ |
| medgemma ICD lstm | $0.805 \pm 1.005e-3$ | $0.94 \pm 1.491e-3$ | $0.808 \pm 8.141e-4$ | $0.94 \pm 1.442e-3$ |
| medgemma ICD mlp | $0.805 \pm 7.957e-4$ | $0.941 \pm 2.369e-3$ | $0.808 \pm 8.444e-4$ | $0.938 \pm 1.166e-3$ |
| medgemma ICD patchtsmixer | $0.803 \pm 8.531e-4$ | $0.939 \pm 3.272e-3$ | $0.805 \pm 4.692e-4$ | $0.94 \pm 3.634e-3$ |
| medgemma ICD timemixer | $0.806 \pm 7.406e-4$ | $0.943 \pm 3.666e-3$ | $0.808 \pm 2.176e-4$ | $0.94 \pm 8.943e-4$ |
| medgemma Trend +TFM lstm | $0.772 \pm 9.801e-4$ | $0.94 \pm 1.071e-2$ | $0.779 \pm 2.408e-3$ | $0.928 \pm 2.555e-3$ |
| medgemma Trend +TFM mlp | $0.768 \pm 4.294e-3$ | $0.945 \pm 3.258e-3$ | $0.781 \pm 1.895e-3$ | $0.934 \pm 1.827e-3$ |
| medgemma Trend +TFM patchtsmixer | $0.75 \pm 6.952e-4$ | $0.943 \pm 4.219e-3$ | $0.769 \pm 1.597e-3$ | $0.927 \pm 8.167e-3$ |
| medgemma Trend +TFM timemixer | $0.764 \pm 2.796e-3$ | $0.94 \pm 5.22e-3$ | $0.776 \pm 1.255e-3$ | $0.934 \pm 2.515e-3$ |
| medgemma Trend lstm | $0.775 \pm 2.571e-3$ | $0.945 \pm 4.016e-3$ | $0.782 \pm 1.714e-3$ | $0.94 \pm 3.073e-3$ |
| medgemma Trend mlp | $0.777 \pm 2.475e-3$ | $0.944 \pm 4.09e-3$ | $0.782 \pm 2.611e-3$ | $0.941 \pm 2.659e-3$ |
| medgemma Trend patchtsmixer | $0.774 \pm 3.23e-3$ | $0.945 \pm 1.613e-3$ | $0.78 \pm 2.85e-3$ | $0.94 \pm 3.268e-3$ |
| medgemma Trend timemixer | $0.777 \pm 3.206e-3$ | $0.948 \pm 3.016e-3$ | $0.782 \pm 2.468e-3$ | $0.942 \pm 4.638e-3$ |
| medgemma zero_shot +TFM lstm | $0.805 \pm 1.895e-3$ | $0.922 \pm 1.735e-3$ | $0.807 \pm 4.661e-4$ | $0.924 \pm 3.651e-3$ |
| medgemma zero_shot +TFM mlp | $0.804 \pm 9.958e-4$ | $0.929 \pm 1.815e-3$ | $0.807 \pm 6.479e-4$ | $0.929 \pm 3.858e-3$ |
| medgemma zero_shot +TFM patchtsmixer | $0.785 \pm 1.413e-3$ | $0.914 \pm 4.07e-3$ | $0.794 \pm 2.091e-3$ | $0.917 \pm 3.716e-3$ |
| medgemma zero_shot +TFM timemixer | $0.792 \pm 8.399e-3$ | $0.918 \pm 5.264e-3$ | $0.803 \pm 2.323e-3$ | $0.919 \pm 1.728e-3$ |
| medgemma zero_shot lstm | $0.805 \pm 9.658e-4$ | $0.927 \pm 2.208e-3$ | $0.807 \pm 7.651e-4$ | $0.927 \pm 1.444e-3$ |
| medgemma zero_shot mlp | $0.807 \pm 7.138e-4$ | $0.931 \pm 2.173e-3$ | $0.807 \pm 9.693e-4$ | $0.926 \pm 1.236e-3$ |
| medgemma zero_shot patchtsmixer | $0.805 \pm 1.884e-4$ | $0.931 \pm 2.154e-3$ | $0.804 \pm 2.671e-3$ | $0.926 \pm 8.539e-4$ |
| medgemma zero_shot timemixer | $0.809 \pm 7.505e-4$ | $0.935 \pm 1.256e-3$ | $0.805 \pm 1.626e-3$ | $0.928 \pm 3.542e-3$ |
| No_sum_zero_shot zero_shot +TFM lstm | $0.83 \pm 2.291e-3$ | $0.953 \pm 3.83e-3$ | $0.789 \pm 2.903e-3$ | $0.909 \pm 7.136e-4$ |
| No_sum_zero_shot zero_shot +TFM patchtsmixer | $0.812 \pm 0e0$ | $0.933 \pm 0e0$ | $0.773 \pm 0e0$ | $0.918 \pm 0e0$ |
| No_sum_zero_shot lstm | $0.829 \pm 1.474e-3$ | $0.955 \pm 2.431e-3$ | $0.813 \pm 2.133e-3$ | $0.943 \pm 3.385e-3$ |
| No_sum_zero_shot mlp | $0.829 \pm 7.801e-4$ | $0.952 \pm 1.673e-3$ | $0.814 \pm 2.352e-3$ | $0.943 \pm 1.866e-3$ |
| No_sum_zero_shot patchtsmixer | $0.832 \pm 1.151e-3$ | $0.958 \pm 3.689e-3$ | $0.816 \pm 1.81e-3$ | $0.945 \pm 3.023e-3$ |
| No_sum_zero_shot timemixer | $0.835 \pm 1.135e-3$ | $0.966 \pm 2.589e-3$ | $0.821 \pm 2.119e-3$ | $0.947 \pm 4.018e-3$ |
| right lstm | $0.725 \pm 2.011e-2$ | $0.769 \pm 9.362e-2$ | $0.526 \pm 1.089e-2$ | $0.41 \pm 1.39e-2$ |
| right mlp | $0.739 \pm 8.853e-4$ | $0.882 \pm 9.386e-3$ | $0.601 \pm 6.53e-3$ | $0.517 \pm 8.03e-3$ |
| right patchtsmixer | $0.578 \pm 2.89e-2$ | $0.529 \pm 5.019e-2$ | $0.346 \pm 2.33e-2$ | $0.234 \pm 2.131e-2$ |
| right timemixer | $0.685 \pm 9.987e-3$ | $0.722 \pm 5.086e-2$ | $0.37 \pm 5.214e-2$ | $0.256 \pm 4.913e-2$ |

Table 29: Cross-site transfer results - Drug. (part 2/2)

| Method | hirid → hirid | | mimic → mimic | | ppicu → ppicu | |
|---|---|---|---|---|---|---|
| | f1_micro | recall_micro | f1_micro | recall_micro | f1_micro | recall_micro |
| Llama-3.1 CoT +TFM lstm | $0.872 \pm 2.001e-3$ | $0.92 \pm 8.798e-4$ | $0.895 \pm 2.53e-3$ | $0.936 \pm 1.722e-3$ | $0.847 \pm 7.912e-4$ | $0.917 \pm 6.809e-4$ |
| Llama-3.1 CoT +TFM mlp | $0.881 \pm 2.75e-3$ | $0.926 \pm 8.832e-4$ | $0.903 \pm 1.691e-3$ | $0.94 \pm 4.133e-4$ | $0.857 \pm 1.235e-3$ | $0.925 \pm 6.144e-4$ |
| Llama-3.1 CoT +TFM patchtsmixer | $0.857 \pm 6.165e-3$ | $0.919 \pm 5.505e-4$ | $0.892 \pm 1.797e-3$ | $0.938 \pm 2.665e-4$ | $0.839 \pm 7.46e-3$ | $0.922 \pm 1.361e-3$ |
| Llama-3.1 CoT +TFM timemixer | $0.856 \pm 6.02e-3$ | $0.923 \pm 1.165e-3$ | $0.87 \pm 1.593e-3$ | $0.935 \pm 8.472e-4$ | $0.811 \pm 3.494e-3$ | $0.928 \pm 1.361e-3$ |
| Llama-3.1 CoT lstm | $0.846 \pm 0e0$ | $0.928 \pm 0e0$ | $0.881 \pm 0e0$ | $0.946 \pm 0e0$ | $0.831 \pm 0e0$ | $0.931 \pm 0e0$ |
| Llama-3.1 CoT mlp | $0.863 \pm 0e0$ | $0.925 \pm 0e0$ | $0.89 \pm 0e0$ | $0.944 \pm 0e0$ | $0.847 \pm 0e0$ | $0.924 \pm 0e0$ |
| Llama-3.1 CoT patchtsmixer | $0.845 \pm 0e0$ | $0.927 \pm 0e0$ | $0.883 \pm 0e0$ | $0.947 \pm 0e0$ | $0.838 \pm 0e0$ | $0.93 \pm 0e0$ |
| Llama-3.1 CoT timemixer | $0.844 \pm 0e0$ | $0.927 \pm 0e0$ | $0.873 \pm 0e0$ | $0.948 \pm 0e0$ | $0.826 \pm 0e0$ | $0.935 \pm 0e0$ |
| Llama-3.1 ICD +TFM lstm | $0.873 \pm 2.458e-4$ | $0.922 \pm 1.169e-3$ | $0.889 \pm 2.848e-3$ | $0.936 \pm 1.269e-3$ | $0.83 \pm 2.191e-3$ | $0.919 \pm 1.804e-3$ |
| Llama-3.1 ICD +TFM mlp | $0.877 \pm 2.737e-3$ | $0.926 \pm 7.136e-4$ | $0.899 \pm 2.145e-3$ | $0.94 \pm 9.086e-4$ | $0.848 \pm 4.533e-3$ | $0.923 \pm 7.047e-4$ |
| Llama-3.1 ICD +TFM patchtsmixer | $0.858 \pm 4.406e-3$ | $0.92 \pm 1.552e-3$ | $0.886 \pm 1.498e-3$ | $0.939 \pm 1.021e-3$ | $0.822 \pm 1.137e-3$ | $0.925 \pm 1.746e-3$ |
| Llama-3.1 ICD +TFM timemixer | $0.851 \pm 4.439e-3$ | $0.923 \pm 9.093e-4$ | $0.861 \pm 2.613e-3$ | $0.936 \pm 1.537e-4$ | $0.811 \pm 6.069e-3$ | $0.93 \pm 7.52e-4$ |
| Llama-3.1 ICD lstm | $0.841 \pm 0e0$ | $0.931 \pm 0e0$ | $0.872 \pm 0e0$ | $0.945 \pm 0e0$ | $0.811 \pm 0e0$ | $0.936 \pm 0e0$ |
| Llama-3.1 ICD mlp | $0.856 \pm 0e0$ | $0.924 \pm 0e0$ | $0.881 \pm 0e0$ | $0.945 \pm 0e0$ | $0.826 \pm 0e0$ | $0.93 \pm 0e0$ |
| Llama-3.1 ICD patchtsmixer | $0.841 \pm 0e0$ | $0.925 \pm 0e0$ | $0.875 \pm 0e0$ | $0.945 \pm 0e0$ | $0.816 \pm 0e0$ | $0.936 \pm 0e0$ |
| Llama-3.1 ICD timemixer | $0.837 \pm 0e0$ | $0.927 \pm 0e0$ | $0.863 \pm 0e0$ | $0.943 \pm 0e0$ | $0.805 \pm 0e0$ | $0.942 \pm 0e0$ |
| Llama-3.1 Trend +TFM lstm | $0.878 \pm 3.433e-3$ | $0.922 \pm 1.257e-3$ | $0.896 \pm 6.165e-4$ | $0.937 \pm 6.24e-4$ | $0.845 \pm 2.67e-3$ | $0.914 \pm 8.072e-4$ |
| Llama-3.1 Trend +TFM mlp | $0.88 \pm 1.597e-3$ | $0.925 \pm 6.38e-4$ | $0.9 \pm 1.567e-3$ | $0.941 \pm 7.65e-4$ | $0.858 \pm 1.94e-3$ | $0.922 \pm 3.614e-4$ |
| Llama-3.1 Trend +TFM patchtsmixer | $0.844 \pm 2.049e-2$ | $0.92 \pm 2.313e-3$ | $0.893 \pm 2.192e-3$ | $0.939 \pm 1.601e-4$ | $0.845 \pm 6.878e-3$ | $0.918 \pm 6.352e-4$ |
| Llama-3.1 Trend +TFM timemixer | $0.848 \pm 2.145e-2$ | $0.929 \pm 1.162e-2$ | $0.875 \pm 4.022e-3$ | $0.937 \pm 2.89e-3$ | $0.819 \pm 1.003e-2$ | $0.926 \pm 1.526e-3$ |
| Llama-3.1 Trend lstm | $0.844 \pm 0e0$ | $0.928 \pm 0e0$ | $0.88 \pm 0e0$ | $0.943 \pm 0e0$ | $0.824 \pm 0e0$ | $0.932 \pm 0e0$ |
| Llama-3.1 Trend mlp | $0.861 \pm 0e0$ | $0.926 \pm 0e0$ | $0.892 \pm 0e0$ | $0.941 \pm 0e0$ | $0.847 \pm 0e0$ | $0.923 \pm 0e0$ |
| Llama-3.1 Trend patchtsmixer | $0.845 \pm 0e0$ | $0.924 \pm 0e0$ | $0.886 \pm 0e0$ | $0.942 \pm 0e0$ | $0.838 \pm 0e0$ | $0.929 \pm 0e0$ |
| Llama-3.1 Trend timemixer | $0.841 \pm 0e0$ | $0.922 \pm 0e0$ | $0.875 \pm 0e0$ | $0.944 \pm 0e0$ | $0.8 \pm 0e0$ | $0.94 \pm 0e0$ |
| Llama-3.1 zero_shot +TFM lstm | $0.883 \pm 2.982e-3$ | $0.924 \pm 7.996e-4$ | $0.895 \pm 1.667e-4$ | $0.938 \pm 1.127e-3$ | $0.849 \pm 1.869e-3$ | $0.919 \pm 1.553e-3$ |
| Llama-3.1 zero_shot +TFM mlp | $0.881 \pm 1.58e-3$ | $0.928 \pm 7.543e-4$ | $0.904 \pm 7.637e-4$ | $0.941 \pm 3.35e-4$ | $0.859 \pm 2.73e-3$ | $0.927 \pm 7.239e-4$ |
| Llama-3.1 zero_shot +TFM patchtsmixer | $0.861 \pm 2.916e-3$ | $0.922 \pm 2.016e-3$ | $0.891 \pm 9.815e-4$ | $0.94 \pm 7.253e-4$ | $0.842 \pm 1.746e-3$ | $0.921 \pm 1.172e-3$ |
| Llama-3.1 zero_shot +TFM timemixer | $0.863 \pm 1.131e-3$ | $0.923 \pm 7.984e-4$ | $0.881 \pm 2.739e-3$ | $0.939 \pm 7.721e-4$ | $0.816 \pm 8.775e-3$ | $0.929 \pm 1.99e-3$ |
| Llama-3.1 zero_shot lstm | $0.854 \pm 0e0$ | $0.929 \pm 0e0$ | $0.883 \pm 0e0$ | $0.944 \pm 0e0$ | $0.839 \pm 0e0$ | $0.931 \pm 0e0$ |
| Llama-3.1 zero_shot mlp | $0.867 \pm 0e0$ | $0.929 \pm 0e0$ | $0.891 \pm 0e0$ | $0.944 \pm 0e0$ | $0.847 \pm 0e0$ | $0.931 \pm 0e0$ |
| Llama-3.1 zero_shot patchtsmixer | $0.856 \pm 0e0$ | $0.927 \pm 0e0$ | $0.884 \pm 0e0$ | $0.945 \pm 0e0$ | $0.841 \pm 0e0$ | $0.933 \pm 0e0$ |
| Llama-3.1 zero_shot timemixer | $0.852 \pm 0e0$ | $0.927 \pm 0e0$ | $0.875 \pm 0e0$ | $0.944 \pm 0e0$ | $0.826 \pm 0e0$ | $0.936 \pm 0e0$ |
| TFM lstm | $0.85 \pm 8.96e-4$ | $0.924 \pm 1.793e-3$ | $0.877 \pm 1.198e-3$ | $0.939 \pm 9.594e-4$ | $0.83 \pm 2.181e-3$ | $0.923 \pm 2.089e-3$ |
| TFM mlp | $0.852 \pm 1.475e-3$ | $0.924 \pm 1.063e-3$ | $0.879 \pm 8.396e-4$ | $0.937 \pm 1.356e-3$ | $0.831 \pm 1.002e-3$ | $0.925 \pm 7.676e-4$ |
| TFM patchtsmixer | $0.842 \pm 2.252e-3$ | $0.925 \pm 2.263e-3$ | $0.869 \pm 2.4e-3$ | $0.94 \pm 1.253e-3$ | $0.82 \pm 5.108e-3$ | $0.925 \pm 1.576e-3$ |
| TFM timemixer | $0.834 \pm 1.109e-2$ | $0.927 \pm 3.234e-3$ | $0.857 \pm 7.105e-3$ | $0.941 \pm 2.644e-3$ | $0.808 \pm 4.268e-3$ | $0.933 \pm 1.69e-3$ |
| TSDE lstm | $0.869 \pm 1.638e-3$ | $0.924 \pm 2.297e-3$ | $0.902 \pm 4.602e-4$ | $0.94 \pm 3.792e-4$ | $0.857 \pm 2.576e-3$ | $0.924 \pm 2.547e-3$ |
| TSDE mlp | $0.875 \pm 4.566e-4$ | $0.928 \pm 5.67e-4$ | $0.908 \pm 1.662e-4$ | $0.941 \pm 1.076e-3$ | $0.874 \pm 1.086e-3$ | $0.93 \pm 5.558e-4$ |
| TSDE patchtsmixer | $0.741 \pm 5e-6$ | $0.902 \pm 6.5e-5$ | $0.763 \pm 1.915e-3$ | $0.899 \pm 2.885e-3$ | $0.733 \pm 5e-6$ | $0.870 \pm 0e0$ |
| TSDE timemixer | $0.744 \pm 3.397e-3$ | $0.913 \pm 1.622e-3$ | $0.763 \pm 3.499e-4$ | $0.919 \pm 1.35e-4$ | $0.734 \pm 7.172e-4$ | $0.939 \pm 3.318e-3$ |
| gemini-2.0-flash CoT +TFM lstm | $0.88 \pm 2.886e-3$ | $0.924 \pm 8.207e-4$ | $0.903 \pm 1.52e-3$ | $0.935 \pm 1.495e-3$ | $0.85 \pm 2.026e-3$ | $0.913 \pm 1.696e-3$ |
| gemini-2.0-flash CoT +TFM mlp | $0.886 \pm 2.639e-3$ | $0.928 \pm 5.889e-4$ | $0.906 \pm 9.991e-4$ | $0.941 \pm 3.564e-4$ | $0.861 \pm 1.612e-3$ | $0.921 \pm 7.276e-4$ |
| gemini-2.0-flash CoT +TFM patchtsmixer | $0.864 \pm 1.437e-3$ | $0.919 \pm 8.718e-4$ | $0.897 \pm 9.968e-4$ | $0.94 \pm 9.819e-4$ | $0.84 \pm 1.2e-2$ | $0.919 \pm 2.696e-3$ |
| gemini-2.0-flash CoT +TFM timemixer | $0.859 \pm 4.816e-3$ | $0.924 \pm 9.548e-4$ | $0.878 \pm 5.403e-3$ | $0.936 \pm 2.136e-3$ | $0.814 \pm 8.623e-3$ | $0.929 \pm 4.58e-4$ |
| gemini-2.0-flash CoT lstm | $0.854 \pm 6.322e-3$ | $0.923 \pm 2.85e-3$ | $0.893 \pm 3.543e-3$ | $0.944 \pm 1.556e-3$ | $0.84 \pm 6.074e-3$ | $0.923 \pm 5.77e-3$ |
| gemini-2.0-flash CoT mlp | $0.865 \pm 0e0$ | $0.924 \pm 0e0$ | $0.896 \pm 0e0$ | $0.944 \pm 0e0$ | $0.854 \pm 0e0$ | $0.921 \pm 0e0$ |
| gemini-2.0-flash CoT patchtsmixer | $0.849 \pm 0e0$ | $0.929 \pm 0e0$ | $0.891 \pm 0e0$ | $0.946 \pm 0e0$ | $0.845 \pm 0e0$ | $0.924 \pm 0e0$ |
| gemini-2.0-flash CoT timemixer | $0.844 \pm 0e0$ | $0.923 \pm 0e0$ | $0.878 \pm 0e0$ | $0.947 \pm 0e0$ | $0.828 \pm 0e0$ | $0.928 \pm 0e0$ |
| gemini-2.0-flash ICD +TFM lstm | $0.882 \pm 4.025e-3$ | $0.923 \pm 1.314e-3$ | $0.9 \pm 5.27e-3$ | $0.935 \pm 1.442e-3$ | $0.85 \pm 1.027e-3$ | $0.917 \pm 1.417e-3$ |
| gemini-2.0-flash ICD +TFM mlp | $0.88 \pm 2.633e-3$ | $0.926 \pm 1.237e-3$ | $0.905 \pm 2.309e-3$ | $0.941 \pm 6.612e-4$ | $0.859 \pm 1.837e-3$ | $0.923 \pm 1.311e-4$ |
| gemini-2.0-flash ICD +TFM patchtsmixer | $0.831 \pm 2.852e-2$ | $0.923 \pm 5.748e-3$ | $0.894 \pm 2.291e-3$ | $0.94 \pm 9.271e-4$ | $0.844 \pm 6.047e-3$ | $0.92 \pm 1.276e-3$ |
| gemini-2.0-flash ICD +TFM timemixer | $0.864 \pm 2.421e-3$ | $0.922 \pm 4.196e-4$ | $0.876 \pm 4.371e-3$ | $0.936 \pm 7.193e-4$ | $0.824 \pm 3.774e-3$ | $0.926 \pm 1.046e-3$ |
| gemini-2.0-flash ICD lstm | $0.851 \pm 7.425e-3$ | $0.923 \pm 1.202e-3$ | $0.887 \pm 7.99e-4$ | $0.945 \pm 7.212e-4$ | $0.838 \pm 7.34e-3$ | $0.925 \pm 4.172e-3$ |
| gemini-2.0-flash ICD mlp | $0.863 \pm 0e0$ | $0.924 \pm 0e0$ | $0.895 \pm 0e0$ | $0.944 \pm 0e0$ | $0.853 \pm 0e0$ | $0.924 \pm 0e0$ |
| gemini-2.0-flash ICD patchtsmixer | $0.847 \pm 0e0$ | $0.925 \pm 0e0$ | $0.888 \pm 0e0$ | $0.945 \pm 0e0$ | $0.841 \pm 0e0$ | $0.928 \pm 0e0$ |
| gemini-2.0-flash ICD timemixer | $0.843 \pm 0e0$ | $0.927 \pm 0e0$ | $0.875 \pm 0e0$ | $0.946 \pm 0e0$ | $0.83 \pm 0e0$ | $0.93 \pm 0e0$ |
| gemini-2.0-flash Trend +TFM lstm | $0.88 \pm 1.391e-3$ | $0.923 \pm 1.364e-3$ | $0.905 \pm 2.185e-3$ | $0.936 \pm 9.111e-4$ | $0.867 \pm 3.576e-4$ | $0.914 \pm 1.105e-3$ |
| gemini-2.0-flash Trend +TFM mlp | $0.883 \pm 1.555e-3$ | $0.928 \pm 7e-5$ | $0.909 \pm 9.385e-4$ | $0.94 \pm 2.974e-4$ | $0.875 \pm 1.33e-3$ | $0.925 \pm 1.106e-3$ |
| gemini-2.0-flash Trend +TFM patchtsmixer | $0.862 \pm 5.485e-3$ | $0.918 \pm 1.886e-3$ | $0.897 \pm 1.479e-3$ | $0.94 \pm 2.685e-4$ | $0.86 \pm 2.408e-3$ | $0.918 \pm 3.602e-4$ |
| gemini-2.0-flash Trend +TFM timemixer | $0.864 \pm 3.814e-3$ | $0.922 \pm 2.445e-3$ | $0.885 \pm 4.067e-3$ | $0.937 \pm 1.064e-3$ | $0.837 \pm 3.347e-3$ | $0.923 \pm 1.102e-3$ |
| gemini-2.0-flash Trend lstm | $0.859 \pm 9.192e-3$ | $0.922 \pm 2.001e-3$ | $0.897 \pm 3.486e-3$ | $0.94 \pm 1.881e-3$ | $0.858 \pm 7.637e-3$ | $0.921 \pm 1.57e-3$ |
| gemini-2.0-flash Trend mlp | $0.869 \pm 0e0$ | $0.923 \pm 0e0$ | $0.904 \pm 0e0$ | $0.94 \pm 0e0$ | $0.87 \pm 0e0$ | $0.924 \pm 0e0$ |
| gemini-2.0-flash Trend patchtsmixer | $0.854 \pm 0e0$ | $0.92 \pm 0e0$ | $0.895 \pm 0e0$ | $0.942 \pm 0e0$ | $0.861 \pm 0e0$ | $0.923 \pm 0e0$ |
| gemini-2.0-flash Trend timemixer | $0.847 \pm 0e0$ | $0.921 \pm 0e0$ | $0.884 \pm 0e0$ | $0.944 \pm 0e0$ | $0.845 \pm 0e0$ | $0.927 \pm 0e0$ |
| gemini-2.0-flash zero_shot +TFM lstm | $0.888 \pm 6.233e-3$ | $0.925 \pm 3.482e-3$ | $0.9 \pm 4.405e-3$ | $0.938 \pm 9.188e-4$ | $0.864 \pm 1.14e-3$ | $0.915 \pm 1.201e-3$ |
| gemini-2.0-flash zero_shot +TFM mlp | $0.886 \pm 2.223e-3$ | $0.927 \pm 6.502e-4$ | $0.906 \pm 2.027e-3$ | $0.942 \pm 6.099e-4$ | $0.87 \pm 5.372e-4$ | $0.924 \pm 1.206e-4$ |
| gemini-2.0-flash zero_shot +TFM patchtsmixer | $0.87 \pm 5.808e-3$ | $0.918 \pm 3.325e-3$ | $0.897 \pm 1.201e-3$ | $0.941 \pm 5.901e-4$ | $0.85 \pm 3.66e-3$ | $0.919 \pm 9.218e-4$ |
| gemini-2.0-flash zero_shot +TFM timemixer | $0.864 \pm 1.514e-3$ | $0.921 \pm 2.516e-3$ | $0.879 \pm 1.897e-3$ | $0.938 \pm 2.642e-3$ | $0.828 \pm 2.471e-3$ | $0.925 \pm 6.238e-4$ |
| gemini-2.0-flash zero_shot lstm | $0.864 \pm 1.317e-2$ | $0.926 \pm 1.541e-3$ | $0.893 \pm 7.425e-4$ | $0.942 \pm 4.455e-4$ | $0.846 \pm 9.744e-3$ | $0.924 \pm 2.984e-3$ |
| gemini-2.0-flash zero_shot mlp | $0.874 \pm 0e0$ | $0.928 \pm 0e0$ | $0.9 \pm 0e0$ | $0.942 \pm 0e0$ | $0.861 \pm 0e0$ | $0.926 \pm 0e0$ |
| gemini-2.0-flash zero_shot patchtsmixer | $0.855 \pm 0e0$ | $0.926 \pm 0e0$ | $0.894 \pm 0e0$ | $0.942 \pm 0e0$ | $0.848 \pm 0e0$ | $0.927 \pm 0e0$ |
| gemini-2.0-flash zero_shot timemixer | $0.853 \pm 0e0$ | $0.923 \pm 0e0$ | $0.881 \pm 0e0$ | $0.944 \pm 0e0$ | $0.834 \pm 0e0$ | $0.93 \pm 0e0$ |
| interp lstm | $0.862 \pm 4.524e-4$ | $0.863 \pm 7.691e-3$ | $0.897 \pm 1.071e-3$ | $0.905 \pm 6.778e-3$ | $0.866 \pm 1.449e-3$ | $0.872 \pm 7.24e-3$ |
| interp mlp | $0.864 \pm 3.877e-4$ | $0.867 \pm 2.024e-3$ | $0.898 \pm 4.303e-4$ | $0.908 \pm 1.671e-3$ | $0.872 \pm 6.682e-4$ | $0.873 \pm 2.105e-3$ |
| interp patchtsmixer | $0.86 \pm 4.162e-3$ | $0.853 \pm 1.984e-2$ | $0.891 \pm 1.864e-3$ | $0.893 \pm 6.767e-3$ | $0.859 \pm 3.309e-3$ | $0.847 \pm 1.608e-2$ |
| interp timemixer | $0.864 \pm 1.179e-3$ | $0.868 \pm 4.322e-3$ | $0.895 \pm 1.871e-4$ | $0.907 \pm 1.987e-3$ | $0.865 \pm 2.192e-3$ | $0.864 \pm 1.057e-2$ |
| mean lstm | $0.857 \pm 4.039e-4$ | $0.859 \pm 4.395e-3$ | $0.889 \pm 5.677e-4$ | $0.893 \pm 4.161e-3$ | $0.84 \pm 1.046e-3$ | $0.833 \pm 6.055e-3$ |
| mean mlp | $0.852 \pm 7.397e-4$ | $0.853 \pm 4.183e-3$ | $0.888 \pm 2.311e-4$ | $0.894 \pm 1.562e-3$ | $0.846 \pm 1.859e-3$ | $0.844 \pm 5.339e-3$ |
| mean patchtsmixer | $0.86 \pm 1.873e-3$ | $0.857 \pm 1.018e-2$ | $0.89 \pm 3.296e-3$ | $0.886 \pm 9.38e-3$ | $0.858 \pm 3.78e-3$ | $0.847 \pm 1.71e-2$ |
| mean timemixer | $0.867 \pm 1.337e-3$ | $0.871 \pm 4.503e-3$ | $0.9 \pm 1.135e-3$ | $0.903 \pm 5.124e-3$ | $0.869 \pm 2.332e-3$ | $0.87 \pm 6.14e-3$ |
| No_sum_CoT CoT +TFM lstm | $0.899 \pm 1.882e-3$ | $0.931 \pm 9.36e-4$ | $0.913 \pm 1.828e-3$ | $0.938 \pm 7.171e-4$ | $0.887 \pm 4.298e-3$ | $0.926 \pm 1.93e-3$ |
| No_sum_CoT CoT +TFM mlp | $0.898 \pm 1.646e-4$ | $0.935 \pm 6.165e-4$ | $0.917 \pm 1.911e-3$ | $0.942 \pm 7.07e-4$ | $0.888 \pm 3.313e-3$ | $0.933 \pm 6.611e-4$ |
| No_sum_CoT CoT +TFM patchtsmixer | $0.88 \pm 4.753e-3$ | $0.923 \pm 1.892e-3$ | $0.907 \pm 1.434e-3$ | $0.942 \pm 6.495e-4$ | $0.871 \pm 2.433e-3$ | $0.924 \pm 1.167e-3$ |
| No_sum_CoT CoT +TFM timemixer | $0.874 \pm 5.09e-3$ | $0.925 \pm 5.211e-3$ | $0.895 \pm 4.736e-3$ | $0.941 \pm 5.895e-4$ | $0.855 \pm 2.225e-3$ | $0.924 \pm 1.743e-3$ |
| No_sum_CoT lstm | $0.878 \pm 3.551e-4$ | $0.929 \pm 9.75e-4$ | $0.905 \pm 9.539e-5$ | $0.942 \pm 3.9e-4$ | $0.873 \pm 2.108e-4$ | $0.932 \pm 7.855e-4$ |
| No_sum_CoT mlp | $0.888 \pm 0e0$ | $0.933 \pm 0e0$ | $0.914 \pm 0e0$ | $0.942 \pm 0e0$ | $0.884 \pm 0e0$ | $0.933 \pm 0e0$ |
| No_sum_CoT patchtsmixer | $0.877 \pm 1.922e-4$ | $0.931 \pm 8.355e-4$ | $0.906 \pm 3.812e-4$ | $0.944 \pm 6.991e-4$ | $0.876 \pm 4.941e-4$ | $0.934 \pm 1.795e-3$ |
| No_sum_CoT timemixer | $0.871 \pm 1.175e-3$ | $0.931 \pm 1.531e-4$ | $0.893 \pm 1.317e-3$ | $0.944 \pm 1.362e-3$ | $0.867 \pm 5.472e-4$ | $0.926 \pm 1.09e-3$ |
| No_sum_ICD ICD +TFM lstm | $0.899 \pm 6.612e-4$ | $0.931 \pm 3.923e-4$ | $0.916 \pm 1.685e-3$ | $0.939 \pm 5.954e-4$ | $0.881 \pm 5.792e-3$ | $0.924 \pm 3.529e-3$ |
| No_sum_ICD ICD +TFM mlp | $0.899 \pm 2.172e-3$ | $0.935 \pm 1.401e-3$ | $0.916 \pm 1.139e-3$ | $0.943 \pm 3.055e-4$ | $0.885 \pm 2.285e-3$ | $0.932 \pm 1.292e-3$ |
| No_sum_ICD ICD +TFM patchtsmixer | $0.882 \pm 3.476e-3$ | $0.923 \pm 1.524e-3$ | $0.906 \pm 1.865e-3$ | $0.941 \pm 5.905e-4$ | $0.871 \pm 2.455e-3$ | $0.925 \pm 1.528e-3$ |
| No_sum_ICD ICD +TFM timemixer | $0.877 \pm 2.363e-3$ | $0.925 \pm 3.242e-3$ | $0.898 \pm 2.777e-3$ | $0.941 \pm 6.577e-4$ | $0.855 \pm 2.207e-3$ | $0.923 \pm 6.616e-4$ |
| No_sum_ICD lstm | $0.877 \pm 4.917e-4$ | $0.93 \pm 7.455e-4$ | $0.903 \pm 4.43e-4$ | $0.944 \pm 6.883e-4$ | $0.871 \pm 2.558e-4$ | $0.931 \pm 1.154e-3$ |
| No_sum_ICD mlp | $0.886 \pm 0e0$ | $0.934 \pm 0e0$ | $0.911 \pm 0e0$ | $0.942 \pm 0e0$ | $0.88 \pm 0e0$ | $0.933 \pm 0e0$ |
| No_sum_ICD patchtsmixer | $0.875 \pm 7.904e-4$ | $0.93 \pm 1.542e-3$ | $0.904 \pm 7.81e-5$ | $0.943 \pm 8.528e-4$ | $0.871 \pm 6.478e-4$ | $0.935 \pm 1.466e-3$ |
| No_sum_ICD timemixer | $0.869 \pm 7.434e-4$ | $0.932 \pm 3.365e-4$ | $0.9 \pm 3.281e-4$ | $0.943 \pm 1.146e-3$ | $0.862 \pm 1.177e-3$ | $0.928 \pm 1.409e-3$ |
| No_sum_Trend Trend +TFM lstm | $0.897 \pm 2.179e-3$ | $0.93 \pm 8.698e-4$ | $0.915 \pm 9.572e-4$ | $0.937 \pm 1.264e-3$ | $0.885 \pm 1.388e-3$ | $0.924 \pm 1.125e-3$ |
| No_sum_Trend Trend +TFM mlp | $0.901 \pm 5.023e-3$ | $0.934 \pm 1.867e-3$ | $0.917 \pm 2.971e-3$ | $0.943 \pm 2.888e-4$ | $0.887 \pm 6e-5$ | $0.933 \pm 4.277e-4$ |
| No_sum_Trend Trend +TFM patchtsmixer | $0.881 \pm 5.503e-3$ | $0.923 \pm 1.202e-3$ | $0.907 \pm 8.208e-4$ | $0.941 \pm 8.731e-4$ | $0.877 \pm 2.42e-3$ | $0.925 \pm 1.415e-3$ |
| No_sum_Trend Trend +TFM timemixer | $0.875 \pm 6.816e-3$ | $0.924 \pm 2.757e-3$ | $0.892 \pm 3.244e-3$ | $0.94 \pm 8.564e-4$ | $0.856 \pm 1.993e-3$ | $0.924 \pm 1.675e-3$ |

Table 30: In-distribution results - Lab. (part 1/2)

| Method | hirid → hirid | | mimic → mimic | | ppicu → ppicu | |
|---|---|---|---|---|---|---|
| | f1_micro | recall_micro | f1_micro | recall_micro | f1_micro | recall_micro |
| No_sum_Trend lstm | $0.878 \pm 2.658e-4$ | $0.928 \pm 6.463e-4$ | $0.906 \pm 2.312e-4$ | $0.943 \pm 6.671e-4$ | $0.875 \pm 1.3e-4$ | $0.932 \pm 1.242e-4$ |
| No_sum_Trend mlp | $0.895 \pm 0e0$ | $0.933 \pm 0e0$ | $0.914 \pm 0e0$ | $0.942 \pm 0e0$ | $0.883 \pm 0e0$ | $0.933 \pm 0e0$ |
| No_sum_Trend patchtsmixer | $0.879 \pm 7.515e-4$ | $0.929 \pm 1.331e-3$ | $0.905 \pm 2.403e-4$ | $0.942 \pm 7.504e-4$ | $0.875 \pm 1.472e-3$ | $0.936 \pm 2.444e-4$ |
| No_sum_Trend timemixer | $0.873 \pm 3.395e-4$ | $0.93 \pm 4.888e-4$ | $0.896 \pm 8.228e-4$ | $0.942 \pm 6.2e-4$ | $0.864 \pm 3.47e-4$ | $0.926 \pm 1.854e-3$ |
| medgemma CoT +TFM lstm | $0.889 \pm 5.13e-3$ | $0.926 \pm 4.828e-4$ | $0.904 \pm 2.248e-3$ | $0.936 \pm 1.293e-3$ | $0.861 \pm 2.565e-3$ | $0.916 \pm 1.212e-3$ |
| medgemma CoT +TFM mlp | $0.889 \pm 2.26e-3$ | $0.93 \pm 4.419e-4$ | $0.909 \pm 1.791e-3$ | $0.942 \pm 3.538e-4$ | $0.869 \pm 7.267e-4$ | $0.927 \pm 7.617e-4$ |
| medgemma CoT +TFM patchtsmixer | $0.843 \pm 2.567e-2$ | $0.921 \pm 4.432e-3$ | $0.899 \pm 2.595e-3$ | $0.941 \pm 1.213e-3$ | $0.844 \pm 1.187e-2$ | $0.922 \pm 8.976e-4$ |
| medgemma CoT +TFM timemixer | $0.87 \pm 2.113e-3$ | $0.923 \pm 7.391e-4$ | $0.883 \pm 7.948e-3$ | $0.938 \pm 1.643e-3$ | $0.833 \pm 2.474e-3$ | $0.926 \pm 8.26e-4$ |
| medgemma CoT lstm | $0.868 \pm 8.252e-3$ | $0.925 \pm 2.256e-3$ | $0.896 \pm 4.759e-3$ | $0.944 \pm 2.836e-3$ | $0.851 \pm 0e0$ | $0.928 \pm 0e0$ |
| medgemma CoT mlp | $0.879 \pm 0e0$ | $0.929 \pm 0e0$ | $0.903 \pm 0e0$ | $0.943 \pm 0e0$ | $0.864 \pm 0e0$ | $0.928 \pm 0e0$ |
| medgemma CoT patchtsmixer | $0.866 \pm 0e0$ | $0.925 \pm 0e0$ | $0.895 \pm 0e0$ | $0.946 \pm 0e0$ | $0.855 \pm 0e0$ | $0.932 \pm 0e0$ |
| medgemma CoT timemixer | $0.86 \pm 0e0$ | $0.927 \pm 0e0$ | $0.888 \pm 0e0$ | $0.946 \pm 0e0$ | $0.84 \pm 0e0$ | $0.927 \pm 0e0$ |
| medgemma ICD +TFM lstm | $0.877 \pm 3.05e-3$ | $0.923 \pm 1.422e-3$ | $0.895 \pm 4.337e-3$ | $0.937 \pm 1.362e-3$ | $0.844 \pm 1.449e-3$ | $0.912 \pm 1.579e-3$ |
| medgemma ICD +TFM mlp | $0.884 \pm 3.329e-3$ | $0.928 \pm 1.335e-3$ | $0.906 \pm 2.99e-3$ | $0.941 \pm 1.008e-3$ | $0.858 \pm 3.321e-3$ | $0.922 \pm 2.843e-4$ |
| medgemma ICD +TFM patchtsmixer | $0.863 \pm 8.107e-3$ | $0.918 \pm 1.423e-3$ | $0.894 \pm 1.985e-3$ | $0.942 \pm 3.724e-4$ | $0.832 \pm 1.02e-2$ | $0.923 \pm 3.201e-3$ |
| medgemma ICD +TFM timemixer | $0.862 \pm 2.257e-3$ | $0.921 \pm 8.469e-4$ | $0.876 \pm 8.269e-3$ | $0.938 \pm 2.769e-3$ | $0.811 \pm 1.756e-3$ | $0.927 \pm 8.819e-4$ |
| medgemma ICD lstm | $0.856 \pm 8.966e-3$ | $0.924 \pm 1.209e-3$ | $0.885 \pm 3.698e-3$ | $0.946 \pm 4.313e-4$ | $0.832 \pm 7.7e-3$ | $0.926 \pm 3.097e-3$ |
| medgemma ICD mlp | $0.868 \pm 0e0$ | $0.926 \pm 0e0$ | $0.895 \pm 0e0$ | $0.944 \pm 0e0$ | $0.843 \pm 0e0$ | $0.925 \pm 0e0$ |
| medgemma ICD patchtsmixer | $0.852 \pm 0e0$ | $0.92 \pm 0e0$ | $0.888 \pm 0e0$ | $0.946 \pm 0e0$ | $0.832 \pm 0e0$ | $0.931 \pm 0e0$ |
| medgemma ICD timemixer | $0.85 \pm 0e0$ | $0.922 \pm 0e0$ | $0.876 \pm 0e0$ | $0.948 \pm 0e0$ | $0.808 \pm 0e0$ | $0.936 \pm 0e0$ |
| medgemma Trend +TFM lstm | $0.88 \pm 4.772e-3$ | $0.924 \pm 5.742e-4$ | $0.903 \pm 8.193e-4$ | $0.936 \pm 5.575e-4$ | $0.861 \pm 1.843e-3$ | $0.917 \pm 2.003e-3$ |
| medgemma Trend +TFM mlp | $0.885 \pm 1.961e-3$ | $0.929 \pm 2.15e-4$ | $0.908 \pm 3.774e-4$ | $0.94 \pm 4.464e-4$ | $0.869 \pm 1.008e-3$ | $0.925 \pm 4.424e-4$ |
| medgemma Trend +TFM patchtsmixer | $0.864 \pm 1.666e-3$ | $0.919 \pm 3.332e-4$ | $0.896 \pm 2.969e-3$ | $0.939 \pm 1.372e-3$ | $0.849 \pm 4.349e-3$ | $0.919 \pm 6.379e-4$ |
| medgemma Trend +TFM timemixer | $0.861 \pm 4.282e-3$ | $0.923 \pm 1.442e-3$ | $0.886 \pm 6.317e-4$ | $0.94 \pm 1.53e-3$ | $0.825 \pm 1.56e-2$ | $0.925 \pm 3.355e-3$ |
| medgemma Trend lstm | $0.861 \pm 6.64e-3$ | $0.925 \pm 2.885e-4$ | $0.896 \pm 3.507e-3$ | $0.94 \pm 2.503e-3$ | $0.85 \pm 6.937e-3$ | $0.924 \pm 7.354e-4$ |
| medgemma Trend mlp | $0.871 \pm 0e0$ | $0.925 \pm 0e0$ | $0.9 \pm 0e0$ | $0.941 \pm 0e0$ | $0.859 \pm 0e0$ | $0.926 \pm 0e0$ |
| medgemma Trend patchtsmixer | $0.855 \pm 0e0$ | $0.924 \pm 0e0$ | $0.894 \pm 0e0$ | $0.942 \pm 0e0$ | $0.851 \pm 0e0$ | $0.925 \pm 0e0$ |
| medgemma Trend timemixer | $0.85 \pm 0e0$ | $0.927 \pm 0e0$ | $0.88 \pm 0e0$ | $0.944 \pm 0e0$ | $0.837 \pm 0e0$ | $0.928 \pm 0e0$ |
| medgemma zero_shot +TFM lstm | $0.886 \pm 1.048e-3$ | $0.923 \pm 6.421e-4$ | $0.902 \pm 2.546e-3$ | $0.937 \pm 7.805e-4$ | $0.853 \pm 5.101e-3$ | $0.915 \pm 1.43e-3$ |
| medgemma zero_shot +TFM mlp | $0.891 \pm 2.723e-3$ | $0.929 \pm 1.725e-3$ | $0.909 \pm 2.35e-3$ | $0.942 \pm 6.171e-4$ | $0.865 \pm 2.928e-3$ | $0.924 \pm 9.504e-5$ |
| medgemma zero_shot +TFM patchtsmixer | $0.866 \pm 1.713e-3$ | $0.917 \pm 3.412e-4$ | $0.896 \pm 8.122e-4$ | $0.942 \pm 3.317e-4$ | $0.843 \pm 1.467e-2$ | $0.921 \pm 2.999e-3$ |
| medgemma zero_shot +TFM timemixer | $0.847 \pm 2.482e-2$ | $0.926 \pm 7.219e-3$ | $0.88 \pm 3.121e-3$ | $0.939 \pm 1.32e-3$ | $0.828 \pm 4.5e-3$ | $0.925 \pm 1.935e-3$ |
| medgemma zero_shot lstm | $0.867 \pm 1.014e-2$ | $0.925 \pm 3.571e-3$ | $0.893 \pm 4.589e-3$ | $0.944 \pm 2.22e-3$ | $0.847 \pm 8.4e-3$ | $0.924 \pm 4.363e-3$ |
| medgemma zero_shot mlp | $0.878 \pm 0e0$ | $0.927 \pm 0e0$ | $0.9 \pm 0e0$ | $0.943 \pm 0e0$ | $0.858 \pm 0e0$ | $0.926 \pm 0e0$ |
| medgemma zero_shot patchtsmixer | $0.858 \pm 0e0$ | $0.923 \pm 0e0$ | $0.894 \pm 0e0$ | $0.945 \pm 0e0$ | $0.852 \pm 0e0$ | $0.93 \pm 0e0$ |
| medgemma zero_shot timemixer | $0.852 \pm 0e0$ | $0.924 \pm 0e0$ | $0.882 \pm 0e0$ | $0.947 \pm 0e0$ | $0.83 \pm 0e0$ | $0.931 \pm 0e0$ |
| No_sum_zero_shot zero_shot +TFM lstm | $0.899 \pm 1.292e-3$ | $0.931 \pm 1.177e-3$ | $0.915 \pm 2.725e-3$ | $0.937 \pm 3.164e-4$ | $0.884 \pm 1.782e-3$ | $0.926 \pm 1.312e-3$ |
| No_sum_zero_shot zero_shot +TFM mlp | $0.899 \pm 3.993e-4$ | $0.935 \pm 9.4e-4$ | $0.917 \pm 1.582e-3$ | $0.942 \pm 3.988e-4$ | $0.888 \pm 2.399e-3$ | $0.933 \pm 8.827e-4$ |
| No_sum_zero_shot zero_shot +TFM patchtsmixer | $0.883 \pm 1.917e-3$ | $0.923 \pm 1.105e-3$ | $0.907 \pm 7.06e-4$ | $0.941 \pm 4.992e-4$ | $0.875 \pm 2.033e-3$ | $0.925 \pm 1.18e-3$ |
| No_sum_zero_shot zero_shot +TFM timemixer | $0.874 \pm 3.213e-3$ | $0.926 \pm 4.367e-3$ | $0.892 \pm 3.519e-3$ | $0.941 \pm 7.862e-4$ | $0.853 \pm 4.922e-3$ | $0.923 \pm 1.54e-3$ |
| No_sum_zero_shot lstm | $0.877 \pm 5.33e-4$ | $0.931 \pm 1.281e-3$ | $0.905 \pm 1.286e-4$ | $0.942 \pm 2.501e-4$ | $0.873 \pm 1.518e-4$ | $0.933 \pm 7.503e-4$ |
| No_sum_zero_shot mlp | $0.891 \pm 0e0$ | $0.934 \pm 0e0$ | $0.912 \pm 0e0$ | $0.942 \pm 0e0$ | $0.881 \pm 0e0$ | $0.933 \pm 0e0$ |
| No_sum_zero_shot patchtsmixer | $0.877 \pm 5.975e-4$ | $0.93 \pm 4.858e-4$ | $0.905 \pm 2.001e-4$ | $0.943 \pm 9.239e-4$ | $0.874 \pm 8.505e-5$ | $0.936 \pm 1.778e-3$ |
| No_sum_zero_shot timemixer | $0.869 \pm 5.951e-4$ | $0.932 \pm 5.065e-4$ | $0.898 \pm 8.837e-4$ | $0.942 \pm 4.535e-4$ | $0.86 \pm 7.794e-4$ | $0.93 \pm 1.172e-3$ |
| right lstm | $0.87 \pm 8.813e-4$ | $0.881 \pm 3.199e-3$ | $0.899 \pm 4.968e-4$ | $0.908 \pm 3.32e-3$ | $0.87 \pm 1.011e-3$ | $0.882 \pm 3.608e-3$ |
| right mlp | $0.873 \pm 4.173e-4$ | $0.88 \pm 2.961e-3$ | $0.9 \pm 2.41e-4$ | $0.91 \pm 3.03e-3$ | $0.876 \pm 6.405e-4$ | $0.879 \pm 4.391e-3$ |
| right patchtsmixer | $0.863 \pm 6.473e-3$ | $0.858 \pm 2.645e-2$ | $0.89 \pm 2.222e-3$ | $0.884 \pm 9.048e-3$ | $0.858 \pm 2.851e-3$ | $0.846 \pm 1.492e-2$ |
| right timemixer | $0.874 \pm 8.987e-4$ | $0.884 \pm 3.664e-3$ | $0.899 \pm 8.577e-4$ | $0.911 \pm 6.399e-3$ | $0.869 \pm 4.859e-4$ | $0.868 \pm 2.748e-3$ |

Table 31: In-distribution results - Lab. (part 2/2)

| Method | hirid → ppicu | | mimic → ppicu | |
|---|---|---|---|---|
| | f1_micro | recall_micro | f1_micro | recall_micro |
| Llama-3.1 CoT +TFM lstm | $0.774 \pm 6.907e-4$ | $0.959 \pm 3.078e-3$ | $0.766 \pm 1.933e-3$ | $0.95 \pm 5.956e-4$ |
| Llama-3.1 CoT +TFM mlp | $0.78 \pm 2.159e-3$ | $0.956 \pm 2.175e-3$ | $0.761 \pm 1.398e-3$ | $0.956 \pm 1.442e-3$ |
| Llama-3.1 CoT +TFM patchtsmixer | $0.777 \pm 2.362e-3$ | $0.959 \pm 4.087e-3$ | $0.755 \pm 1.652e-3$ | $0.961 \pm 2.694e-3$ |
| Llama-3.1 CoT +TFM timemixer | $0.773 \pm 1.304e-3$ | $0.962 \pm 2.365e-3$ | $0.775 \pm 3.476e-3$ | $0.94 \pm 5.293e-3$ |
| Llama-3.1 CoT lstm | $0.786 \pm 0e0$ | $0.954 \pm 0e0$ | $0.775 \pm 0e0$ | $0.952 \pm 0e0$ |
| Llama-3.1 CoT mlp | $0.781 \pm 0e0$ | $0.955 \pm 0e0$ | $0.765 \pm 0e0$ | $0.951 \pm 0e0$ |
| Llama-3.1 CoT patchtsmixer | $0.785 \pm 0e0$ | $0.959 \pm 0e0$ | $0.775 \pm 0e0$ | $0.953 \pm 0e0$ |
| Llama-3.1 CoT timemixer | $0.786 \pm 0e0$ | $0.957 \pm 0e0$ | $0.78 \pm 0e0$ | $0.949 \pm 0e0$ |
| Llama-3.1 ICD +TFM lstm | $0.769 \pm 2.365e-3$ | $0.967 \pm 2.211e-3$ | $0.77 \pm 1.332e-3$ | $0.944 \pm 1.466e-3$ |
| Llama-3.1 ICD +TFM mlp | $0.776 \pm 2.135e-3$ | $0.96 \pm 2.754e-3$ | $0.761 \pm 1.713e-3$ | $0.955 \pm 1.754e-3$ |
| Llama-3.1 ICD +TFM patchtsmixer | $0.775 \pm 2.289e-3$ | $0.959 \pm 3.655e-3$ | $0.76 \pm 1.61e-3$ | $0.956 \pm 3.691e-3$ |
| Llama-3.1 ICD +TFM timemixer | $0.774 \pm 7.144e-4$ | $0.962 \pm 3.057e-4$ | $0.773 \pm 1.563e-3$ | $0.941 \pm 1.077e-3$ |
| Llama-3.1 ICD lstm | $0.788 \pm 0e0$ | $0.948 \pm 0e0$ | $0.775 \pm 0e0$ | $0.947 \pm 0e0$ |
| Llama-3.1 ICD mlp | $0.788 \pm 0e0$ | $0.947 \pm 0e0$ | $0.77 \pm 0e0$ | $0.945 \pm 0e0$ |
| Llama-3.1 ICD patchtsmixer | $0.789 \pm 0e0$ | $0.954 \pm 0e0$ | $0.776 \pm 0e0$ | $0.95 \pm 0e0$ |
| Llama-3.1 ICD timemixer | $0.787 \pm 0e0$ | $0.951 \pm 0e0$ | $0.777 \pm 0e0$ | $0.947 \pm 0e0$ |
| Llama-3.1 Trend +TFM lstm | $0.773 \pm 4.59e-3$ | $0.962 \pm 6.862e-3$ | $0.771 \pm 2.454e-3$ | $0.946 \pm 1.418e-3$ |
| Llama-3.1 Trend +TFM mlp | $0.778 \pm 1.86e-3$ | $0.96 \pm 1.412e-3$ | $0.77 \pm 2.548e-3$ | $0.948 \pm 1.384e-3$ |
| Llama-3.1 Trend +TFM patchtsmixer | $0.778 \pm 1.755e-3$ | $0.957 \pm 7.842e-4$ | $0.762 \pm 2.552e-3$ | $0.956 \pm 3.489e-3$ |
| Llama-3.1 Trend +TFM timemixer | $0.779 \pm 5.526e-3$ | $0.959 \pm 6.212e-3$ | $0.771 \pm 4.247e-3$ | $0.945 \pm 7.923e-3$ |
| Llama-3.1 Trend lstm | $0.79 \pm 0e0$ | $0.952 \pm 0e0$ | $0.77 \pm 0e0$ | $0.948 \pm 0e0$ |
| Llama-3.1 Trend mlp | $0.788 \pm 0e0$ | $0.949 \pm 0e0$ | $0.768 \pm 0e0$ | $0.947 \pm 0e0$ |
| Llama-3.1 Trend patchtsmixer | $0.787 \pm 0e0$ | $0.956 \pm 0e0$ | $0.769 \pm 0e0$ | $0.953 \pm 0e0$ |
| Llama-3.1 Trend timemixer | $0.789 \pm 0e0$ | $0.953 \pm 0e0$ | $0.769 \pm 0e0$ | $0.95 \pm 0e0$ |
| Llama-3.1 zero_shot +TFM lstm | $0.774 \pm 1.044e-4$ | $0.962 \pm 1.739e-4$ | $0.77 \pm 4.499e-3$ | $0.946 \pm 1.427e-3$ |
| Llama-3.1 zero_shot +TFM mlp | $0.779 \pm 5.577e-4$ | $0.959 \pm 2.524e-4$ | $0.766 \pm 4.885e-3$ | $0.957 \pm 3.664e-3$ |
| Llama-3.1 zero_shot +TFM patchtsmixer | $0.78 \pm 4.661e-4$ | $0.954 \pm 1.319e-3$ | $0.765 \pm 4.887e-4$ | $0.954 \pm 6.074e-4$ |
| Llama-3.1 zero_shot +TFM timemixer | $0.78 \pm 2.402e-3$ | $0.957 \pm 3.382e-3$ | $0.769 \pm 2.373e-3$ | $0.948 \pm 2.451e-3$ |
| Llama-3.1 zero_shot lstm | $0.791 \pm 0e0$ | $0.949 \pm 0e0$ | $0.781 \pm 0e0$ | $0.945 \pm 0e0$ |
| Llama-3.1 zero_shot mlp | $0.791 \pm 0e0$ | $0.946 \pm 0e0$ | $0.775 \pm 0e0$ | $0.945 \pm 0e0$ |
| Llama-3.1 zero_shot patchtsmixer | $0.793 \pm 0e0$ | $0.953 \pm 0e0$ | $0.78 \pm 0e0$ | $0.948 \pm 0e0$ |
| Llama-3.1 zero_shot timemixer | $0.794 \pm 0e0$ | $0.949 \pm 0e0$ | $0.781 \pm 0e0$ | $0.945 \pm 0e0$ |
| TFM lstm | $0.799 \pm 2.24e-3$ | $0.94 \pm 2.723e-3$ | $0.787 \pm 1.781e-3$ | $0.934 \pm 1.136e-3$ |
| TFM mlp | $0.798 \pm 8.308e-4$ | $0.942 \pm 2.808e-3$ | $0.786 \pm 1.065e-3$ | $0.934 \pm 2.918e-3$ |
| TFM patchtsmixer | $0.794 \pm 1.727e-3$ | $0.946 \pm 4.452e-4$ | $0.778 \pm 1.301e-3$ | $0.945 \pm 1.541e-3$ |
| TFM timemixer | $0.802 \pm 2.974e-3$ | $0.943 \pm 2.743e-3$ | $0.782 \pm 2.503e-3$ | $0.941 \pm 6.271e-3$ |
| TSDE lstm | $0.746 \pm 4.069e-4$ | $0.898 \pm 3.86e-4$ | $0.738 \pm 6.399e-3$ | $0.929 \pm 6.829e-3$ |
| TSDE mlp | $0.742 \pm 9.998e-4$ | $0.898 \pm 5.68e-4$ | $0.734 \pm 4.081e-3$ | $0.908 \pm 1.485e-3$ |
| TSDE patchtsmixer | $0.734 \pm 0e0$ | $0.909 \pm 2.5e-5$ | $0.721 \pm 2.765e-3$ | $0.899 \pm 1.2e-4$ |
| TSDE timemixer | $0.734 \pm 4.349e-5$ | $0.889 \pm 0e0$ | $0.72 \pm 4.75e-4$ | $0.879 \pm 4.7e-4$ |
| gemini-2.0-flash CoT +TFM lstm | $0.779 \pm 1.157e-3$ | $0.952 \pm 1.509e-3$ | $0.77 \pm 1.024e-3$ | $0.941 \pm 1.253e-3$ |
| gemini-2.0-flash CoT +TFM mlp | $0.781 \pm 2.019e-3$ | $0.955 \pm 3.218e-3$ | $0.765 \pm 1.35e-3$ | $0.951 \pm 1.08e-3$ |
| gemini-2.0-flash CoT +TFM patchtsmixer | $0.781 \pm 1.446e-3$ | $0.955 \pm 2.37e-3$ | $0.758 \pm 7.211e-4$ | $0.961 \pm 1.573e-3$ |
| gemini-2.0-flash CoT +TFM timemixer | $0.778 \pm 5.45e-4$ | $0.958 \pm 1.473e-3$ | $0.771 \pm 5.743e-3$ | $0.945 \pm 5.898e-3$ |
| gemini-2.0-flash CoT lstm | $0.793 \pm 1.584e-3$ | $0.945 \pm 3.267e-3$ | $0.773 \pm 5.346e-3$ | $0.941 \pm 4.391e-3$ |
| gemini-2.0-flash CoT mlp | $0.792 \pm 0e0$ | $0.946 \pm 0e0$ | $0.772 \pm 0e0$ | $0.945 \pm 0e0$ |
| gemini-2.0-flash CoT patchtsmixer | $0.791 \pm 0e0$ | $0.952 \pm 0e0$ | $0.774 \pm 0e0$ | $0.948 \pm 0e0$ |
| gemini-2.0-flash CoT timemixer | $0.792 \pm 0e0$ | $0.95 \pm 0e0$ | $0.777 \pm 0e0$ | $0.945 \pm 0e0$ |
| gemini-2.0-flash ICD +TFM lstm | $0.775 \pm 3.041e-3$ | $0.96 \pm 4.06e-3$ | $0.77 \pm 2.096e-3$ | $0.945 \pm 3.438e-3$ |
| gemini-2.0-flash ICD +TFM mlp | $0.777 \pm 1.871e-3$ | $0.96 \pm 1.075e-3$ | $0.764 \pm 2.623e-3$ | $0.954 \pm 2.623e-3$ |
| gemini-2.0-flash ICD +TFM patchtsmixer | $0.771 \pm 5.129e-4$ | $0.965 \pm 1.344e-3$ | $0.759 \pm 3.777e-3$ | $0.962 \pm 5.819e-3$ |
| gemini-2.0-flash ICD +TFM timemixer | $0.771 \pm 2.899e-3$ | $0.966 \pm 5.001e-3$ | $0.772 \pm 7.667e-3$ | $0.949 \pm 8.648e-3$ |
| gemini-2.0-flash ICD lstm | $0.772 \pm 7.707e-4$ | $0.974 \pm 8.825e-3$ | $0.772 \pm 1.838e-4$ | $0.948 \pm 1.344e-4$ |
| gemini-2.0-flash ICD mlp | $0.772 \pm 0e0$ | $0.965 \pm 0e0$ | $0.772 \pm 0e0$ | $0.946 \pm 0e0$ |
| gemini-2.0-flash ICD patchtsmixer | $0.777 \pm 0e0$ | $0.969 \pm 0e0$ | $0.772 \pm 0e0$ | $0.953 \pm 0e0$ |
| gemini-2.0-flash ICD timemixer | $0.772 \pm 0e0$ | $0.975 \pm 0e0$ | $0.776 \pm 0e0$ | $0.949 \pm 0e0$ |
| gemini-2.0-flash Trend +TFM lstm | $0.784 \pm 4.197e-3$ | $0.942 \pm 7.127e-3$ | $0.776 \pm 5.278e-4$ | $0.94 \pm 1.476e-3$ |
| gemini-2.0-flash Trend +TFM mlp | $0.789 \pm 2.121e-3$ | $0.947 \pm 2.406e-3$ | $0.773 \pm 6.574e-4$ | $0.944 \pm 8.593e-4$ |
| gemini-2.0-flash Trend +TFM patchtsmixer | $0.789 \pm 1.383e-3$ | $0.944 \pm 1.347e-3$ | $0.765 \pm 2.892e-3$ | $0.951 \pm 3.808e-3$ |
| gemini-2.0-flash Trend +TFM timemixer | $0.781 \pm 1.092e-3$ | $0.954 \pm 2.578e-3$ | $0.773 \pm 1.482e-3$ | $0.946 \pm 4.805e-3$ |
| gemini-2.0-flash Trend lstm | $0.801 \pm 3.111e-3$ | $0.938 \pm 3.458e-3$ | $0.781 \pm 2.178e-3$ | $0.935 \pm 6.838e-3$ |
| gemini-2.0-flash Trend mlp | $0.799 \pm 0e0$ | $0.939 \pm 0e0$ | $0.777 \pm 0e0$ | $0.941 \pm 0e0$ |
| gemini-2.0-flash Trend patchtsmixer | $0.802 \pm 0e0$ | $0.945 \pm 0e0$ | $0.783 \pm 0e0$ | $0.94 \pm 0e0$ |
| gemini-2.0-flash Trend timemixer | $0.8 \pm 0e0$ | $0.942 \pm 0e0$ | $0.785 \pm 0e0$ | $0.938 \pm 0e0$ |
| gemini-2.0-flash zero_shot +TFM lstm | $0.778 \pm 2.447e-3$ | $0.952 \pm 3.203e-3$ | $0.77 \pm 1.228e-3$ | $0.945 \pm 2.536e-3$ |
| gemini-2.0-flash zero_shot +TFM mlp | $0.778 \pm 1.966e-3$ | $0.957 \pm 9.836e-4$ | $0.761 \pm 3.949e-3$ | $0.956 \pm 3.297e-3$ |
| gemini-2.0-flash zero_shot +TFM patchtsmixer | $0.775 \pm 1.757e-3$ | $0.959 \pm 9.166e-4$ | $0.758 \pm 2.899e-3$ | $0.962 \pm 4.923e-3$ |
| gemini-2.0-flash zero_shot +TFM timemixer | $0.774 \pm 3.55e-3$ | $0.964 \pm 6.95e-3$ | $0.768 \pm 3.997e-3$ | $0.952 \pm 4.818e-3$ |
| gemini-2.0-flash zero_shot lstm | $0.793 \pm 7.071e-4$ | $0.943 \pm 5.19e-3$ | $0.765 \pm 3.309e-3$ | $0.955 \pm 9.115e-3$ |
| gemini-2.0-flash zero_shot mlp | $0.794 \pm 0e0$ | $0.942 \pm 0e0$ | $0.762 \pm 0e0$ | $0.957 \pm 0e0$ |
| gemini-2.0-flash zero_shot patchtsmixer | $0.795 \pm 0e0$ | $0.953 \pm 0e0$ | $0.767 \pm 0e0$ | $0.954 \pm 0e0$ |
| gemini-2.0-flash zero_shot timemixer | $0.787 \pm 0e0$ | $0.957 \pm 0e0$ | $0.767 \pm 0e0$ | $0.957 \pm 0e0$ |
| interp lstm | $0.708 \pm 1.932e-3$ | $0.886 \pm 6.243e-3$ | $0.657 \pm 9.491e-3$ | $0.813 \pm 2.445e-2$ |
| interp mlp | $0.733 \pm 1.943e-3$ | $0.955 \pm 3.448e-3$ | $0.702 \pm 7.303e-3$ | $0.911 \pm 2.16e-2$ |
| interp patchtsmixer | $0.702 \pm 9.913e-3$ | $0.808 \pm 1.83e-2$ | $0.678 \pm 7.828e-3$ | $0.8 \pm 1.262e-2$ |
| interp timemixer | $0.641 \pm 8.843e-3$ | $0.654 \pm 1.769e-2$ | $0.638 \pm 3.367e-3$ | $0.733 \pm 8.544e-3$ |
| mean lstm | $0.649 \pm 4.088e-3$ | $0.8 \pm 1.151e-2$ | $0.623 \pm 9.661e-3$ | $0.726 \pm 1.935e-2$ |
| mean mlp | $0.691 \pm 3.43e-3$ | $0.885 \pm 6.064e-3$ | $0.642 \pm 1.942e-3$ | $0.772 \pm 2.956e-3$ |
| mean patchtsmixer | $0.686 \pm 1.272e-2$ | $0.775 \pm 2.779e-2$ | $0.679 \pm 7.18e-3$ | $0.81 \pm 1.601e-2$ |
| mean timemixer | $0.667 \pm 5.537e-3$ | $0.708 \pm 1.183e-2$ | $0.649 \pm 6.057e-3$ | $0.754 \pm 1.358e-2$ |
| No_sum_CoT CoT +TFM lstm | $0.795 \pm 4.985e-3$ | $0.936 \pm 5.18e-3$ | $0.775 \pm 1.182e-3$ | $0.943 \pm 2.763e-3$ |
| No_sum_CoT CoT +TFM mlp | $0.793 \pm 2.168e-3$ | $0.951 \pm 5.7e-4$ | $0.771 \pm 1.56e-3$ | $0.951 \pm 2.646e-3$ |
| No_sum_CoT CoT +TFM patchtsmixer | $0.788 \pm 1.265e-3$ | $0.947 \pm 7.808e-4$ | $0.769 \pm 1.909e-3$ | $0.945 \pm 2.385e-3$ |
| No_sum_CoT CoT +TFM timemixer | $0.794 \pm 8.153e-4$ | $0.94 \pm 3.354e-3$ | $0.772 \pm 1.13e-3$ | $0.948 \pm 2.62e-3$ |
| No_sum_CoT lstm | $0.806 \pm 6.248e-4$ | $0.948 \pm 4.444e-4$ | $0.786 \pm 8.689e-4$ | $0.951 \pm 5.341e-4$ |
| No_sum_CoT mlp | $0.803 \pm 0e0$ | $0.948 \pm 0e0$ | $0.778 \pm 0e0$ | $0.948 \pm 0e0$ |
| No_sum_CoT patchtsmixer | $0.805 \pm 1.274e-4$ | $0.95 \pm 1.266e-4$ | $0.781 \pm 4.933e-4$ | $0.95 \pm 1.188e-3$ |
| No_sum_CoT timemixer | $0.807 \pm 9.986e-4$ | $0.943 \pm 5.289e-4$ | $0.796 \pm 1.695e-3$ | $0.938 \pm 4.941e-4$ |
| No_sum_ICD ICD +TFM lstm | $0.798 \pm 3.219e-3$ | $0.937 \pm 1.662e-3$ | $0.783 \pm 7.751e-4$ | $0.938 \pm 1.566e-3$ |
| No_sum_ICD ICD +TFM mlp | $0.793 \pm 6.338e-4$ | $0.951 \pm 1.252e-3$ | $0.771 \pm 2.683e-3$ | $0.951 \pm 2.63e-3$ |
| No_sum_ICD ICD +TFM patchtsmixer | $0.787 \pm 1.883e-3$ | $0.95 \pm 2.141e-3$ | $0.773 \pm 8.41e-4$ | $0.942 \pm 1.64e-3$ |
| No_sum_ICD ICD +TFM timemixer | $0.792 \pm 3.412e-4$ | $0.945 \pm 1.421e-3$ | $0.777 \pm 1.567e-3$ | $0.941 \pm 5.876e-3$ |
| No_sum_ICD lstm | $0.805 \pm 3.356e-4$ | $0.942 \pm 6.258e-4$ | $0.79 \pm 1.22e-3$ | $0.941 \pm 2.926e-4$ |
| No_sum_ICD mlp | $0.802 \pm 0e0$ | $0.948 \pm 0e0$ | $0.782 \pm 0e0$ | $0.945 \pm 0e0$ |
| No_sum_ICD patchtsmixer | $0.805 \pm 9.485e-4$ | $0.953 \pm 8.13e-4$ | $0.785 \pm 1.948e-3$ | $0.947 \pm 7.304e-4$ |
| No_sum_ICD timemixer | $0.811 \pm 3.35e-4$ | $0.942 \pm 3.166e-4$ | $0.794 \pm 3.853e-4$ | $0.936 \pm 7.954e-4$ |
| No_sum_Trend Trend +TFM lstm | $0.8 \pm 7.995e-4$ | $0.936 \pm 1.19e-3$ | $0.776 \pm 2.982e-3$ | $0.939 \pm 6.048e-3$ |
| No_sum_Trend Trend +TFM mlp | $0.791 \pm 2.368e-3$ | $0.951 \pm 1.239e-3$ | $0.771 \pm 4.85e-3$ | $0.947 \pm 2.446e-3$ |
| No_sum_Trend Trend +TFM patchtsmixer | $0.789 \pm 1.329e-3$ | $0.946 \pm 2.136e-3$ | $0.769 \pm 1.466e-3$ | $0.945 \pm 1.707e-3$ |
| No_sum_Trend Trend +TFM timemixer | $0.791 \pm 3.189e-3$ | $0.943 \pm 2.13e-3$ | $0.777 \pm 4.916e-3$ | $0.939 \pm 6.115e-3$ |

Table 32: Cross-site transfer results - Lab. (part 1/2)

| Method | hirid → ppicu | | mimic → ppicu | |
|---|---|---|---|---|
| | f1_micro | recall_micro | f1_micro | recall_micro |
| No_sum_Trend lstm | $0.805 \pm 8.298e-4$ | $0.944 \pm 1.776e-3$ | $0.78 \pm 3.98e-4$ | $0.953 \pm 5.442e-4$ |
| No_sum_Trend mlp | $0.8 \pm 0e0$ | $0.946 \pm 0e0$ | $0.779 \pm 0e0$ | $0.946 \pm 0e0$ |
| No_sum_Trend patchtsmixer | $0.807 \pm 7.472e-4$ | $0.95 \pm 9.666e-4$ | $0.779 \pm 2.562e-3$ | $0.948 \pm 1.084e-3$ |
| No_sum_Trend timemixer | $0.808 \pm 1.079e-3$ | $0.941 \pm 4.553e-4$ | $0.789 \pm 7.168e-4$ | $0.937 \pm 1.581e-3$ |
| medgemma CoT +TFM lstm | $0.779 \pm 3.345e-3$ | $0.955 \pm 5.752e-3$ | $0.775 \pm 1.652e-3$ | $0.94 \pm 2.151e-3$ |
| medgemma CoT +TFM mlp | $0.783 \pm 4.335e-4$ | $0.957 \pm 7.516e-4$ | $0.771 \pm 1.904e-3$ | $0.949 \pm 2.446e-3$ |
| medgemma CoT +TFM patchtsmixer | $0.778 \pm 4.118e-3$ | $0.958 \pm 2.585e-3$ | $0.766 \pm 2.887e-3$ | $0.952 \pm 4.626e-3$ |
| medgemma CoT +TFM timemixer | $0.78 \pm 1.849e-3$ | $0.956 \pm 3.352e-3$ | $0.772 \pm 5.854e-3$ | $0.944 \pm 8.061e-3$ |
| medgemma CoT lstm | $0.797 \pm 2.765e-3$ | $0.946 \pm 1.407e-3$ | $0.774 \pm 2.348e-3$ | $0.951 \pm 9.214e-3$ |
| medgemma CoT mlp | $0.797 \pm 0e0$ | $0.947 \pm 0e0$ | $0.774 \pm 0e0$ | $0.949 \pm 0e0$ |
| medgemma CoT patchtsmixer | $0.799 \pm 0e0$ | $0.954 \pm 0e0$ | $0.774 \pm 0e0$ | $0.949 \pm 0e0$ |
| medgemma CoT timemixer | $0.796 \pm 0e0$ | $0.952 \pm 0e0$ | $0.776 \pm 0e0$ | $0.946 \pm 0e0$ |
| medgemma ICD +TFM lstm | $0.77 \pm 8.732e-4$ | $0.966 \pm 2.587e-3$ | $0.772 \pm 1.314e-3$ | $0.944 \pm 2.011e-3$ |
| medgemma ICD +TFM mlp | $0.774 \pm 2.889e-3$ | $0.963 \pm 2.689e-3$ | $0.764 \pm 7.353e-4$ | $0.957 \pm 1.034e-3$ |
| medgemma ICD +TFM patchtsmixer | $0.771 \pm 8.649e-4$ | $0.964 \pm 1.031e-3$ | $0.76 \pm 3.556e-3$ | $0.958 \pm 4.775e-3$ |
| medgemma ICD +TFM timemixer | $0.77 \pm 6.629e-4$ | $0.968 \pm 8.445e-4$ | $0.771 \pm 2.011e-3$ | $0.949 \pm 1.917e-3$ |
| medgemma ICD lstm | $0.791 \pm 3.62e-3$ | $0.946 \pm 1.039e-3$ | $0.777 \pm 5.636e-3$ | $0.945 \pm 4.801e-3$ |
| medgemma ICD mlp | $0.789 \pm 0e0$ | $0.95 \pm 0e0$ | $0.771 \pm 0e0$ | $0.95 \pm 0e0$ |
| medgemma ICD patchtsmixer | $0.788 \pm 0e0$ | $0.957 \pm 0e0$ | $0.775 \pm 0e0$ | $0.952 \pm 0e0$ |
| medgemma ICD timemixer | $0.786 \pm 0e0$ | $0.958 \pm 0e0$ | $0.776 \pm 0e0$ | $0.949 \pm 0e0$ |
| medgemma Trend +TFM lstm | $0.779 \pm 5.745e-3$ | $0.954 \pm 1.007e-2$ | $0.773 \pm 4.056e-3$ | $0.94 \pm 2.083e-3$ |
| medgemma Trend +TFM mlp | $0.78 \pm 1.057e-3$ | $0.958 \pm 1.437e-3$ | $0.773 \pm 1.767e-3$ | $0.944 \pm 8.65e-4$ |
| medgemma Trend +TFM patchtsmixer | $0.784 \pm 6.558e-4$ | $0.951 \pm 1.35e-4$ | $0.764 \pm 1.626e-3$ | $0.952 \pm 3.448e-3$ |
| medgemma Trend +TFM timemixer | $0.775 \pm 5.554e-4$ | $0.963 \pm 1.068e-3$ | $0.77 \pm 1.111e-3$ | $0.947 \pm 2.364e-4$ |
| medgemma Trend lstm | $0.797 \pm 2.411e-3$ | $0.942 \pm 2.284e-3$ | $0.784 \pm 6.873e-3$ | $0.938 \pm 1.287e-3$ |
| medgemma Trend mlp | $0.793 \pm 0e0$ | $0.946 \pm 0e0$ | $0.781 \pm 0e0$ | $0.936 \pm 0e0$ |
| medgemma Trend patchtsmixer | $0.796 \pm 0e0$ | $0.951 \pm 0e0$ | $0.785 \pm 0e0$ | $0.941 \pm 0e0$ |
| medgemma Trend timemixer | $0.796 \pm 0e0$ | $0.95 \pm 0e0$ | $0.787 \pm 0e0$ | $0.94 \pm 0e0$ |
| medgemma zero_shot +TFM lstm | $0.772 \pm 2.638e-3$ | $0.964 \pm 5.331e-3$ | $0.775 \pm 3.988e-3$ | $0.942 \pm 3.711e-3$ |
| medgemma zero_shot +TFM mlp | $0.777 \pm 9.697e-4$ | $0.962 \pm 1.42e-3$ | $0.769 \pm 1.61e-3$ | $0.953 \pm 1.657e-3$ |
| medgemma zero_shot +TFM patchtsmixer | $0.773 \pm 3.995e-4$ | $0.963 \pm 5.138e-4$ | $0.761 \pm 1.3e-3$ | $0.96 \pm 2.928e-4$ |
| medgemma zero_shot +TFM timemixer | $0.774 \pm 6.375e-3$ | $0.961 \pm 7.042e-3$ | $0.768 \pm 3.265e-3$ | $0.952 \pm 2.319e-3$ |
| medgemma zero_shot lstm | $0.793 \pm 2.093e-3$ | $0.947 \pm 7.361e-3$ | $0.777 \pm 8.91e-4$ | $0.943 \pm 8.026e-3$ |
| medgemma zero_shot mlp | $0.791 \pm 0e0$ | $0.952 \pm 0e0$ | $0.773 \pm 0e0$ | $0.95 \pm 0e0$ |
| medgemma zero_shot patchtsmixer | $0.791 \pm 0e0$ | $0.958 \pm 0e0$ | $0.776 \pm 0e0$ | $0.951 \pm 0e0$ |
| medgemma zero_shot timemixer | $0.793 \pm 0e0$ | $0.953 \pm 0e0$ | $0.78 \pm 0e0$ | $0.946 \pm 0e0$ |
| No_sum_zero_shot zero_shot +TFM lstm | $0.797 \pm 5.88e-4$ | $0.936 \pm 4.818e-4$ | $0.772 \pm 9.725e-4$ | $0.945 \pm 2.726e-4$ |
| No_sum_zero_shot zero_shot +TFM mlp | $0.787 \pm 3.481e-3$ | $0.954 \pm 2.038e-3$ | $0.772 \pm 5.512e-3$ | $0.943 \pm 4.855e-3$ |
| No_sum_zero_shot zero_shot +TFM patchtsmixer | $0.787 \pm 1.369e-3$ | $0.948 \pm 7.006e-4$ | $0.768 \pm 6.24e-4$ | $0.947 \pm 1.247e-3$ |
| No_sum_zero_shot zero_shot +TFM timemixer | $0.793 \pm 4.514e-3$ | $0.941 \pm 3.638e-3$ | $0.774 \pm 2.981e-3$ | $0.942 \pm 4.471e-3$ |
| No_sum_zero_shot lstm | $0.808 \pm 9.673e-4$ | $0.939 \pm 1.164e-3$ | $0.787 \pm 4.993e-4$ | $0.95 \pm 8.082e-4$ |
| No_sum_zero_shot mlp | $0.8 \pm 0e0$ | $0.947 \pm 0e0$ | $0.775 \pm 0e0$ | $0.946 \pm 0e0$ |
| No_sum_zero_shot patchtsmixer | $0.807 \pm 1.189e-3$ | $0.948 \pm 5.173e-4$ | $0.781 \pm 1.305e-3$ | $0.946 \pm 1.944e-3$ |
| No_sum_zero_shot timemixer | $0.807 \pm 9.704e-4$ | $0.941 \pm 1.385e-3$ | $0.789 \pm 5.1e-4$ | $0.943 \pm 1.549e-3$ |
| right lstm | $0.711 \pm 1.256e-3$ | $0.913 \pm 1.471e-3$ | $0.715 \pm 7.657e-3$ | $0.955 \pm 1.734e-2$ |
| right mlp | $0.723 \pm 8.389e-3$ | $0.938 \pm 2.397e-2$ | $0.663 \pm 1.991e-2$ | $0.813 \pm 5.145e-2$ |
| right patchtsmixer | $0.719 \pm 3.188e-3$ | $0.879 \pm 1.593e-2$ | $0.692 \pm 1.36e-2$ | $0.823 \pm 2.478e-2$ |
| right timemixer | $0.689 \pm 9.4e-3$ | $0.8 \pm 4.114e-2$ | $0.671 \pm 8.889e-3$ | $0.797 \pm 2.827e-2$ |

Table 33: Cross-site transfer results - Lab. (part 2/2)

| Method | hirid → hirid | | mimic → mimic | | ppicu → ppicu | |
|---|---|---|---|---|---|---|
| | masked_mae | masked_mse | masked_mae | masked_mse | masked_mae | masked_mse |
| Llama-3.1 CoT +TFM lstm | $0.073 \pm 1.4e-3$ | $0.014 \pm 3.704e-4$ | $0.093 \pm 8.415e-4$ | $0.018 \pm 2.152e-4$ | $0.074 \pm 8.799e-4$ | $0.015 \pm 2.152e-4$ |
| Llama-3.1 CoT +TFM mlp | $0.078 \pm 2.696e-4$ | $0.013 \pm 7.234e-5$ | $0.096 \pm 3.704e-4$ | $0.018 \pm 1.193e-4$ | $0.08 \pm 8.415e-4$ | $0.015 \pm 2.801e-4$ |
| Llama-3.1 CoT +TFM patchtsmixer | $0.125 \pm 7.892e-4$ | $0.028 \pm 3.819e-4$ | $0.139 \pm 9.019e-4$ | $0.033 \pm 4.1e-4$ | $0.455 \pm 7.315e-2$ | $0.339 \pm 9.426e-2$ |
| Llama-3.1 CoT +TFM timemixer | $0.077 \pm 7.422e-4$ | $0.013 \pm 2.196e-4$ | $0.094 \pm 3.646e-4$ | $0.018 \pm 2.452e-4$ | $0.109 \pm 6.269e-4$ | $0.021 \pm 2.065e-4$ |
| Llama-3.1 CoT lstm | $0.075 \pm 4.255e-4$ | $0.015 \pm 1.569e-4$ | $0.094 \pm 4.84e-4$ | $0.019 \pm 1.787e-4$ | $0.077 \pm 6.466e-4$ | $0.017 \pm 1.758e-4$ |
| Llama-3.1 CoT mlp | $0.077 \pm 6.316e-4$ | $0.015 \pm 1.471e-4$ | $0.095 \pm 5.053e-4$ | $0.019 \pm 1.637e-4$ | $0.078 \pm 4.155e-4$ | $0.016 \pm 1.392e-4$ |
| Llama-3.1 CoT patchtsmixer | $0.114 \pm 1.501e-3$ | $0.024 \pm 4.802e-4$ | $0.131 \pm 1.895e-4$ | $0.03 \pm 2.291e-4$ | $0.102 \pm 5.067e-3$ | $0.021 \pm 1.382e-3$ |
| Llama-3.1 CoT timemixer | $0.079 \pm 1.151e-3$ | $0.015 \pm 3.293e-4$ | $0.098 \pm 1.675e-3$ | $0.02 \pm 5.097e-4$ | $0.09 \pm 9.862e-4$ | $0.018 \pm 2.485e-4$ |
| Llama-3.1 ICD +TFM lstm | $0.074 \pm 7.703e-4$ | $0.014 \pm 1.793e-4$ | $0.094 \pm 3.676e-4$ | $0.019 \pm 1.301e-4$ | $0.074 \pm 1.087e-3$ | $0.015 \pm 2.811e-4$ |
| Llama-3.1 ICD +TFM mlp | $0.079 \pm 1.656e-4$ | $0.014 \pm 3.786e-5$ | $0.097 \pm 4.725e-4$ | $0.019 \pm 1.769e-4$ | $0.08 \pm 1.747e-4$ | $0.015 \pm 6.11e-5$ |
| Llama-3.1 ICD +TFM patchtsmixer | $0.125 \pm 2.646e-5$ | $0.028 \pm 1.473e-4$ | $0.14 \pm 1.908e-4$ | $0.034 \pm 1.058e-4$ | $0.453 \pm 9.85e-2$ | $0.338 \pm 1.324e-1$ |
| Llama-3.1 ICD +TFM timemixer | $0.077 \pm 4.784e-4$ | $0.013 \pm 1.877e-4$ | $0.095 \pm 2.836e-4$ | $0.018 \pm 2.001e-4$ | $0.109 \pm 8.773e-4$ | $0.021 \pm 3.139e-4$ |
| Llama-3.1 ICD lstm | $0.074 \pm 3.806e-4$ | $0.015 \pm 7.848e-5$ | $0.095 \pm 5.013e-4$ | $0.02 \pm 1.673e-4$ | $0.076 \pm 7.454e-4$ | $0.016 \pm 2.512e-4$ |
| Llama-3.1 ICD mlp | $0.077 \pm 6.549e-4$ | $0.015 \pm 2.034e-4$ | $0.096 \pm 5.709e-4$ | $0.02 \pm 1.431e-4$ | $0.078 \pm 4.151e-4$ | $0.016 \pm 1.066e-4$ |
| Llama-3.1 ICD patchtsmixer | $0.115 \pm 9.215e-4$ | $0.024 \pm 2.721e-4$ | $0.13 \pm 1.749e-3$ | $0.03 \pm 6.951e-4$ | $0.1 \pm 5.492e-3$ | $0.021 \pm 1.42e-3$ |
| Llama-3.1 ICD timemixer | $0.079 \pm 5.826e-4$ | $0.015 \pm 2.47e-4$ | $0.098 \pm 8.147e-4$ | $0.02 \pm 1.876e-4$ | $0.088 \pm 4.189e-4$ | $0.017 \pm 1.237e-4$ |
| Llama-3.1 Trend +TFM lstm | $0.074 \pm 1.042e-3$ | $0.014 \pm 1.662e-4$ | $0.094 \pm 4.75e-4$ | $0.019 \pm 1.803e-4$ | $0.073 \pm 8.15e-4$ | $0.014 \pm 2.974e-4$ |
| Llama-3.1 Trend +TFM mlp | $0.079 \pm 6.755e-4$ | $0.014 \pm 2.344e-4$ | $0.097 \pm 3.592e-4$ | $0.019 \pm 1.127e-4$ | $0.08 \pm 1.446e-3$ | $0.015 \pm 4.917e-4$ |
| Llama-3.1 Trend +TFM patchtsmixer | $0.125 \pm 1.351e-3$ | $0.028 \pm 4.681e-4$ | $0.14 \pm 1.02e-3$ | $0.034 \pm 3.47e-4$ | $0.39 \pm 1.236e-2$ | $0.275 \pm 1.994e-2$ |
| Llama-3.1 Trend +TFM timemixer | $0.077 \pm 3.602e-4$ | $0.013 \pm 4e-5$ | $0.095 \pm 2.159e-4$ | $0.018 \pm 2.517e-5$ | $0.108 \pm 1.084e-3$ | $0.021 \pm 4.368e-4$ |
| Llama-3.1 Trend lstm | $0.074 \pm 3.203e-4$ | $0.014 \pm 9.574e-6$ | $0.094 \pm 4.096e-4$ | $0.019 \pm 1.576e-4$ | $0.076 \pm 1.132e-3$ | $0.016 \pm 2.965e-4$ |
| Llama-3.1 Trend mlp | $0.076 \pm 1.679e-4$ | $0.014 \pm 9.179e-5$ | $0.096 \pm 2.505e-4$ | $0.019 \pm 1.044e-4$ | $0.078 \pm 6.638e-4$ | $0.016 \pm 1.73e-4$ |
| Llama-3.1 Trend patchtsmixer | $0.114 \pm 1.396e-3$ | $0.024 \pm 3.739e-4$ | $0.13 \pm 9.261e-4$ | $0.029 \pm 3.314e-4$ | $0.095 \pm 4.376e-3$ | $0.019 \pm 9.721e-4$ |
| Llama-3.1 Trend timemixer | $0.078 \pm 1.254e-3$ | $0.015 \pm 2.831e-4$ | $0.097 \pm 1.196e-3$ | $0.02 \pm 2.822e-4$ | $0.089 \pm 9.46e-4$ | $0.017 \pm 2.888e-4$ |
| Llama-3.1 zero_shot +TFM lstm | $0.072 \pm 7.624e-4$ | $0.013 \pm 2.021e-4$ | $0.089 \pm 1.266e-4$ | $0.017 \pm 6.807e-5$ | $0.073 \pm 4.245e-4$ | $0.014 \pm 1.677e-4$ |
| Llama-3.1 zero_shot +TFM mlp | $0.075 \pm 1.989e-3$ | $0.012 \pm 6.351e-4$ | $0.092 \pm 1.021e-3$ | $0.017 \pm 2.914e-4$ | $0.077 \pm 8.995e-4$ | $0.014 \pm 3.153e-4$ |
| Llama-3.1 zero_shot +TFM patchtsmixer | $0.126 \pm 9.015e-4$ | $0.028 \pm 2.178e-4$ | $0.14 \pm 1.091e-3$ | $0.034 \pm 3.46e-4$ | $0.276 \pm 1.287e-1$ | $0.149 \pm 1.026e-1$ |
| Llama-3.1 zero_shot +TFM timemixer | $0.074 \pm 3.95e-4$ | $0.012 \pm 1.015e-4$ | $0.091 \pm 2.234e-4$ | $0.017 \pm 1.274e-4$ | $0.107 \pm 1.173e-3$ | $0.02 \pm 5.33e-4$ |
| Llama-3.1 zero_shot lstm | $0.072 \pm 3.809e-4$ | $0.014 \pm 9.106e-5$ | $0.091 \pm 2.923e-4$ | $0.018 \pm 1.134e-4$ | $0.076 \pm 1.052e-3$ | $0.016 \pm 2.92e-4$ |
| Llama-3.1 zero_shot mlp | $0.074 \pm 4.196e-4$ | $0.014 \pm 9.946e-5$ | $0.093 \pm 3.957e-4$ | $0.019 \pm 1.445e-4$ | $0.076 \pm 4.853e-4$ | $0.016 \pm 1.447e-4$ |
| Llama-3.1 zero_shot patchtsmixer | $0.114 \pm 1.306e-3$ | $0.024 \pm 5.302e-4$ | $0.129 \pm 2.36e-3$ | $0.029 \pm 8.739e-4$ | $0.093 \pm 3.283e-3$ | $0.019 \pm 5.885e-4$ |
| Llama-3.1 zero_shot timemixer | $0.076 \pm 1.425e-3$ | $0.014 \pm 3.643e-4$ | $0.095 \pm 1.243e-3$ | $0.019 \pm 2.466e-4$ | $0.088 \pm 1.194e-3$ | $0.017 \pm 3.195e-4$ |
| TFM lstm | $0.075 \pm 6.816e-4$ | $0.015 \pm 1.303e-4$ | $0.094 \pm 6.191e-5$ | $0.019 \pm 5.715e-5$ | $0.075 \pm 6.752e-4$ | $0.016 \pm 1.608e-4$ |
| TFM mlp | $0.077 \pm 3.99e-4$ | $0.015 \pm 8.921e-5$ | $0.095 \pm 3.025e-4$ | $0.019 \pm 5.737e-5$ | $0.076 \pm 4.834e-4$ | $0.016 \pm 6.683e-5$ |
| TFM patchtsmixer | $0.126 \pm 1.558e-3$ | $0.028 \pm 5.498e-4$ | $0.132 \pm 7.469e-4$ | $0.03 \pm 3.232e-4$ | $0.142 \pm 3.861e-3$ | $0.036 \pm 1.391e-3$ |
| TFM timemixer | $0.076 \pm 5.576e-4$ | $0.014 \pm 1.573e-4$ | $0.095 \pm 3.951e-4$ | $0.019 \pm 4.546e-5$ | $0.102 \pm 3.706e-3$ | $0.02 \pm 1.618e-3$ |
| TSDE lstm | $0.066 \pm 3.204e-4$ | $0.012 \pm 7.047e-5$ | $0.072 \pm 1.936e-4$ | $0.012 \pm 3.916e-5$ | $0.052 \pm 4.738e-4$ | $0.009 \pm 3.317e-5$ |
| TSDE mlp | $0.067 \pm 2.139e-4$ | $0.011 \pm 2.16e-5$ | $0.073 \pm 2.428e-4$ | $0.012 \pm 1.414e-5$ | $0.054 \pm 1.744e-4$ | $0.008 \pm 2.754e-5$ |
| TSDE patchtsmixer | $0.127 \pm 1.484e-3$ | $0.029 \pm 5.634e-4$ | $0.103 \pm 1.141e-2$ | $0.02 \pm 3.797e-3$ | $0.177 \pm 1.168e-2$ | $0.053 \pm 6.641e-3$ |
| TSDE timemixer | $0.069 \pm 8.416e-4$ | $0.012 \pm 7.416e-5$ | $0.073 \pm 2.546e-4$ | $0.012 \pm 2.944e-5$ | $0.091 \pm 4.927e-4$ | $0.015 \pm 1.533e-4$ |
| gemini-2.0-flash CoT +TFM lstm | $0.07 \pm 4.102e-4$ | $0.012 \pm 1.716e-4$ | $0.091 \pm 8.102e-4$ | $0.017 \pm 1.607e-4$ | $0.072 \pm 2.073e-3$ | $0.014 \pm 5.859e-4$ |
| gemini-2.0-flash CoT +TFM mlp | $0.076 \pm 2.954e-4$ | $0.013 \pm 8.888e-5$ | $0.095 \pm 1.234e-4$ | $0.018 \pm 6.429e-5$ | $0.079 \pm 2.364e-4$ | $0.014 \pm 4.933e-5$ |
| gemini-2.0-flash CoT +TFM patchtsmixer | $0.125 \pm 1.115e-3$ | $0.028 \pm 3.247e-4$ | $0.133 \pm 1.085e-2$ | $0.031 \pm 4.64e-3$ | $0.425 \pm 3.751e-2$ | $0.288 \pm 5.694e-2$ |
| gemini-2.0-flash CoT +TFM timemixer | $0.075 \pm 4.59e-4$ | $0.012 \pm 1.058e-4$ | $0.092 \pm 8.083e-4$ | $0.017 \pm 2.371e-4$ | $0.107 \pm 8.752e-4$ | $0.021 \pm 4.05e-4$ |
| gemini-2.0-flash CoT lstm | $0.072 \pm 6.472e-4$ | $0.013 \pm 1.982e-4$ | $0.091 \pm 3.33e-4$ | $0.018 \pm 8.884e-5$ | $0.072 \pm 8.141e-4$ | $0.014 \pm 2.108e-4$ |
| gemini-2.0-flash CoT mlp | $0.073 \pm 4.444e-4$ | $0.013 \pm 1.389e-4$ | $0.092 \pm 2.359e-4$ | $0.018 \pm 9.815e-5$ | $0.073 \pm 3.309e-4$ | $0.014 \pm 6.551e-5$ |
| gemini-2.0-flash CoT patchtsmixer | $0.112 \pm 8.465e-4$ | $0.023 \pm 2.616e-4$ | $0.128 \pm 8.426e-4$ | $0.029 \pm 3.549e-4$ | $0.097 \pm 1.014e-2$ | $0.019 \pm 3.068e-3$ |
| gemini-2.0-flash CoT timemixer | $0.076 \pm 1.872e-3$ | $0.014 \pm 4.269e-4$ | $0.093 \pm 8.366e-4$ | $0.018 \pm 1.911e-4$ | $0.084 \pm 1.191e-3$ | $0.015 \pm 2.974e-4$ |
| gemini-2.0-flash ICD +TFM lstm | $0.072 \pm 6.045e-4$ | $0.013 \pm 2.179e-4$ | $0.091 \pm 5.595e-4$ | $0.017 \pm 8.963e-5$ | $0.073 \pm 2.059e-3$ | $0.014 \pm 6.668e-4$ |
| gemini-2.0-flash ICD +TFM mlp | $0.078 \pm 4.842e-4$ | $0.013 \pm 1.528e-4$ | $0.094 \pm 2.538e-4$ | $0.017 \pm 7.55e-5$ | $0.08 \pm 1.065e-3$ | $0.014 \pm 3.148e-4$ |
| gemini-2.0-flash ICD +TFM patchtsmixer | $0.124 \pm 1.068e-3$ | $0.028 \pm 3.609e-4$ | $0.139 \pm 7.686e-4$ | $0.033 \pm 4.2e-4$ | $0.379 \pm 2.04e-2$ | $0.229 \pm 2.503e-2$ |
| gemini-2.0-flash ICD +TFM timemixer | $0.074 \pm 4.486e-4$ | $0.012 \pm 2.301e-4$ | $0.093 \pm 7.472e-4$ | $0.017 \pm 2.74e-4$ | $0.107 \pm 1.184e-3$ | $0.021 \pm 5.147e-4$ |
| gemini-2.0-flash ICD lstm | $0.072 \pm 1.111e-3$ | $0.013 \pm 2.629e-4$ | $0.092 \pm 1.443e-4$ | $0.018 \pm 8.287e-5$ | $0.073 \pm 4.526e-4$ | $0.015 \pm 9.142e-5$ |
| gemini-2.0-flash ICD mlp | $0.074 \pm 5.099e-4$ | $0.013 \pm 1.425e-4$ | $0.093 \pm 3.266e-4$ | $0.018 \pm 9.032e-5$ | $0.074 \pm 4.188e-4$ | $0.015 \pm 1.209e-4$ |
| gemini-2.0-flash ICD patchtsmixer | $0.113 \pm 5.735e-4$ | $0.023 \pm 1.782e-4$ | $0.129 \pm 5.676e-4$ | $0.029 \pm 3.551e-4$ | $0.093 \pm 1.037e-2$ | $0.018 \pm 2.615e-3$ |
| gemini-2.0-flash ICD timemixer | $0.078 \pm 2.33e-3$ | $0.014 \pm 4.438e-4$ | $0.095 \pm 1.849e-4$ | $0.019 \pm 4.16e-4$ | $0.085 \pm 7.941e-4$ | $0.016 \pm 1.919e-4$ |
| gemini-2.0-flash Trend +TFM lstm | $0.072 \pm 9.879e-4$ | $0.013 \pm 2.553e-4$ | $0.093 \pm 3.412e-4$ | $0.018 \pm 9e-5$ | $0.073 \pm 1.085e-3$ | $0.014 \pm 3.667e-4$ |
| gemini-2.0-flash Trend +TFM mlp | $0.077 \pm 1.128e-3$ | $0.013 \pm 3.963e-4$ | $0.096 \pm 3.723e-4$ | $0.018 \pm 1.345e-4$ | $0.08 \pm 9.753e-4$ | $0.014 \pm 3.279e-4$ |
| gemini-2.0-flash Trend +TFM patchtsmixer | $0.126 \pm 4.194e-4$ | $0.028 \pm 2.052e-4$ | $0.139 \pm 3.086e-4$ | $0.034 \pm 2.179e-4$ | $0.339 \pm 3.58e-2$ | $0.201 \pm 3.404e-2$ |
| gemini-2.0-flash Trend +TFM timemixer | $0.074 \pm 1.176e-3$ | $0.012 \pm 3.675e-4$ | $0.093 \pm 8.504e-4$ | $0.017 \pm 1.986e-4$ | $0.108 \pm 3.383e-4$ | $0.021 \pm 1.557e-4$ |
| gemini-2.0-flash Trend lstm | $0.072 \pm 8.606e-4$ | $0.014 \pm 3.057e-4$ | $0.092 \pm 4.894e-4$ | $0.018 \pm 1.394e-4$ | $0.073 \pm 6.635e-4$ | $0.015 \pm 1.718e-4$ |
| gemini-2.0-flash Trend mlp | $0.073 \pm 2.54e-4$ | $0.013 \pm 9.747e-5$ | $0.093 \pm 6.64e-4$ | $0.018 \pm 2.25e-4$ | $0.074 \pm 3.727e-4$ | $0.015 \pm 1.139e-4$ |
| gemini-2.0-flash Trend patchtsmixer | $0.114 \pm 1.558e-3$ | $0.024 \pm 5.443e-4$ | $0.129 \pm 7.93e-4$ | $0.029 \pm 3.429e-4$ | $0.089 \pm 2.222e-3$ | $0.017 \pm 4.8e-4$ |
| gemini-2.0-flash Trend timemixer | $0.077 \pm 9.486e-4$ | $0.014 \pm 2.533e-4$ | $0.094 \pm 5.857e-4$ | $0.018 \pm 1.921e-4$ | $0.086 \pm 3.859e-4$ | $0.016 \pm 2.121e-4$ |
| gemini-2.0-flash zero_shot +TFM lstm | $0.07 \pm 1.208e-3$ | $0.012 \pm 5.456e-4$ | $0.091 \pm 5.216e-4$ | $0.017 \pm 1.674e-4$ | $0.071 \pm 9.767e-4$ | $0.013 \pm 3.25e-4$ |
| gemini-2.0-flash zero_shot +TFM mlp | $0.076 \pm 7.953e-4$ | $0.012 \pm 2.754e-4$ | $0.093 \pm 5.311e-4$ | $0.017 \pm 1.45e-4$ | $0.078 \pm 6.503e-4$ | $0.014 \pm 1.682e-4$ |
| gemini-2.0-flash zero_shot +TFM patchtsmixer | $0.125 \pm 1.584e-3$ | $0.028 \pm 4.748e-4$ | $0.138 \pm 1.492e-3$ | $0.033 \pm 6.374e-4$ | $0.316 \pm 1.156e-2$ | $0.182 \pm 1.268e-2$ |
| gemini-2.0-flash zero_shot +TFM timemixer | $0.073 \pm 4.336e-4$ | $0.012 \pm 1.443e-4$ | $0.091 \pm 2.318e-4$ | $0.017 \pm 1.015e-4$ | $0.105 \pm 9e-5$ | $0.02 \pm 7.506e-5$ |
| gemini-2.0-flash zero_shot lstm | $0.07 \pm 5.497e-4$ | $0.013 \pm 1.687e-4$ | $0.09 \pm 4.867e-4$ | $0.018 \pm 1.187e-4$ | $0.071 \pm 6.809e-4$ | $0.014 \pm 7.455e-5$ |
| gemini-2.0-flash zero_shot mlp | $0.073 \pm 1.195e-3$ | $0.013 \pm 3.206e-4$ | $0.091 \pm 4.109e-4$ | $0.018 \pm 1.258e-4$ | $0.072 \pm 7.889e-4$ | $0.014 \pm 2.163e-4$ |
| gemini-2.0-flash zero_shot patchtsmixer | $0.112 \pm 7.7e-4$ | $0.023 \pm 3.372e-4$ | $0.128 \pm 8.403e-4$ | $0.028 \pm 2.666e-4$ | $0.088 \pm 6.003e-3$ | $0.017 \pm 1.257e-3$ |
| gemini-2.0-flash zero_shot timemixer | $0.074 \pm 1.52e-3$ | $0.013 \pm 3.207e-4$ | $0.093 \pm 1.556e-3$ | $0.018 \pm 3.88e-4$ | $0.083 \pm 1.11e-3$ | $0.015 \pm 3.31e-4$ |
| interp lstm | $0.051 \pm 5.768e-4$ | $0.008 \pm 7.958e-5$ | $0.068 \pm 1.974e-4$ | $0.011 \pm 4.992e-5$ | $0.048 \pm 2.052e-4$ | $0.008 \pm 2.217e-5$ |
| interp mlp | $0.06 \pm 2.003e-3$ | $0.009 \pm 2.888e-4$ | $0.075 \pm 1.632e-4$ | $0.013 \pm 3.916e-5$ | $0.054 \pm 2.749e-4$ | $0.009 \pm 4.243e-5$ |
| interp patchtsmixer | $0.184 \pm 4.418e-3$ | $0.435 \pm 2.706e-2$ | $0.142 \pm 2.662e-2$ | $0.103 \pm 7.948e-2$ | $0.117 \pm 1.12e-2$ | $0.133 \pm 4.628e-2$ |
| interp timemixer | $0.231 \pm 1.652e-2$ | $0.121 \pm 1.241e-2$ | $0.19 \pm 4.754e-3$ | $0.082 \pm 3.223e-3$ | $0.319 \pm 3.802e-3$ | $0.22 \pm 4.306e-3$ |
| mean lstm | $0.05 \pm 3.677e-4$ | $0.007 \pm 6.557e-5$ | $0.069 \pm 1.621e-4$ | $0.012 \pm 6.455e-5$ | $0.049 \pm 4.51e-4$ | $0.008 \pm 4.193e-5$ |
| mean mlp | $0.054 \pm 3.937e-4$ | $0.008 \pm 3.697e-5$ | $0.072 \pm 6.137e-5$ | $0.012 \pm 2.16e-5$ | $0.051 \pm 7.141e-5$ | $0.008 \pm 1.414e-5$ |
| mean patchtsmixer | $0.106 \pm 3.928e-4$ | $0.04 \pm 2.229e-4$ | $0.108 \pm 2.447e-4$ | $0.035 \pm 4.372e-4$ | $0.082 \pm 4.109e-4$ | $0.028 \pm 2.391e-4$ |
| mean timemixer | $0.195 \pm 1.005e-2$ | $0.09 \pm 7.819e-3$ | $0.165 \pm 1.271e-3$ | $0.06 \pm 7.447e-4$ | $0.262 \pm 3.148e-2$ | $0.167 \pm 2.992e-2$ |
| No_sum_CoT CoT +TFM lstm | $0.065 \pm 5.46e-4$ | $0.011 \pm 9.165e-5$ | $0.083 \pm 5.05e-4$ | $0.015 \pm 1.514e-4$ | $0.064 \pm 1.559e-4$ | $0.011 \pm 4.406e-4$ |
| No_sum_CoT CoT +TFM mlp | $0.068 \pm 1.489e-3$ | $0.01 \pm 3.9e-4$ | $0.084 \pm 1.162e-3$ | $0.014 \pm 2.957e-4$ | $0.067 \pm 1.8e-3$ | $0.011 \pm 4.652e-4$ |
| No_sum_CoT CoT +TFM patchtsmixer | $0.124 \pm 4.937e-4$ | $0.027 \pm 5.429e-4$ | $0.13 \pm 1.521e-2$ | $0.03 \pm 6.112e-3$ | $0.095 \pm 2.757e-2$ | $0.02 \pm 1.131e-2$ |
| No_sum_CoT CoT +TFM timemixer | $0.069 \pm 1.213e-3$ | $0.011 \pm 2.967e-4$ | $0.084 \pm 3.821e-4$ | $0.014 \pm 8.387e-5$ | $0.1 \pm 3.803e-4$ | $0.018 \pm 1.442e-4$ |
| No_sum_CoT lstm | $0.068 \pm 1.919e-4$ | $0.012 \pm 2.517e-5$ | $0.087 \pm 9.539e-4$ | $0.017 \pm 1.947e-4$ | $0.066 \pm 7.127e-4$ | $0.013 \pm 1.836e-4$ |
| No_sum_CoT mlp | $0.068 \pm 4.33e-4$ | $0.011 \pm 1.034e-4$ | $0.087 \pm 2.246e-4$ | $0.016 \pm 6.245e-5$ | $0.068 \pm 5.267e-4$ | $0.013 \pm 1.916e-4$ |
| No_sum_CoT patchtsmixer | $0.11 \pm 4.16e-4$ | $0.022 \pm 2.835e-4$ | $0.126 \pm 3.01e-4$ | $0.028 \pm 1.877e-4$ | $0.091 \pm 7.053e-3$ | $0.017 \pm 1.781e-3$ |
| No_sum_CoT timemixer | $0.071 \pm 1.913e-3$ | $0.012 \pm 3.332e-4$ | $0.088 \pm 1.554e-3$ | $0.016 \pm 4.282e-4$ | $0.078 \pm 5.167e-4$ | $0.014 \pm 1.441e-4$ |
| No_sum_ICD ICD +TFM lstm | $0.064 \pm 1.066e-3$ | $0.011 \pm 2.75e-4$ | $0.082 \pm 3.418e-4$ | $0.015 \pm 1.05e-4$ | $0.063 \pm 9.255e-4$ | $0.011 \pm 1.724e-4$ |
| No_sum_ICD ICD +TFM mlp | $0.068 \pm 4.036e-4$ | $0.01 \pm 1.106e-4$ | $0.084 \pm 3.98e-4$ | $0.014 \pm 1.531e-4$ | $0.067 \pm 1.261e-3$ | $0.011 \pm 3.371e-4$ |
| No_sum_ICD ICD +TFM patchtsmixer | $0.123 \pm 4.822e-4$ | $0.027 \pm 3.707e-4$ | $0.123 \pm 1.306e-2$ | $0.027 \pm 5.384e-3$ | $0.282 \pm 1.625e-1$ | $0.164 \pm 1.377e-1$ |
| No_sum_ICD ICD +TFM timemixer | $0.068 \pm 3.485e-4$ | $0.01 \pm 1.882e-4$ | $0.084 \pm 1.168e-3$ | $0.014 \pm 3.993e-4$ | $0.099 \pm 1.084e-3$ | $0.018 \pm 3.851e-4$ |
| No_sum_ICD lstm | $0.068 \pm 8.225e-4$ | $0.012 \pm 2.161e-4$ | $0.086 \pm 1.076e-3$ | $0.016 \pm 2.791e-4$ | $0.066 \pm 5.363e-4$ | $0.013 \pm 1.179e-4$ |
| No_sum_ICD mlp | $0.068 \pm 1.764e-3$ | $0.012 \pm 4.181e-4$ | $0.087 \pm 8.077e-4$ | $0.016 \pm 2.247e-4$ | $0.068 \pm 6.574e-4$ | $0.013 \pm 1.47e-4$ |
| No_sum_ICD patchtsmixer | $0.112 \pm 1.053e-3$ | $0.023 \pm 3.8e-4$ | $0.126 \pm 3.79e-4$ | $0.028 \pm 1.564e-4$ | $0.089 \pm 3.511e-3$ | $0.017 \pm 6.877e-4$ |
| No_sum_ICD timemixer | $0.071 \pm 1.266e-3$ | $0.012 \pm 3.096e-4$ | $0.087 \pm 8.752e-4$ | $0.016 \pm 2.513e-4$ | $0.079 \pm 7.487e-4$ | $0.014 \pm 2.395e-4$ |
| No_sum_Trend Trend +TFM lstm | $0.064 \pm 3e-3$ | $0.01 \pm 7.238e-4$ | $0.082 \pm 4.706e-4$ | $0.015 \pm 1.732e-4$ | $0.064 \pm 1.618e-3$ | $0.011 \pm 4.07e-4$ |
| No_sum_Trend Trend +TFM mlp | $0.067 \pm 1.328e-3$ | $0.01 \pm 3.251e-4$ | $0.084 \pm 8.851e-4$ | $0.014 \pm 2.506e-4$ | $0.067 \pm 4.674e-4$ | $0.011 \pm 1.365e-4$ |
| No_sum_Trend Trend +TFM patchtsmixer | $0.124 \pm 1.31e-3$ | $0.027 \pm 3.134e-4$ | $0.131 \pm 7.397e-3$ | $0.03 \pm 3.107e-3$ | $0.264 \pm 1.202e-1$ | $0.145 \pm 1.013e-1$ |
| No_sum_Trend Trend +TFM timemixer | $0.069 \pm 9.413e-4$ | $0.01 \pm 1.45e-4$ | $0.084 \pm 6.694e-4$ | $0.014 \pm 1.818e-4$ | $0.1 \pm 1.34e-3$ | $0.018 \pm 5.112e-4$ |

Table 34: In-distribution results - Forecast. (part 1/2)

| Method | hirid $\rightarrow$ hirid | | mimic $\rightarrow$ mimic | | ppicu $\rightarrow$ ppicu | |
|---|---|---|---|---|---|---|
| | masked_mae | masked_mse | masked_mae | masked_mse | masked_mae | masked_mse |
| No_sum_Trend lstm | $0.068 \pm 1.317e-3$ | $0.012 \pm 3.116e-4$ | $0.085 \pm 4.437e-4$ | $0.016 \pm 1.175e-4$ | $0.066 \pm 5.187e-4$ | $0.013 \pm 1.835e-4$ |
| No_sum_Trend mlp | $0.068 \pm 6.357e-4$ | $0.011 \pm 1.617e-4$ | $0.087 \pm 1.297e-3$ | $0.016 \pm 3.724e-4$ | $0.067 \pm 8.032e-4$ | $0.012 \pm 2.634e-4$ |
| No_sum_Trend patchtsmixer | $0.111 \pm 8.596e-4$ | $0.022 \pm 2.076e-4$ | $0.126 \pm 1.205e-3$ | $0.028 \pm 5.758e-4$ | $0.089 \pm 6.671e-3$ | $0.017 \pm 1.798e-3$ |
| No_sum_Trend timemixer | $0.072 \pm 2.463e-3$ | $0.012 \pm 5.846e-4$ | $0.088 \pm 1.163e-3$ | $0.016 \pm 3.008e-4$ | $0.079 \pm 1.129e-3$ | $0.014 \pm 3.756e-4$ |
| medgemma CoT +TFM lstm | $0.069 \pm 2.873e-4$ | $0.012 \pm 5.686e-5$ | $0.088 \pm 6.381e-4$ | $0.016 \pm 1.65e-4$ | $0.069 \pm 1.11e-3$ | $0.013 \pm 3.317e-4$ |
| medgemma CoT +TFM mlp | $0.073 \pm 5.965e-4$ | $0.012 \pm 1.747e-4$ | $0.093 \pm 3.061e-4$ | $0.017 \pm 1.136e-4$ | $0.077 \pm 1.177e-3$ | $0.013 \pm 3.378e-4$ |
| medgemma CoT +TFM patchtsmixer | $0.126 \pm 1.701e-4$ | $0.028 \pm 2.136e-4$ | $0.138 \pm 1.191e-3$ | $0.033 \pm 4.423e-4$ | $0.426 \pm 2.746e-2$ | $0.295 \pm 3.72e-2$ |
| medgemma CoT +TFM timemixer | $0.072 \pm 1.163e-3$ | $0.011 \pm 2.166e-4$ | $0.09 \pm 7.181e-4$ | $0.016 \pm 2.615e-4$ | $0.106 \pm 1.041e-3$ | $0.02 \pm 4.748e-4$ |
| medgemma CoT lstm | $0.069 \pm 6.06e-4$ | $0.013 \pm 1.515e-4$ | $0.088 \pm 6.359e-4$ | $0.017 \pm 1.708e-4$ | $0.07 \pm 6.807e-4$ | $0.014 \pm 1.799e-4$ |
| medgemma CoT mlp | $0.071 \pm 5.875e-4$ | $0.013 \pm 1.431e-4$ | $0.09 \pm 7.285e-4$ | $0.017 \pm 2.234e-4$ | $0.072 \pm 4.666e-4$ | $0.014 \pm 1.45e-4$ |
| medgemma CoT patchtsmixer | $0.112 \pm 1.6e-3$ | $0.023 \pm 4.853e-4$ | $0.128 \pm 6.378e-4$ | $0.028 \pm 1.309e-4$ | $0.092 \pm 4.877e-3$ | $0.018 \pm 1.024e-3$ |
| medgemma CoT timemixer | $0.075 \pm 2.101e-3$ | $0.013 \pm 3.557e-4$ | $0.091 \pm 8.831e-4$ | $0.017 \pm 2.701e-4$ | $0.083 \pm 8.314e-4$ | $0.015 \pm 3.74e-4$ |
| medgemma ICD +TFM lstm | $0.072 \pm 1.752e-3$ | $0.013 \pm 6.251e-4$ | $0.09 \pm 1.406e-3$ | $0.017 \pm 3.769e-4$ | $0.072 \pm 1.13e-3$ | $0.014 \pm 3.703e-4$ |
| medgemma ICD +TFM mlp | $0.076 \pm 6.813e-4$ | $0.013 \pm 1.93e-4$ | $0.094 \pm 2.524e-4$ | $0.018 \pm 9.504e-5$ | $0.079 \pm 1.08e-3$ | $0.014 \pm 3.46e-4$ |
| medgemma ICD +TFM patchtsmixer | $0.125 \pm 1.208e-3$ | $0.028 \pm 3.5e-4$ | $0.139 \pm 3.731e-4$ | $0.033 \pm 2.201e-4$ | $0.419 \pm 3.58e-2$ | $0.279 \pm 4.63e-2$ |
| medgemma ICD +TFM timemixer | $0.074 \pm 1.771e-3$ | $0.012 \pm 3.913e-4$ | $0.092 \pm 7.543e-4$ | $0.017 \pm 2.829e-4$ | $0.107 \pm 3.934e-4$ | $0.02 \pm 1.861e-4$ |
| medgemma ICD lstm | $0.071 \pm 4.095e-4$ | $0.013 \pm 1.193e-4$ | $0.091 \pm 3.183e-4$ | $0.018 \pm 1.303e-4$ | $0.072 \pm 7.333e-4$ | $0.015 \pm 1.498e-4$ |
| medgemma ICD mlp | $0.073 \pm 9.869e-4$ | $0.013 \pm 2.471e-4$ | $0.091 \pm 5.297e-4$ | $0.018 \pm 1.841e-4$ | $0.074 \pm 4.049e-4$ | $0.015 \pm 1.207e-4$ |
| medgemma ICD patchtsmixer | $0.113 \pm 3.482e-4$ | $0.023 \pm 1.571e-4$ | $0.128 \pm 5.202e-4$ | $0.029 \pm 2.037e-4$ | $0.093 \pm 6.342e-3$ | $0.018 \pm 1.394e-3$ |
| medgemma ICD timemixer | $0.076 \pm 1.618e-3$ | $0.014 \pm 2.87e-4$ | $0.094 \pm 1.292e-3$ | $0.018 \pm 3.181e-4$ | $0.085 \pm 8.767e-4$ | $0.016 \pm 1.905e-4$ |
| medgemma Trend +TFM lstm | $0.07 \pm 4.701e-4$ | $0.012 \pm 1.778e-4$ | $0.09 \pm 9.335e-4$ | $0.017 \pm 2.538e-4$ | $0.071 \pm 1.38e-3$ | $0.014 \pm 4.46e-4$ |
| medgemma Trend +TFM mlp | $0.075 \pm 5.048e-4$ | $0.012 \pm 1.498e-4$ | $0.092 \pm 6.178e-4$ | $0.017 \pm 1.562e-4$ | $0.078 \pm 1.407e-3$ | $0.014 \pm 4.779e-4$ |
| medgemma Trend +TFM patchtsmixer | $0.126 \pm 1.025e-3$ | $0.028 \pm 1.95e-4$ | $0.134 \pm 1.056e-2$ | $0.031 \pm 4.599e-3$ | $0.375 \pm 4.543e-2$ | $0.238 \pm 4.575e-2$ |
| medgemma Trend +TFM timemixer | $0.073 \pm 1.142e-3$ | $0.012 \pm 4.309e-4$ | $0.091 \pm 3.443e-4$ | $0.017 \pm 1.277e-4$ | $0.107 \pm 1.32e-3$ | $0.021 \pm 5.525e-4$ |
| medgemma Trend lstm | $0.071 \pm 4.996e-4$ | $0.013 \pm 1.388e-4$ | $0.09 \pm 2.858e-4$ | $0.018 \pm 7.789e-5$ | $0.073 \pm 3.869e-4$ | $0.015 \pm 1.958e-4$ |
| medgemma Trend mlp | $0.072 \pm 7.023e-4$ | $0.013 \pm 1.992e-4$ | $0.092 \pm 3.242e-4$ | $0.018 \pm 7.724e-5$ | $0.074 \pm 4.473e-4$ | $0.015 \pm 1.184e-4$ |
| medgemma Trend patchtsmixer | $0.113 \pm 1.433e-3$ | $0.023 \pm 3.029e-4$ | $0.128 \pm 1.315e-3$ | $0.029 \pm 4.507e-4$ | $0.095 \pm 4.215e-3$ | $0.019 \pm 9.673e-4$ |
| medgemma Trend timemixer | $0.075 \pm 1.406e-3$ | $0.014 \pm 3.356e-4$ | $0.094 \pm 9.093e-4$ | $0.018 \pm 3.172e-4$ | $0.086 \pm 2.637e-4$ | $0.016 \pm 1.499e-4$ |
| medgemma zero_shot +TFM lstm | $0.069 \pm 2.157e-4$ | $0.012 \pm 7.095e-5$ | $0.09 \pm 2.8e-4$ | $0.017 \pm 8.718e-5$ | $0.071 \pm 2.487e-3$ | $0.014 \pm 6.589e-4$ |
| medgemma zero_shot +TFM mlp | $0.075 \pm 1.15e-3$ | $0.012 \pm 3.724e-4$ | $0.093 \pm 8.479e-4$ | $0.017 \pm 2.829e-4$ | $0.078 \pm 5.486e-4$ | $0.014 \pm 1.484e-4$ |
| medgemma zero_shot +TFM patchtsmixer | $0.125 \pm 1.021e-3$ | $0.028 \pm 5.116e-4$ | $0.13 \pm 1.54e-2$ | $0.03 \pm 6.322e-3$ | $0.439 \pm 4.731e-2$ | $0.31 \pm 4.83e-2$ |
| medgemma zero_shot +TFM timemixer | $0.073 \pm 7.679e-4$ | $0.012 \pm 2.409e-4$ | $0.091 \pm 7.674e-4$ | $0.017 \pm 2.663e-4$ | $0.105 \pm 1.015e-3$ | $0.02 \pm 3.544e-4$ |
| medgemma zero_shot lstm | $0.07 \pm 3.414e-4$ | $0.013 \pm 6.782e-5$ | $0.09 \pm 1.015e-3$ | $0.018 \pm 2.858e-4$ | $0.071 \pm 5.2e-4$ | $0.014 \pm 1.564e-4$ |
| medgemma zero_shot mlp | $0.072 \pm 2.818e-4$ | $0.013 \pm 1.03e-4$ | $0.091 \pm 6.299e-4$ | $0.018 \pm 1.88e-4$ | $0.073 \pm 9.287e-4$ | $0.014 \pm 2.866e-4$ |
| medgemma zero_shot patchtsmixer | $0.112 \pm 1.105e-3$ | $0.023 \pm 2.934e-4$ | $0.128 \pm 9.704e-4$ | $0.028 \pm 2.298e-4$ | $0.092 \pm 8.414e-3$ | $0.018 \pm 2.275e-3$ |
| medgemma zero_shot timemixer | $0.074 \pm 2.111e-3$ | $0.013 \pm 4.833e-4$ | $0.093 \pm 5.915e-4$ | $0.018 \pm 1.072e-4$ | $0.084 \pm 7.86e-4$ | $0.016 \pm 1.688e-4$ |
| No_sum_zero_shot zero_shot +TFM lstm | $0.065 \pm 1.6e-3$ | $0.011 \pm 4.01e-4$ | $0.083 \pm 1.504e-3$ | $0.015 \pm 3.972e-4$ | $0.064 \pm 1.191e-3$ | $0.011 \pm 2.845e-4$ |
| No_sum_zero_shot zero_shot +TFM mlp | $0.067 \pm 6.681e-4$ | $0.01 \pm 1.626e-4$ | $0.084 \pm 8.605e-4$ | $0.015 \pm 2.631e-4$ | $0.068 \pm 1.01e-3$ | $0.011 \pm 2.639e-4$ |
| No_sum_zero_shot zero_shot +TFM patchtsmixer | $0.124 \pm 1.164e-3$ | $0.027 \pm 5.45e-4$ | $0.124 \pm 1.319e-2$ | $0.027 \pm 5.462e-3$ | $0.256 \pm 1.429e-1$ | $0.144 \pm 1.174e-1$ |
| No_sum_zero_shot zero_shot +TFM timemixer | $0.068 \pm 1.837e-3$ | $0.01 \pm 4.328e-4$ | $0.085 \pm 1.501e-3$ | $0.015 \pm 3.487e-4$ | $0.1 \pm 7.862e-4$ | $0.018 \pm 2.641e-4$ |
| No_sum_zero_shot lstm | $0.067 \pm 1.073e-3$ | $0.012 \pm 2.413e-4$ | $0.086 \pm 1.129e-3$ | $0.016 \pm 3.381e-4$ | $0.066 \pm 8.154e-4$ | $0.013 \pm 2.403e-4$ |
| No_sum_zero_shot mlp | $0.068 \pm 8.283e-4$ | $0.012 \pm 2.077e-4$ | $0.086 \pm 9.896e-4$ | $0.016 \pm 2.701e-4$ | $0.068 \pm 4.37e-4$ | $0.013 \pm 1.22e-4$ |
| No_sum_zero_shot patchtsmixer | $0.11 \pm 4.895e-4$ | $0.022 \pm 2.851e-4$ | $0.126 \pm 5.631e-4$ | $0.028 \pm 2.757e-4$ | $0.089 \pm 8.783e-3$ | $0.017 \pm 2.339e-3$ |
| No_sum_zero_shot timemixer | $0.072 \pm 2.877e-3$ | $0.012 \pm 7.297e-4$ | $0.088 \pm 8.869e-4$ | $0.016 \pm 2.265e-4$ | $0.08 \pm 1.347e-3$ | $0.016 \pm 3.821e-4$ |
| right lstm | $0.049 \pm 3.421e-4$ | $0.007 \pm 5.62e-5$ | $0.067 \pm 9.956e-4$ | $0.011 \pm 1.228e-4$ | $0.047 \pm 1.752e-4$ | $0.008 \pm 2.646e-5$ |
| right mlp | $0.056 \pm 1.771e-4$ | $0.008 \pm 2.646e-5$ | $0.072 \pm 1.945e-4$ | $0.012 \pm 1.258e-5$ | $0.052 \pm 2.052e-4$ | $0.008 \pm 1.893e-5$ |
| right patchtsmixer | $0.109 \pm 5.173e-4$ | $0.041 \pm 3.377e-4$ | $0.11 \pm 3.675e-4$ | $0.037 \pm 5.371e-4$ | $0.083 \pm 9.554e-4$ | $0.031 \pm 6.364e-4$ |
| right timemixer | $0.156 \pm 2.139e-3$ | $0.068 \pm 1.209e-3$ | $0.134 \pm 7.387e-4$ | $0.046 \pm 9.389e-4$ | $0.228 \pm 4.642e-3$ | $0.141 \pm 3.671e-3$ |

Table 35: In-distribution results - Forecast. (part 2/2)

| Method | hirid → ppicu | | mimic → ppicu | |
|---|---|---|---|---|
| | masked_mae | masked_mse | masked_mae | masked_mse |
| Llama-3.1 CoT +TFM lstm | $0.233 \pm 1.151e-2$ | $0.12 \pm 8.28e-3$ | $0.263 \pm 5.289e-4$ | $0.151 \pm 2.991e-3$ |
| Llama-3.1 CoT +TFM mlp | $0.257 \pm 1.285e-3$ | $0.138 \pm 2.024e-3$ | $0.266 \pm 8.977e-4$ | $0.155 \pm 1.573e-3$ |
| Llama-3.1 CoT +TFM patchtsmixer | $0.309 \pm 1.042e-2$ | $0.209 \pm 2.272e-2$ | $0.36 \pm 6.736e-3$ | $0.267 \pm 1.614e-2$ |
| Llama-3.1 CoT +TFM timemixer | $0.264 \pm 5.936e-3$ | $0.141 \pm 4.233e-3$ | $0.27 \pm 1.086e-3$ | $0.155 \pm 1.749e-3$ |
| Llama-3.1 CoT lstm | $0.249 \pm 3.484e-3$ | $0.14 \pm 4.978e-3$ | $0.265 \pm 1.101e-3$ | $0.159 \pm 1.279e-3$ |
| Llama-3.1 CoT mlp | $0.254 \pm 1.175e-3$ | $0.144 \pm 1.125e-3$ | $0.266 \pm 4.836e-4$ | $0.16 \pm 5.539e-4$ |
| Llama-3.1 CoT patchtsmixer | $0.301 \pm 3.518e-3$ | $0.203 \pm 1.327e-2$ | $0.35 \pm 1.025e-2$ | $0.255 \pm 7.716e-3$ |
| Llama-3.1 CoT timemixer | $0.257 \pm 2.158e-3$ | $0.147 \pm 1.634e-3$ | $0.273 \pm 1.845e-3$ | $0.165 \pm 1.135e-3$ |
| Llama-3.1 ICD +TFM lstm | $0.244 \pm 3.883e-3$ | $0.129 \pm 2.99e-3$ | $0.259 \pm 6.222e-3$ | $0.146 \pm 7.989e-3$ |
| Llama-3.1 ICD +TFM mlp | $0.255 \pm 7.535e-4$ | $0.139 \pm 8.615e-4$ | $0.262 \pm 7.825e-4$ | $0.148 \pm 1.253e-3$ |
| Llama-3.1 ICD +TFM patchtsmixer | $0.302 \pm 7.799e-3$ | $0.19 \pm 1.528e-2$ | $0.368 \pm 1.021e-2$ | $0.279 \pm 2.077e-2$ |
| Llama-3.1 ICD +TFM timemixer | $0.258 \pm 3.675e-3$ | $0.139 \pm 2.935e-3$ | $0.268 \pm 4.168e-3$ | $0.154 \pm 6.667e-3$ |
| Llama-3.1 ICD lstm | $0.248 \pm 6.559e-4$ | $0.14 \pm 9.291e-4$ | $0.265 \pm 8.688e-4$ | $0.16 \pm 1.093e-3$ |
| Llama-3.1 ICD mlp | $0.248 \pm 8.898e-4$ | $0.14 \pm 9.803e-4$ | $0.267 \pm 9.823e-4$ | $0.161 \pm 9.952e-4$ |
| Llama-3.1 ICD patchtsmixer | $0.296 \pm 4.522e-3$ | $0.191 \pm 1.087e-2$ | $0.349 \pm 5.127e-3$ | $0.259 \pm 1.508e-2$ |
| Llama-3.1 ICD timemixer | $0.254 \pm 1.418e-3$ | $0.144 \pm 1.737e-3$ | $0.272 \pm 1.191e-3$ | $0.165 \pm 1.344e-3$ |
| Llama-3.1 Trend +TFM lstm | $0.241 \pm 9.205e-3$ | $0.127 \pm 8.831e-3$ | $0.265 \pm 2.777e-3$ | $0.154 \pm 4.278e-3$ |
| Llama-3.1 Trend +TFM mlp | $0.255 \pm 2.103e-3$ | $0.136 \pm 3.044e-3$ | $0.264 \pm 1.337e-3$ | $0.152 \pm 1.552e-3$ |
| Llama-3.1 Trend +TFM patchtsmixer | $0.309 \pm 2.944e-3$ | $0.212 \pm 1.078e-2$ | $0.35 \pm 6.284e-3$ | $0.244 \pm 1.247e-2$ |
| Llama-3.1 Trend +TFM timemixer | $0.255 \pm 3.634e-3$ | $0.135 \pm 4.983e-3$ | $0.27 \pm 7.901e-4$ | $0.153 \pm 2.791e-3$ |
| Llama-3.1 Trend lstm | $0.249 \pm 2.012e-3$ | $0.141 \pm 2.458e-3$ | $0.265 \pm 9.183e-4$ | $0.16 \pm 6.172e-4$ |
| Llama-3.1 Trend mlp | $0.251 \pm 6.926e-4$ | $0.142 \pm 7.046e-4$ | $0.268 \pm 1.058e-3$ | $0.161 \pm 8.152e-4$ |
| Llama-3.1 Trend patchtsmixer | $0.299 \pm 7.225e-3$ | $0.207 \pm 2.734e-2$ | $0.343 \pm 6.525e-3$ | $0.242 \pm 8.377e-3$ |
| Llama-3.1 Trend timemixer | $0.254 \pm 2.033e-3$ | $0.144 \pm 1.564e-3$ | $0.273 \pm 1.936e-3$ | $0.165 \pm 1.44e-3$ |
| Llama-3.1 zero_shot +TFM lstm | $0.236 \pm 9.481e-3$ | $0.122 \pm 9.524e-3$ | $0.263 \pm 2.426e-3$ | $0.151 \pm 2.222e-3$ |
| Llama-3.1 zero_shot +TFM mlp | $0.254 \pm 9.672e-4$ | $0.133 \pm 2.668e-3$ | $0.259 \pm 1.655e-3$ | $0.142 \pm 3.406e-3$ |
| Llama-3.1 zero_shot +TFM patchtsmixer | $0.307 \pm 1.225e-2$ | $0.208 \pm 5.803e-3$ | $0.359 \pm 1.014e-2$ | $0.259 \pm 1.957e-2$ |
| Llama-3.1 zero_shot +TFM timemixer | $0.258 \pm 4.549e-3$ | $0.135 \pm 3.992e-3$ | $0.27 \pm 2.229e-3$ | $0.155 \pm 3.77e-3$ |
| Llama-3.1 zero_shot lstm | $0.248 \pm 2.096e-3$ | $0.139 \pm 3.128e-3$ | $0.265 \pm 8.146e-4$ | $0.16 \pm 8.607e-4$ |
| Llama-3.1 zero_shot mlp | $0.252 \pm 1.013e-3$ | $0.143 \pm 1.042e-3$ | $0.266 \pm 3.319e-4$ | $0.16 \pm 4.706e-4$ |
| Llama-3.1 zero_shot patchtsmixer | $0.293 \pm 3.942e-3$ | $0.183 \pm 5.569e-3$ | $0.353 \pm 4.407e-3$ | $0.254 \pm 1.169e-2$ |
| Llama-3.1 zero_shot timemixer | $0.254 \pm 1.517e-3$ | $0.143 \pm 1.546e-3$ | $0.274 \pm 3.347e-3$ | $0.165 \pm 2.725e-3$ |
| TFM lstm | $0.256 \pm 1.187e-3$ | $0.148 \pm 8.907e-4$ | $0.271 \pm 2.313e-3$ | $0.165 \pm 1.201e-3$ |
| TFM mlp | $0.252 \pm 2.536e-4$ | $0.142 \pm 3.369e-4$ | $0.273 \pm 7.091e-4$ | $0.167 \pm 7.996e-4$ |
| TFM patchtsmixer | $0.313 \pm 5.429e-3$ | $0.217 \pm 1.902e-2$ | $0.365 \pm 1.016e-2$ | $0.284 \pm 2.297e-2$ |
| TFM timemixer | $0.255 \pm 1.952e-3$ | $0.144 \pm 2.339e-3$ | $0.272 \pm 1.852e-3$ | $0.165 \pm 1.846e-3$ |
| TSDE lstm | $0.441 \pm 5.705e-2$ | $0.284 \pm 7.197e-2$ | $0.317 \pm 7.954e-2$ | $0.172 \pm 7.826e-2$ |
| TSDE mlp | $0.329 \pm 3.907e-2$ | $0.169 \pm 3.646e-2$ | $0.425 \pm 3.644e-2$ | $0.313 \pm 4.37e-2$ |
| TSDE patchtsmixer | $0.311 \pm 4.882e-3$ | $0.209 \pm 9.028e-3$ | $0.361 \pm 4.177e-3$ | $0.269 \pm 1.2e-2$ |
| TSDE timemixer | $0.379 \pm 4.678e-2$ | $0.23 \pm 6.628e-2$ | $0.309 \pm 1.989e-2$ | $0.167 \pm 1.677e-2$ |
| gemini-2.0-flash CoT +TFM lstm | $0.245 \pm 7.668e-3$ | $0.129 \pm 8.157e-3$ | $0.264 \pm 7.638e-3$ | $0.152 \pm 1.099e-2$ |
| gemini-2.0-flash CoT +TFM mlp | $0.256 \pm 2.95e-4$ | $0.137 \pm 6.799e-4$ | $0.271 \pm 5.314e-4$ | $0.158 \pm 4.912e-4$ |
| gemini-2.0-flash CoT +TFM patchtsmixer | $0.303 \pm 5.928e-3$ | $0.194 \pm 1.536e-2$ | $0.35 \pm 1.437e-2$ | $0.249 \pm 1.319e-2$ |
| gemini-2.0-flash CoT +TFM timemixer | $0.262 \pm 4.926e-3$ | $0.14 \pm 1.162e-3$ | $0.271 \pm 4.388e-3$ | $0.152 \pm 3.192e-3$ |
| gemini-2.0-flash CoT lstm | $0.248 \pm 2.401e-3$ | $0.137 \pm 2.332e-3$ | $0.266 \pm 6.472e-4$ | $0.16 \pm 1.095e-3$ |
| gemini-2.0-flash CoT mlp | $0.251 \pm 1.123e-4$ | $0.142 \pm 1.458e-4$ | $0.267 \pm 8.293e-4$ | $0.16 \pm 8.07e-4$ |
| gemini-2.0-flash CoT patchtsmixer | $0.297 \pm 1.961e-3$ | $0.194 \pm 6.499e-3$ | $0.345 \pm 8.73e-3$ | $0.243 \pm 1.871e-2$ |
| gemini-2.0-flash CoT timemixer | $0.259 \pm 1.743e-3$ | $0.147 \pm 1.713e-3$ | $0.272 \pm 3.148e-3$ | $0.163 \pm 3.351e-3$ |
| gemini-2.0-flash ICD +TFM lstm | $0.242 \pm 1.895e-3$ | $0.128 \pm 1.697e-3$ | $0.263 \pm 5.395e-3$ | $0.148 \pm 6.053e-3$ |
| gemini-2.0-flash ICD +TFM mlp | $0.254 \pm 1.227e-3$ | $0.131 \pm 1.766e-3$ | $0.269 \pm 1.092e-3$ | $0.156 \pm 1.055e-3$ |
| gemini-2.0-flash ICD +TFM patchtsmixer | $0.309 \pm 2.339e-3$ | $0.202 \pm 6.27e-3$ | $0.359 \pm 1.048e-2$ | $0.258 \pm 1.893e-2$ |
| gemini-2.0-flash ICD +TFM timemixer | $0.263 \pm 1.018e-3$ | $0.136 \pm 1.011e-3$ | $0.267 \pm 5.24e-3$ | $0.149 \pm 6.799e-3$ |
| gemini-2.0-flash ICD lstm | $0.247 \pm 5.091e-3$ | $0.138 \pm 6.117e-3$ | $0.263 \pm 1.068e-3$ | $0.157 \pm 1.731e-3$ |
| gemini-2.0-flash ICD mlp | $0.25 \pm 1.183e-3$ | $0.14 \pm 1.361e-3$ | $0.266 \pm 8.483e-4$ | $0.159 \pm 8.67e-4$ |
| gemini-2.0-flash ICD patchtsmixer | $0.299 \pm 8.339e-3$ | $0.2 \pm 1.925e-2$ | $0.342 \pm 6.903e-3$ | $0.243 \pm 1.72e-2$ |
| gemini-2.0-flash ICD timemixer | $0.259 \pm 2.312e-3$ | $0.147 \pm 2.326e-3$ | $0.272 \pm 3.99e-3$ | $0.164 \pm 3.513e-3$ |
| gemini-2.0-flash Trend +TFM lstm | $0.223 \pm 2.934e-3$ | $0.112 \pm 1.287e-3$ | $0.258 \pm 5.156e-3$ | $0.144 \pm 6.115e-3$ |
| gemini-2.0-flash Trend +TFM mlp | $0.255 \pm 7.559e-4$ | $0.132 \pm 2.128e-3$ | $0.265 \pm 6.888e-4$ | $0.15 \pm 1.064e-3$ |
| gemini-2.0-flash Trend +TFM patchtsmixer | $0.304 \pm 4.473e-3$ | $0.195 \pm 9.066e-3$ | $0.362 \pm 9.5e-3$ | $0.267 \pm 1.868e-2$ |
| gemini-2.0-flash Trend +TFM timemixer | $0.259 \pm 2.829e-3$ | $0.141 \pm 5.123e-3$ | $0.271 \pm 2.502e-3$ | $0.152 \pm 4.551e-3$ |
| gemini-2.0-flash Trend lstm | $0.246 \pm 2.201e-3$ | $0.138 \pm 2.734e-3$ | $0.264 \pm 6.985e-4$ | $0.158 \pm 8.871e-4$ |
| gemini-2.0-flash Trend mlp | $0.248 \pm 9.524e-4$ | $0.14 \pm 9.351e-4$ | $0.268 \pm 2.051e-3$ | $0.161 \pm 1.963e-3$ |
| gemini-2.0-flash Trend patchtsmixer | $0.304 \pm 2.698e-3$ | $0.216 \pm 1.305e-2$ | $0.347 \pm 1.166e-2$ | $0.244 \pm 1.992e-2$ |
| gemini-2.0-flash Trend timemixer | $0.254 \pm 1.467e-3$ | $0.144 \pm 1.907e-3$ | $0.272 \pm 2.424e-3$ | $0.164 \pm 2.789e-3$ |
| gemini-2.0-flash zero_shot +TFM lstm | $0.237 \pm 8.483e-3$ | $0.123 \pm 8.268e-3$ | $0.269 \pm 7.437e-3$ | $0.157 \pm 8.831e-3$ |
| gemini-2.0-flash zero_shot +TFM mlp | $0.257 \pm 2.318e-4$ | $0.14 \pm 3.055e-4$ | $0.27 \pm 1.445e-3$ | $0.158 \pm 1.491e-3$ |
| gemini-2.0-flash zero_shot +TFM patchtsmixer | $0.322 \pm 1.483e-3$ | $0.237 \pm 1.498e-2$ | $0.349 \pm 2.874e-3$ | $0.241 \pm 7.466e-3$ |
| gemini-2.0-flash zero_shot +TFM timemixer | $0.257 \pm 2.596e-3$ | $0.137 \pm 4.234e-3$ | $0.271 \pm 1.195e-3$ | $0.152 \pm 1.976e-3$ |
| gemini-2.0-flash zero_shot lstm | $0.25 \pm 2.993e-3$ | $0.141 \pm 3.455e-3$ | $0.267 \pm 5.898e-4$ | $0.16 \pm 5.229e-4$ |
| gemini-2.0-flash zero_shot mlp | $0.253 \pm 1.152e-3$ | $0.144 \pm 8.367e-4$ | $0.27 \pm 1.568e-3$ | $0.163 \pm 1.278e-3$ |
| gemini-2.0-flash zero_shot patchtsmixer | $0.297 \pm 8.402e-3$ | $0.193 \pm 2.072e-2$ | $0.353 \pm 1.37e-2$ | $0.261 \pm 2.067e-2$ |
| gemini-2.0-flash zero_shot timemixer | $0.256 \pm 6.253e-4$ | $0.146 \pm 4.215e-4$ | $0.272 \pm 1.866e-3$ | $0.164 \pm 1.651e-3$ |
| interp lstm | $0.326 \pm 7.263e-3$ | $0.217 \pm 5.938e-3$ | $0.383 \pm 4.215e-3$ | $0.276 \pm 2.743e-3$ |
| interp mlp | $0.509 \pm 5.545e-2$ | $0.466 \pm 9.495e-2$ | $0.507 \pm 1.317e-2$ | $0.427 \pm 1.943e-2$ |
| interp patchtsmixer | $0.601 \pm 4.662e-2$ | $24.8 \pm 7.965e0$ | $0.83 \pm 3.299e-2$ | $13.411 \pm 3.412e0$ |
| interp timemixer | $0.59 \pm 2.228e-2$ | $0.614 \pm 7.488e-2$ | $0.56 \pm 1.332e-2$ | $0.561 \pm 4.048e-2$ |
| mean lstm | $0.297 \pm 3.83e-3$ | $0.197 \pm 1.58e-3$ | $0.38 \pm 9.415e-4$ | $0.261 \pm 1.091e-3$ |
| mean mlp | $0.412 \pm 9.945e-3$ | $0.292 \pm 9.692e-3$ | $0.402 \pm 1.338e-2$ | $0.282 \pm 1.586e-2$ |
| mean patchtsmixer | $0.259 \pm 4.115e-3$ | $0.306 \pm 1.698e-2$ | $0.448 \pm 3.38e-2$ | $4.839 \pm 1.345e0$ |
| mean timemixer | $0.575 \pm 3.447e-2$ | $0.541 \pm 1.402e-2$ | $0.557 \pm 7.886e-3$ | $0.51 \pm 1.44e-2$ |
| No_sum_CoT CoT +TFM lstm | $0.241 \pm 4.677e-3$ | $0.125 \pm 5.111e-3$ | $0.263 \pm 2.178e-3$ | $0.15 \pm 2.378e-3$ |
| No_sum_CoT CoT +TFM mlp | $0.26 \pm 3.366e-3$ | $0.144 \pm 4.77e-3$ | $0.271 \pm 3.026e-3$ | $0.155 \pm 4.368e-3$ |
| No_sum_CoT CoT +TFM patchtsmixer | $0.31 \pm 6.41e-3$ | $0.208 \pm 1.751e-2$ | $0.361 \pm 1.783e-3$ | $0.269 \pm 5.946e-3$ |
| No_sum_CoT CoT +TFM timemixer | $0.258 \pm 5.222e-3$ | $0.133 \pm 2.969e-3$ | $0.267 \pm 6.478e-3$ | $0.145 \pm 9.273e-3$ |
| No_sum_CoT lstm | $0.245 \pm 1.906e-3$ | $0.138 \pm 1.487e-3$ | $0.271 \pm 2.812e-3$ | $0.163 \pm 3.417e-3$ |
| No_sum_CoT mlp | $0.252 \pm 9.53e-4$ | $0.145 \pm 6.666e-4$ | $0.283 \pm 2.47e-3$ | $0.174 \pm 2.155e-3$ |
| No_sum_CoT patchtsmixer | $0.302 \pm 3.796e-3$ | $0.214 \pm 1.484e-2$ | $0.351 \pm 1.451e-2$ | $0.25 \pm 3.05e-2$ |
| No_sum_CoT timemixer | $0.251 \pm 2.86e-3$ | $0.141 \pm 2.135e-3$ | $0.274 \pm 4.468e-3$ | $0.163 \pm 5.064e-3$ |
| No_sum_ICD ICD +TFM lstm | $0.245 \pm 2.003e-3$ | $0.132 \pm 2.688e-3$ | $0.265 \pm 3.819e-3$ | $0.152 \pm 5.225e-3$ |
| No_sum_ICD ICD +TFM mlp | $0.264 \pm 6.594e-3$ | $0.148 \pm 7.287e-3$ | $0.277 \pm 2.85e-3$ | $0.163 \pm 2.906e-3$ |
| No_sum_ICD ICD +TFM patchtsmixer | $0.311 \pm 9.231e-3$ | $0.22 \pm 3.153e-2$ | $0.353 \pm 2.39e-3$ | $0.267 \pm 1.882e-2$ |
| No_sum_ICD ICD +TFM timemixer | $0.258 \pm 3.285e-3$ | $0.139 \pm 5.799e-3$ | $0.272 \pm 5.074e-3$ | $0.155 \pm 8.076e-3$ |
| No_sum_ICD lstm | $0.243 \pm 10e-4$ | $0.138 \pm 1.574e-3$ | $0.272 \pm 1.379e-3$ | $0.163 \pm 9.315e-4$ |
| No_sum_ICD mlp | $0.248 \pm 1.423e-3$ | $0.141 \pm 1.563e-3$ | $0.276 \pm 2.058e-3$ | $0.168 \pm 1.711e-3$ |
| No_sum_ICD patchtsmixer | $0.3 \pm 5.824e-3$ | $0.216 \pm 2.673e-2$ | $0.356 \pm 7.101e-3$ | $0.263 \pm 1.785e-2$ |
| No_sum_ICD timemixer | $0.248 \pm 2.058e-3$ | $0.14 \pm 1.885e-3$ | $0.28 \pm 4.925e-3$ | $0.17 \pm 3.804e-3$ |
| No_sum_Trend Trend +TFM lstm | $0.243 \pm 2.09e-3$ | $0.129 \pm 3.814e-3$ | $0.266 \pm 1.709e-3$ | $0.154 \pm 8.954e-4$ |
| No_sum_Trend Trend +TFM mlp | $0.261 \pm 2.61e-3$ | $0.145 \pm 3.731e-3$ | $0.277 \pm 3.419e-3$ | $0.163 \pm 3.859e-3$ |
| No_sum_Trend Trend +TFM patchtsmixer | $0.308 \pm 2.055e-3$ | $0.204 \pm 4.42e-3$ | $0.362 \pm 1.132e-2$ | $0.277 \pm 3.123e-2$ |
| No_sum_Trend Trend +TFM timemixer | $0.262 \pm 2.033e-3$ | $0.138 \pm 4.333e-3$ | $0.272 \pm 1.25e-3$ | $0.149 \pm 1.171e-3$ |

Table 36: Cross-site transfer results - Forecast. (part 1/2)

| Method | hirid → ppicu | | mimic → ppicu | |
|---|---|---|---|---|
| | masked_mae | masked_mse | masked_mae | masked_mse |
| No_sum_Trend lstm | $0.249 \pm 2.623e-3$ | $0.143 \pm 2.817e-3$ | $0.271 \pm 1.111e-3$ | $0.163 \pm 1.111e-3$ |
| No_sum_Trend mlp | $0.253 \pm 2.225e-3$ | $0.145 \pm 2.026e-3$ | $0.284 \pm 2.136e-3$ | $0.175 \pm 1.803e-3$ |
| No_sum_Trend patchtsmixer | $0.293 \pm 5.256e-3$ | $0.19 \pm 1.041e-2$ | $0.345 \pm 9.259e-3$ | $0.243 \pm 1.885e-2$ |
| No_sum_Trend timemixer | $0.253 \pm 2.217e-3$ | $0.144 \pm 2.074e-3$ | $0.277 \pm 2.751e-3$ | $0.169 \pm 3.138e-3$ |
| medgemma CoT +TFM lstm | $0.24 \pm 6.816e-3$ | $0.125 \pm 8.342e-3$ | $0.269 \pm 7.465e-4$ | $0.157 \pm 3.317e-3$ |
| medgemma CoT +TFM mlp | $0.257 \pm 1.504e-3$ | $0.137 \pm 9.943e-4$ | $0.268 \pm 1.119e-3$ | $0.154 \pm 1.309e-3$ |
| medgemma CoT +TFM patchtsmixer | $0.305 \pm 1.024e-2$ | $0.193 \pm 2.069e-2$ | $0.365 \pm 1.438e-2$ | $0.281 \pm 3.208e-2$ |
| medgemma CoT +TFM timemixer | $0.262 \pm 3.617e-3$ | $0.138 \pm 2.563e-3$ | $0.267 \pm 3.583e-3$ | $0.15 \pm 5.676e-3$ |
| medgemma CoT lstm | $0.249 \pm 7.646e-4$ | $0.141 \pm 4.029e-4$ | $0.265 \pm 1.296e-3$ | $0.16 \pm 1.675e-3$ |
| medgemma CoT mlp | $0.251 \pm 2.272e-4$ | $0.142 \pm 1.186e-4$ | $0.267 \pm 1.201e-3$ | $0.16 \pm 1.024e-3$ |
| medgemma CoT patchtsmixer | $0.302 \pm 5.299e-3$ | $0.2 \pm 5.499e-3$ | $0.353 \pm 9.822e-3$ | $0.261 \pm 1.398e-2$ |
| medgemma CoT timemixer | $0.258 \pm 2.634e-3$ | $0.147 \pm 7.614e-4$ | $0.271 \pm 1.594e-3$ | $0.163 \pm 1.808e-3$ |
| medgemma ICD +TFM lstm | $0.242 \pm 3.343e-3$ | $0.127 \pm 5.492e-3$ | $0.268 \pm 2.749e-3$ | $0.156 \pm 3.411e-3$ |
| medgemma ICD +TFM mlp | $0.256 \pm 1.712e-3$ | $0.137 \pm 1.293e-3$ | $0.269 \pm 5.604e-4$ | $0.156 \pm 6.755e-4$ |
| medgemma ICD +TFM patchtsmixer | $0.309 \pm 4.25e-3$ | $0.204 \pm 1.313e-2$ | $0.358 \pm 1.76e-3$ | $0.259 \pm 3.414e-3$ |
| medgemma ICD +TFM timemixer | $0.264 \pm 8.753e-3$ | $0.143 \pm 3.938e-3$ | $0.265 \pm 1.113e-3$ | $0.149 \pm 3.272e-3$ |
| medgemma ICD lstm | $0.247 \pm 1.964e-3$ | $0.138 \pm 3.509e-3$ | $0.265 \pm 1.511e-3$ | $0.159 \pm 1.315e-3$ |
| medgemma ICD mlp | $0.249 \pm 6.003e-4$ | $0.141 \pm 6.605e-4$ | $0.267 \pm 1.313e-3$ | $0.161 \pm 1.376e-3$ |
| medgemma ICD patchtsmixer | $0.301 \pm 8.083e-3$ | $0.204 \pm 2.75e-2$ | $0.351 \pm 1.059e-2$ | $0.257 \pm 2.034e-2$ |
| medgemma ICD timemixer | $0.255 \pm 2.058e-3$ | $0.145 \pm 2.054e-3$ | $0.272 \pm 2.285e-3$ | $0.164 \pm 2.24e-3$ |
| medgemma Trend +TFM lstm | $0.245 \pm 4.502e-3$ | $0.129 \pm 5.574e-3$ | $0.268 \pm 8.775e-4$ | $0.157 \pm 3.522e-3$ |
| medgemma Trend +TFM mlp | $0.254 \pm 4.188e-4$ | $0.133 \pm 1.285e-3$ | $0.267 \pm 1.403e-3$ | $0.154 \pm 1.471e-3$ |
| medgemma Trend +TFM patchtsmixer | $0.312 \pm 3.208e-3$ | $0.221 \pm 1.173e-2$ | $0.352 \pm 1.223e-3$ | $0.253 \pm 1.052e-2$ |
| medgemma Trend +TFM timemixer | $0.265 \pm 5.529e-3$ | $0.139 \pm 1.168e-3$ | $0.271 \pm 1.898e-3$ | $0.154 \pm 4.041e-3$ |
| medgemma Trend lstm | $0.25 \pm 1.1e-3$ | $0.142 \pm 1.041e-3$ | $0.264 \pm 4.763e-4$ | $0.158 \pm 3.932e-4$ |
| medgemma Trend mlp | $0.251 \pm 7.036e-4$ | $0.143 \pm 5.268e-4$ | $0.267 \pm 8.651e-4$ | $0.161 \pm 6.607e-4$ |
| medgemma Trend patchtsmixer | $0.299 \pm 6.688e-4$ | $0.195 \pm 4.897e-3$ | $0.346 \pm 1.116e-2$ | $0.246 \pm 2.384e-2$ |
| medgemma Trend timemixer | $0.256 \pm 1.805e-3$ | $0.146 \pm 1.074e-3$ | $0.272 \pm 2.892e-3$ | $0.164 \pm 2.426e-3$ |
| medgemma zero_shot +TFM lstm | $0.25 \pm 2.708e-3$ | $0.139 \pm 4.651e-3$ | $0.264 \pm 9.359e-3$ | $0.151 \pm 9.994e-3$ |
| medgemma zero_shot +TFM mlp | $0.257 \pm 1.904e-3$ | $0.138 \pm 2.683e-3$ | $0.268 \pm 9.304e-4$ | $0.155 \pm 1.016e-3$ |
| medgemma zero_shot +TFM patchtsmixer | $0.309 \pm 8.308e-3$ | $0.207 \pm 1.436e-2$ | $0.356 \pm 2.022e-2$ | $0.26 \pm 2.808e-2$ |
| medgemma zero_shot +TFM timemixer | $0.259 \pm 4.9e-3$ | $0.139 \pm 7.448e-3$ | $0.271 \pm 3.455e-3$ | $0.153 \pm 4.72e-3$ |
| medgemma zero_shot lstm | $0.248 \pm 1.353e-3$ | $0.141 \pm 1.412e-3$ | $0.267 \pm 1.275e-3$ | $0.16 \pm 8.217e-4$ |
| medgemma zero_shot mlp | $0.249 \pm 7.299e-4$ | $0.141 \pm 5.935e-4$ | $0.267 \pm 1.14e-3$ | $0.16 \pm 1.277e-3$ |
| medgemma zero_shot patchtsmixer | $0.295 \pm 4.016e-3$ | $0.192 \pm 1.876e-2$ | $0.349 \pm 7.039e-3$ | $0.247 \pm 1.202e-2$ |
| medgemma zero_shot timemixer | $0.254 \pm 4.558e-3$ | $0.144 \pm 3.587e-3$ | $0.271 \pm 2.25e-3$ | $0.163 \pm 2.291e-3$ |
| No_sum_zero_shot zero_shot +TFM lstm | $0.245 \pm 4.863e-3$ | $0.131 \pm 6.715e-3$ | $0.264 \pm 2.851e-3$ | $0.15 \pm 4.289e-3$ |
| No_sum_zero_shot zero_shot +TFM mlp | $0.262 \pm 4.131e-3$ | $0.146 \pm 4.051e-3$ | $0.273 \pm 3.846e-3$ | $0.158 \pm 4.645e-3$ |
| No_sum_zero_shot zero_shot +TFM patchtsmixer | $0.304 \pm 2.445e-3$ | $0.194 \pm 9.183e-3$ | $0.356 \pm 2.551e-3$ | $0.265 \pm 1.22e-2$ |
| No_sum_zero_shot zero_shot +TFM timemixer | $0.26 \pm 9.804e-3$ | $0.134 \pm 3.106e-3$ | $0.273 \pm 3.046e-3$ | $0.156 \pm 4.517e-3$ |
| No_sum_zero_shot lstm | $0.246 \pm 1.023e-3$ | $0.139 \pm 4.635e-4$ | $0.27 \pm 1.73e-3$ | $0.162 \pm 2.37e-3$ |
| No_sum_zero_shot mlp | $0.251 \pm 2.359e-3$ | $0.143 \pm 1.861e-3$ | $0.282 \pm 2.943e-3$ | $0.173 \pm 2.678e-3$ |
| No_sum_zero_shot patchtsmixer | $0.296 \pm 1.026e-2$ | $0.195 \pm 2.108e-2$ | $0.351 \pm 3.333e-3$ | $0.259 \pm 2.272e-2$ |
| No_sum_zero_shot timemixer | $0.249 \pm 3.272e-3$ | $0.14 \pm 3.473e-3$ | $0.28 \pm 2.941e-3$ | $0.169 \pm 1.838e-3$ |
| right lstm | $0.32 \pm 9.703e-3$ | $0.212 \pm 5.96e-3$ | $0.379 \pm 3.51e-3$ | $0.27 \pm 2.655e-3$ |
| right mlp | $0.432 \pm 3.094e-2$ | $0.317 \pm 3.616e-2$ | $0.409 \pm 9.342e-3$ | $0.28 \pm 1.177e-2$ |
| right patchtsmixer | $0.303 \pm 1.23e-2$ | $0.417 \pm 4.725e-2$ | $0.547 \pm 3.435e-2$ | $6.344 \pm 1.676e0$ |
| right timemixer | $0.537 \pm 4.212e-2$ | $0.593 \pm 2.114e-1$ | $0.487 \pm 1.123e-2$ | $0.431 \pm 2.56e-2$ |

Table 37: Cross-site transfer results - Forecast. (part 2/2)

| Method | hirid → hirid | | mimic → mimic | | ppicu → ppicu | |
|---|---|---|---|---|---|---|
| | mae | mse | mae | mse | mae | mse |
| Llama-3.1 CoT +TFM lstm | $0.302 \pm 1.48e-2$ | $0.379 \pm 3.635e-2$ | $0.322 \pm 8.197e-3$ | $0.567 \pm 4.639e-3$ | $0.351 \pm 4.62e-3$ | $0.937 \pm 8.33e-3$ |
| Llama-3.1 CoT +TFM mlp | $0.304 \pm 9.901e-3$ | $0.372 \pm 1.076e-2$ | $0.304 \pm 3.403e-3$ | $0.505 \pm 1.562e-2$ | $0.343 \pm 9.457e-3$ | $0.922 \pm 1.08e-2$ |
| Llama-3.1 CoT +TFM patchtsmixer | $0.524 \pm 2.093e-2$ | $0.691 \pm 3.479e-2$ | $0.47 \pm 2.87e-2$ | $0.766 \pm 1.297e-1$ | $0.416 \pm 9.236e-3$ | $1.011 \pm 3.858e-3$ |
| Llama-3.1 CoT +TFM timemixer | $0.348 \pm 3.617e-3$ | $0.38 \pm 1.789e-3$ | $0.338 \pm 1.84e-2$ | $0.519 \pm 3.783e-2$ | $0.337 \pm 5.955e-3$ | $0.924 \pm 7.734e-3$ |
| Llama-3.1 CoT lstm | $0.378 \pm 1.686e-2$ | $0.527 \pm 1.747e-2$ | $0.396 \pm 1.467e-2$ | $0.692 \pm 9.725e-3$ | $0.358 \pm 6.408e-3$ | $0.958 \pm 7.056e-3$ |
| Llama-3.1 CoT mlp | $0.378 \pm 1.342e-2$ | $0.537 \pm 1.634e-2$ | $0.383 \pm 7.868e-3$ | $0.679 \pm 1.381e-2$ | $0.353 \pm 5.766e-3$ | $0.957 \pm 7.114e-3$ |
| Llama-3.1 CoT patchtsmixer | $0.45 \pm 9.321e-3$ | $0.603 \pm 1.17e-2$ | $0.436 \pm 1.907e-2$ | $0.726 \pm 1.54e-2$ | $0.371 \pm 5.441e-3$ | $0.948 \pm 4.508e-3$ |
| Llama-3.1 CoT timemixer | $0.428 \pm 9.948e-3$ | $0.608 \pm 1.768e-2$ | $0.4 \pm 1.577e-2$ | $0.745 \pm 4.069e-2$ | $0.348 \pm 1.209e-2$ | $0.959 \pm 3.893e-3$ |
| Llama-3.1 ICD +TFM lstm | $0.305 \pm 3.202e-3$ | $0.38 \pm 6.206e-3$ | $0.315 \pm 7.487e-3$ | $0.54 \pm 2.976e-2$ | $0.346 \pm 1.467e-3$ | $0.93 \pm 6.941e-4$ |
| Llama-3.1 ICD +TFM mlp | $0.291 \pm 1.751e-2$ | $0.338 \pm 4.194e-2$ | $0.307 \pm 1.124e-2$ | $0.512 \pm 2.612e-2$ | $0.338 \pm 5.783e-3$ | $0.907 \pm 2.877e-2$ |
| Llama-3.1 ICD +TFM patchtsmixer | $0.467 \pm 6.059e-2$ | $0.549 \pm 1.337e-1$ | $0.443 \pm 3.829e-2$ | $0.667 \pm 1.191e-1$ | $0.409 \pm 7.221e-3$ | $0.998 \pm 9.691e-3$ |
| Llama-3.1 ICD +TFM timemixer | $0.352 \pm 2.81e-3$ | $0.393 \pm 3.153e-2$ | $0.343 \pm 9.235e-3$ | $0.544 \pm 2.314e-2$ | $0.339 \pm 8.263e-3$ | $0.931 \pm 1.72e-3$ |
| Llama-3.1 ICD lstm | $0.373 \pm 1.35e-2$ | $0.507 \pm 1.465e-2$ | $0.396 \pm 4.652e-3$ | $0.671 \pm 1.096e-2$ | $0.343 \pm 1.374e-2$ | $0.943 \pm 1.831e-2$ |
| Llama-3.1 ICD mlp | $0.375 \pm 7.018e-3$ | $0.519 \pm 1.186e-2$ | $0.386 \pm 1.069e-2$ | $0.662 \pm 2.48e-2$ | $0.346 \pm 1.124e-2$ | $0.945 \pm 7.691e-3$ |
| Llama-3.1 ICD patchtsmixer | $0.44 \pm 1.366e-2$ | $0.567 \pm 2.372e-2$ | $0.414 \pm 3.289e-3$ | $0.694 \pm 1.099e-2$ | $0.363 \pm 8.136e-3$ | $0.951 \pm 7.857e-3$ |
| Llama-3.1 ICD timemixer | $0.415 \pm 7.205e-3$ | $0.598 \pm 1.578e-2$ | $0.394 \pm 7.977e-3$ | $0.735 \pm 1.667e-2$ | $0.346 \pm 8.294e-3$ | $0.952 \pm 5.613e-3$ |
| Llama-3.1 Trend +TFM lstm | $0.288 \pm 4.85e-3$ | $0.357 \pm 2.131e-2$ | $0.294 \pm 9.581e-3$ | $0.519 \pm 2.705e-2$ | $0.352 \pm 1.563e-3$ | $0.94 \pm 9.196e-4$ |
| Llama-3.1 Trend +TFM mlp | $0.291 \pm 1.039e-2$ | $0.342 \pm 2.252e-2$ | $0.304 \pm 8e-3$ | $0.539 \pm 1.464e-2$ | $0.354 \pm 6.17e-3$ | $0.919 \pm 2.627e-3$ |
| Llama-3.1 Trend +TFM patchtsmixer | $0.488 \pm 3.853e-2$ | $0.598 \pm 6.89e-2$ | $0.469 \pm 2.486e-2$ | $0.724 \pm 7.989e-2$ | $0.414 \pm 1.17e-2$ | $0.993 \pm 4.875e-3$ |
| Llama-3.1 Trend +TFM timemixer | $0.335 \pm 3.325e-3$ | $0.374 \pm 9.667e-3$ | $0.33 \pm 8.606e-3$ | $0.532 \pm 1.417e-2$ | $0.341 \pm 1.1e-2$ | $0.935 \pm 1.23e-2$ |
| Llama-3.1 Trend lstm | $0.317 \pm 7.011e-3$ | $0.417 \pm 1.042e-2$ | $0.32 \pm 7.124e-3$ | $0.587 \pm 1.364e-2$ | $0.349 \pm 4.507e-3$ | $0.944 \pm 1.046e-2$ |
| Llama-3.1 Trend mlp | $0.313 \pm 1.212e-2$ | $0.411 \pm 2.107e-2$ | $0.311 \pm 4.869e-3$ | $0.579 \pm 1.221e-2$ | $0.351 \pm 3.812e-3$ | $0.938 \pm 1.169e-2$ |
| Llama-3.1 Trend patchtsmixer | $0.383 \pm 9.965e-3$ | $0.473 \pm 1.785e-2$ | $0.369 \pm 7.19e-3$ | $0.616 \pm 1.752e-2$ | $0.374 \pm 9.925e-3$ | $0.943 \pm 1.132e-2$ |
| Llama-3.1 Trend timemixer | $0.359 \pm 7.483e-3$ | $0.479 \pm 1.367e-2$ | $0.338 \pm 6.444e-3$ | $0.625 \pm 7.808e-3$ | $0.353 \pm 6.539e-3$ | $0.952 \pm 9.052e-3$ |
| Llama-3.1 zero_shot +TFM lstm | $0.31 \pm 9.545e-3$ | $0.398 \pm 2.841e-2$ | $0.32 \pm 4.846e-3$ | $0.57 \pm 1.255e-2$ | $0.342 \pm 1.128e-3$ | $0.93 \pm 1.504e-4$ |
| Llama-3.1 zero_shot +TFM mlp | $0.307 \pm 5.449e-3$ | $0.383 \pm 2.041e-2$ | $0.315 \pm 7.42e-3$ | $0.545 \pm 1.806e-2$ | $0.348 \pm 8.502e-3$ | $0.917 \pm 1.885e-2$ |
| Llama-3.1 zero_shot +TFM patchtsmixer | $0.489 \pm 2.735e-2$ | $0.615 \pm 6.752e-2$ | $0.398 \pm 1.701e-2$ | $0.547 \pm 3.993e-2$ | $0.408 \pm 6.968e-3$ | $1.002 \pm 2.556e-3$ |
| Llama-3.1 zero_shot +TFM timemixer | $0.343 \pm 5.445e-3$ | $0.387 \pm 1.905e-2$ | $0.348 \pm 2.792e-3$ | $0.565 \pm 2.677e-2$ | $0.333 \pm 8.801e-3$ | $0.935 \pm 9.816e-3$ |
| Llama-3.1 zero_shot lstm | $0.353 \pm 7.614e-3$ | $0.493 \pm 1.717e-2$ | $0.391 \pm 1.006e-2$ | $0.701 \pm 2.508e-2$ | $0.345 \pm 6.4e-3$ | $0.944 \pm 4.13e-3$ |
| Llama-3.1 zero_shot mlp | $0.36 \pm 1.207e-2$ | $0.495 \pm 1.645e-2$ | $0.377 \pm 1.374e-2$ | $0.659 \pm 5.663e-3$ | $0.338 \pm 8.451e-3$ | $0.944 \pm 3.195e-3$ |
| Llama-3.1 zero_shot patchtsmixer | $0.421 \pm 4.832e-3$ | $0.549 \pm 1.19e-2$ | $0.411 \pm 1.348e-2$ | $0.709 \pm 1.73e-2$ | $0.36 \pm 6.219e-3$ | $0.947 \pm 2.512e-3$ |
| Llama-3.1 zero_shot timemixer | $0.397 \pm 1.102e-2$ | $0.568 \pm 2.333e-2$ | $0.392 \pm 1.076e-2$ | $0.707 \pm 6.456e-3$ | $0.333 \pm 8.444e-3$ | $0.95 \pm 9.248e-3$ |
| TFM lstm | $0.313 \pm 6.465e-3$ | $0.438 \pm 6.413e-3$ | $0.322 \pm 4.787e-3$ | $0.823 \pm 1.426e-3$ | $0.419 \pm 1.588e-2$ | $0.889 \pm 1.232e-2$ |
| TFM mlp | $0.319 \pm 9.517e-3$ | $0.442 \pm 5.719e-3$ | $0.336 \pm 7.05e-3$ | $0.837 \pm 3.961e-3$ | $0.416 \pm 7.481e-3$ | $0.9 \pm 9.927e-3$ |
| TFM patchtsmixer | $0.44 \pm 3.117e-2$ | $0.565 \pm 4.867e-2$ | $0.413 \pm 8.228e-3$ | $0.882 \pm 1.314e-2$ | $0.494 \pm 1.073e-2$ | $0.957 \pm 2.818e-2$ |
| TFM timemixer | $0.345 \pm 2.447e-2$ | $0.49 \pm 5.165e-2$ | $0.355 \pm 1.015e-2$ | $0.848 \pm 4.638e-3$ | $0.418 \pm 5.579e-3$ | $0.89 \pm 1.031e-2$ |
| TSDE lstm | $0.293 \pm 7.301e-3$ | $0.382 \pm 4.77e-3$ | $0.325 \pm 1.299e-2$ | $0.743 \pm 1.35e-2$ | $0.399 \pm 5.013e-3$ | $0.806 \pm 4.848e-3$ |
| TSDE mlp | $0.298 \pm 6.89e-3$ | $0.378 \pm 5.964e-3$ | $0.326 \pm 1.834e-2$ | $0.729 \pm 4.613e-3$ | $0.412 \pm 6.807e-3$ | $0.806 \pm 6.799e-3$ |
| TSDE patchtsmixer | $0.411 \pm 2.723e-2$ | $0.484 \pm 5.127e-2$ | $0.406 \pm 1.342e-2$ | $0.801 \pm 2.698e-2$ | $0.485 \pm 5.896e-3$ | $0.877 \pm 1.835e-2$ |
| TSDE timemixer | $0.331 \pm 1.209e-2$ | $0.403 \pm 1.16e-2$ | $0.343 \pm 1.109e-2$ | $0.742 \pm 8.187e-3$ | $0.4 \pm 9.397e-3$ | $0.821 \pm 2.762e-2$ |
| gemini-2.0-flash CoT +TFM lstm | $0.311 \pm 5.765e-3$ | $0.401 \pm 1.682e-2$ | $0.299 \pm 4.825e-3$ | $0.501 \pm 2.849e-3$ | $0.346 \pm 1.309e-3$ | $0.934 \pm 5.717e-4$ |
| gemini-2.0-flash CoT +TFM mlp | $0.293 \pm 1.353e-2$ | $0.341 \pm 2.886e-2$ | $0.311 \pm 1.27e-2$ | $0.519 \pm 3.552e-2$ | $0.353 \pm 6.034e-3$ | $0.916 \pm 1.346e-2$ |
| gemini-2.0-flash CoT +TFM patchtsmixer | $0.515 \pm 5.591e-2$ | $0.648 \pm 1.144e-1$ | $0.473 \pm 2.76e-2$ | $0.744 \pm 7.319e-2$ | $0.417 \pm 2.119e-3$ | $0.994 \pm 4.372e-3$ |
| gemini-2.0-flash CoT +TFM timemixer | $0.342 \pm 2.814e-3$ | $0.392 \pm 7.927e-3$ | $0.353 \pm 1.326e-2$ | $0.537 \pm 3.793e-2$ | $0.333 \pm 2.434e-3$ | $0.916 \pm 3.93e-3$ |
| gemini-2.0-flash CoT lstm | $0.364 \pm 9.354e-3$ | $0.503 \pm 1.154e-2$ | $0.364 \pm 5.208e-3$ | $0.641 \pm 6.085e-3$ | $0.348 \pm 2.818e-3$ | $0.944 \pm 5.052e-3$ |
| gemini-2.0-flash CoT mlp | $0.375 \pm 8.365e-3$ | $0.514 \pm 1.245e-2$ | $0.358 \pm 1.419e-2$ | $0.614 \pm 1.426e-2$ | $0.345 \pm 7.564e-3$ | $0.943 \pm 8.451e-3$ |
| gemini-2.0-flash CoT patchtsmixer | $0.438 \pm 9.928e-3$ | $0.563 \pm 5.535e-3$ | $0.396 \pm 2.093e-3$ | $0.646 \pm 4.094e-2$ | $0.363 \pm 6.436e-3$ | $0.935 \pm 1.274e-2$ |
| gemini-2.0-flash CoT timemixer | $0.398 \pm 3.311e-3$ | $0.555 \pm 9.959e-3$ | $0.366 \pm 1.037e-2$ | $0.651 \pm 3.209e-2$ | $0.341 \pm 7.12e-3$ | $0.943 \pm 5.792e-3$ |
| gemini-2.0-flash ICD +TFM lstm | $0.314 \pm 1.145e-2$ | $0.406 \pm 2.45e-2$ | $0.304 \pm 1.338e-2$ | $0.526 \pm 4.775e-2$ | $0.347 \pm 2.497e-3$ | $0.932 \pm 1.617e-3$ |
| gemini-2.0-flash ICD +TFM mlp | $0.295 \pm 1.323e-2$ | $0.341 \pm 3.493e-2$ | $0.309 \pm 7.168e-3$ | $0.511 \pm 3.592e-2$ | $0.353 \pm 5.594e-3$ | $0.916 \pm 1.94e-3$ |
| gemini-2.0-flash ICD +TFM patchtsmixer | $0.499 \pm 3.616e-2$ | $0.626 \pm 8.963e-2$ | $0.445 \pm 3.464e-2$ | $0.661 \pm 1.563e-1$ | $0.409 \pm 5.918e-3$ | $0.998 \pm 8.959e-3$ |
| gemini-2.0-flash ICD +TFM timemixer | $0.351 \pm 8.9e-3$ | $0.395 \pm 1.877e-2$ | $0.342 \pm 3.224e-2$ | $0.508 \pm 4.002e-2$ | $0.331 \pm 1.488e-2$ | $0.937 \pm 3.433e-3$ |
| gemini-2.0-flash ICD lstm | $0.377 \pm 7.773e-3$ | $0.51 \pm 1.163e-2$ | $0.379 \pm 9.69e-3$ | $0.655 \pm 2.066e-2$ | $0.345 \pm 4.67e-3$ | $0.943 \pm 6.364e-3$ |
| gemini-2.0-flash ICD mlp | $0.37 \pm 6.721e-3$ | $0.508 \pm 8.322e-3$ | $0.384 \pm 1.658e-2$ | $0.666 \pm 2.546e-2$ | $0.347 \pm 7.267e-3$ | $0.932 \pm 1.746e-2$ |
| gemini-2.0-flash ICD patchtsmixer | $0.425 \pm 1.65e-2$ | $0.554 \pm 2.298e-2$ | $0.436 \pm 5.942e-3$ | $0.72 \pm 1.483e-2$ | $0.368 \pm 1.732e-2$ | $0.943 \pm 1.599e-2$ |
| gemini-2.0-flash ICD timemixer | $0.412 \pm 4.38e-3$ | $0.578 \pm 1.692e-2$ | $0.395 \pm 1.056e-2$ | $0.709 \pm 1.757e-2$ | $0.337 \pm 1.046e-2$ | $0.943 \pm 4.178e-3$ |
| gemini-2.0-flash Trend +TFM lstm | $0.301 \pm 4.708e-3$ | $0.384 \pm 1.742e-2$ | $0.307 \pm 6.442e-3$ | $0.55 \pm 2.156e-2$ | $0.344 \pm 6.558e-3$ | $0.92 \pm 8.059e-3$ |
| gemini-2.0-flash Trend +TFM mlp | $0.305 \pm 7.051e-3$ | $0.385 \pm 3.715e-3$ | $0.307 \pm 1.223e-2$ | $0.523 \pm 2.544e-2$ | $0.347 \pm 6.139e-3$ | $0.903 \pm 1.67e-3$ |
| gemini-2.0-flash Trend +TFM patchtsmixer | $0.492 \pm 6.942e-2$ | $0.6 \pm 1.178e-1$ | $0.491 \pm 3.313e-2$ | $0.798 \pm 9.88e-2$ | $0.41 \pm 4.623e-3$ | $0.999 \pm 7.724e-3$ |
| gemini-2.0-flash Trend +TFM timemixer | $0.336 \pm 1.257e-2$ | $0.396 \pm 2.114e-2$ | $0.337 \pm 7.226e-3$ | $0.54 \pm 1.294e-2$ | $0.338 \pm 4.963e-3$ | $0.928 \pm 9.852e-4$ |
| gemini-2.0-flash Trend lstm | $0.321 \pm 4.403e-3$ | $0.436 \pm 4.094e-3$ | $0.324 \pm 8.987e-3$ | $0.603 \pm 6.258e-3$ | $0.343 \pm 5.967e-3$ | $0.937 \pm 7.98e-4$ |
| gemini-2.0-flash Trend mlp | $0.333 \pm 5.068e-3$ | $0.453 \pm 6.094e-3$ | $0.32 \pm 4.738e-3$ | $0.592 \pm 1.639e-2$ | $0.348 \pm 1.226e-2$ | $0.929 \pm 1.13e-2$ |
| gemini-2.0-flash Trend patchtsmixer | $0.379 \pm 3.72e-3$ | $0.479 \pm 3.638e-3$ | $0.37 \pm 5.582e-3$ | $0.615 \pm 9.403e-3$ | $0.366 \pm 8.984e-3$ | $0.934 \pm 9.468e-3$ |
| gemini-2.0-flash Trend timemixer | $0.35 \pm 8.609e-3$ | $0.471 \pm 1.277e-2$ | $0.349 \pm 1.801e-2$ | $0.615 \pm 1.997e-2$ | $0.326 \pm 5.127e-3$ | $0.921 \pm 6.568e-3$ |
| gemini-2.0-flash zero_shot +TFM lstm | $0.31 \pm 6.208e-3$ | $0.399 \pm 1.677e-2$ | $0.308 \pm 8.873e-3$ | $0.556 \pm 1.445e-2$ | $0.348 \pm 3.44e-3$ | $0.93 \pm 5.467e-4$ |
| gemini-2.0-flash zero_shot +TFM mlp | $0.299 \pm 4.766e-3$ | $0.351 \pm 4.897e-3$ | $0.302 \pm 7.655e-3$ | $0.52 \pm 3.221e-2$ | $0.344 \pm 1.348e-2$ | $0.891 \pm 3.461e-2$ |
| gemini-2.0-flash zero_shot +TFM patchtsmixer | $0.516 \pm 4.704e-2$ | $0.659 \pm 9.077e-2$ | $0.438 \pm 3.328e-2$ | $0.635 \pm 1.248e-1$ | $0.407 \pm 1.049e-2$ | $0.975 \pm 1.336e-2$ |
| gemini-2.0-flash zero_shot +TFM timemixer | $0.339 \pm 6.364e-3$ | $0.391 \pm 3.247e-2$ | $0.333 \pm 4.202e-3$ | $0.544 \pm 9.495e-3$ | $0.325 \pm 8.249e-3$ | $0.944 \pm 1.064e-2$ |
| gemini-2.0-flash zero_shot lstm | $0.343 \pm 6.846e-3$ | $0.464 \pm 5.354e-3$ | $0.338 \pm 2.042e-3$ | $0.597 \pm 6.937e-3$ | $0.352 \pm 1.166e-2$ | $0.936 \pm 9.862e-3$ |
| gemini-2.0-flash zero_shot mlp | $0.347 \pm 8.917e-3$ | $0.459 \pm 1.166e-2$ | $0.339 \pm 3.727e-3$ | $0.59 \pm 8.758e-3$ | $0.347 \pm 1.1e-2$ | $0.941 \pm 5.464e-3$ |
| gemini-2.0-flash zero_shot patchtsmixer | $0.414 \pm 1.191e-2$ | $0.509 \pm 2.031e-2$ | $0.378 \pm 1.284e-2$ | $0.631 \pm 2.204e-2$ | $0.364 \pm 6.278e-3$ | $0.922 \pm 8.645e-3$ |
| gemini-2.0-flash zero_shot timemixer | $0.378 \pm 1.103e-2$ | $0.51 \pm 2.382e-2$ | $0.35 \pm 1.269e-2$ | $0.624 \pm 1.051e-2$ | $0.347 \pm 8.378e-3$ | $0.939 \pm 8.521e-3$ |
| interp lstm | $0.29 \pm 1.648e-3$ | $0.38 \pm 6.123e-3$ | $0.309 \pm 1.547e-2$ | $0.756 \pm 4.973e-3$ | $0.412 \pm 5.721e-3$ | $0.861 \pm 3.341e-2$ |
| interp mlp | $0.301 \pm 7.762e-3$ | $0.366 \pm 5.698e-3$ | $0.332 \pm 7.038e-3$ | $0.775 \pm 3.284e-3$ | $0.425 \pm 1.774e-2$ | $0.79 \pm 1.397e-2$ |
| interp patchtsmixer | $0.37 \pm 1.346e-3$ | $0.46 \pm 1.034e-2$ | $0.43 \pm 1.757e-2$ | $0.914 \pm 1.501e-2$ | $0.493 \pm 9.374e-3$ | $0.891 \pm 2.526e-2$ |
| interp timemixer | $0.337 \pm 1.468e-2$ | $0.426 \pm 1.816e-3$ | $0.373 \pm 9.808e-3$ | $0.86 \pm 9.56e-3$ | $0.437 \pm 1.081e-2$ | $0.889 \pm 2.57e-2$ |
| mean lstm | $0.278 \pm 6.395e-3$ | $0.364 \pm 5.147e-3$ | $0.336 \pm 1.411e-2$ | $0.786 \pm 9.499e-3$ | $0.402 \pm 9.46e-3$ | $0.858 \pm 2.289e-2$ |
| mean mlp | $0.289 \pm 7.444e-3$ | $0.357 \pm 2.237e-3$ | $0.343 \pm 7.519e-3$ | $0.822 \pm 6.117e-3$ | $0.393 \pm 2.354e-3$ | $0.801 \pm 1.183e-2$ |
| mean patchtsmixer | $0.38 \pm 4.793e-3$ | $0.486 \pm 1.508e-2$ | $0.447 \pm 9.717e-3$ | $0.928 \pm 9.854e-3$ | $0.528 \pm 8.409e-3$ | $0.954 \pm 1.272e-2$ |
| mean timemixer | $0.328 \pm 8.1e-3$ | $0.457 \pm 3.781e-3$ | $0.357 \pm 8.922e-3$ | $0.82 \pm 1.152e-2$ | $0.439 \pm 1.902e-2$ | $0.848 \pm 1.813e-2$ |
| No_sum_CoT CoT +TFM lstm | $0.29 \pm 7.114e-3$ | $0.369 \pm 1.166e-2$ | $0.281 \pm 1.775e-3$ | $0.532 \pm 2.267e-3$ | $0.327 \pm 8.959e-4$ | $0.908 \pm 1.299e-3$ |
| No_sum_CoT CoT +TFM mlp | $0.283 \pm 9.061e-3$ | $0.337 \pm 1.212e-2$ | $0.265 \pm 2.853e-3$ | $0.45 \pm 1.279e-2$ | $0.332 \pm 5.21e-3$ | $0.849 \pm 2.953e-3$ |
| No_sum_CoT CoT +TFM patchtsmixer | $0.5 \pm 4.685e-2$ | $0.628 \pm 1.227e-1$ | $0.44 \pm 4.554e-2$ | $0.686 \pm 1.143e-1$ | $0.423 \pm 1.273e-2$ | $0.986 \pm 1.659e-2$ |
| No_sum_CoT CoT +TFM timemixer | $0.324 \pm 4.28e-3$ | $0.377 \pm 7.914e-3$ | $0.31 \pm 6.657e-3$ | $0.545 \pm 1.486e-2$ | $0.336 \pm 1.661e-2$ | $0.894 \pm 1.348e-2$ |
| No_sum_CoT lstm | $0.301 \pm 5.112e-3$ | $0.4 \pm 1.102e-2$ | $0.288 \pm 1.314e-2$ | $0.532 \pm 1.009e-2$ | $0.34 \pm 8.861e-3$ | $0.919 \pm 7.784e-3$ |
| No_sum_CoT mlp | $0.299 \pm 3.81e-3$ | $0.396 \pm 7.231e-3$ | $0.28 \pm 3.344e-3$ | $0.537 \pm 7.668e-3$ | $0.342 \pm 4.967e-3$ | $0.917 \pm 1.435e-2$ |
| No_sum_CoT patchtsmixer | $0.353 \pm 1.188e-2$ | $0.432 \pm 1.909e-2$ | $0.328 \pm 7.745e-3$ | $0.562 \pm 1.088e-2$ | $0.364 \pm 9.251e-3$ | $0.927 \pm 2.058e-2$ |
| No_sum_CoT timemixer | $0.338 \pm 1.152e-2$ | $0.436 \pm 7.67e-3$ | $0.308 \pm 1.879e-2$ | $0.56 \pm 2.424e-2$ | $0.321 \pm 6.086e-3$ | $0.92 \pm 7.261e-3$ |
| No_sum_ICD ICD +TFM lstm | $0.256 \pm 1.28e-2$ | $0.284 \pm 2.642e-2$ | $0.274 \pm 1.719e-2$ | $0.494 \pm 1.944e-2$ | $0.324 \pm 3.583e-3$ | $0.905 \pm 2.221e-3$ |
| No_sum_ICD ICD +TFM mlp | $0.269 \pm 9.898e-3$ | $0.309 \pm 1.739e-2$ | $0.249 \pm 1.39e-2$ | $0.39 \pm 8.124e-2$ | $0.331 \pm 1.195e-2$ | $0.846 \pm 2.891e-2$ |
| No_sum_ICD ICD +TFM patchtsmixer | $0.506 \pm 5.949e-2$ | $0.628 \pm 1.194e-1$ | $0.448 \pm 4.995e-2$ | $0.701 \pm 1.086e-1$ | $0.428 \pm 1.259e-2$ | $0.999 \pm 1.091e-2$ |
| No_sum_ICD ICD +TFM timemixer | $0.324 \pm 6.1e-4$ | $0.37 \pm 1.659e-2$ | $0.313 \pm 1.024e-2$ | $0.522 \pm 1.691e-2$ | $0.338 \pm 1.838e-2$ | $0.909 \pm 4.097e-3$ |
| No_sum_ICD lstm | $0.297 \pm 5.216e-3$ | $0.386 \pm 6.314e-3$ | $0.298 \pm 1.215e-2$ | $0.546 \pm 1.543e-2$ | $0.336 \pm 1.041e-2$ | $0.918 \pm 5.768e-3$ |
| No_sum_ICD mlp | $0.309 \pm 4.016e-3$ | $0.411 \pm 6.651e-3$ | $0.291 \pm 8.175e-3$ | $0.547 \pm 7.513e-3$ | $0.348 \pm 1.28e-2$ | $0.915 \pm 9.047e-3$ |
| No_sum_ICD patchtsmixer | $0.347 \pm 5.559e-3$ | $0.432 \pm 7.443e-3$ | $0.339 \pm 9.858e-3$ | $0.565 \pm 3.923e-3$ | $0.364 \pm 7.45e-3$ | $0.915 \pm 1.019e-2$ |
| No_sum_ICD timemixer | $0.362 \pm 5.646e-3$ | $0.459 \pm 1.286e-2$ | $0.32 \pm 6.855e-3$ | $0.574 \pm 8.25e-3$ | $0.321 \pm 1.326e-2$ | $0.931 \pm 5.679e-3$ |
| No_sum_Trend Trend +TFM lstm | $0.279 \pm 9.944e-3$ | $0.343 \pm 1.686e-2$ | $0.283 \pm 8.561e-3$ | $0.53 \pm 1.1e-2$ | $0.329 \pm 1.937e-3$ | $0.908 \pm 9.978e-4$ |
| No_sum_Trend Trend +TFM mlp | $0.271 \pm 1.802e-2$ | $0.307 \pm 4.645e-2$ | $0.267 \pm 5.543e-3$ | $0.469 \pm 2.384e-2$ | $0.33 \pm 7.231e-3$ | $0.819 \pm 5.575e-2$ |
| No_sum_Trend Trend +TFM patchtsmixer | $0.522 \pm 7.286e-2$ | $0.65 \pm 1.428e-1$ | $0.478 \pm 5.07e-2$ | $0.731 \pm 8.215e-2$ | $0.404 \pm 8.409e-3$ | $0.985 \pm 1.298e-2$ |
| No_sum_Trend Trend +TFM timemixer | $0.321 \pm 6.699e-3$ | $0.349 \pm 1.516e-2$ | $0.307 \pm 4.901e-3$ | $0.531 \pm 1.704e-2$ | $0.347 \pm 7.004e-3$ | $0.904 \pm 1.368e-3$ |

Table 38: In-distribution results - Los. (part 1/2)

| Method | hirid → hirid | | mimic → mimic | | ppicu → ppicu | |
|---|---|---|---|---|---|---|
| | mae | mse | mae | mse | mae | mse |
| No_sum_Trend lstm | $0.301 \pm 6.829e-3$ | $0.399 \pm 1.76e-2$ | $0.283 \pm 5.464e-3$ | $0.538 \pm 7.988e-3$ | $0.339 \pm 5.337e-3$ | $0.92 \pm 6.172e-3$ |
| No_sum_Trend mlp | $0.3 \pm 9.631e-3$ | $0.386 \pm 2.206e-2$ | $0.294 \pm 1.381e-2$ | $0.539 \pm 1.816e-2$ | $0.343 \pm 1.163e-2$ | $0.929 \pm 2.489e-3$ |
| No_sum_Trend patchtsmixer | $0.348 \pm 3.213e-3$ | $0.428 \pm 8.989e-3$ | $0.329 \pm 9.072e-3$ | $0.563 \pm 6.203e-3$ | $0.37 \pm 1.121e-2$ | $0.93 \pm 1.805e-2$ |
| No_sum_Trend timemixer | $0.363 \pm 1.044e-2$ | $0.457 \pm 1.057e-2$ | $0.326 \pm 1.872e-2$ | $0.575 \pm 8.601e-3$ | $0.325 \pm 9.516e-3$ | $0.916 \pm 6.562e-3$ |
| medgemma CoT +TFM lstm | $0.312 \pm 6.866e-3$ | $0.405 \pm 2.334e-2$ | $0.312 \pm 1.294e-2$ | $0.569 \pm 2.844e-2$ | $0.346 \pm 3.115e-3$ | $0.929 \pm 4.153e-4$ |
| medgemma CoT +TFM mlp | $0.299 \pm 1.399e-2$ | $0.355 \pm 3.591e-2$ | $0.288 \pm 9.295e-3$ | $0.486 \pm 3.863e-2$ | $0.339 \pm 2.546e-2$ | $0.912 \pm 1.611e-2$ |
| medgemma CoT +TFM patchtsmixer | $0.485 \pm 3.325e-2$ | $0.601 \pm 9.309e-2$ | $0.403 \pm 1.52e-2$ | $0.543 \pm 1.788e-2$ | $0.412 \pm 6.571e-3$ | $0.993 \pm 1.227e-2$ |
| medgemma CoT +TFM timemixer | $0.344 \pm 5.178e-3$ | $0.405 \pm 1.659e-2$ | $0.339 \pm 2.904e-3$ | $0.542 \pm 7.279e-3$ | $0.32 \pm 6.392e-3$ | $0.933 \pm 1.025e-2$ |
| medgemma CoT lstm | $0.341 \pm 4.005e-3$ | $0.463 \pm 7.062e-3$ | $0.332 \pm 7.19e-3$ | $0.603 \pm 8.312e-3$ | $0.347 \pm 5.449e-3$ | $0.939 \pm 6.47e-3$ |
| medgemma CoT mlp | $0.343 \pm 6.417e-3$ | $0.477 \pm 1.229e-2$ | $0.333 \pm 1.285e-2$ | $0.609 \pm 8.441e-3$ | $0.343 \pm 5.861e-3$ | $0.937 \pm 5.103e-3$ |
| medgemma CoT patchtsmixer | $0.408 \pm 8.074e-3$ | $0.518 \pm 2.414e-2$ | $0.386 \pm 3.992e-3$ | $0.647 \pm 9.512e-3$ | $0.359 \pm 3.952e-3$ | $0.929 \pm 8.375e-3$ |
| medgemma CoT timemixer | $0.387 \pm 1.108e-2$ | $0.529 \pm 2.851e-2$ | $0.347 \pm 7.617e-3$ | $0.615 \pm 1.594e-2$ | $0.336 \pm 7.898e-3$ | $0.946 \pm 2.82e-3$ |
| medgemma ICD +TFM lstm | $0.319 \pm 1.329e-2$ | $0.422 \pm 2.758e-2$ | $0.307 \pm 6.127e-3$ | $0.528 \pm 2.618e-2$ | $0.348 \pm 2.002e-3$ | $0.935 \pm 6.341e-4$ |
| medgemma ICD +TFM mlp | $0.322 \pm 3.012e-3$ | $0.421 \pm 1.392e-2$ | $0.3 \pm 1.038e-2$ | $0.498 \pm 3.683e-2$ | $0.349 \pm 6.55e-3$ | $0.916 \pm 1.648e-3$ |
| medgemma ICD +TFM patchtsmixer | $0.484 \pm 4.294e-2$ | $0.587 \pm 9.281e-2$ | $0.492 \pm 2.757e-2$ | $0.799 \pm 6.441e-2$ | $0.415 \pm 5.771e-3$ | $1.012 \pm 9.506e-3$ |
| medgemma ICD +TFM timemixer | $0.345 \pm 6.85e-3$ | $0.405 \pm 2.674e-2$ | $0.343 \pm 3.796e-3$ | $0.543 \pm 1.663e-2$ | $0.344 \pm 6.094e-3$ | $0.949 \pm 8.038e-3$ |
| medgemma ICD lstm | $0.355 \pm 2.561e-3$ | $0.494 \pm 1.149e-2$ | $0.356 \pm 6.237e-3$ | $0.609 \pm 1.937e-2$ | $0.35 \pm 9.343e-3$ | $0.938 \pm 6.572e-3$ |
| medgemma ICD mlp | $0.352 \pm 5.407e-3$ | $0.499 \pm 1.063e-2$ | $0.349 \pm 6.459e-3$ | $0.618 \pm 1.178e-2$ | $0.349 \pm 6.77e-3$ | $0.944 \pm 8.925e-3$ |
| medgemma ICD patchtsmixer | $0.424 \pm 1.153e-2$ | $0.549 \pm 1.159e-2$ | $0.399 \pm 2.139e-2$ | $0.653 \pm 3.121e-2$ | $0.368 \pm 1.218e-2$ | $0.942 \pm 1.61e-2$ |
| medgemma ICD timemixer | $0.396 \pm 4.963e-3$ | $0.549 \pm 8.559e-3$ | $0.381 \pm 6.995e-3$ | $0.679 \pm 1.414e-2$ | $0.334 \pm 9.54e-3$ | $0.934 \pm 2.482e-3$ |
| medgemma Trend +TFM lstm | $0.291 \pm 1.051e-2$ | $0.366 \pm 1.907e-2$ | $0.301 \pm 1.494e-2$ | $0.528 \pm 3.878e-2$ | $0.346 \pm 5.352e-4$ | $0.931 \pm 8.166e-4$ |
| medgemma Trend +TFM mlp | $0.287 \pm 1.007e-2$ | $0.346 \pm 2.704e-2$ | $0.281 \pm 6.273e-3$ | $0.474 \pm 1.746e-2$ | $0.344 \pm 1.023e-2$ | $0.902 \pm 1.618e-2$ |
| medgemma Trend +TFM patchtsmixer | $0.512 \pm 3.909e-2$ | $0.657 \pm 8.499e-2$ | $0.456 \pm 3.573e-2$ | $0.727 \pm 1.408e-1$ | $0.412 \pm 3.71e-3$ | $0.999 \pm 2.558e-3$ |
| medgemma Trend +TFM timemixer | $0.345 \pm 5.339e-3$ | $0.412 \pm 3.851e-3$ | $0.33 \pm 1.176e-2$ | $0.508 \pm 2.546e-2$ | $0.333 \pm 1.622e-2$ | $0.93 \pm 5.742e-3$ |
| medgemma Trend lstm | $0.316 \pm 2.651e-3$ | $0.437 \pm 3.236e-3$ | $0.316 \pm 7.792e-3$ | $0.601 \pm 1.346e-2$ | $0.35 \pm 8.86e-3$ | $0.94 \pm 6.694e-3$ |
| medgemma Trend mlp | $0.321 \pm 2.304e-3$ | $0.433 \pm 9.588e-3$ | $0.322 \pm 8.1e-3$ | $0.599 \pm 1.109e-2$ | $0.342 \pm 9.51e-3$ | $0.933 \pm 7.875e-3$ |
| medgemma Trend patchtsmixer | $0.383 \pm 9.782e-3$ | $0.478 \pm 8.87e-3$ | $0.37 \pm 1.515e-2$ | $0.637 \pm 2.943e-2$ | $0.36 \pm 9.033e-3$ | $0.935 \pm 1.177e-2$ |
| medgemma Trend timemixer | $0.357 \pm 6.686e-3$ | $0.481 \pm 8.074e-3$ | $0.349 \pm 2.561e-2$ | $0.636 \pm 2.673e-2$ | $0.33 \pm 6.434e-3$ | $0.935 \pm 9.01e-3$ |
| medgemma zero_shot +TFM lstm | $0.306 \pm 1.331e-2$ | $0.391 \pm 3.787e-2$ | $0.31 \pm 9.843e-3$ | $0.555 \pm 1.442e-2$ | $0.343 \pm 7.858e-4$ | $0.932 \pm 1.795e-3$ |
| medgemma zero_shot +TFM mlp | $0.291 \pm 6.487e-3$ | $0.333 \pm 1.188e-2$ | $0.304 \pm 1.345e-2$ | $0.53 \pm 3.875e-2$ | $0.343 \pm 1.195e-2$ | $0.902 \pm 2.005e-2$ |
| medgemma zero_shot +TFM patchtsmixer | $0.516 \pm 6.151e-2$ | $0.631 \pm 1.278e-1$ | $0.452 \pm 2.158e-2$ | $0.706 \pm 7.779e-2$ | $0.411 \pm 1.256e-2$ | $0.995 \pm 1.312e-2$ |
| medgemma zero_shot +TFM timemixer | $0.345 \pm 2.781e-2$ | $0.412 \pm 1.626e-2$ | $0.33 \pm 4.14e-3$ | $0.553 \pm 2.514e-2$ | $0.335 \pm 6.177e-3$ | $0.918 \pm 1.864e-2$ |
| medgemma zero_shot lstm | $0.345 \pm 8.007e-3$ | $0.469 \pm 5.448e-3$ | $0.348 \pm 4.531e-3$ | $0.608 \pm 1.119e-2$ | $0.343 \pm 4.79e-3$ | $0.933 \pm 3.1e-3$ |
| medgemma zero_shot mlp | $0.35 \pm 8.235e-3$ | $0.473 \pm 2.061e-2$ | $0.343 \pm 9.584e-3$ | $0.596 \pm 9.739e-3$ | $0.341 \pm 8.908e-3$ | $0.937 \pm 1.26e-2$ |
| medgemma zero_shot patchtsmixer | $0.422 \pm 8.425e-3$ | $0.53 \pm 3.072e-2$ | $0.384 \pm 6.472e-3$ | $0.633 \pm 1.235e-2$ | $0.364 \pm 4.605e-3$ | $0.927 \pm 8.179e-3$ |
| medgemma zero_shot timemixer | $0.399 \pm 6.156e-3$ | $0.557 \pm 1.604e-2$ | $0.359 \pm 1.415e-2$ | $0.622 \pm 2.514e-2$ | $0.333 \pm 9.835e-3$ | $0.935 \pm 4.769e-3$ |
| No_sum_zero_shot zero_shot +TFM lstm | $0.297 \pm 5.776e-3$ | $0.373 \pm 1.361e-2$ | $0.282 \pm 8.444e-3$ | $0.521 \pm 2.444e-2$ | $0.327 \pm 1.642e-3$ | $0.906 \pm 2.524e-3$ |
| No_sum_zero_shot zero_shot +TFM mlp | $0.267 \pm 1.307e-2$ | $0.308 \pm 3.37e-2$ | $0.272 \pm 1.423e-2$ | $0.478 \pm 2.572e-2$ | $0.333 \pm 4.563e-3$ | $0.847 \pm 4.354e-3$ |
| No_sum_zero_shot zero_shot +TFM patchtsmixer | $0.511 \pm 6.612e-2$ | $0.648 \pm 1.607e-1$ | $0.406 \pm 1.438e-2$ | $0.597 \pm 4.365e-2$ | $0.415 \pm 1.439e-2$ | $0.986 \pm 1.714e-2$ |
| No_sum_zero_shot zero_shot +TFM timemixer | $0.324 \pm 4.604e-3$ | $0.395 \pm 1.705e-2$ | $0.31 \pm 3.484e-3$ | $0.52 \pm 1.671e-2$ | $0.323 \pm 3.841e-3$ | $0.897 \pm 1.225e-2$ |
| No_sum_zero_shot lstm | $0.313 \pm 5.513e-3$ | $0.419 \pm 1.07e-2$ | $0.28 \pm 1.149e-2$ | $0.534 \pm 1.308e-2$ | $0.337 \pm 6.522e-3$ | $0.921 \pm 5.607e-3$ |
| No_sum_zero_shot mlp | $0.303 \pm 4.134e-3$ | $0.398 \pm 8.519e-3$ | $0.281 \pm 1.109e-2$ | $0.528 \pm 9.66e-3$ | $0.338 \pm 1.255e-2$ | $0.92 \pm 4.317e-3$ |
| No_sum_zero_shot patchtsmixer | $0.364 \pm 1.151e-2$ | $0.445 \pm 1.28e-2$ | $0.331 \pm 1.281e-2$ | $0.562 \pm 1.346e-2$ | $0.358 \pm 9.935e-3$ | $0.914 \pm 1.942e-2$ |
| No_sum_zero_shot timemixer | $0.348 \pm 1.641e-2$ | $0.451 \pm 1.311e-2$ | $0.313 \pm 1.207e-2$ | $0.566 \pm 9.761e-3$ | $0.324 \pm 6.311e-3$ | $0.917 \pm 4.715e-3$ |
| right lstm | $0.28 \pm 7.312e-3$ | $0.36 \pm 6.177e-3$ | $0.299 \pm 1.479e-2$ | $0.751 \pm 8.077e-3$ | $0.41 \pm 1.044e-2$ | $0.847 \pm 7.172e-3$ |
| right mlp | $0.276 \pm 4.021e-3$ | $0.341 \pm 1.953e-3$ | $0.295 \pm 1.189e-2$ | $0.752 \pm 4.509e-3$ | $0.408 \pm 1.236e-2$ | $0.786 \pm 1.062e-2$ |
| right patchtsmixer | $0.342 \pm 2.115e-3$ | $0.44 \pm 6.682e-3$ | $0.409 \pm 2.039e-2$ | $0.897 \pm 5.624e-3$ | $0.49 \pm 1.099e-2$ | $0.918 \pm 8.367e-3$ |
| right timemixer | $0.299 \pm 5.797e-3$ | $0.405 \pm 5.026e-3$ | $0.333 \pm 1.573e-2$ | $0.852 \pm 7.719e-3$ | $0.426 \pm 1.061e-2$ | $0.881 \pm 1.795e-2$ |

Table 39: In-distribution results - Los. (part 2/2)

| Method | hirid → ppicu | | mimic → ppicu | |
|---|---|---|---|---|
| | mae | mse | mae | mse |
| Llama-3.1 CoT +TFM lstm | $0.343 \pm 6.473e-3$ | $1.017 \pm 4.164e-3$ | $0.347 \pm 1.144e-3$ | $0.988 \pm 2.541e-3$ |
| Llama-3.1 CoT +TFM mlp | $0.373 \pm 3.751e-3$ | $1.047 \pm 9.349e-3$ | $0.351 \pm 2.08e-3$ | $1.014 \pm 1.484e-3$ |
| Llama-3.1 CoT +TFM patchtsmixer | $0.516 \pm 1.446e-2$ | $1.113 \pm 1.521e-2$ | $0.489 \pm 1.7e-2$ | $1.091 \pm 1.069e-2$ |
| Llama-3.1 CoT +TFM timemixer | $0.425 \pm 6.04e-3$ | $1.043 \pm 3.039e-3$ | $0.392 \pm 2.021e-2$ | $1.015 \pm 1.283e-2$ |
| Llama-3.1 CoT lstm | $0.795 \pm 3.91e-2$ | $1.544 \pm 5.866e-2$ | $0.558 \pm 2.002e-2$ | $1.267 \pm 5.593e-2$ |
| Llama-3.1 CoT mlp | $0.788 \pm 9.331e-3$ | $1.522 \pm 1.23e-2$ | $0.57 \pm 1.623e-2$ | $1.299 \pm 3.296e-2$ |
| Llama-3.1 CoT patchtsmixer | $0.736 \pm 2.019e-2$ | $1.365 \pm 3.218e-2$ | $0.598 \pm 4.047e-2$ | $1.255 \pm 5.565e-2$ |
| Llama-3.1 CoT timemixer | $0.736 \pm 2.372e-2$ | $1.402 \pm 3.504e-2$ | $0.541 \pm 1.175e-2$ | $1.21 \pm 1.633e-2$ |
| Llama-3.1 ICD +TFM lstm | $0.355 \pm 7.647e-3$ | $1 \pm 1.835e-3$ | $0.345 \pm 2.242e-3$ | $0.972 \pm 5.06e-3$ |
| Llama-3.1 ICD +TFM mlp | $0.386 \pm 2.413e-2$ | $1.042 \pm 1.282e-2$ | $0.347 \pm 5.083e-3$ | $0.993 \pm 4.004e-3$ |
| Llama-3.1 ICD +TFM patchtsmixer | $0.516 \pm 1.217e-2$ | $1.11 \pm 5.535e-3$ | $0.535 \pm 3.187e-3$ | $1.154 \pm 1.118e-2$ |
| Llama-3.1 ICD +TFM timemixer | $0.427 \pm 7.748e-3$ | $1.037 \pm 6.871e-3$ | $0.425 \pm 7.496e-3$ | $1.016 \pm 4.586e-3$ |
| Llama-3.1 ICD lstm | $0.824 \pm 2.439e-2$ | $1.715 \pm 4.504e-2$ | $0.683 \pm 2.758e-2$ | $1.474 \pm 7.487e-2$ |
| Llama-3.1 ICD mlp | $0.803 \pm 2.404e-2$ | $1.632 \pm 5.696e-2$ | $0.677 \pm 5.667e-2$ | $1.452 \pm 1.538e-1$ |
| Llama-3.1 ICD patchtsmixer | $0.807 \pm 2.36e-2$ | $1.583 \pm 4.903e-2$ | $0.645 \pm 2.377e-2$ | $1.29 \pm 4.463e-2$ |
| Llama-3.1 ICD timemixer | $0.756 \pm 1.606e-2$ | $1.515 \pm 2.635e-2$ | $0.577 \pm 7.892e-3$ | $1.208 \pm 1.812e-2$ |
| Llama-3.1 Trend +TFM lstm | $0.366 \pm 1.907e-3$ | $1.004 \pm 5.751e-3$ | $0.363 \pm 1.113e-2$ | $0.976 \pm 4.142e-3$ |
| Llama-3.1 Trend +TFM mlp | $0.393 \pm 3.876e-3$ | $1.037 \pm 1.453e-2$ | $0.378 \pm 4.262e-3$ | $0.995 \pm 1.128e-3$ |
| Llama-3.1 Trend +TFM patchtsmixer | $0.563 \pm 1.006e-2$ | $1.152 \pm 1.473e-2$ | $0.543 \pm 3.772e-2$ | $1.125 \pm 3.512e-2$ |
| Llama-3.1 Trend +TFM timemixer | $0.463 \pm 1.657e-2$ | $1.049 \pm 1.219e-2$ | $0.441 \pm 6.405e-3$ | $1.023 \pm 6.917e-3$ |
| Llama-3.1 Trend lstm | $1.075 \pm 1.984e-2$ | $2.233 \pm 5.925e-2$ | $0.677 \pm 7.822e-3$ | $1.389 \pm 1.461e-2$ |
| Llama-3.1 Trend mlp | $1.041 \pm 2.621e-2$ | $2.128 \pm 5.849e-2$ | $0.621 \pm 1.222e-2$ | $1.278 \pm 3.591e-2$ |
| Llama-3.1 Trend patchtsmixer | $0.985 \pm 2.71e-2$ | $1.874 \pm 6.315e-2$ | $0.663 \pm 2.518e-2$ | $1.285 \pm 4.798e-2$ |
| Llama-3.1 Trend timemixer | $0.971 \pm 4.939e-2$ | $1.889 \pm 1.276e-1$ | $0.62 \pm 1.656e-2$ | $1.228 \pm 2.602e-2$ |
| Llama-3.1 zero_shot +TFM lstm | $0.347 \pm 3.392e-3$ | $0.991 \pm 1.087e-2$ | $0.355 \pm 3.006e-3$ | $0.976 \pm 4.357e-3$ |
| Llama-3.1 zero_shot +TFM mlp | $0.373 \pm 5.138e-3$ | $1.018 \pm 7.934e-3$ | $0.349 \pm 6.696e-3$ | $0.992 \pm 3.06e-3$ |
| Llama-3.1 zero_shot +TFM patchtsmixer | $0.517 \pm 1.5e-2$ | $1.107 \pm 8.694e-3$ | $0.482 \pm 1.316e-2$ | $1.071 \pm 1.003e-2$ |
| Llama-3.1 zero_shot +TFM timemixer | $0.418 \pm 9.84e-3$ | $1.025 \pm 8.069e-3$ | $0.429 \pm 6.42e-3$ | $1.013 \pm 7.363e-3$ |
| Llama-3.1 zero_shot lstm | $0.834 \pm 1.674e-2$ | $1.618 \pm 4.316e-2$ | $0.722 \pm 4.193e-2$ | $1.446 \pm 6.508e-2$ |
| Llama-3.1 zero_shot mlp | $0.872 \pm 2.293e-2$ | $1.673 \pm 3.933e-2$ | $0.721 \pm 5.726e-2$ | $1.477 \pm 1.215e-1$ |
| Llama-3.1 zero_shot patchtsmixer | $0.824 \pm 9.857e-3$ | $1.509 \pm 2.648e-2$ | $0.693 \pm 3.927e-2$ | $1.314 \pm 6.808e-2$ |
| Llama-3.1 zero_shot timemixer | $0.776 \pm 2.656e-2$ | $1.448 \pm 5.02e-2$ | $0.703 \pm 4.272e-2$ | $1.381 \pm 8.666e-2$ |
| TFM lstm | $0.908 \pm 4.184e-2$ | $1.523 \pm 9.286e-2$ | $1.082 \pm 3.76e-2$ | $2.178 \pm 1.497e-1$ |
| TFM mlp | $0.895 \pm 1.446e-2$ | $1.427 \pm 3.894e-2$ | $1.082 \pm 4.853e-2$ | $2.196 \pm 2.061e-1$ |
| TFM patchtsmixer | $0.797 \pm 6.325e-2$ | $1.206 \pm 1.132e-1$ | $0.929 \pm 8.13e-2$ | $1.585 \pm 2.215e-1$ |
| TFM timemixer | $0.814 \pm 1.252e-1$ | $1.28 \pm 2.451e-1$ | $0.99 \pm 7.457e-2$ | $1.944 \pm 2.091e-1$ |
| TSDE lstm | $0.551 \pm 2.465e-2$ | $1.205 \pm 1.036e-1$ | $0.766 \pm 1.867e-1$ | $1.144 \pm 2.156e-1$ |
| TSDE mlp | $1.454 \pm 4.611e-1$ | $3.135 \pm 1.268e0$ | $0.85 \pm 2.359e-1$ | $1.323 \pm 3.066e-1$ |
| TSDE patchtsmixer | $0.729 \pm 7.139e-2$ | $1.147 \pm 5.025e-2$ | $0.758 \pm 3.708e-2$ | $1.163 \pm 3.816e-2$ |
| TSDE timemixer | $0.888 \pm 2.711e-1$ | $1.446 \pm 3.929e-1$ | $1.42 \pm 6.94e-2$ | $2.369 \pm 1.759e-1$ |
| gemini-2.0-flash CoT +TFM lstm | $0.357 \pm 3.759e-3$ | $1.016 \pm 4.572e-3$ | $0.333 \pm 4.676e-3$ | $0.986 \pm 4.933e-3$ |
| gemini-2.0-flash CoT +TFM mlp | $0.383 \pm 5.032e-4$ | $1.07 \pm 1.139e-2$ | $0.348 \pm 4.266e-3$ | $0.998 \pm 4.574e-3$ |
| gemini-2.0-flash CoT +TFM patchtsmixer | $0.524 \pm 7.045e-3$ | $1.119 \pm 4.735e-3$ | $0.506 \pm 1.762e-2$ | $1.093 \pm 1.268e-2$ |
| gemini-2.0-flash CoT +TFM timemixer | $0.413 \pm 6.516e-3$ | $1.034 \pm 8.604e-3$ | $0.418 \pm 3.945e-3$ | $1.023 \pm 2.46e-3$ |
| gemini-2.0-flash CoT lstm | $1.031 \pm 1.665e-2$ | $2.036 \pm 3.139e-2$ | $0.51 \pm 9.334e-3$ | $1.167 \pm 1.83e-2$ |
| gemini-2.0-flash CoT mlp | $1.053 \pm 3.363e-2$ | $2.053 \pm 7.408e-2$ | $0.525 \pm 2.298e-2$ | $1.178 \pm 2.304e-2$ |
| gemini-2.0-flash CoT patchtsmixer | $0.981 \pm 4.818e-2$ | $1.82 \pm 1.047e-1$ | $0.537 \pm 2.603e-2$ | $1.138 \pm 4.755e-2$ |
| gemini-2.0-flash CoT timemixer | $1.043 \pm 3.713e-2$ | $2.007 \pm 7.049e-2$ | $0.52 \pm 2.384e-2$ | $1.15 \pm 3.186e-2$ |
| gemini-2.0-flash ICD +TFM lstm | $0.359 \pm 2.204e-3$ | $1.016 \pm 1.085e-2$ | $0.331 \pm 2.327e-3$ | $0.979 \pm 4.021e-3$ |
| gemini-2.0-flash ICD +TFM mlp | $0.385 \pm 5.553e-3$ | $1.062 \pm 2.03e-2$ | $0.343 \pm 4.33e-3$ | $0.986 \pm 4.732e-3$ |
| gemini-2.0-flash ICD +TFM patchtsmixer | $0.521 \pm 1.868e-2$ | $1.132 \pm 2.618e-2$ | $0.491 \pm 7.453e-3$ | $1.071 \pm 1.968e-2$ |
| gemini-2.0-flash ICD +TFM timemixer | $0.464 \pm 1.75e-2$ | $1.072 \pm 1.602e-2$ | $0.397 \pm 2.892e-2$ | $0.997 \pm 6.933e-3$ |
| gemini-2.0-flash ICD lstm | $0.67 \pm 7.942e-2$ | $1.369 \pm 1.18e-1$ | $0.507 \pm 8.186e-3$ | $1.15 \pm 1.919e-2$ |
| gemini-2.0-flash ICD mlp | $0.657 \pm 1.122e-1$ | $1.317 \pm 1.588e-1$ | $0.502 \pm 1.481e-2$ | $1.123 \pm 1.053e-2$ |
| gemini-2.0-flash ICD patchtsmixer | $0.688 \pm 4.246e-2$ | $1.277 \pm 4.596e-2$ | $0.545 \pm 1.764e-2$ | $1.125 \pm 2.547e-2$ |
| gemini-2.0-flash ICD timemixer | $0.706 \pm 3.749e-2$ | $1.324 \pm 6.35e-2$ | $0.506 \pm 2.247e-2$ | $1.113 \pm 3.125e-2$ |
| gemini-2.0-flash Trend +TFM lstm | $0.366 \pm 3.201e-3$ | $1.004 \pm 4.439e-3$ | $0.356 \pm 3.593e-3$ | $0.972 \pm 6.498e-3$ |
| gemini-2.0-flash Trend +TFM mlp | $0.384 \pm 1.284e-2$ | $1.035 \pm 5.205e-3$ | $0.361 \pm 1.506e-2$ | $0.987 \pm 4.302e-3$ |
| gemini-2.0-flash Trend +TFM patchtsmixer | $0.527 \pm 2.383e-2$ | $1.12 \pm 9.917e-3$ | $0.496 \pm 1.71e-2$ | $1.076 \pm 1.597e-2$ |
| gemini-2.0-flash Trend +TFM timemixer | $0.448 \pm 1.2e-2$ | $1.042 \pm 8.75e-3$ | $0.433 \pm 1.054e-2$ | $1.016 \pm 7.269e-3$ |
| gemini-2.0-flash Trend lstm | $1.078 \pm 2.93e-2$ | $2.209 \pm 7.425e-2$ | $0.52 \pm 3.029e-2$ | $1.157 \pm 4.75e-2$ |
| gemini-2.0-flash Trend mlp | $1.115 \pm 3.582e-2$ | $2.265 \pm 9.766e-2$ | $0.545 \pm 2.363e-2$ | $1.187 \pm 3.69e-2$ |
| gemini-2.0-flash Trend patchtsmixer | $1.059 \pm 4.256e-2$ | $2.041 \pm 1.149e-1$ | $0.576 \pm 2.366e-2$ | $1.165 \pm 3.035e-2$ |
| gemini-2.0-flash Trend timemixer | $1.083 \pm 5.359e-2$ | $2.152 \pm 1.607e-1$ | $0.539 \pm 1.149e-2$ | $1.141 \pm 2.25e-2$ |
| gemini-2.0-flash zero_shot +TFM lstm | $0.361 \pm 5.824e-3$ | $1.017 \pm 1.433e-3$ | $0.353 \pm 4.76e-3$ | $0.971 \pm 3.582e-3$ |
| gemini-2.0-flash zero_shot +TFM mlp | $0.375 \pm 2.239e-3$ | $1.054 \pm 5.46e-3$ | $0.348 \pm 4.035e-3$ | $0.982 \pm 1.185e-3$ |
| gemini-2.0-flash zero_shot +TFM patchtsmixer | $0.544 \pm 1.187e-2$ | $1.144 \pm 2.683e-2$ | $0.511 \pm 2.272e-2$ | $1.092 \pm 2.521e-2$ |
| gemini-2.0-flash zero_shot +TFM timemixer | $0.429 \pm 5.633e-3$ | $1.045 \pm 5.555e-3$ | $0.405 \pm 9.436e-4$ | $0.995 \pm 3.062e-3$ |
| gemini-2.0-flash zero_shot lstm | $0.927 \pm 3.896e-2$ | $1.869 \pm 8.006e-2$ | $0.476 \pm 1.601e-2$ | $1.084 \pm 2.417e-2$ |
| gemini-2.0-flash zero_shot mlp | $0.973 \pm 1.181e-2$ | $1.93 \pm 3.46e-2$ | $0.481 \pm 4.471e-3$ | $1.067 \pm 6.673e-3$ |
| gemini-2.0-flash zero_shot patchtsmixer | $0.912 \pm 1.512e-2$ | $1.692 \pm 3.892e-2$ | $0.491 \pm 1.926e-2$ | $1.046 \pm 1.578e-2$ |
| gemini-2.0-flash zero_shot timemixer | $0.888 \pm 3.834e-2$ | $1.699 \pm 9.643e-2$ | $0.484 \pm 2.025e-2$ | $1.063 \pm 2.581e-2$ |
| interp lstm | $2.313 \pm 1.543e-1$ | $6.444 \pm 9.518e-1$ | $1.01 \pm 1.029e-1$ | $2.33 \pm 3.167e-1$ |
| interp mlp | $2.398 \pm 6.509e-2$ | $7.078 \pm 3.312e-1$ | $1.465 \pm 1.149e-1$ | $3.839 \pm 6.104e-1$ |
| interp patchtsmixer | $0.908 \pm 8.378e-2$ | $1.395 \pm 1.569e-1$ | $0.71 \pm 5.453e-2$ | $1.219 \pm 5.595e-2$ |
| interp timemixer | $0.845 \pm 3.561e-2$ | $1.263 \pm 4.676e-2$ | $0.556 \pm 5.435e-3$ | $1.122 \pm 4.275e-2$ |
| mean lstm | $3.13 \pm 2.202e-1$ | $10.631 \pm 1.473e0$ | $1.293 \pm 3.09e-1$ | $3.012 \pm 1.482e0$ |
| mean mlp | $2.745 \pm 6.051e-2$ | $8.177 \pm 3.628e-1$ | $2.204 \pm 1.503e-1$ | $6.225 \pm 6.756e-1$ |
| mean patchtsmixer | $1.092 \pm 6.323e-2$ | $1.822 \pm 1.841e-1$ | $0.849 \pm 5.7e-2$ | $1.393 \pm 9.83e-2$ |
| mean timemixer | $0.982 \pm 4.65e-2$ | $1.503 \pm 1.194e-1$ | $0.61 \pm 3.178e-2$ | $0.978 \pm 3.169e-2$ |
| No_sum_CoT CoT +TFM lstm | $0.404 \pm 9.289e-3$ | $0.994 \pm 3.007e-3$ | $0.467 \pm 1.512e-2$ | $1.021 \pm 1.776e-2$ |
| No_sum_CoT CoT +TFM mlp | $0.411 \pm 1.161e-2$ | $1.019 \pm 6.541e-3$ | $0.435 \pm 2.834e-2$ | $1.032 \pm 3.57e-2$ |
| No_sum_CoT CoT +TFM patchtsmixer | $0.585 \pm 2.176e-2$ | $1.181 \pm 4.043e-2$ | $0.594 \pm 6.436e-2$ | $1.176 \pm 1.01e-1$ |
| No_sum_CoT CoT +TFM timemixer | $0.527 \pm 2.021e-2$ | $1.099 \pm 2.189e-2$ | $0.581 \pm 4.402e-2$ | $1.194 \pm 7.407e-2$ |
| No_sum_CoT lstm | $0.844 \pm 5.067e-2$ | $1.724 \pm 1.236e-1$ | $0.967 \pm 3.653e-2$ | $2.018 \pm 1.216e-1$ |
| No_sum_CoT mlp | $0.931 \pm 4.334e-2$ | $1.89 \pm 1.096e-1$ | $0.915 \pm 2.2e-2$ | $1.851 \pm 5.932e-2$ |
| No_sum_CoT patchtsmixer | $0.959 \pm 4.675e-2$ | $1.843 \pm 1.004e-1$ | $0.84 \pm 4.783e-2$ | $1.58 \pm 1.046e-1$ |
| No_sum_CoT timemixer | $0.917 \pm 2.341e-2$ | $1.848 \pm 5.82e-2$ | $0.81 \pm 8.9e-2$ | $1.578 \pm 2.312e-1$ |
| No_sum_ICD ICD +TFM lstm | $0.367 \pm 1.105e-2$ | $0.989 \pm 7.262e-4$ | $0.387 \pm 1.494e-2$ | $0.962 \pm 9.855e-4$ |
| No_sum_ICD ICD +TFM mlp | $0.414 \pm 2.747e-2$ | $1.024 \pm 2.568e-2$ | $0.374 \pm 6.777e-3$ | $1.005 \pm 2.132e-2$ |
| No_sum_ICD ICD +TFM patchtsmixer | $0.605 \pm 1.834e-2$ | $1.208 \pm 5.107e-2$ | $0.565 \pm 2.557e-2$ | $1.13 \pm 3.813e-2$ |
| No_sum_ICD ICD +TFM timemixer | $0.533 \pm 1.226e-2$ | $1.112 \pm 1.019e-2$ | $0.508 \pm 2.526e-2$ | $1.065 \pm 2.561e-2$ |
| No_sum_ICD lstm | $0.886 \pm 1.93e-2$ | $1.811 \pm 4.878e-2$ | $0.897 \pm 5.587e-2$ | $1.733 \pm 1.361e-1$ |
| No_sum_ICD mlp | $0.989 \pm 2.553e-2$ | $2.024 \pm 7.196e-2$ | $0.873 \pm 3.112e-2$ | $1.675 \pm 6.143e-2$ |
| No_sum_ICD patchtsmixer | $0.926 \pm 3.175e-2$ | $1.804 \pm 8.536e-2$ | $0.887 \pm 3.278e-2$ | $1.639 \pm 6.911e-2$ |
| No_sum_ICD timemixer | $0.796 \pm 8.071e-2$ | $1.562 \pm 1.722e-1$ | $0.766 \pm 5.493e-2$ | $1.435 \pm 1.114e-1$ |
| No_sum_Trend Trend +TFM lstm | $0.379 \pm 1.126e-2$ | $0.978 \pm 2.533e-3$ | $0.452 \pm 2.965e-2$ | $1.01 \pm 3.527e-2$ |
| No_sum_Trend Trend +TFM mlp | $0.403 \pm 2.194e-2$ | $1.012 \pm 2.18e-3$ | $0.423 \pm 1.988e-2$ | $1.013 \pm 4.05e-2$ |
| No_sum_Trend Trend +TFM patchtsmixer | $0.594 \pm 1.385e-2$ | $1.177 \pm 1.88e-3$ | $0.605 \pm 1.428e-2$ | $1.179 \pm 2.521e-2$ |
| No_sum_Trend Trend +TFM timemixer | $0.519 \pm 4.6e-3$ | $1.089 \pm 1.016e-2$ | $0.522 \pm 2.55e-2$ | $1.086 \pm 3.986e-2$ |

Table 40: Cross-site transfer results - Los. (part 1/2)

| Method | hirid → ppicu | | mimic → ppicu | |
|---|---|---|---|---|
| | mae | mse | mae | mse |
| No_sum_Trend lstm | $0.704 \pm 8.339e-2$ | $1.389 \pm 1.546e-1$ | $0.742 \pm 4.071e-2$ | $1.461 \pm 9.049e-2$ |
| No_sum_Trend mlp | $0.754 \pm 3.342e-2$ | $1.496 \pm 4.605e-2$ | $0.794 \pm 3.188e-2$ | $1.568 \pm 6.687e-2$ |
| No_sum_Trend patchtsmixer | $0.853 \pm 3.164e-2$ | $1.606 \pm 6.786e-2$ | $0.776 \pm 6.463e-2$ | $1.468 \pm 1.277e-1$ |
| No_sum_Trend timemixer | $0.802 \pm 5.126e-2$ | $1.562 \pm 1.056e-1$ | $0.753 \pm 3.791e-2$ | $1.451 \pm 8.137e-2$ |
| medgemma CoT +TFM lstm | $0.369 \pm 5.866e-3$ | $1.007 \pm 5.832e-3$ | $0.352 \pm 6.795e-3$ | $0.963 \pm 3.323e-3$ |
| medgemma CoT +TFM mlp | $0.382 \pm 3.831e-3$ | $1.042 \pm 1.398e-2$ | $0.343 \pm 6.876e-3$ | $0.988 \pm 1.181e-2$ |
| medgemma CoT +TFM patchtsmixer | $0.528 \pm 1.409e-3$ | $1.126 \pm 3.894e-3$ | $0.48 \pm 6.525e-3$ | $1.07 \pm 2.149e-2$ |
| medgemma CoT +TFM timemixer | $0.428 \pm 5.554e-3$ | $1.025 \pm 9.87e-3$ | $0.416 \pm 1.092e-2$ | $1 \pm 6.901e-3$ |
| medgemma CoT lstm | $1.116 \pm 2.962e-2$ | $2.313 \pm 8.62e-2$ | $0.535 \pm 9.297e-3$ | $1.181 \pm 3.171e-2$ |
| medgemma CoT mlp | $1.063 \pm 2.839e-2$ | $2.121 \pm 8.412e-2$ | $0.539 \pm 2.816e-2$ | $1.153 \pm 3.959e-2$ |
| medgemma CoT patchtsmixer | $1 \pm 5.994e-2$ | $1.878 \pm 1.566e-1$ | $0.554 \pm 2.224e-2$ | $1.118 \pm 2.79e-2$ |
| medgemma CoT timemixer | $1.018 \pm 8.012e-2$ | $1.97 \pm 2.173e-1$ | $0.556 \pm 1.472e-2$ | $1.165 \pm 1.833e-2$ |
| medgemma ICD +TFM lstm | $0.367 \pm 2.197e-3$ | $1.021 \pm 1.355e-2$ | $0.334 \pm 8.282e-3$ | $0.972 \pm 2.582e-3$ |
| medgemma ICD +TFM mlp | $0.375 \pm 5.707e-3$ | $1.042 \pm 6.535e-3$ | $0.345 \pm 7.011e-3$ | $0.987 \pm 3.264e-3$ |
| medgemma ICD +TFM patchtsmixer | $0.547 \pm 1.076e-2$ | $1.16 \pm 2.469e-2$ | $0.507 \pm 6.239e-3$ | $1.093 \pm 2.527e-3$ |
| medgemma ICD +TFM timemixer | $0.424 \pm 4.215e-3$ | $1.065 \pm 7.981e-3$ | $0.4 \pm 1.227e-3$ | $1 \pm 6.85e-3$ |
| medgemma ICD lstm | $0.969 \pm 3.308e-2$ | $1.936 \pm 8.443e-2$ | $0.607 \pm 2.19e-2$ | $1.315 \pm 3.69e-2$ |
| medgemma ICD mlp | $0.9 \pm 2.985e-2$ | $1.773 \pm 6.322e-2$ | $0.59 \pm 1.878e-2$ | $1.247 \pm 3.427e-2$ |
| medgemma ICD patchtsmixer | $0.904 \pm 3.217e-2$ | $1.681 \pm 5.134e-2$ | $0.608 \pm 3.347e-2$ | $1.213 \pm 3.086e-2$ |
| medgemma ICD timemixer | $0.892 \pm 2.006e-2$ | $1.71 \pm 4.965e-2$ | $0.585 \pm 3.202e-2$ | $1.21 \pm 4.16e-2$ |
| medgemma Trend +TFM lstm | $0.364 \pm 5.091e-2$ | $0.993 \pm 8.769e-3$ | $0.37 \pm 2.093e-2$ | $0.98 \pm 1.068e-2$ |
| medgemma Trend +TFM mlp | $0.385 \pm 1.412e-3$ | $1.03 \pm 8.041e-3$ | $0.349 \pm 2.607e-3$ | $0.99 \pm 7.04e-3$ |
| medgemma Trend +TFM patchtsmixer | $0.565 \pm 2.129e-2$ | $1.149 \pm 1.421e-2$ | $0.502 \pm 1.755e-3$ | $1.082 \pm 3.905e-3$ |
| medgemma Trend +TFM timemixer | $0.453 \pm 6.806e-3$ | $1.033 \pm 6.152e-3$ | $0.424 \pm 1.422e-2$ | $1.014 \pm 9.086e-3$ |
| medgemma Trend lstm | $0.979 \pm 3.445e-2$ | $1.975 \pm 1.007e-1$ | $0.628 \pm 1.822e-2$ | $1.334 \pm 2.749e-2$ |
| medgemma Trend mlp | $1.054 \pm 2.43e-2$ | $2.158 \pm 8.046e-2$ | $0.646 \pm 2.782e-2$ | $1.358 \pm 6.475e-2$ |
| medgemma Trend patchtsmixer | $0.998 \pm 3.545e-2$ | $1.899 \pm 7.864e-2$ | $0.641 \pm 1.909e-2$ | $1.267 \pm 4.206e-2$ |
| medgemma Trend timemixer | $0.974 \pm 2.663e-2$ | $1.885 \pm 5.929e-2$ | $0.627 \pm 2.08e-2$ | $1.271 \pm 1.604e-2$ |
| medgemma zero_shot +TFM lstm | $0.351 \pm 6.479e-3$ | $1.023 \pm 6.784e-3$ | $0.342 \pm 4.256e-3$ | $0.968 \pm 4.758e-3$ |
| medgemma zero_shot +TFM mlp | $0.385 \pm 8.864e-3$ | $1.077 \pm 1.426e-2$ | $0.343 \pm 6.275e-4$ | $0.979 \pm 1.358e-2$ |
| medgemma zero_shot +TFM patchtsmixer | $0.533 \pm 8.593e-3$ | $1.131 \pm 1.83e-2$ | $0.501 \pm 2.225e-2$ | $1.083 \pm 2.194e-2$ |
| medgemma zero_shot +TFM timemixer | $0.413 \pm 6.214e-3$ | $1.045 \pm 2.5e-3$ | $0.394 \pm 4.21e-3$ | $0.987 \pm 4.679e-3$ |
| medgemma zero_shot lstm | $1.094 \pm 4.891e-2$ | $2.266 \pm 1.103e-1$ | $0.51 \pm 8.577e-2$ | $1.137 \pm 1.391e-2$ |
| medgemma zero_shot mlp | $1.075 \pm 1.523e-2$ | $2.132 \pm 6.644e-2$ | $0.518 \pm 2.548e-2$ | $1.125 \pm 4.125e-2$ |
| medgemma zero_shot patchtsmixer | $1.003 \pm 7.105e-2$ | $1.86 \pm 1.89e-1$ | $0.52 \pm 1.337e-2$ | $1.087 \pm 1.572e-2$ |
| medgemma zero_shot timemixer | $0.956 \pm 7.827e-2$ | $1.802 \pm 1.906e-1$ | $0.492 \pm 6.012e-3$ | $1.088 \pm 7.534e-3$ |
| No_sum_zero_shot zero_shot +TFM lstm | $0.408 \pm 1.453e-2$ | $0.996 \pm 1.043e-2$ | $0.485 \pm 3.59e-2$ | $1.06 \pm 4.806e-2$ |
| No_sum_zero_shot zero_shot +TFM mlp | $0.409 \pm 9.729e-3$ | $1.025 \pm 1.152e-2$ | $0.456 \pm 3.514e-2$ | $1.054 \pm 3.852e-2$ |
| No_sum_zero_shot zero_shot +TFM patchtsmixer | $0.573 \pm 1.885e-2$ | $1.159 \pm 2.325e-2$ | $0.576 \pm 3.064e-2$ | $1.144 \pm 4.679e-2$ |
| No_sum_zero_shot zero_shot +TFM timemixer | $0.533 \pm 1.621e-2$ | $1.118 \pm 1.32e-2$ | $0.56 \pm 3.409e-2$ | $1.15 \pm 5.566e-2$ |
| No_sum_zero_shot lstm | $0.907 \pm 2.519e-2$ | $1.867 \pm 5.855e-2$ | $0.973 \pm 3.373e-2$ | $2.067 \pm 9.612e-2$ |
| No_sum_zero_shot mlp | $0.978 \pm 2.412e-2$ | $2.032 \pm 7.958e-2$ | $0.979 \pm 2.353e-2$ | $2.071 \pm 7.347e-2$ |
| No_sum_zero_shot patchtsmixer | $0.983 \pm 5.443e-2$ | $1.898 \pm 1.248e-1$ | $0.931 \pm 4.189e-2$ | $1.836 \pm 9.39e-2$ |
| No_sum_zero_shot timemixer | $0.824 \pm 5.054e-2$ | $1.623 \pm 1.238e-1$ | $0.873 \pm 3.693e-2$ | $1.753 \pm 9.175e-2$ |
| right lstm | $2.116 \pm 4.78e-2$ | $6.35 \pm 2.965e-1$ | $0.692 \pm 6.11e-2$ | $1.542 \pm 2.375e-1$ |
| right mlp | $2.011 \pm 1.23e-1$ | $5.654 \pm 7.344e-1$ | $1.09 \pm 8.253e-2$ | $2.449 \pm 3.128e-1$ |
| right patchtsmixer | $1.03 \pm 6.371e-2$ | $1.643 \pm 1.166e-1$ | $0.759 \pm 7.229e-2$ | $1.268 \pm 5.882e-2$ |
| right timemixer | $0.963 \pm 6.711e-2$ | $1.487 \pm 9.134e-2$ | $0.599 \pm 3.729e-2$ | $1.03 \pm 2.153e-2$ |

Table 41: Cross-site transfer results - Los. (part 2/2)

| Method | hirid → hirid | | mimic → mimic | | ppicu → ppicu | |
|---|---|---|---|---|---|---|
| | auprc | auroc | auprc | auroc | auprc | auroc |
| Llama-3.1 CoT +TFM lstm | $0.434 \pm 4.047e-2$ | $0.888 \pm 1.253e-2$ | $0.539 \pm 1.934e-2$ | $0.851 \pm 8.073e-3$ | $0.573 \pm 2.554e-2$ | $0.796 \pm 1.254e-2$ |
| Llama-3.1 CoT +TFM mlp | $0.443 \pm 2.504e-2$ | $0.89 \pm 1.262e-2$ | $0.514 \pm 2.709e-2$ | $0.847 \pm 1.626e-2$ | $0.546 \pm 4.932e-2$ | $0.789 \pm 3.076e-2$ |
| Llama-3.1 CoT +TFM patchtsmixer | $0.351 \pm 4.992e-3$ | $0.847 \pm 2.807e-3$ | $0.412 \pm 4.796e-3$ | $0.776 \pm 4.092e-3$ | $0.247 \pm 1.763e-3$ | $0.506 \pm 1.556e-3$ |
| Llama-3.1 CoT +TFM timemixer | $0.343 \pm 1.168e-2$ | $0.847 \pm 4.997e-3$ | $0.488 \pm 4.005e-3$ | $0.834 \pm 2.875e-3$ | $0.537 \pm 3.13e-2$ | $0.788 \pm 1.751e-2$ |
| Llama-3.1 CoT lstm | $0.341 \pm 9.298e-3$ | $0.843 \pm 5.597e-3$ | $0.451 \pm 5.876e-3$ | $0.815 \pm 3.777e-3$ | $0.457 \pm 2.569e-2$ | $0.731 \pm 1.79e-2$ |
| Llama-3.1 CoT mlp | $0.346 \pm 1.218e-2$ | $0.844 \pm 7.496e-3$ | $0.441 \pm 7.156e-3$ | $0.81 \pm 4.386e-3$ | $0.481 \pm 1.72e-2$ | $0.751 \pm 1.171e-2$ |
| Llama-3.1 CoT patchtsmixer | $0.271 \pm 4.8e-2$ | $0.779 \pm 4.136e-2$ | $0.27 \pm 8.385e-2$ | $0.671 \pm 6.974e-2$ | $0.254 \pm 1.391e-2$ | $0.519 \pm 2.568e-2$ |
| Llama-3.1 CoT timemixer | $0.346 \pm 8.836e-3$ | $0.838 \pm 3.985e-3$ | $0.438 \pm 1.006e-2$ | $0.801 \pm 6.004e-3$ | $0.427 \pm 3.391e-2$ | $0.707 \pm 2.318e-2$ |
| Llama-3.1 ICD +TFM lstm | $0.413 \pm 1.303e-2$ | $0.882 \pm 5.344e-3$ | $0.499 \pm 2.606e-2$ | $0.829 \pm 1.53e-2$ | $0.589 \pm 8.374e-3$ | $0.793 \pm 2.423e-3$ |
| Llama-3.1 ICD +TFM mlp | $0.431 \pm 1.249e-2$ | $0.885 \pm 5.047e-3$ | $0.494 \pm 2.567e-2$ | $0.832 \pm 1.578e-2$ | $0.563 \pm 2.534e-2$ | $0.795 \pm 1.56e-2$ |
| Llama-3.1 ICD +TFM patchtsmixer | $0.367 \pm 1.159e-2$ | $0.85 \pm 4.922e-3$ | $0.414 \pm 1.644e-2$ | $0.764 \pm 1.01e-2$ | $0.252 \pm 2.609e-3$ | $0.515 \pm 3.33e-3$ |
| Llama-3.1 ICD +TFM timemixer | $0.316 \pm 4.6e-2$ | $0.833 \pm 3.057e-2$ | $0.467 \pm 9.826e-3$ | $0.816 \pm 7.494e-3$ | $0.529 \pm 1.666e-2$ | $0.772 \pm 9.703e-3$ |
| Llama-3.1 ICD lstm | $0.358 \pm 5.344e-3$ | $0.851 \pm 3.623e-3$ | $0.426 \pm 5.02e-3$ | $0.786 \pm 4.125e-3$ | $0.485 \pm 1.091e-2$ | $0.735 \pm 6.719e-3$ |
| Llama-3.1 ICD mlp | $0.36 \pm 1.804e-2$ | $0.85 \pm 9.91e-3$ | $0.405 \pm 3.751e-3$ | $0.777 \pm 2.018e-3$ | $0.484 \pm 1.949e-2$ | $0.74 \pm 1.339e-2$ |
| Llama-3.1 ICD patchtsmixer | $0.184 \pm 1.02e-1$ | $0.668 \pm 1.099e-1$ | $0.265 \pm 6.944e-2$ | $0.672 \pm 4.457e-2$ | $0.276 \pm 2.049e-2$ | $0.56 \pm 3.181e-2$ |
| Llama-3.1 ICD timemixer | $0.373 \pm 5.533e-3$ | $0.848 \pm 2.639e-3$ | $0.416 \pm 8.855e-3$ | $0.777 \pm 5.424e-3$ | $0.437 \pm 1.335e-2$ | $0.707 \pm 7.682e-3$ |
| Llama-3.1 Trend +TFM lstm | $0.477 \pm 1.975e-2$ | $0.905 \pm 5.623e-3$ | $0.516 \pm 1.086e-2$ | $0.848 \pm 6.583e-3$ | $0.553 \pm 1.78e-2$ | $0.782 \pm 1.068e-2$ |
| Llama-3.1 Trend +TFM mlp | $0.488 \pm 1.448e-2$ | $0.907 \pm 5.459e-3$ | $0.5 \pm 1.744e-3$ | $0.845 \pm 3.669e-4$ | $0.55 \pm 1.294e-2$ | $0.786 \pm 9.126e-3$ |
| Llama-3.1 Trend +TFM patchtsmixer | $0.397 \pm 2.262e-2$ | $0.875 \pm 1.143e-2$ | $0.422 \pm 7.062e-3$ | $0.786 \pm 4.757e-3$ | $0.249 \pm 1.022e-3$ | $0.507 \pm 3.733e-3$ |
| Llama-3.1 Trend +TFM timemixer | $0.383 \pm 3.465e-2$ | $0.867 \pm 1.702e-2$ | $0.451 \pm 6.875e-3$ | $0.819 \pm 6.305e-3$ | $0.529 \pm 1.622e-2$ | $0.776 \pm 8.389e-3$ |
| Llama-3.1 Trend lstm | $0.379 \pm 9.086e-3$ | $0.861 \pm 5.678e-3$ | $0.432 \pm 9.522e-3$ | $0.808 \pm 8.114e-3$ | $0.509 \pm 1.237e-2$ | $0.755 \pm 9.144e-3$ |
| Llama-3.1 Trend mlp | $0.383 \pm 1.052e-2$ | $0.86 \pm 5.351e-3$ | $0.437 \pm 5.673e-3$ | $0.813 \pm 4.773e-3$ | $0.494 \pm 1.749e-2$ | $0.748 \pm 1.242e-2$ |
| Llama-3.1 Trend patchtsmixer | $0.32 \pm 6.834e-2$ | $0.817 \pm 5.205e-2$ | $0.246 \pm 1.042e-2$ | $0.669 \pm 1.561e-2$ | $0.256 \pm 1.661e-2$ | $0.525 \pm 2.969e-2$ |
| Llama-3.1 Trend timemixer | $0.414 \pm 2.107e-2$ | $0.866 \pm 9.814e-3$ | $0.429 \pm 1.088e-2$ | $0.8 \pm 8e-3$ | $0.487 \pm 8.158e-3$ | $0.739 \pm 5.146e-3$ |
| Llama-3.1 zero_shot +TFM lstm | $0.479 \pm 1.003e-2$ | $0.904 \pm 2.183e-3$ | $0.57 \pm 1.765e-2$ | $0.867 \pm 6.438e-3$ | $0.564 \pm 1.818e-2$ | $0.787 \pm 7.777e-3$ |
| Llama-3.1 zero_shot +TFM mlp | $0.493 \pm 7.385e-3$ | $0.907 \pm 1.986e-3$ | $0.498 \pm 1.421e-2$ | $0.84 \pm 8.639e-3$ | $0.54 \pm 3.381e-2$ | $0.783 \pm 1.955e-2$ |
| Llama-3.1 zero_shot +TFM patchtsmixer | $0.359 \pm 2.186e-2$ | $0.863 \pm 1.464e-2$ | $0.428 \pm 8.876e-3$ | $0.786 \pm 5.617e-3$ | $0.251 \pm 2.901e-3$ | $0.511 \pm 7.165e-3$ |
| Llama-3.1 zero_shot +TFM timemixer | $0.37 \pm 5.431e-3$ | $0.867 \pm 1.671e-2$ | $0.479 \pm 2.191e-2$ | $0.833 \pm 1.254e-2$ | $0.48 \pm 1.758e-2$ | $0.749 \pm 1.067e-2$ |
| Llama-3.1 zero_shot lstm | $0.398 \pm 2.089e-2$ | $0.873 \pm 4.317e-2$ | $0.437 \pm 1.317e-2$ | $0.813 \pm 9.215e-3$ | $0.492 \pm 2.239e-2$ | $0.746 \pm 1.226e-2$ |
| Llama-3.1 zero_shot mlp | $0.407 \pm 3.909e-3$ | $0.876 \pm 1.509e-3$ | $0.441 \pm 6.955e-3$ | $0.816 \pm 4.737e-3$ | $0.465 \pm 2.571e-2$ | $0.736 \pm 1.543e-2$ |
| Llama-3.1 zero_shot patchtsmixer | $0.316 \pm 8.626e-2$ | $0.823 \pm 6.232e-2$ | $0.281 \pm 8.333e-2$ | $0.689 \pm 6.299e-2$ | $0.262 \pm 1.707e-2$ | $0.537 \pm 3.236e-2$ |
| Llama-3.1 zero_shot timemixer | $0.394 \pm 8.982e-3$ | $0.866 \pm 1.953e-3$ | $0.434 \pm 1.654e-2$ | $0.805 \pm 8.895e-3$ | $0.473 \pm 1.179e-2$ | $0.736 \pm 6.462e-3$ |
| TFM lstm | $0.363 \pm 8.238e-3$ | $0.856 \pm 1.688e-3$ | $0.485 \pm 2.578e-2$ | $0.797 \pm 2.742e-3$ | $0.402 \pm 1.312e-2$ | $0.676 \pm 1.54e-2$ |
| TFM mlp | $0.361 \pm 1.466e-2$ | $0.851 \pm 3.699e-3$ | $0.474 \pm 6.051e-2$ | $0.792 \pm 3.642e-3$ | $0.412 \pm 1.503e-2$ | $0.689 \pm 1.612e-2$ |
| TFM patchtsmixer | $0.266 \pm 8.085e-2$ | $0.793 \pm 3.754e-3$ | $0.435 \pm 6.064e-3$ | $0.753 \pm 3.762e-3$ | $0.279 \pm 1.976e-3$ | $0.511 \pm 3.903e-3$ |
| TFM timemixer | $0.263 \pm 7.734e-3$ | $0.815 \pm 3.212e-3$ | $0.417 \pm 4.331e-2$ | $0.77 \pm 1.75e-2$ | $0.377 \pm 1.22e-3$ | $0.664 \pm 2.151e-3$ |
| TSDE lstm | $0.539 \pm 1.805e-2$ | $0.92 \pm 3.811e-3$ | $0.708 \pm 2.806e-3$ | $0.913 \pm 3.496e-4$ | $0.745 \pm 7.439e-3$ | $0.885 \pm 4.187e-3$ |
| TSDE mlp | $0.528 \pm 1.921e-2$ | $0.921 \pm 2.489e-3$ | $0.705 \pm 2.821e-3$ | $0.913 \pm 1.258e-3$ | $0.751 \pm 4.275e-3$ | $0.888 \pm 1.822e-3$ |
| TSDE patchtsmixer | $0.324 \pm 1.226e-1$ | $0.806 \pm 1.361e-1$ | $0.494 \pm 2.136e-1$ | $0.793 \pm 1.772e-1$ | $0.583 \pm 2.046e-1$ | $0.765 \pm 1.73e-1$ |
| TSDE timemixer | $0.48 \pm 2.2e-2$ | $0.909 \pm 2.952e-3$ | $0.679 \pm 1.438e-2$ | $0.906 \pm 4.163e-3$ | $0.741 \pm 3.528e-3$ | $0.884 \pm 3.234e-3$ |
| gemini-2.0-flash CoT +TFM lstm | $0.545 \pm 1.623e-2$ | $0.92 \pm 5.141e-3$ | $0.57 \pm 3.002e-3$ | $0.868 \pm 1.086e-3$ | $0.667 \pm 2.369e-2$ | $0.843 \pm 1.084e-2$ |
| gemini-2.0-flash CoT +TFM mlp | $0.533 \pm 1.684e-2$ | $0.913 \pm 5.039e-3$ | $0.556 \pm 1.724e-2$ | $0.865 \pm 8.301e-3$ | $0.629 \pm 2.831e-3$ | $0.83 \pm 1.045e-3$ |
| gemini-2.0-flash CoT +TFM patchtsmixer | $0.464 \pm 2.736e-2$ | $0.894 \pm 6.52e-3$ | $0.469 \pm 2.338e-2$ | $0.811 \pm 1.472e-2$ | $0.568 \pm 2.321e-2$ | $0.782 \pm 9.886e-3$ |
| gemini-2.0-flash CoT +TFM timemixer | $0.463 \pm 2.377e-2$ | $0.897 \pm 7.053e-3$ | $0.506 \pm 1.608e-2$ | $0.845 \pm 6.596e-3$ | $0.611 \pm 1.684e-2$ | $0.827 \pm 7.697e-3$ |
| gemini-2.0-flash CoT lstm | $0.488 \pm 1.318e-2$ | $0.898 \pm 5.601e-3$ | $0.535 \pm 1.128e-2$ | $0.849 \pm 6.038e-3$ | $0.609 \pm 2.214e-2$ | $0.813 \pm 1.378e-2$ |
| gemini-2.0-flash CoT mlp | $0.495 \pm 5.355e-3$ | $0.9 \pm 2.406e-3$ | $0.543 \pm 4.201e-3$ | $0.855 \pm 2.708e-3$ | $0.599 \pm 2.296e-2$ | $0.811 \pm 1.515e-2$ |
| gemini-2.0-flash CoT patchtsmixer | $0.46 \pm 7.063e-3$ | $0.886 \pm 2.332e-3$ | $0.352 \pm 8.72e-2$ | $0.737 \pm 5.631e-2$ | $0.358 \pm 1.421e-1$ | $0.627 \pm 1.101e-1$ |
| gemini-2.0-flash CoT timemixer | $0.515 \pm 1.074e-2$ | $0.897 \pm 3.754e-3$ | $0.527 \pm 5.431e-3$ | $0.84 \pm 3.986e-3$ | $0.583 \pm 4.305e-3$ | $0.795 \pm 2.4e-3$ |
| gemini-2.0-flash ICD +TFM lstm | $0.542 \pm 1.859e-2$ | $0.92 \pm 5.373e-3$ | $0.575 \pm 3.127e-3$ | $0.869 \pm 1.216e-3$ | $0.617 \pm 1.63e-2$ | $0.819 \pm 8.116e-3$ |
| gemini-2.0-flash ICD +TFM mlp | $0.539 \pm 1.848e-2$ | $0.917 \pm 5.173e-3$ | $0.558 \pm 1.31e-2$ | $0.865 \pm 7.865e-3$ | $0.656 \pm 1.286e-2$ | $0.849 \pm 9.153e-3$ |
| gemini-2.0-flash ICD +TFM patchtsmixer | $0.463 \pm 4.079e-2$ | $0.893 \pm 1.05e-2$ | $0.494 \pm 4.833e-3$ | $0.819 \pm 5.417e-3$ | $0.446 \pm 1.682e-1$ | $0.683 \pm 1.509e-1$ |
| gemini-2.0-flash ICD +TFM timemixer | $0.414 \pm 2.549e-2$ | $0.884 \pm 1.156e-2$ | $0.521 \pm 1.206e-2$ | $0.849 \pm 8.343e-3$ | $0.592 \pm 2.55e-2$ | $0.817 \pm 1.571e-2$ |
| gemini-2.0-flash ICD lstm | $0.491 \pm 9.256e-3$ | $0.903 \pm 3.847e-3$ | $0.519 \pm 2.924e-3$ | $0.844 \pm 1.835e-3$ | $0.601 \pm 1.177e-2$ | $0.812 \pm 9.131e-3$ |
| gemini-2.0-flash ICD mlp | $0.488 \pm 9.744e-3$ | $0.902 \pm 4.768e-3$ | $0.522 \pm 6.623e-3$ | $0.847 \pm 4.112e-3$ | $0.582 \pm 2.098e-2$ | $0.804 \pm 1.415e-2$ |
| gemini-2.0-flash ICD patchtsmixer | $0.372 \pm 1.737e-1$ | $0.798 \pm 1.752e-1$ | $0.384 \pm 1.383e-1$ | $0.749 \pm 9.318e-2$ | $0.265 \pm 1.408e-2$ | $0.542 \pm 2.714e-2$ |
| gemini-2.0-flash ICD timemixer | $0.507 \pm 1.638e-2$ | $0.9 \pm 5.618e-3$ | $0.518 \pm 1.333e-2$ | $0.839 \pm 7.76e-3$ | $0.552 \pm 2.526e-2$ | $0.781 \pm 1.502e-2$ |
| gemini-2.0-flash Trend +TFM lstm | $0.506 \pm 4.189e-3$ | $0.911 \pm 1.254e-3$ | $0.547 \pm 1.392e-2$ | $0.86 \pm 5.845e-3$ | $0.659 \pm 2.17e-2$ | $0.839 \pm 7.601e-3$ |
| gemini-2.0-flash Trend +TFM mlp | $0.526 \pm 2.121e-2$ | $0.917 \pm 7.035e-3$ | $0.541 \pm 6.958e-3$ | $0.863 \pm 2.501e-3$ | $0.611 \pm 1.525e-2$ | $0.827 \pm 9.683e-3$ |
| gemini-2.0-flash Trend +TFM patchtsmixer | $0.389 \pm 3.568e-2$ | $0.866 \pm 1.646e-2$ | $0.454 \pm 5.615e-3$ | $0.805 \pm 6.018e-3$ | $0.448 \pm 1.73e-1$ | $0.686 \pm 1.545e-1$ |
| gemini-2.0-flash Trend +TFM timemixer | $0.428 \pm 6.544e-2$ | $0.883 \pm 2.307e-2$ | $0.504 \pm 1.568e-2$ | $0.848 \pm 8.365e-3$ | $0.569 \pm 3.539e-2$ | $0.809 \pm 1.971e-2$ |
| gemini-2.0-flash Trend lstm | $0.455 \pm 2.345e-2$ | $0.891 \pm 1.043e-2$ | $0.502 \pm 8.363e-3$ | $0.845 \pm 3.571e-3$ | $0.557 \pm 2.188e-2$ | $0.791 \pm 1.465e-2$ |
| gemini-2.0-flash Trend mlp | $0.453 \pm 3.441e-2$ | $0.888 \pm 1.365e-2$ | $0.496 \pm 2.96e-3$ | $0.842 \pm 1.715e-3$ | $0.556 \pm 2.6e-2$ | $0.793 \pm 1.715e-2$ |
| gemini-2.0-flash Trend patchtsmixer | $0.345 \pm 1.025e-1$ | $0.826 \pm 7.356e-2$ | $0.304 \pm 1.152e-1$ | $0.701 \pm 8.825e-2$ | $0.318 \pm 1.327e-1$ | $0.581 \pm 1.204e-1$ |
| gemini-2.0-flash Trend timemixer | $0.46 \pm 2.204e-2$ | $0.884 \pm 8.693e-3$ | $0.499 \pm 9.007e-3$ | $0.838 \pm 4.022e-3$ | $0.537 \pm 7.438e-3$ | $0.775 \pm 4.428e-3$ |
| gemini-2.0-flash zero_shot +TFM lstm | $0.559 \pm 1.48e-2$ | $0.92 \pm 4.149e-3$ | $0.59 \pm 1.774e-2$ | $0.877 \pm 8.688e-3$ | $0.671 \pm 1.047e-2$ | $0.848 \pm 3.791e-3$ |
| gemini-2.0-flash zero_shot +TFM mlp | $0.588 \pm 1.514e-2$ | $0.926 \pm 4.635e-3$ | $0.577 \pm 2.385e-3$ | $0.875 \pm 9.789e-4$ | $0.661 \pm 1.407e-2$ | $0.85 \pm 9.313e-3$ |
| gemini-2.0-flash zero_shot +TFM patchtsmixer | $0.529 \pm 3.47e-2$ | $0.913 \pm 7.443e-3$ | $0.481 \pm 3.513e-2$ | $0.818 \pm 1.429e-2$ | $0.585 \pm 3.289e-2$ | $0.801 \pm 1.514e-2$ |
| gemini-2.0-flash zero_shot +TFM timemixer | $0.482 \pm 4.511e-2$ | $0.897 \pm 1.444e-2$ | $0.545 \pm 1.157e-2$ | $0.859 \pm 1.057e-2$ | $0.61 \pm 1.279e-2$ | $0.825 \pm 6.485e-3$ |
| gemini-2.0-flash zero_shot lstm | $0.52 \pm 6.046e-3$ | $0.909 \pm 1.438e-3$ | $0.548 \pm 5.271e-3$ | $0.858 \pm 3.121e-3$ | $0.611 \pm 1.238e-2$ | $0.822 \pm 6.855e-3$ |
| gemini-2.0-flash zero_shot mlp | $0.514 \pm 8.721e-3$ | $0.905 \pm 2.253e-3$ | $0.55 \pm 5.146e-3$ | $0.862 \pm 3.469e-3$ | $0.59 \pm 1.971e-2$ | $0.811 \pm 1.093e-2$ |
| gemini-2.0-flash zero_shot patchtsmixer | $0.502 \pm 6.089e-3$ | $0.903 \pm 3.096e-3$ | $0.367 \pm 1.597e-1$ | $0.724 \pm 1.452e-1$ | $0.252 \pm 6.236e-3$ | $0.518 \pm 1.4e-2$ |
| gemini-2.0-flash zero_shot timemixer | $0.53 \pm 7.372e-3$ | $0.904 \pm 2.18e-3$ | $0.556 \pm 8.705e-3$ | $0.86 \pm 5.334e-3$ | $0.588 \pm 3.082e-3$ | $0.804 \pm 1.6e-3$ |
| interp lstm | $0.531 \pm 8.932e-2$ | $0.891 \pm 4.505e-2$ | $0.635 \pm 9.586e-3$ | $0.888 \pm 5.944e-3$ | $0.704 \pm 4.107e-2$ | $0.844 \pm 4.157e-2$ |
| interp mlp | $0.552 \pm 9.912e-3$ | $0.909 \pm 1.24e-2$ | $0.631 \pm 5.08e-3$ | $0.89 \pm 2.156e-3$ | $0.715 \pm 7.563e-3$ | $0.863 \pm 4.351e-3$ |
| interp patchtsmixer | $0.09 \pm 0e0$ | $0.5 \pm 0e0$ | $0.161 \pm 0e0$ | $0.5 \pm 0e0$ | $0.273 \pm 0e0$ | $0.5 \pm 0e0$ |
| interp timemixer | $0.44 \pm 5.277e-2$ | $0.873 \pm 1.352e-2$ | $0.536 \pm 1.41e-2$ | $0.842 \pm 9.574e-3$ | $0.614 \pm 7.715e-3$ | $0.804 \pm 9.152e-3$ |
| mean lstm | $0.522 \pm 1.259e-1$ | $0.898 \pm 4.499e-2$ | $0.59 \pm 6.604e-2$ | $0.862 \pm 3.451e-2$ | $0.678 \pm 1.46e-2$ | $0.843 \pm 1.883e-2$ |
| mean mlp | $0.549 \pm 2.305e-2$ | $0.912 \pm 5.273e-3$ | $0.583 \pm 4.494e-2$ | $0.863 \pm 2.267e-2$ | $0.683 \pm 6.58e-3$ | $0.852 \pm 1.164e-2$ |
| mean patchtsmixer | $0.09 \pm 7.05e-4$ | $0.5 \pm 3.5e-4$ | $0.161 \pm 0e0$ | $0.5 \pm 0e0$ | $0.273 \pm 0e0$ | $0.5 \pm 0e0$ |
| mean timemixer | $0.59 \pm 8.689e-3$ | $0.927 \pm 2.573e-3$ | $0.62 \pm 1.842e-2$ | $0.881 \pm 9.053e-3$ | $0.654 \pm 1.856e-2$ | $0.827 \pm 1.538e-2$ |
| No_sum_CoT lstm | $0.634 \pm 8.476e-3$ | $0.93 \pm 2.699e-3$ | $0.621 \pm 3.847e-3$ | $0.888 \pm 1.819e-3$ | $0.681 \pm 5.061e-3$ | $0.852 \pm 3.018e-3$ |
| No_sum_CoT mlp | $0.603 \pm 1.558e-2$ | $0.925 \pm 3.711e-3$ | $0.615 \pm 6.819e-3$ | $0.888 \pm 2.711e-3$ | $0.682 \pm 8.652e-3$ | $0.856 \pm 5.444e-3$ |
| No_sum_CoT patchtsmixer | $0.427 \pm 1.858e-1$ | $0.829 \pm 1.538e-1$ | $0.405 \pm 1.571e-1$ | $0.768 \pm 1.107e-1$ | $0.38 \pm 1.891e-1$ | $0.638 \pm 1.454e-1$ |
| No_sum_CoT timemixer | $0.595 \pm 1.814e-2$ | $0.92 \pm 4.36e-3$ | $0.593 \pm 4.734e-3$ | $0.876 \pm 2.179e-3$ | $0.653 \pm 6.419e-3$ | $0.833 \pm 4.429e-3$ |
| No_sum_ICD lstm | $0.601 \pm 1.16e-2$ | $0.924 \pm 2.787e-3$ | $0.611 \pm 3.797e-3$ | $0.885 \pm 2.014e-3$ | $0.664 \pm 5.689e-3$ | $0.843 \pm 3.831e-3$ |
| No_sum_ICD mlp | $0.587 \pm 8.258e-3$ | $0.922 \pm 1.93e-3$ | $0.592 \pm 1.31e-2$ | $0.88 \pm 5.193e-3$ | $0.665 \pm 1.535e-2$ | $0.847 \pm 9.774e-3$ |
| No_sum_ICD patchtsmixer | $0.44 \pm 1.527e-1$ | $0.868 \pm 7.703e-2$ | $0.365 \pm 1.131e-1$ | $0.751 \pm 6.852e-2$ | $0.28 \pm 2.339e-2$ | $0.562 \pm 2.774e-2$ |
| No_sum_ICD timemixer | $0.58 \pm 4.962e-3$ | $0.918 \pm 1.255e-3$ | $0.588 \pm 7.78e-3$ | $0.873 \pm 3.758e-3$ | $0.651 \pm 1.578e-2$ | $0.834 \pm 9.552e-3$ |
| No_sum_Trend lstm | $0.629 \pm 4.602e-3$ | $0.928 \pm 1.935e-3$ | $0.614 \pm 5.217e-3$ | $0.887 \pm 2.218e-3$ | $0.687 \pm 9.539e-3$ | $0.856 \pm 5.409e-3$ |
| No_sum_Trend mlp | $0.598 \pm 7.807e-3$ | $0.924 \pm 1.88e-3$ | $0.613 \pm 1.319e-2$ | $0.889 \pm 5.065e-3$ | $0.679 \pm 1.49e-2$ | $0.853 \pm 8.805e-3$ |
| No_sum_Trend patchtsmixer | $0.467 \pm 1.18e-1$ | $0.883 \pm 4.873e-2$ | $0.359 \pm 1.188e-1$ | $0.746 \pm 7.483e-2$ | $0.258 \pm 1.191e-2$ | $0.532 \pm 2.675e-2$ |
| No_sum_Trend timemixer | $0.602 \pm 8.593e-3$ | $0.921 \pm 2.787e-3$ | $0.605 \pm 1.238e-2$ | $0.883 \pm 5.221e-3$ | $0.642 \pm 2.953e-3$ | $0.827 \pm 1.822e-3$ |
| medgemma CoT +TFM lstm | $0.592 \pm 1.745e-2$ | $0.927 \pm 4.496e-3$ | $0.624 \pm 1.678e-2$ | $0.89 \pm 6.438e-3$ | $0.68 \pm 2.922e-2$ | $0.853 \pm 1.525e-2$ |
| medgemma CoT +TFM mlp | $0.6 \pm 3.052e-2$ | $0.929 \pm 9.352e-3$ | $0.593 \pm 1.427e-3$ | $0.882 \pm 1.475e-3$ | $0.638 \pm 1.244e-2$ | $0.839 \pm 7.595e-3$ |
| medgemma CoT +TFM patchtsmixer | $0.485 \pm 2.384e-2$ | $0.903 \pm 7.847e-3$ | $0.503 \pm 8.196e-3$ | $0.833 \pm 5.26e-3$ | $0.562 \pm 3.049e-2$ | $0.786 \pm 1.93e-2$ |
| medgemma CoT +TFM timemixer | $0.523 \pm 1.243e-2$ | $0.91 \pm 2.409e-3$ | $0.568 \pm 1.704e-2$ | $0.874 \pm 8.405e-3$ | $0.615 \pm 2.055e-2$ | $0.832 \pm 1.059e-2$ |
| medgemma CoT lstm | $0.556 \pm 9.648e-3$ | $0.917 \pm 3.49e-3$ | $0.577 \pm 8.543e-3$ | $0.874 \pm 3.529e-3$ | $0.604 \pm 1.558e-2$ | $0.818 \pm 1.096e-2$ |
| medgemma CoT mlp | $0.542 \pm 2.072e-2$ | $0.912 \pm 6.416e-3$ | $0.583 \pm 1.125e-2$ | $0.879 \pm 5.863e-3$ | $0.589 \pm 6.352e-3$ | $0.81 \pm 4.491e-3$ |
| medgemma CoT patchtsmixer | $0.407 \pm 1.298e-1$ | $0.854 \pm 7.869e-2$ | $0.357 \pm 9.654e-2$ | $0.742 \pm 6.527e-2$ | $0.304 \pm 1.14e-2$ | $0.591 \pm 1.576e-2$ |
| medgemma CoT timemixer | $0.543 \pm 1.328e-2$ | $0.906 \pm 3.944e-3$ | $0.57 \pm 1.39e-2$ | $0.866 \pm 6.891e-3$ | $0.568 \pm 1.264e-2$ | $0.789 \pm 7.697e-3$ |

Table 42: In-distribution results - Mort. (part 1/2)

effPublished as a conference paper at ICLR 2026

| Method | hirid → hirid | | mimic → mimic | | ppicu → ppicu | |
|---|---|---|---|---|---|---|
| | auprc | auroc | auprc | auroc | auprc | auroc |
| medgemma ICD +TFM lstm | $0.582 \pm 6.357e-3$ | $0.927 \pm 2.878e-3$ | $0.572 \pm 1.236e-2$ | $0.87 \pm 5.52e-3$ | $0.66 \pm 2.533e-2$ | $0.844 \pm 1.297e-2$ |
| medgemma ICD +TFM mlp | $0.582 \pm 2.505e-2$ | $0.929 \pm 8.287e-3$ | $0.571 \pm 4.68e-3$ | $0.874 \pm 4.162e-4$ | $0.634 \pm 1.234e-2$ | $0.837 \pm 7.092e-3$ |
| medgemma ICD +TFM patchtsmixer | $0.467 \pm 5.098e-3$ | $0.899 \pm 2.98e-3$ | $0.489 \pm 2.237e-2$ | $0.821 \pm 1.109e-2$ | $0.582 \pm 1.334e-2$ | $0.786 \pm 8.672e-3$ |
| medgemma ICD +TFM timemixer | $0.453 \pm 6.038e-2$ | $0.894 \pm 1.792e-2$ | $0.526 \pm 1.474e-2$ | $0.853 \pm 7.344e-3$ | $0.609 \pm 1.786e-2$ | $0.827 \pm 1.059e-2$ |
| medgemma ICD lstm | $0.506 \pm 2.012e-2$ | $0.906 \pm 7.309e-3$ | $0.547 \pm 1.449e-2$ | $0.858 \pm 7.275e-3$ | $0.594 \pm 8.807e-3$ | $0.81 \pm 5.606e-3$ |
| medgemma ICD mlp | $0.492 \pm 1.938e-2$ | $0.901 \pm 7.272e-3$ | $0.538 \pm 5.229e-3$ | $0.858 \pm 2.935e-3$ | $0.586 \pm 1.356e-2$ | $0.807 \pm 9.684e-3$ |
| medgemma ICD patchtsmixer | $0.381 \pm 1.088e-1$ | $0.85 \pm 6.58e-2$ | $0.385 \pm 1.089e-1$ | $0.764 \pm 6.721e-2$ | $0.28 \pm 1.13e-2$ | $0.564 \pm 1.614e-2$ |
| medgemma ICD timemixer | $0.504 \pm 1.269e-2$ | $0.899 \pm 4.643e-3$ | $0.53 \pm 1.274e-2$ | $0.847 \pm 6.164e-3$ | $0.574 \pm 5.149e-3$ | $0.794 \pm 2.91e-3$ |
| medgemma Trend +TFM lstm | $0.524 \pm 3.32e-2$ | $0.911 \pm 1.015e-2$ | $0.549 \pm 1.372e-2$ | $0.864 \pm 5.829e-3$ | $0.643 \pm 3.179e-2$ | $0.827 \pm 1.416e-2$ |
| medgemma Trend +TFM mlp | $0.573 \pm 1.708e-2$ | $0.928 \pm 6.287e-3$ | $0.532 \pm 1.378e-2$ | $0.86 \pm 7.304e-3$ | $0.595 \pm 1.673e-2$ | $0.813 \pm 9.188e-3$ |
| medgemma Trend +TFM patchtsmixer | $0.371 \pm 5.494e-3$ | $0.863 \pm 3.222e-3$ | $0.456 \pm 2.658e-2$ | $0.818 \pm 1.328e-2$ | $0.25 \pm 5.324e-3$ | $0.509 \pm 4.151e-3$ |
| medgemma Trend +TFM timemixer | $0.396 \pm 3.892e-2$ | $0.87 \pm 1.642e-2$ | $0.499 \pm 1.572e-2$ | $0.85 \pm 7.941e-3$ | $0.56 \pm 2.041e-2$ | $0.796 \pm 1.352e-2$ |
| medgemma Trend lstm | $0.44 \pm 2.584e-2$ | $0.884 \pm 8.082e-3$ | $0.501 \pm 6.582e-3$ | $0.846 \pm 3.181e-3$ | $0.565 \pm 1.96e-2$ | $0.788 \pm 1.193e-2$ |
| medgemma Trend mlp | $0.454 \pm 2.115e-2$ | $0.886 \pm 6.986e-3$ | $0.492 \pm 1.114e-2$ | $0.843 \pm 5.195e-3$ | $0.522 \pm 2.362e-2$ | $0.767 \pm 1.326e-2$ |
| medgemma Trend patchtsmixer | $0.365 \pm 4.961e-3$ | $0.858 \pm 3.198e-3$ | $0.369 \pm 8.7e-2$ | $0.769 \pm 6.259e-2$ | $0.334 \pm 1.194e-1$ | $0.608 \pm 1.03e-1$ |
| medgemma Trend timemixer | $0.441 \pm 1.48e-2$ | $0.875 \pm 4.638e-3$ | $0.495 \pm 3.069e-3$ | $0.843 \pm 2.361e-3$ | $0.522 \pm 8.734e-3$ | $0.762 \pm 6.247e-3$ |
| medgemma zero_shot +TFM lstm | $0.579 \pm 2.183e-2$ | $0.924 \pm 5.858e-3$ | $0.602 \pm 1.126e-2$ | $0.884 \pm 5.661e-3$ | $0.644 \pm 1.1e-2$ | $0.836 \pm 4.788e-3$ |
| medgemma zero_shot +TFM mlp | $0.57 \pm 1.762e-2$ | $0.921 \pm 5.704e-3$ | $0.579 \pm 2.067e-2$ | $0.876 \pm 1.088e-2$ | $0.65 \pm 2.438e-2$ | $0.846 \pm 1.564e-2$ |
| medgemma zero_shot +TFM patchtsmixer | $0.418 \pm 6.342e-2$ | $0.881 \pm 2.308e-2$ | $0.478 \pm 2.752e-2$ | $0.818 \pm 1.679e-2$ | $0.572 \pm 1.182e-2$ | $0.79 \pm 7.234e-3$ |
| medgemma zero_shot +TFM timemixer | $0.459 \pm 4.196e-2$ | $0.892 \pm 1.27e-2$ | $0.553 \pm 1.822e-2$ | $0.868 \pm 1.061e-2$ | $0.614 \pm 1.171e-2$ | $0.829 \pm 4.433e-3$ |
| medgemma zero_shot lstm | $0.546 \pm 6.857e-3$ | $0.913 \pm 2.359e-3$ | $0.551 \pm 9.915e-3$ | $0.863 \pm 4.472e-3$ | $0.606 \pm 1.032e-2$ | $0.818 \pm 6.156e-3$ |
| medgemma zero_shot mlp | $0.545 \pm 8.764e-3$ | $0.913 \pm 3.428e-3$ | $0.559 \pm 6.325e-3$ | $0.868 \pm 2.58e-3$ | $0.598 \pm 1.61e-2$ | $0.815 \pm 1.05e-2$ |
| medgemma zero_shot patchtsmixer | $0.462 \pm 7.816e-3$ | $0.887 \pm 4.008e-3$ | $0.463 \pm 8.225e-2$ | $0.814 \pm 5.34e-2$ | $0.3 \pm 1.767e-2$ | $0.592 \pm 1.908e-2$ |
| medgemma zero_shot timemixer | $0.543 \pm 6.09e-3$ | $0.908 \pm 2.577e-3$ | $0.562 \pm 1.155e-2$ | $0.862 \pm 6.347e-3$ | $0.585 \pm 5.476e-3$ | $0.801 \pm 3.374e-3$ |
| No_sum_zero_shot lstm | $0.615 \pm 2.522e-2$ | $0.925 \pm 6.661e-3$ | $0.606 \pm 6.168e-3$ | $0.883 \pm 2.376e-3$ | $0.686 \pm 9.426e-3$ | $0.855 \pm 5.876e-3$ |
| No_sum_zero_shot mlp | $0.603 \pm 9.87e-3$ | $0.924 \pm 2.557e-3$ | $0.616 \pm 1.695e-2$ | $0.89 \pm 6.911e-3$ | $0.669 \pm 2.192e-2$ | $0.847 \pm 1.442e-2$ |
| No_sum_zero_shot patchtsmixer | $0.468 \pm 1.222e-1$ | $0.882 \pm 5.065e-2$ | $0.479 \pm 1.207e-1$ | $0.823 \pm 7.118e-2$ | $0.267 \pm 3.135e-2$ | $0.54 \pm 4.644e-2$ |
| No_sum_zero_shot timemixer | $0.587 \pm 1.445e-2$ | $0.917 \pm 4.193e-3$ | $0.579 \pm 2.141e-2$ | $0.871 \pm 9.394e-3$ | $0.647 \pm 1.746e-2$ | $0.83 \pm 1.127e-2$ |
| right lstm | $0.572 \pm 1.18e-1$ | $0.898 \pm 6.297e-2$ | $0.675 \pm 1.337e-2$ | $0.901 \pm 4.097e-3$ | $0.726 \pm 2.893e-2$ | $0.863 \pm 2.959e-2$ |
| right mlp | $0.625 \pm 2.227e-2$ | $0.918 \pm 6.474e-3$ | $0.665 \pm 9.48e-3$ | $0.896 \pm 4.352e-3$ | $0.742 \pm 7.615e-3$ | $0.874 \pm 7.008e-3$ |
| right patchtsmixer | $0.09 \pm 0e0$ | $0.5 \pm 1.15e-4$ | $0.161 \pm 0e0$ | $0.5 \pm 0e0$ | $0.273 \pm 0e0$ | $0.5 \pm 0e0$ |
| right timemixer | $0.54 \pm 2.507e-2$ | $0.907 \pm 1.645e-2$ | $0.589 \pm 1.16e-2$ | $0.867 \pm 3.511e-3$ | $0.648 \pm 5.734e-3$ | $0.821 \pm 2.964e-3$ |

Table 43: In-distribution results - Mort. (part 2/2)

| Method | hirid → ppicu | | mimic → ppicu | |
|---|---|---|---|---|
| | auprc | auroc | auprc | auroc |
| Llama-3.1 CoT +TFM lstm | $0.365 \pm 5.073e-3$ | $0.656 \pm 2.368e-3$ | $0.392 \pm 2.422e-3$ | $0.681 \pm 9.8e-4$ |
| Llama-3.1 CoT +TFM mlp | $0.373 \pm 8.213e-4$ | $0.664 \pm 1.107e-3$ | $0.383 \pm 7.763e-3$ | $0.67 \pm 6.792e-3$ |
| Llama-3.1 CoT +TFM patchtsmixer | $0.342 \pm 2.198e-3$ | $0.625 \pm 1.408e-3$ | $0.361 \pm 4.654e-3$ | $0.633 \pm 5.435e-3$ |
| Llama-3.1 CoT +TFM timemixer | $0.369 \pm 4.226e-3$ | $0.662 \pm 3.855e-3$ | $0.376 \pm 2.861e-3$ | $0.665 \pm 2.893e-3$ |
| Llama-3.1 CoT lstm | $0.362 \pm 3.588e-3$ | $0.653 \pm 1.988e-3$ | $0.365 \pm 1.458e-3$ | $0.652 \pm 1.682e-3$ |
| Llama-3.1 CoT mlp | $0.361 \pm 4.063e-3$ | $0.65 \pm 4.195e-3$ | $0.365 \pm 2.772e-3$ | $0.652 \pm 2.356e-3$ |
| Llama-3.1 CoT patchtsmixer | $0.32 \pm 3.425e-2$ | $0.602 \pm 4.628e-2$ | $0.314 \pm 3.019e-2$ | $0.6 \pm 3.221e-2$ |
| Llama-3.1 CoT timemixer | $0.357 \pm 1.49e-3$ | $0.647 \pm 1.388e-3$ | $0.357 \pm 1.373e-3$ | $0.647 \pm 1.679e-3$ |
| Llama-3.1 ICD +TFM lstm | $0.36 \pm 1.051e-3$ | $0.656 \pm 1.813e-3$ | $0.389 \pm 4.147e-3$ | $0.674 \pm 4.169e-3$ |
| Llama-3.1 ICD +TFM mlp | $0.367 \pm 3.983e-3$ | $0.661 \pm 3.915e-3$ | $0.393 \pm 3.627e-3$ | $0.676 \pm 3.508e-3$ |
| Llama-3.1 ICD +TFM patchtsmixer | $0.347 \pm 9.868e-4$ | $0.632 \pm 2.735e-3$ | $0.361 \pm 2.17e-3$ | $0.637 \pm 2.782e-3$ |
| Llama-3.1 ICD +TFM timemixer | $0.368 \pm 2.004e-3$ | $0.664 \pm 6.352e-3$ | $0.388 \pm 1.479e-3$ | $0.675 \pm 1.013e-3$ |
| Llama-3.1 ICD lstm | $0.359 \pm 1.49e-3$ | $0.657 \pm 9.47e-4$ | $0.385 \pm 4.528e-3$ | $0.674 \pm 2.962e-3$ |
| Llama-3.1 ICD mlp | $0.361 \pm 2.11e-3$ | $0.657 \pm 7.309e-4$ | $0.386 \pm 1.053e-3$ | $0.674 \pm 4.826e-4$ |
| Llama-3.1 ICD patchtsmixer | $0.291 \pm 4.389e-2$ | $0.578 \pm 5.288e-2$ | $0.31 \pm 3.568e-2$ | $0.604 \pm 3.397e-2$ |
| Llama-3.1 ICD timemixer | $0.362 \pm 1.846e-3$ | $0.658 \pm 1.467e-4$ | $0.388 \pm 3.129e-3$ | $0.672 \pm 1.952e-3$ |
| Llama-3.1 Trend +TFM lstm | $0.355 \pm 3.702e-3$ | $0.655 \pm 3.368e-3$ | $0.388 \pm 7.469e-3$ | $0.677 \pm 5.3e-3$ |
| Llama-3.1 Trend +TFM mlp | $0.36 \pm 4.772e-3$ | $0.66 \pm 3.541e-3$ | $0.391 \pm 9.703e-3$ | $0.677 \pm 4.798e-3$ |
| Llama-3.1 Trend +TFM patchtsmixer | $0.341 \pm 5.777e-3$ | $0.624 \pm 4.799e-3$ | $0.375 \pm 2.775e-3$ | $0.652 \pm 3.665e-3$ |
| Llama-3.1 Trend +TFM timemixer | $0.364 \pm 4.571e-3$ | $0.663 \pm 1.511e-3$ | $0.385 \pm 3.674e-3$ | $0.672 \pm 4.213e-3$ |
| Llama-3.1 Trend lstm | $0.366 \pm 2.342e-3$ | $0.663 \pm 4.979e-4$ | $0.385 \pm 7.211e-3$ | $0.669 \pm 3.003e-3$ |
| Llama-3.1 Trend mlp | $0.369 \pm 1.656e-3$ | $0.667 \pm 2.262e-4$ | $0.384 \pm 2.798e-3$ | $0.669 \pm 1.983e-3$ |
| Llama-3.1 Trend patchtsmixer | $0.344 \pm 3.142e-2$ | $0.642 \pm 4.051e-2$ | $0.282 \pm 7.166e-3$ | $0.565 \pm 1.115e-2$ |
| Llama-3.1 Trend timemixer | $0.364 \pm 6.515e-4$ | $0.662 \pm 9.032e-4$ | $0.385 \pm 3.325e-3$ | $0.665 \pm 1.772e-3$ |
| Llama-3.1 zero_shot +TFM lstm | $0.346 \pm 2.309e-3$ | $0.65 \pm 8.195e-4$ | $0.398 \pm 3.998e-3$ | $0.674 \pm 1.957e-3$ |
| Llama-3.1 zero_shot +TFM mlp | $0.348 \pm 4.011e-3$ | $0.653 \pm 8.969e-4$ | $0.403 \pm 2.312e-3$ | $0.688 \pm 2.442e-3$ |
| Llama-3.1 zero_shot +TFM patchtsmixer | $0.333 \pm 1.884e-3$ | $0.612 \pm 5.934e-3$ | $0.381 \pm 2.406e-3$ | $0.652 \pm 2.807e-3$ |
| Llama-3.1 zero_shot +TFM timemixer | $0.361 \pm 1.536e-3$ | $0.665 \pm 1.248e-3$ | $0.397 \pm 1.288e-3$ | $0.683 \pm 5.83e-3$ |
| Llama-3.1 zero_shot lstm | $0.362 \pm 2.002e-3$ | $0.654 \pm 5.816e-4$ | $0.39 \pm 9.319e-4$ | $0.671 \pm 1.87e-3$ |
| Llama-3.1 zero_shot mlp | $0.362 \pm 1.834e-3$ | $0.654 \pm 1.988e-3$ | $0.39 \pm 2.984e-3$ | $0.672 \pm 1.651e-3$ |
| Llama-3.1 zero_shot patchtsmixer | $0.336 \pm 3.817e-2$ | $0.632 \pm 4.132e-2$ | $0.321 \pm 4.617e-2$ | $0.614 \pm 4.224e-2$ |
| Llama-3.1 zero_shot timemixer | $0.361 \pm 2.158e-3$ | $0.657 \pm 1.538e-3$ | $0.389 \pm 2.366e-3$ | $0.673 \pm 2.809e-3$ |
| TFM lstm | $0.408 \pm 1.68e-3$ | $0.686 \pm 1.911e-3$ | $0.393 \pm 3.643e-3$ | $0.683 \pm 1.977e-3$ |
| TFM mlp | $0.409 \pm 4.157e-3$ | $0.678 \pm 3.606e-3$ | $0.409 \pm 4.364e-3$ | $0.694 \pm 2.94e-3$ |
| TFM patchtsmixer | $0.375 \pm 1.74e-3$ | $0.637 \pm 8.51e-3$ | $0.415 \pm 9.513e-3$ | $0.688 \pm 2.668e-3$ |
| TFM timemixer | $0.39 \pm 1.764e-3$ | $0.662 \pm 1.212e-3$ | $0.397 \pm 1.115e-2$ | $0.679 \pm 1.147e-2$ |
| TSDE lstm | $0.311 \pm 9.39e-2$ | $0.521 \pm 1.192e-1$ | $0.231 \pm 2.691e-2$ | $0.399 \pm 6.764e-2$ |
| TSDE mlp | $0.412 \pm 1.143e-1$ | $0.596 \pm 1.218e-1$ | $0.273 \pm 1.937e-2$ | $0.466 \pm 2.423e-2$ |
| TSDE patchtsmixer | $0.265 \pm 6.186e-3$ | $0.49 \pm 8.424e-3$ | $0.274 \pm 2.561e-3$ | $0.503 \pm 2.375e-3$ |
| TSDE timemixer | $0.255 \pm 1.379e-2$ | $0.466 \pm 2.595e-2$ | $0.276 \pm 4.016e-2$ | $0.5 \pm 6.036e-2$ |
| gemini-2.0-flash CoT +TFM lstm | $0.44 \pm 5.224e-3$ | $0.711 \pm 2.302e-3$ | $0.481 \pm 3.247e-3$ | $0.743 \pm 1.26e-3$ |
| gemini-2.0-flash CoT +TFM mlp | $0.443 \pm 8.222e-3$ | $0.714 \pm 6.119e-3$ | $0.478 \pm 8.296e-3$ | $0.742 \pm 3.99e-3$ |
| gemini-2.0-flash CoT +TFM patchtsmixer | $0.404 \pm 2.595e-3$ | $0.672 \pm 3.237e-3$ | $0.436 \pm 2e-3$ | $0.699 \pm 7.007e-3$ |
| gemini-2.0-flash CoT +TFM timemixer | $0.439 \pm 3.451e-3$ | $0.717 \pm 1.775e-3$ | $0.466 \pm 4.676e-3$ | $0.736 \pm 8.95e-4$ |
| gemini-2.0-flash CoT lstm | $0.447 \pm 2.148e-3$ | $0.715 \pm 1.988e-3$ | $0.492 \pm 4.484e-3$ | $0.741 \pm 3.722e-3$ |
| gemini-2.0-flash CoT mlp | $0.453 \pm 1.266e-3$ | $0.716 \pm 1.248e-3$ | $0.489 \pm 3.167e-3$ | $0.739 \pm 1.827e-3$ |
| gemini-2.0-flash CoT patchtsmixer | $0.443 \pm 1.368e-3$ | $0.714 \pm 1.423e-3$ | $0.382 \pm 5.961e-2$ | $0.662 \pm 4.915e-2$ |
| gemini-2.0-flash CoT timemixer | $0.455 \pm 2.492e-3$ | $0.714 \pm 5.74e-4$ | $0.493 \pm 2.823e-3$ | $0.741 \pm 1.451e-3$ |
| gemini-2.0-flash ICD +TFM lstm | $0.372 \pm 1.011e-2$ | $0.662 \pm 8.436e-3$ | $0.484 \pm 3.49e-3$ | $0.739 \pm 1.642e-3$ |
| gemini-2.0-flash ICD +TFM mlp | $0.373 \pm 5.795e-3$ | $0.668 \pm 2.982e-3$ | $0.479 \pm 4.541e-3$ | $0.738 \pm 4.24e-3$ |
| gemini-2.0-flash ICD +TFM patchtsmixer | $0.35 \pm 4.79e-3$ | $0.631 \pm 3.774e-3$ | $0.431 \pm 5.018e-3$ | $0.689 \pm 4.109e-3$ |
| gemini-2.0-flash ICD +TFM timemixer | $0.393 \pm 1.423e-2$ | $0.675 \pm 8.435e-3$ | $0.47 \pm 3.658e-3$ | $0.73 \pm 2.716e-3$ |
| gemini-2.0-flash ICD lstm | $0.385 \pm 1.991e-3$ | $0.671 \pm 2.031e-3$ | $0.478 \pm 2.194e-3$ | $0.735 \pm 1.021e-3$ |
| gemini-2.0-flash ICD mlp | $0.386 \pm 4.339e-3$ | $0.67 \pm 3.934e-3$ | $0.476 \pm 2.665e-3$ | $0.734 \pm 1.832e-3$ |
| gemini-2.0-flash ICD patchtsmixer | $0.359 \pm 7.454e-2$ | $0.633 \pm 8.33e-2$ | $0.394 \pm 7.93e-2$ | $0.671 \pm 6.795e-2$ |
| gemini-2.0-flash ICD timemixer | $0.392 \pm 1.708e-3$ | $0.671 \pm 1.699e-3$ | $0.478 \pm 2.165e-3$ | $0.735 \pm 1.052e-3$ |
| gemini-2.0-flash Trend +TFM lstm | $0.406 \pm 1.599e-3$ | $0.693 \pm 8.598e-4$ | $0.436 \pm 7.652e-3$ | $0.708 \pm 5.305e-3$ |
| gemini-2.0-flash Trend +TFM mlp | $0.402 \pm 2.571e-3$ | $0.69 \pm 1.739e-3$ | $0.423 \pm 4.072e-3$ | $0.696 \pm 2.111e-3$ |
| gemini-2.0-flash Trend +TFM patchtsmixer | $0.376 \pm 1.085e-3$ | $0.659 \pm 2.212e-3$ | $0.402 \pm 2.707e-3$ | $0.674 \pm 1.87e-3$ |
| gemini-2.0-flash Trend +TFM timemixer | $0.403 \pm 2.033e-3$ | $0.692 \pm 7.048e-3$ | $0.422 \pm 7.139e-3$ | $0.698 \pm 5.21e-3$ |
| gemini-2.0-flash Trend lstm | $0.4 \pm 4.035e-3$ | $0.687 \pm 5.04e-3$ | $0.405 \pm 8.401e-3$ | $0.692 \pm 6.798e-3$ |
| gemini-2.0-flash Trend mlp | $0.411 \pm 1.205e-3$ | $0.693 \pm 4.034e-3$ | $0.412 \pm 4.631e-3$ | $0.696 \pm 3.027e-3$ |
| gemini-2.0-flash Trend patchtsmixer | $0.378 \pm 7.402e-2$ | $0.662 \pm 7.818e-2$ | $0.326 \pm 6.116e-2$ | $0.613 \pm 5.775e-2$ |
| gemini-2.0-flash Trend timemixer | $0.409 \pm 2.988e-3$ | $0.694 \pm 2.309e-3$ | $0.398 \pm 7.758e-3$ | $0.689 \pm 5.446e-3$ |
| gemini-2.0-flash zero_shot +TFM lstm | $0.46 \pm 2.592e-3$ | $0.724 \pm 1.031e-3$ | $0.486 \pm 4.692e-3$ | $0.75 \pm 2.913e-3$ |
| gemini-2.0-flash zero_shot +TFM mlp | $0.455 \pm 1.591e-3$ | $0.721 \pm 1.511e-3$ | $0.486 \pm 6.096e-3$ | $0.747 \pm 4.234e-3$ |
| gemini-2.0-flash zero_shot +TFM patchtsmixer | $0.414 \pm 2.556e-3$ | $0.68 \pm 2.316e-3$ | $0.453 \pm 5.93e-3$ | $0.717 \pm 4.403e-3$ |
| gemini-2.0-flash zero_shot +TFM timemixer | $0.446 \pm 2.195e-3$ | $0.716 \pm 2.61e-3$ | $0.469 \pm 3.844e-3$ | $0.742 \pm 6.946e-4$ |
| gemini-2.0-flash zero_shot lstm | $0.462 \pm 3.317e-3$ | $0.725 \pm 2.439e-3$ | $0.492 \pm 7.376e-4$ | $0.75 \pm 6.335e-4$ |
| gemini-2.0-flash zero_shot mlp | $0.467 \pm 1.358e-3$ | $0.726 \pm 1.781e-3$ | $0.491 \pm 3.966e-3$ | $0.745 \pm 2.92e-3$ |
| gemini-2.0-flash zero_shot patchtsmixer | $0.454 \pm 4.39e-3$ | $0.723 \pm 2.019e-3$ | $0.39 \pm 1.129e-1$ | $0.66 \pm 1.138e-1$ |
| gemini-2.0-flash zero_shot timemixer | $0.463 \pm 1.015e-3$ | $0.722 \pm 9.899e-4$ | $0.487 \pm 4.665e-3$ | $0.741 \pm 4.708e-3$ |
| interp lstm | $0.336 \pm 2.386e-2$ | $0.612 \pm 3.01e-2$ | $0.327 \pm 3.616e-2$ | $0.557 \pm 3.184e-2$ |
| interp mlp | $0.308 \pm 8.673e-3$ | $0.573 \pm 2.125e-2$ | $0.233 \pm 5.897e-3$ | $0.445 \pm 1.573e-2$ |
| interp patchtsmixer | $0.273 \pm 1.45e-4$ | $0.5 \pm 9.5e-5$ | $0.273 \pm 0e0$ | $0.5 \pm 0e0$ |
| interp timemixer | $0.299 \pm 2.289e-2$ | $0.536 \pm 4.692e-2$ | $0.272 \pm 1.66e-2$ | $0.496 \pm 2.654e-2$ |
| mean lstm | $0.35 \pm 8.786e-3$ | $0.6 \pm 9.442e-3$ | $0.358 \pm 5.789e-2$ | $0.573 \pm 4.881e-2$ |
| mean mlp | $0.341 \pm 2.137e-2$ | $0.588 \pm 2.935e-2$ | $0.313 \pm 3.221e-2$ | $0.505 \pm 5.15e-2$ |
| mean patchtsmixer | $0.273 \pm 1.7e-4$ | $0.5 \pm 2.05e-4$ | $0.273 \pm 1.45e-4$ | $0.5 \pm 9.5e-5$ |
| mean timemixer | $0.348 \pm 1.908e-2$ | $0.615 \pm 1.567e-2$ | $0.284 \pm 1.763e-2$ | $0.516 \pm 2.608e-2$ |
| No_sum_CoT lstm | $0.522 \pm 7.255e-3$ | $0.757 \pm 3.119e-3$ | $0.477 \pm 3.339e-3$ | $0.735 \pm 3.721e-3$ |
| No_sum_CoT mlp | $0.522 \pm 3.581e-4$ | $0.76 \pm 7.204e-4$ | $0.449 \pm 7.005e-3$ | $0.7 \pm 6.358e-3$ |
| No_sum_CoT patchtsmixer | $0.408 \pm 9.767e-2$ | $0.68 \pm 9.853e-2$ | $0.396 \pm 1.008e-1$ | $0.674 \pm 8.421e-2$ |
| No_sum_CoT timemixer | $0.502 \pm 8.977e-3$ | $0.75 \pm 2.818e-3$ | $0.446 \pm 1.245e-2$ | $0.702 \pm 1.019e-2$ |
| No_sum_ICD lstm | $0.509 \pm 4.305e-3$ | $0.753 \pm 7.202e-4$ | $0.492 \pm 6.297e-3$ | $0.744 \pm 1.324e-3$ |
| No_sum_ICD mlp | $0.504 \pm 7.52e-3$ | $0.755 \pm 2.455e-3$ | $0.512 \pm 1.189e-2$ | $0.751 \pm 6.433e-3$ |
| No_sum_ICD patchtsmixer | $0.407 \pm 7.366e-2$ | $0.7 \pm 7.045e-2$ | $0.359 \pm 8.137e-2$ | $0.655 \pm 7.15e-2$ |
| No_sum_ICD timemixer | $0.482 \pm 5.592e-3$ | $0.747 \pm 1.144e-3$ | $0.483 \pm 1.094e-2$ | $0.739 \pm 5.264e-3$ |
| No_sum_Trend lstm | $0.549 \pm 2.184e-3$ | $0.768 \pm 1.26e-3$ | $0.489 \pm 8.991e-3$ | $0.752 \pm 4.14e-3$ |
| No_sum_Trend mlp | $0.551 \pm 1.711e-3$ | $0.772 \pm 3.869e-4$ | $0.489 \pm 1.771e-2$ | $0.739 \pm 1.1e-2$ |
| No_sum_Trend patchtsmixer | $0.461 \pm 7.405e-2$ | $0.723 \pm 4.69e-2$ | $0.379 \pm 8.544e-2$ | $0.672 \pm 6.206e-2$ |
| No_sum_Trend timemixer | $0.525 \pm 6.96e-3$ | $0.759 \pm 3.27e-3$ | $0.453 \pm 1.791e-2$ | $0.723 \pm 1.081e-2$ |
| medgemma CoT +TFM lstm | $0.461 \pm 2.095e-3$ | $0.721 \pm 6.483e-4$ | $0.495 \pm 7.993e-3$ | $0.737 \pm 4.285e-3$ |
| medgemma CoT +TFM mlp | $0.46 \pm 2.703e-3$ | $0.722 \pm 5.972e-4$ | $0.491 \pm 3.25e-3$ | $0.727 \pm 3.272e-3$ |
| medgemma CoT +TFM patchtsmixer | $0.419 \pm 3.118e-3$ | $0.677 \pm 6.207e-4$ | $0.453 \pm 1.679e-3$ | $0.698 \pm 7.701e-4$ |
| medgemma CoT +TFM timemixer | $0.451 \pm 2.571e-3$ | $0.721 \pm 4.86e-4$ | $0.481 \pm 2.674e-3$ | $0.722 \pm 3.019e-3$ |
| medgemma CoT lstm | $0.482 \pm 1.635e-3$ | $0.731 \pm 1.264e-3$ | $0.501 \pm 6.132e-3$ | $0.737 \pm 4.484e-3$ |
| medgemma CoT mlp | $0.477 \pm 2.855e-3$ | $0.73 \pm 1.213e-3$ | $0.497 \pm 6.07e-3$ | $0.733 \pm 3.784e-3$ |
| medgemma CoT patchtsmixer | $0.416 \pm 8.737e-2$ | $0.683 \pm 7.765e-2$ | $0.391 \pm 6.036e-2$ | $0.658 \pm 5.204e-2$ |
| medgemma CoT timemixer | $0.473 \pm 2.42e-3$ | $0.726 \pm 1.035e-3$ | $0.486 \pm 4.948e-3$ | $0.728 \pm 2.753e-3$ |

Table 44: Cross-site transfer results - Mort. (part 1/2)

| Method | hirid → ppicu | | mimic → ppicu | |
|---|---|---|---|---|
| | auprc | auroc | auprc | auroc |
| medgemma ICD +TFM lstm | $0.459 \pm 2.197e-3$ | $0.715 \pm 1.475e-3$ | $0.489 \pm 4.21e-3$ | $0.741 \pm 2.302e-3$ |
| medgemma ICD +TFM mlp | $0.46 \pm 7.566e-3$ | $0.717 \pm 3.012e-3$ | $0.484 \pm 9.579e-3$ | $0.736 \pm 6.33e-3$ |
| medgemma ICD +TFM patchtsmixer | $0.408 \pm 2.124e-3$ | $0.675 \pm 2.087e-3$ | $0.449 \pm 2.893e-3$ | $0.7 \pm 5.408e-4$ |
| medgemma ICD +TFM timemixer | $0.438 \pm 1.577e-2$ | $0.712 \pm 4.393e-3$ | $0.482 \pm 3.808e-3$ | $0.735 \pm 6.012e-4$ |
| medgemma ICD lstm | $0.455 \pm 7.277e-3$ | $0.715 \pm 1.228e-3$ | $0.49 \pm 4.828e-3$ | $0.741 \pm 2.204e-3$ |
| medgemma ICD mlp | $0.461 \pm 2.622e-3$ | $0.717 \pm 1.017e-3$ | $0.487 \pm 2.734e-3$ | $0.74 \pm 1.132e-3$ |
| medgemma ICD patchtsmixer | $0.416 \pm 5.818e-2$ | $0.692 \pm 4.581e-2$ | $0.403 \pm 7.541e-2$ | $0.685 \pm 5.847e-2$ |
| medgemma ICD timemixer | $0.453 \pm 2.641e-3$ | $0.714 \pm 7.776e-4$ | $0.485 \pm 3.375e-3$ | $0.737 \pm 1.735e-3$ |
| medgemma Trend +TFM lstm | $0.411 \pm 3.69e-3$ | $0.694 \pm 2.313e-3$ | $0.422 \pm 7.35e-3$ | $0.695 \pm 4.394e-3$ |
| medgemma Trend +TFM mlp | $0.407 \pm 8.281e-3$ | $0.691 \pm 7.451e-3$ | $0.423 \pm 4.594e-3$ | $0.697 \pm 4.121e-3$ |
| medgemma Trend +TFM patchtsmixer | $0.368 \pm 2.699e-3$ | $0.641 \pm 5.89e-3$ | $0.382 \pm 2.051e-3$ | $0.653 \pm 3.541e-3$ |
| medgemma Trend +TFM timemixer | $0.401 \pm 1.04e-2$ | $0.696 \pm 3.943e-3$ | $0.409 \pm 7.085e-3$ | $0.686 \pm 9.073e-3$ |
| medgemma Trend lstm | $0.417 \pm 5.283e-3$ | $0.701 \pm 2.298e-3$ | $0.396 \pm 8.327e-3$ | $0.694 \pm 2.259e-3$ |
| medgemma Trend mlp | $0.425 \pm 1.544e-3$ | $0.705 \pm 1.227e-3$ | $0.402 \pm 4.629e-3$ | $0.696 \pm 1.47e-3$ |
| medgemma Trend patchtsmixer | $0.411 \pm 1.628e-3$ | $0.7 \pm 1.69e-3$ | $0.373 \pm 4.873e-2$ | $0.666 \pm 4.055e-2$ |
| medgemma Trend timemixer | $0.419 \pm 8.68e-4$ | $0.701 \pm 6.009e-4$ | $0.389 \pm 2.712e-3$ | $0.693 \pm 1.612e-3$ |
| medgemma zero_shot +TFM lstm | $0.463 \pm 3.495e-3$ | $0.718 \pm 2.008e-3$ | $0.483 \pm 4.484e-3$ | $0.736 \pm 1.781e-3$ |
| medgemma zero_shot +TFM mlp | $0.468 \pm 4.851e-4$ | $0.726 \pm 7.529e-4$ | $0.49 \pm 6.615e-3$ | $0.737 \pm 3.856e-3$ |
| medgemma zero_shot +TFM patchtsmixer | $0.424 \pm 1.662e-2$ | $0.681 \pm 9.491e-3$ | $0.446 \pm 7.676e-3$ | $0.698 \pm 7.654e-3$ |
| medgemma zero_shot +TFM timemixer | $0.446 \pm 3.23e-3$ | $0.72 \pm 1.113e-3$ | $0.476 \pm 1.064e-2$ | $0.728 \pm 7.495e-3$ |
| medgemma zero_shot lstm | $0.455 \pm 2.794e-3$ | $0.717 \pm 2.637e-4$ | $0.496 \pm 6.549e-3$ | $0.74 \pm 3.214e-3$ |
| medgemma zero_shot mlp | $0.462 \pm 2.986e-3$ | $0.715 \pm 2.225e-3$ | $0.492 \pm 2.232e-3$ | $0.735 \pm 1.473e-3$ |
| medgemma zero_shot patchtsmixer | $0.458 \pm 3.019e-3$ | $0.716 \pm 2.53e-3$ | $0.447 \pm 4.454e-2$ | $0.709 \pm 3.68e-2$ |
| medgemma zero_shot timemixer | $0.454 \pm 1.164e-3$ | $0.713 \pm 1.441e-3$ | $0.485 \pm 8.093e-3$ | $0.731 \pm 4.914e-3$ |
| No_sum_zero_shot lstm | $0.54 \pm 1.203e-2$ | $0.764 \pm 6.374e-3$ | $0.477 \pm 3.603e-3$ | $0.734 \pm 4.113e-3$ |
| No_sum_zero_shot mlp | $0.548 \pm 6.611e-3$ | $0.77 \pm 3.377e-3$ | $0.46 \pm 7.96e-3$ | $0.718 \pm 6.132e-3$ |
| No_sum_zero_shot patchtsmixer | $0.434 \pm 6.888e-2$ | $0.709 \pm 5.228e-2$ | $0.44 \pm 7.164e-2$ | $0.719 \pm 5.262e-2$ |
| No_sum_zero_shot timemixer | $0.508 \pm 1.576e-2$ | $0.75 \pm 6.17e-3$ | $0.444 \pm 2.188e-2$ | $0.713 \pm 1.891e-2$ |
| right lstm | $0.377 \pm 3.521e-2$ | $0.632 \pm 3.204e-2$ | $0.369 \pm 4.221e-2$ | $0.605 \pm 4.497e-2$ |
| right mlp | $0.38 \pm 1.781e-2$ | $0.646 \pm 2.192e-2$ | $0.216 \pm 9.502e-3$ | $0.383 \pm 2.84e-2$ |
| right patchtsmixer | $0.273 \pm 1.85e-4$ | $0.5 \pm 1.2e-4$ | $0.273 \pm 0e0$ | $0.5 \pm 0e0$ |
| right timemixer | $0.332 \pm 2.488e-2$ | $0.59 \pm 3.827e-2$ | $0.231 \pm 4.167e-3$ | $0.424 \pm 1.226e-2$ |

Table 45: Cross-site transfer results - Mort. (part 2/2)

