# OpenReview forum: "Can we generate portable representations for clinical time series data using LLMs?"
_ICLR.cc/2026/Conference — ICLR 2026 Poster_

### Official Review · Reviewer_rGkM · 2025-10-26

**Soundness:** 3
**Presentation:** 3
**Contribution:** 3
**Rating:** 8
**Confidence:** 3

**Summary:**

This paper asks a crisp, deployment-relevant question: Can frozen large language models (LLMs) produce portable patient representations from irregular ICU time series that permit downstream predictors trained on one hospital to generalize to another with minimal re-training?

To address this, the authors propose Record2Vec, a novel "summarize-then-embed" pipeline. This approach first maps a 48-hour irregular patient record to a concise natural-language summary by using a frozen LLM. This summary is then converted into a fixed-length vector using a frozen text embedder. The authors evaluate Record2Vec across three distinct cohorts (MIMIC-IV, HiRID, and PPICU) and seven clinical tasks (including forecasting, length of stay (LOS), mortality, and predictions of treatment or lab values). The method is compared against multiple established baselines, including classical grid imputation, a time-series diffusion embedding (TSDE), and a time-series foundation model (TimesFM). Record2Vec is reported to be competitive with these baselines for in-distribution performance. Crucially, the approach demonstrates substantially greater robustness under cross-site transfer. Furthermore, the authors show that it is highly data-efficient in few-shot settings and, importantly, does not increase demographic recoverability relative to the baselines. The full set of empirical claims, the experimental setup, and the detailed results are documented in the submitted draft.

**Strengths:**

Strengths:
- treat language as a harmonizing interface between heterogeneous EHR encodings and generic predictors. Though some other works have described similar trends. Please cite (https://www.nature.com/articles/s41746-025-01777-x). Also cite MEDS which is trying to build transferable data schema in tabular domain (https://openreview.net/forum?id=IsHy2ebjIG)
- The multi-site empirical evaluation and transfer (public and private cohorts) is a major asset so the cross-hospital experiments carry real weight.
- The authors also do a good job exploring multiple LLM variants, prompt styles, and downstream tasks; the finding that summaries reduce token costs by ~25× (page 6) while improving transfer is practically important.
- Methodologically, the pipeline is simple, engineering-light, and compatible with frozen models, which is attractive for real systems where re-training large models is infeasible.
- The inclusion of privacy probes and sanity checks about demographic leakage shows the authors are thinking beyond pure accuracy

**Weaknesses:**

- A number of ablations and diagnostics could strenghten the work that would make the story rigorous rather than suggestive. The first is a controlled ablation separating the effects of (a) canonicalization of names/units, (b) summarization (compression of time series), and (c) the choice of text embedder. Without this, one cannot conclude whether “language” per se is necessary, or whether a structured canonicalizer plus a shallow encoder would suffice.

- the privacy evaluation is narrowly scoped to demographic inference for age/sex. This is insufficient to claim "no additional privacy risk." Embedding inversion, membership inference against the summarizer+embedder pipeline, and probing for rare/highly identifying events (e.g., unusual lab values, timestamps, or free-text tokens) are missing. Additionally, the LLM summarizer itself could hallucinate or introduce artifacts; there is no human evaluation of summary faithfulness to the numeric inputs, nor an automated fidelity metric (e.g., check that numeric extrema referenced in summaries match the underlying series).

- operational costs and latency are discussed qualitatively but lack quantitative benchmarks. Reporting inference latency (ms per window), token counts, embedding dimensionality, and cost per 1k patients would give readers a concrete sense of real-world viability.

- figures can be of higher quality.

**Questions:**

- How much of Record2Vec’s transfer advantage remains if you replace the LLM summarizer with a deterministic template based approach that normalizes variable names, units, and missingness and emits a short structured template? This would clarify whether LLM semantic abstraction is essential or merely convenient and be a direct comparison with previous literature (https://www.nature.com/articles/s41746-025-01777-x)
- Why is TSDE excluded from few-shot adaptation? Could you adapt TimesFM and TSDE with the same small labeled target budgets (or simulate a light adaptation procedure) to make the few-shot comparison symmetric?
- Can you report concrete latency and cost numbers (tokens, GPU time, ms per example) for Gemini-2.0, MedGemma, Llama-3.1 summarizers and for Qwen3 embeddings, so practitioners can weigh the tradeoffs?
- Have you looked at worst-group performance (e.g., by race/ethnicity, rare diagnoses, or admission types)? The aggregate AUROC can mask severe degradation for clinically important subpopulations.

---

> ### Author Response · Authors · 2025-11-21
> **Rebuttal Part 1**
>
> We thank the reviewer for the detailed and valuable feedback and for the positive assessment of our work.
>
> We appreciate the reviewer’s recognition that our paper poses a crisp, deployment-relevant question and that Record2Vec offers a simple, engineering-light pipeline with strong cross-hospital transfer, data-efficient performance, and careful consideration of privacy. We will be sure to cite MEME (Lee et al.) and the MEDS paper in our revision. Below, we respond to each of the reviewer’s points one by one.
>
>
> **Weakness 1**
> > A number of ablations and diagnostics could strengthen the work …
>
> We thank the reviewer for the valuable suggestions regarding additional ablations.
>
> (a) The standardization and preprocessing steps for our three datasets were designed and implemented in close consultation with our clinical collaborators, who are experienced ICU clinicians. We followed their guidance to preserve original site-specific units while harmonizing variable names when merging drug classes (e.g., grouping liver-toxic drugs, vasopressors, etc.). Because of this clinically driven design, we did not prioritize additional ablations specifically on units/names, but we agree this is an important axis and will clarify these choices more explicitly in the revision.
>
> (b) We agree with the reviewer that summarization should be ablated. In the current draft, our summarization ablation varies four prompts and three distinct frozen LLMs, alongside a no-summarization baseline (Sections 4.3 and 4.4). We will highlight this more clearly.
>
> (c)  We also agree that embedder ablation is crucial. Here, line with Reviewer xPBC’s suggestion, we performed additional ablations on the embedding model and observed:
>
>
> 1. Better embedding models generally yield better performance, but the improvements are modest across SOTA models. A non-SOTA model degrades performance, but still remains higher than baseline.
>
> 2. Our method is robust to changes in pooling and normalization methods. But using the suggested pooling + normalization methods by the model documentation can obtain the most consistent performance across benchmarks.
>
>
> On our base embedder Qwen3, we tested the following pooling + normalization pairs:
> mean + L2 (base), mean + none, CLS + L2, last + L2, and max + L2.
>
> In addition, we replaced Qwen3 with the current MTEB leader nvidia/llama-embed-nemotron-8b and the prior SOTA gte-Qwen2-7B-instruct.
>
>
> Here are the results:
> Ablations |Forecast | LoS |Mort | Drug | Lab|
> ---:|---:|---:| ---:| ---:|---:|
> Qwen3-mean-l2 | 0.021 |0.347| 0.90| 0.911| 0.931 |
> Qwen3-mean-none |0.027 | 0.371 | 0.88 | 0.871 | 0.919 |
> Qwen3-cls-l2 |0.024 | 0.530 | 0.89 | 0.90 | 0.940 |
> Qwen3-last-l2 |0.022 | 0.423 | 0.89 | 0.920 | 0.923 |
> gte-Qwen2-instruct |0.028 | 0.480 | 0.88 | 0.886 | 0.918 |
> llama-embed-nemotron|0.021| 0.379| 0.89|0.9097| 0.929|
> Baseline | 0.040 | 0.378 | 0.52 | 0.878 | 0.857 |
>
>
> Hirid -> Ppicu transfer
> Ablations |Forecast | LoS |Mort | Drug | Lab|
> ---:|---:|---:| ---:| ---:|---:|
> Qwen3-mean-l2 | 0.183 | 0.69 | 0.72 |0.97| 0.97 |
> Qwen3-mean-none | 0.209 | 0.66 | 0.71| 0.94 | 0.96
> Qwen3-cls-l2 |0.23 | 0.78 | 0.50 | 0.995 | 0.98
> Qwen3-last-l2 | 0.19 | 0.78| 0.73 | 0.94| 0.96
> gte-Qwen2-instruct|0.25 |0.82 | 0.50| 0.93 | 0.96 |
> llama-embed-nemotron| 0.190| 0.72| 0.713|0.96| 0.998|
> Baseline | 0.306 | 1.09 | 0.50 | 0.42 | 0.77 |
>
>
> We will expand and better document the embedder ablation in the revision.
>
> **Weakness 2**
> > the privacy evaluation is narrowly scoped to demographic inference for age/sex … Additionally, the LLM summarizer itself could hallucinate or introduce artifacts …
>
> We agree with the reviewer’s concerns about the scope of our privacy evaluation. In the revision, we will tone down our claims and explicitly limit them to demographic leakage along age and sex, rather than suggesting the absence of broader privacy risks. We will also clarify that a more comprehensive privacy analysis (e.g., embedding inversion, membership inference, rare-event leakage) is an important direction for future work.
>
> We also agree that human evaluation of the summaries is crucial. We currently include an example case study in Appendix E, and in the revision we will make this more prominent and expand the limitations section to explicitly discuss the risk that the LLM-generated summaries may introduce hallucinations or artifacts.
>
>
> **Weakness 3**
> > Reporting inference latency … would give readers a concrete sense of real-world viability.
>
> We thank the reviewer for this helpful suggestion. We agree that concrete operational metrics are important, and we will report inference latency, token counts, embedding dimensionality, and cost per 1k sample  in our response to Question 3 and incorporate them into the revised manuscript.

---

> ### Author Response · Authors · 2025-11-21
> **Rebuttal Part 2**
>
> **Weakness 4**
> > figures can be of higher quality.
>
> We thank the reviewer for pointing this out and apologize for the oversight. We have also noted similar comments from other reviewers. In the revised manuscript, we will replace Figures 3 and 4 with higher-resolution versions to improve readability.
>
>
> **Question 1**
> > How much of Record2Vec’s transfer advantage remains if you replace the LLM summarizer with a deterministic template?
>
> We thank the reviewer for this insightful question.
>
> A fixed template can align medical codes and units but does not address distributional shifts induced by differing patient populations and site-specific policies, and therefore performs worse in transfer across all tasks.
>
>
> We didn’t fully reproduce MEME, but our experiments include a no-summarization “templated text” baseline in which normalized variable names, units, and feature values are wrapped in natural language (though not in exactly the same fashion as MEME, which maps different EHR components into separate embeddings). This no-summarization method performs comparably in-distribution but suffers substantial degradation under cross-site transfer. In addition, templated text yields far more tokens than summaries, and this results in much higher computation cost and latency.
> No summarization methods, like GenHPF attempts to resolve the issue of unaligned medical codes, units, but did not resolve the distributional shifts induced by population, site-specific protocols.
>
> Here are the results we try to compare when using a deterministic template as another baseline:
>
> | **Method ↓**           | **HiRID→PPICU Forecast (MSE)** | **HiRID→PPICU LoS (MAE)** | **HiRID→PPICU Mort (AUROC)** | **HiRID→PPICU Drug (Recall)** | **HiRID→PPICU Lab (Recall)** |
> |------------------------|-------------------------------:|---------------------------:|------------------------------:|-------------------------------:|------------------------------:|
> | Best baseline          | 0.209                          | 0.73                       | 0.49                          | 0.93                          | 0.95                          |
> | Deterministic Template | 0.195                          | 0.98                       | **0.72**                      | 0.96                          | 0.96                          |
> | Record2Vec             | **0.183**                      | **0.69**                   | **0.72**                      | **0.97**                      | **0.97**                      |
>
> | **Method ↓**           | **MIMIC→PPICU Forecast (MSE)** | **MIMIC→PPICU LoS (MAE)** | **MIMIC→PPICU Mort (AUROC)** | **MIMIC→PPICU Drug (Recall)** | **MIMIC→PPICU Lab (Recall)** |
> |------------------------|--------------------------------:|---------------------------:|------------------------------:|-------------------------------:|------------------------------:|
> | Best baseline          | 0.269                           | 0.71                      | 0.69                          | 0.90                           | 0.90                          |
> | Deterministic Template | 0.263                           | 0.77                      | 0.71                          | 0.95                           | 0.95                          |
> | Record2Vec             | **0.242**                       | **0.49**                  | **0.72**                      | **0.95**                       | **0.95**                      |
>
>
> Table: Transfer Learning Result (Question1). Best per column in **bold**. The above table shows the result trained on HIRID and test on PPICU and the below table shows the result

---

> ### Author Response · Authors · 2025-11-21
> **Rebuttal Part 3**
>
> **Question 2**
> > Why is TSDE excluded …? Could you adapt TimesFM and TSDE … make the few-shot comparison symmetric?
>
> We thank the reviewer for raising this point.
> TSDE is a self-supervised representation learning method. We initially excluded TSDE from the few-shot comparison to avoid conflating self-supervised pretraining on the target distribution with few-shot supervised adaptation. To obtain meaningful TSDE representations, one typically needs access to the full (or at least a large) unlabeled target dataset, not just 16 labeled samples. In the strict 16-sample few-shot setting, we cannot reasonably train a TSDE model on the target site.
> In the current draft, TimesFM results are reported in Figure 5. For better symmetry, we now additionally extract cross-site TSDE embeddings by inferring with a TSDE model trained on the source (in-distribution) data and then fitting a small head with 16 labeled target examples (with source data trained TSDE inferred 16-sample representations). We will update Figure 5 to include these results, which perform substantially worse than the other methods. Please also refer to the following table where we include TSDE and the mean/std results.
>
> **HiRID→PPICU**
>
> | **Method** | **Forecast**   | **LoS**        | **Mortality**   |
> |-------------|---------------:|---------------:|----------------:|
> | mean        | 0.133±0.001    | 2.678±0.229    | 0.600±0.007     |
> | right       | 0.136±0.002    | 1.803±0.041    | 0.621±0.036     |
> | interp      | 0.139±0.001    | 1.999±0.123    | 0.605±0.033     |
> | TFM         | 0.144±0.005    | 0.896±0.042    | 0.686±0.003     |
> | TSDE        | 0.302±0.002    | 0.795±0.036    | 0.507±0.017     |
> | Record2Vec  | **0.022±0.000**| **0.500±0.005**| **0.792±0.002** |
>
>
> **MIMIC→PPICU**
>
> | **Method** | **Forecast**   | **LoS**        | **Mortality**   |
> |-------------|---------------:|---------------:|----------------:|
> | mean        | 0.183±0.003    | 1.171±0.284    | 0.565±0.050     |
> | right       | 0.181±0.003    | 0.662±0.056    | 0.606±0.036     |
> | interp      | 0.183±0.003    | 0.972±0.090    | 0.563±0.028     |
> | TFM         | 0.163±0.001    | 1.067±0.028    | 0.683±0.002     |
> | TSDE        | 0.392±0.016    | 0.643±0.170    | 0.586±0.025     |
> | Record2Vec  | **0.022±0.000**| **0.448±0.003**| **0.750±0.008** |
>
>
>
> **Question 3**
> > Can you report concrete latency and cost numbers … so practitioners can weigh the tradeoffs?
>
> Balancing the results and costs, we find Gemini-2.0-flash results the least latency/costs while maintaining high performance, with a cost of 0.7 dollar per 1000 samples and 0.26s latency per patient.
> In the revision, we will report inference + embedding latency, GPU time per batch, token counts, and cost per patient.
> We report inference + embedding latency, GPU time per batch, token counts and costs per patient. Specifically,  the language models are served on 4 NVIDIA L40S GPUs using vLLM inference; the embedding model is served on 1 L40S for the summarization-based methods, while the no-summarization baseline uses 2 L40S.
> Additionally, we compare the inference time using a deterministic template (no summarization).
> Here’s the detailed table comparing different models.
>
> | Model                 | Latency (per sample)           | GPU time (per batch)             | Token counts                 | Cost (Per 1k samples)                         |
> |-----------------------|--------------------------------|----------------------------------|-----------------------------|------------------------------|
> | MedGemma + Qwen       | $0.784 + 0.026\,\mathrm{s}$    | $3959.41 + 119.61\,\mathrm{ms}$  | $234.03~\text{tks/sample}$  | --                           |
> | Llama + Qwen          | $0.25406 + 0.023\,\mathrm{s}$  | $1299.4 + 117.24\,\mathrm{ms}$   | $254.17~\text{tks/sample}$  | --                           |
> | Gemini + Qwen         | $0.28 + 0.036\,\mathrm{s}$     | --                               | $293.44~\text{tks/sample}$  | \$ 0.7  |
> | Qwen (no summarization) | $2.9675\,\mathrm{s}$         | $23608.14\,\mathrm{ms}$         | $6106.56~\text{tks/sample}$ | --                           |
>
> Table: Detailed latency, GPU time, token counts, and cost for different model configurations.
>
>
>
> **Question 4**
> > Have you looked at the worst-group performance … ?
>
> We appreciate the reviewer’s suggestion to examine the worst-group performance. While we did not initially focus on worst-group analyses, following your comment we performed an additional evaluation using diagnosis codes (not used in training and representation generation) to identify minority diagnostic subgroups of ICU stays. Comparing different representation methods against Record2Vec, we did not observe any fairness concerns.  The performance trends for each minority diagnosis subgroup broadly follow those of the overall population. We will describe this analysis and its limitations in the revised manuscript.

---

> ### Comment · Reviewer_rGkM · 2025-11-25
> **Thank you Authors**
>
> Thanks to the authors for addressing my questions and concerns. I think the authors should take some time to submit a revision in addition to the rebuttals. This way the reviewers feel as if our feedback will be incorperated into the final manuscript. I am already high on this work, but we ask the authors to take all necessary steps to make this discussion period beneficial for both sides.

---

> > ### Author Response · Authors · 2025-11-26
> > **Thank you**
> >
> > We sincerely appreciate your strong support and insightful suggestions. We have submitted the full revision to the manuscript and hope these changes further clarify and strengthen the work.

---

### Official Review · Reviewer_xPpy · 2025-10-27

**Soundness:** 4
**Presentation:** 3
**Contribution:** 2
**Rating:** 4
**Confidence:** 4

**Summary:**

This paper introduces Record2Vec, a novel "summarize-then-embed" pipeline designed to create portable representations of clinical time-series data from the Intensive Care Unit (ICU). The central claim is that by first using a frozen Large Language Model (LLM) to generate a natural language summary of serialized patient data, and then embedding this summary into a fixed-length vector, the resulting representation enables downstream predictive models to generalize across different hospital systems with minimal retraining. The authors evaluate this approach on three distinct ICU cohorts (MIMIC-IV, HiRID, and PPICU) across a range of forecasting and classification tasks. They demonstrate that their method achieves competitive in-distribution performance while significantly outperforming baselines in cross-site transfer and few-shot learning settings, without introducing additional privacy risks related to demographic recoverability.

**Strengths:**

Important and Practical Problem: The paper tackles the problem of model portability in clinical machine learning. The degradation of model performance across different hospital systems is a major barrier to the real-world deployment of AI in healthcare.

Novel "Summarize-then-Embed" Architecture: The main contribution is the two-stage process that uses an LLM for summarization as an intermediate step. The paper provides compelling evidence (Figure 3) that this summarization step is crucial for achieving portability, acting as an intelligent compression and harmonization layer that abstracts away site-specific noise.

Rigorous Evaluation of Portability: The experimental design is strong, with a clear focus on testing cross-site transfer as the primary outcome.

**Weaknesses:**

The primary weakness of the submission is its failure to properly contextualize itself within the rapidly evolving literature on using LLMs for structured EHR data. Several highly relevant, recent works (see below) that explore similar "textualization" and embedding-based approaches are not cited or discussed. This significantly overstates the novelty of the paper's general premise and misses a critical opportunity to highlight its more specific, architectural contribution. This prior works also also demonstrates cross-site generalization with a direct embedding approach, which directly challenges some of the implicit claims in this paper. By not engaging with several recent papers, the authors overstate the novelty of their core idea and fail to properly frame their specific contributions. A substantial revision of the manuscript is necessary to accurately position this work in the current landscape.

The paper's core narrative sometimes feels repetitive, and the most novel and important findings, specifically, the crucial role of the summarization step for portability and efficiency, can get buried. Additionally, several figures in the submission (e.g., Figures 3 and 4) are low-resolution.

Simon A. Lee, Sujay Jain, Alex Chen, Kyoka Ono, Arabdha Biswas, Akos Rudas, Jennifer Fang, and Jeffrey N. Chiang. "Clinical decision support using pseudo-notes from multiple streams of EHR data". npj Digital Medicine, 8(1):394, 2025.

Stefan Hegselmann, Georg von Arnim, Tillmann Rheude, Noel Kronenberg, David Sontag, Gerhard Hindricks, Roland Eils, and Benjamin Wild. "Large Language Models are Powerful EHR Encoders". arXiv preprint arXiv:2502.17403, 2025.

Yanjun Gao, Skatje Myers, Shan Chen, Dmitriy Dligach, Timothy A. Miller, Danielle Bitterman, Matthew Churpek, and Majid Afshar. "When Raw Data Prevails: Are Large Language Model Embeddings Effective in Numerical Data Representation for Medical Machine Learning Applications?". In Findings of the Association for Computational Linguistics: EMNLP 2024, pages 5414–5428, 2024.

Contreras, M., Kapoor, S., Zhang, J., Davidson, A., Ren, Y., Guan, Z., Ozrazgat-Baslanti, T., Nerella, S., Bihorac, A., Rashidi, P.: DeLLiriuM: A large language model for delirium prediction in the ICU using structured EHR. arXiv. arXiv:2410.17363 [cs] (2024).

Stefan Hegselmann, Alejandro Buendia, Hunter Lang, Monica Agrawal, Xiaoyi Jiang, and David Sontag. "TabLLM: Few-shot Classification of Tabular Data with Large Language Models". In Proceedings of The 26th International Conference on Artificial Intelligence and Statistics (AISTATS), PMLR 206:5549-5581, 2023.

**Questions:**

Recent work, such as Hegselmann et al. (2025), has shown that a direct "serialize-then-embed" approach can also achieve strong generalization across different cohorts using off-the-shelf embedding models. Your "no-summary" baseline, however, shows poor transfer performance. Could you please discuss this discrepancy?

---

In general I think this is a well motivated and nicely executed work, but I'm not sure how much novelty is left once other recent works are properly taken into account. Can the authors please clearly state how their work goes beyond the current state of the art?

---

> ### Author Response · Authors · 2025-11-21
> **Rebuttal Part 1**
>
> We thank the reviewer for the detailed and valuable feedback and for highlighting several recent works that use LLMs to process structured EHR data.
>
> After carefully reading the suggested prior work, we continue to believe that our paper offers distinct contributions and novelty compared to these state-of-the-art approaches.
>
> In the following, we first clarify the differences between our paper and each referenced work individually, and then conclude with a summary of how our contributions relate to these works.
>
> For readability, we number the works mentioned by the reviewer in chronological order and discuss the differences between each prior work and our paper accordingly.
>
> > [1] Hegselmann et al. "TabLLM: Few-shot Classification of Tabular Data with Large Language Models"
>
> Our paper differs from TabLLM in data source, tasks, and methodology. TabLLM serializes tabular data into text and then prompts and optionally fine-tunes an LLM (T0) to directly output class labels for tabular classification tasks. It focuses on designing serialization methods (text templates) and leveraging labeled tabular data for zero-shot and especially few-shot supervised classification on a variety of benchmark datasets.
>
> In contrast, our work targets irregular, high-dimensional ICU time-series data and studies how a summarize-then-embed pipeline affects cross-site transfer. Our no-summary baseline is conceptually related to TabLLM’s serialize-then-predict approach, but our method builds task-agnostic representations using frozen LLMs and text encoders (no task-specific fine-tuning of the LLM), and then trains lightweight downstream models. While TabLLM emphasizes few-shot classification, we place a strong emphasis on portability and transfer learning across hospitals and tasks.
>
> > [2] Gao et al. "When Raw Data Prevails: Are Large Language Model Embeddings Effective in Numerical Data Representation for Medical Machine Learning Applications?"
>
>
> Our paper differs from Gao et al.’s work in data preparation, motivation, evaluation, and findings. Gao et al. serialize numerical EHR features from the MIMIC-Extract pipeline (built on MIMIC-III) into text descriptions (e.g., narrative templates) and obtain embeddings from the last hidden layer of generalist and medical LLMs. This is conceptually similar to our in-distribution no-summarization baseline.
>
> Their motivation is to study whether LLM embeddings, obtained from instruction-tuned LLMs, can effectively replace raw numerical EHR features for standard diagnostic tasks.  In addition, our motivation is more deployment-driven: we seek input representations that support portability and transfer across hospitals.
>
> The main difference is that we explicitly assess the transfer learning ability of ICU stay representations across different hospitals. We find that a no-summarization serialize-then-embed pipeline is less suitable for transfer than a summarize-then-embed approach, and it also incurs substantially higher computational cost for the embedding model. In contrast, Gao et al. focus on single-site in-distribution performance and do not study cross-site transfer.
>
> Gao et al. derive embeddings from the last hidden states of text generation models rather than from dedicated text embedding models, and they evaluate primarily on classification tasks. Our work, on the other hand, uses frozen text encoders, covers both classification and regression, and additionally includes multivariate forecasting tasks.
>
> More importantly, Gao et al. find that LLM embeddings generally underperform traditional models trained on raw features, whereas our paper demonstrates updated and competitive performance even for in-distribution tasks, while also improving cross-site transfer.
>
> The preprocessing pipelines also differ substantially. Gao et al. use MIMIC-Extract with 104 time-varying features, while our pipeline starts from 241 raw MIMIC-IV features, curated with practicing ICU clinicians and merged into 60 clinically meaningful features.  This leads to **much longer raw text sequences** per ICU stay in our setting: our raw serialization has roughly three times as many tokens per stay as theirs on average. This token length difference partly explains why summarization before embedding is more critical in our setting, and why it improves both in-distribution and out-of-distribution performance.
>
> Finally, in their discussion, Gao et al. highlight recent work such as LLM2Vec, which trains decoder-only LLMs as text encoders with unsupervised objectives and “merits further investigation.” Our experiments directly explore this direction by using frozen decoder-based LLMs as text encoders within a summarize-then-embed pipeline and, importantly, by evaluating whether these representations transfer across different hospitals.

---

> ### Author Response · Authors · 2025-11-21
> **Rebuttal Part 2**
>
> > [3] Lee et al. "Clinical decision support using pseudo-notes from multiple streams of EHR data". npj Digital Medicine, 8(1):394, 2025.
>
> Lee et al. propose MEME, which is another special case of our no-summarization baseline. MEME converts different categories of records into pseudo-notes (a manual text-serialization method) and uses an LLM multiple times to generate several embeddings, one per EHR domain, followed by a learned aggregation module. Our method is more straightforward: we serialize each ICU stay once and do not separate records into different parts before representation learning. As for the transfer learning or portability assessment, MEME yields the same result as we do, where raw serialization and embedding does not transfer well.
> MEME is trained end-to-end and requires a special aggregation design in the downstream model, whereas our framework keeps the representation model frozen and uses simple downstream predictors. The data focus is also different: we work with irregular, number-intensive ICU time-series, while MEME focuses on emergency department visits where most features appear once with limited timestamp structure. In addition, MEME evaluates cross-site behavior on two ED cohorts, whereas we evaluate transfer across three distinct ICU cohorts, enabling a more systematic study of portability.
>
> > [4] Contreras et al. DeLLiriM: A large language model for delirium prediction in the ICU using structured EHR
>
> The goal of Contreras et al.’s work is substantially different from ours. Contreras et al. propose DeLLiriuM, an end-to-end LLM predictor that is pre-trained on serialized ICU EHR text and then fine-tuned on binary labels to predict delirium risk during the ICU stay. This line of work is best compared with methods that directly use LLMs to predict specific medical events. In contrast, our paper uses LLMs to generate general-purpose input representations that can be reused across multiple downstream models and tasks.
>
> Moreover, DeLLiriuM focuses on a single clinical endpoint (delirium) and, although it includes external validation on additional ICU datasets, it does not study representation portability or transfer across different tasks and hospital systems in the way our work does.

---

> ### Author Response · Authors · 2025-11-21
> **Rebuttal Part 3**
>
> > [5] Stefan Hegselmann, Georg von Arnim, Tillmann Rheude, Noel Kronenberg, David Sontag, Gerhard Hindricks, Roland Eils, and Benjamin Wild. "Large Language Models are Powerful EHR Encoders".
>
> Hegselmann et al.’s work can again be viewed as closely related to our no-summarization baseline, with a strong focus on embedding model ablations: they serialize patient records into structured Markdown text and then compare several general-purpose LLM embedding models against an EHR-specific foundation model (CLMBR-T-Base) and traditional baselines.
>
> Instead of studying summarization or different summarization strategies, Hegselmann et al. focus on how different ways of generating the raw serialization (e.g., which sections and aggregates to include, context length, chunking) affect downstream embeddings and prediction performance.
>
> Another key difference is the type of data. Their experiments use the EHRSHOT benchmark, which contains longitudinal hospital visits with relatively sparse events over long timelines, and their serialization aggregates lab and vitals into a small set of recent measurements plus rich natural language descriptors.
> In contrast, we work with dense, hourly (or even more frequent) ICU time-series, where each stay has many numeric measurements and irregular sampling across sites. This naturally makes our raw serialization more number-heavy and more sensitive to differences in recording practices across ICUs. (See their Figure 2 and our Appendix E case study.)
> In EHRSHOT, the same health system, coding scheme, and serialization format are used across all 15 prediction tasks, which reduces covariate shift and helps their direct serialize-then-embed approach generalize well across tasks.  In the ICU setting, even when the feature set is aligned, recording frequency and clinical practice can differ substantially between cohorts (e.g., minute-level vitals in HiRID versus hourly or even daily charting in MIMIC), which introduces stronger distribution shifts for raw serializations.
> This helps explain the discrepancy in transfer performance: in our ICU cross-site experiments, no-summary serialize-then-embed representations are more sensitive to site-specific patterns, whereas summarize-then-embed reduces this sensitivity and leads to better portability. Our findings therefore highlight that summarization becomes especially important when the underlying records are dense, number-heavy ICU time-series, and that summaries can improve both in-distribution and cross-hospital generalization.
> Finally, we note that Hegselmann et al. is currently available as a preprint and their most updated version was released around the same time as our work.  Despite this similar timeline, our paper addresses a different question: how to construct portable input representations for ICU irregular time-series and when summarize-then-embed pipelines improve portability across hospitals, which is not the focus of their study.

---

> ### Author Response · Authors · 2025-11-21
> **Rebuttal Part 4**
>
> All of the above works mentioned by the reviewer indeed convert EHR data into text and then use LLMs for generation, prediction, or embedding. However, they either do not systematically explore the transfer learning ability of the learned representations [1, 2], or are designed as model- and task-specific predictive pipelines [4], or do not directly address the challenges posed by irregular, number-intensive ICU time-series data [3, 5]. With our extensive experiments across three distinct ICU cohorts from different hospitals, our paper still stands out and provides its own contributions.
>
> Compared with these prior works, our paper shows that summarization is especially useful when raw serialization produces very long token sequences. We demonstrate that direct raw serialization (no summary) is not suitable for cross-site transfer when data density and missingness patterns differ between hospitals. We further show that, given current LLM generation and embedding capabilities, one can already obtain high-quality, portable representations in a purely zero-shot fashion.
> Moreover, most of the above work considers patient EHR snapshots or short duration [1, 3, 4], which are less similar to the dense, irregular ICU time-series studied in our paper. In such ICU settings, direct serialization leads to extremely long sequences due to many numeric values and timestamps. As the reviewer notes, our work is explicitly deployment-driven: we aim to improve representation learning for clinical time-series prediction under realistic cross-site shifts. Beyond the data and time-series differences, we highlight our empirical findings on (i) the necessity and effectiveness of summarization, (ii) the impact of different summarization options, and (iii) the portability of Record2Vec representations under our carefully designed transfer experiments.

---

> > ### Comment · Reviewer_xPpy · 2025-11-21
> >
> > Thank you for the comprehensive rebuttal. Your explanations clarify important distinctions. However, I maintain that a major revision of the manuscript is necessary to properly position this work. Are you planning to update your submission during the discussion phase to address this contextualization concern?

---

> > > ### Author Response · Authors · 2025-11-21
> > >
> > > Thank you for the quick follow-up and for engaging carefully with our rebuttal. We are glad that you found it helpful in clarifying important distinctions.
> > >
> > > Yes, we plan to update the manuscript during the discussion phase to better contextualize our contributions, including adding and integrating the missing related work and citations, and once we upload the revised PDF, we will also add a general comment to inform all reviewers where we have made changes.

---

> ### Author Response · Authors · 2025-11-26
>
> Thank you again for your constructive feedback and for engaging with our rebuttal. We have uploaded a revised manuscript where we performed a major revision to properly contextualize our contributions within the literature you highlighted. Specifically, we rewrote the Related Work section to clearly distinguish our work from the provided references by emphasizing the difference between optimizing model pipelines versus creating transferable input representations. We also formally introduced the additional baseline in Section 3, reported its full results in Section 4, and included a direct comparison between the no-summary method and our proposed pipeline to empirically demonstrate the impact of summarization on dense ICU data. We would appreciate it if you could take a look to see if these revisions align with your thoughts and properly address your concerns regarding the positioning of the work. We remain fully open to further feedback and are happy to make additional edits to improve the manuscript.

---

### Official Review · Reviewer_xPBC · 2025-10-29

**Soundness:** 3
**Presentation:** 3
**Contribution:** 3
**Rating:** 8
**Confidence:** 3

**Summary:**

This paper proposes Record2Vec a pipeline that first asks an LLM to write a brief handoff of an ICU patient’s irregular time‑series record and then feeds that text to a text embedder to obtain a fixed‑length vector for standard predictors. Across three ICU cohorts and seven tasks, the authors report competitive in‑distribution results, improved cross‑site transfer, better few‑shot adaptation, and no increase in demographic leakage compared to strong baselines.

**Strengths:**

- Clear, deployment‑motivated idea summarize irregular numeric streams into natural language, then embed once to standardize inputs across hospitals, figure 2 nicely explains the study.
- Extensive benchmarking across three datasets and seven various outcomes, with record2vec winning most in‑distribution tasks and most cross‑site transfer columns.
- Systematic ablations on summarizer choice and prompt design, good findings: summarization generally boosts portability and structured prompts modestly reduce variance.
- Few‑shot study shows that with only 16 labeled target examples, finetuning on record2vec closes much of the cross‑site gap for mortality.
- Privacy and representation diagnostics are included: demographic recoverability stays comparable or lower than baselines, and UMAP/silhouette analyses provide an embedding‑quality proxy.

**Weaknesses:**

-The paper does not quantify information loss end‑to‑end or analyze the embedding geometry needed for downstream tasks, proxies like silhouette and task‑relative gains are informative but do not measure retained mutual information or per‑feature fidelity, and both the summarizer and embedder remain frozen rather than being optimized for downstream geometry. This makes it unclear whether key clinical signals are systematically discarded or distorted during the summarize-embed pipeline.

- Absolute performance on mortality is mixed: in‑distribution AUROC reaches about 0.90 on HiRID but is lower on MIMIC and PPICU, and cross‑site mortality AUROC is ~0.72, which may be below what many deployments would select.

- The embedder is used off‑the‑shelf and frozen, the paper does not test task‑aware metric learning, embedder finetuning, or geometry‑regularized objectives that could improve linear probe performance and calibration.

**Questions:**

- Please position results against task-specific SOTA (e.g., strong ICU mortality baselines) so readers can judge absolute utility, not only portability. You do include mortality among your tasks; a direct SOTA comparison table would help.


- How sensitive are outcomes to the choice and dimensionality of the frozen text embedder? You mainly note Qwen3, an embedder ablation (dim, normalization, pooling) would clarify it.


- The normalization choice differs across baselines (grids normalized with train-split stats, language keeps raw magnitudes/units). Can you justify fairness here or add a control where text summaries are unit-standardized too?


- Figures 3–4 rely on “rank distributions.” Please report the exact metric used to compute ranks in the captions and include a small table of corresponding absolute numbers per method/prompt.


- Few-shot: what was the selection protocol for the 16 examples (random vs stratified), and how variable are results across seeds? A CI/SD per setting would help.


Minor:
- In line 21, “surprisingly competitive with in-distribution with grid imputation …” consider “competitive in-distribution with grid imputation …”.


- In line 1088 the sentence is slightly off: “All clinical features are selected in discussion with real clinician have prominent experience in ICU units.


- In Figure 6 caption, “Age results have less than 0.5% gap and can be inferred from appendix 4”, should be table 4?


- Figure captions especially figures 4 and 5 could explicitly state the evaluation metric used for ranking/few-shot plots, right now the captions don’t name a metric.

---

> ### Author Response · Authors · 2025-11-21
> **Rebuttal Part 1**
>
> We thank the reviewer for the detailed and valuable feedback and for the positive overall assessment of our work.
>
> We appreciate that the reviewer finds our motivation clear and deployment-oriented, and that they commend our experimental design, including the extensive benchmarking across three ICU cohorts and seven tasks, the systematic ablations, the few-shot adaptation study, and the privacy and representation diagnostics.
>
> In the following, we respond to each weakness and question point by point. In our revised manuscript, we will also incorporate all suggested minor edits from the reviewer.
> We are very grateful for all the careful suggestions.
>
>
> **Weakness 1**
> > The paper does not quantify information loss end‑to‑end …
>
> We agree with the reviewer that we currently do not directly quantify information loss end-to-end. In this version, we rely on downstream task performance as a proxy, which suggests that any information loss is not critical, since Record2Vec representations still perform strongly for in-distribution tasks and competitively for cross-site transfer.
> We also agree that assessing per-feature fidelity and mutual information would be very valuable for more formally characterizing information loss in the Record2Vec pipeline. We will explicitly highlight the lack of such quantified information loss assessment as a limitation in the revised manuscript and briefly discuss this direction as important future work.
>
> **Weakness 2 & Question 1**
> > mortality performance; task-specific SOTA methods
>
> We thank the reviewer for suggesting a comparison to task-specific SOTA mortality models and we also think that we should use strong mortality baseline models to replace the results from running a simple model.
>
> We have tested our embeddings and baselines using the strong ICU mortality model reported in the HiRID-ICU-Benchmark (Hugo et al., 2021). From these comparisons, we draw two main conclusions:
>
> 1. Our embeddings are comparable to strong baselines for in-distribution mortality prediction under these stronger models, demonstrating substantial practical utility.
> 2. For transportability, our embeddings substantially outperform the non-language baselines, but remain slightly below our original architecture such as PatchTSMixer (for reference, AUROC around 0.72). This suggests that while Record2Vec provides highly portable representations, there is still room to design downstream models that more fully exploit the structure of these summary-then-embed generated representations.
>
>
>
> |              | LSTM            | GRU             | TCN              | TRANS            |
> |--------------|-----------------|-----------------|------------------|------------------|
> | Baselines    | 0.903±0.002     | 0.900±0.004     | 0.897±0.004      | 0.908±0.002      |
> | TSDE         | **0.915±0.006** | 0.891±0.005     | 0.8925±0.004     | **0.920±0.003**  |
> | TFM          | 0.850±0.001     | 0.820±0.003     | 0.850±0.002      | 0.814±0.001      |
> | Record2Vec   | 0.905±0.003     | **0.9035±0.004**| **0.9033±0.001** | 0.90315±0.002    |
>
>
>
> | Model        | H→P Baseline | H→P Record2Vec | M→P  Baseline | M→P Record2Vec |
> |-------------|--------------|----------------|-----------------|-------------------|
> | LSTM        | 0.640        | 0.69788        | 0.5235          | 0.722             |
> | GRU         | 0.687        | 0.6838         | 0.5476          | 0.685             |
> | TRANSFORMER | 0.57858      | 0.679          | 0.564           | 0.682             |
> | TCN         | 0.719        | 0.684          | 0.719           | 0.695             |
>
> Table: Strong Mortality model baseline results comparison. The top table shows the in-distribution and the bottom shows the cross-site results.
>
> We will add these results to the revised manuscript to more clearly position our pipeline relative to task-specific SOTA mortality methods.

---

> ### Author Response · Authors · 2025-11-21
> **Rebuttal Part 2**
>
> **Weakness 3 & Question 2**
> > embedder ablation
>
> We thank the reviewer for mentioning the embedding model ablation experiment and we agree this is crucial to strengthen our paper.
>
> We performed additional ablations on the embedding model and observed:
> 1. Better embedding models generally yield better performance, but the improvements are modest across SOTA models. A non-SOTA model degrades performance, but still remains higher than baseline.
> 2. Our method is mostly robust to changes in pooling and normalization methods. But using the suggested pooling + normalization methods by the model documentation can obtain the most consistent performance across benchmarks.
>
> On our base embedder Qwen3, we tested the following pooling + normalization pairs:
> mean + L2 (base), mean + none, CLS + L2, last + L2, and max + L2.
>
> In addition, we replaced Qwen3 with the current MTEB leader nvidia/llama-embed-nemotron-8b and the prior SOTA gte-Qwen2-7B-instruct.
>
>
> Here are the results:
>
> Ablations |Forecast | LoS |Mort | Drug | Lab|
> ---:|---:|---:| ---:| ---:|---:|
> Qwen3-mean-l2 | 0.021 |0.347| 0.90| 0.911| 0.931 |
> Qwen3-mean-none |0.027 | 0.371 | 0.88 | 0.871 | 0.919 |
> Qwen3-cls-l2 |0.024 | 0.530 | 0.89 | 0.90 | 0.940 |
> Qwen3-last-l2 |0.022 | 0.423 | 0.89 | 0.920 | 0.923 |
> gte-Qwen2-instruct |0.028 | 0.480 | 0.88 | 0.886 | 0.918 |
> llama-embed-nemotron|0.021| 0.379| 0.89|0.9097| 0.929|
> Baseline | 0.040 | 0.378 | 0.52 | 0.878 | 0.857 |
>
>
>
>
>
>
>
>
> Hirid -> Ppicu transfer
> Ablations |Forecast | LoS |Mort | Drug | Lab|
> ---:|---:|---:| ---:| ---:|---:|
> Qwen3-mean-l2 | 0.183 | 0.69 | 0.72 |0.97| 0.97 |
> Qwen3-mean-none | 0.209 | 0.66 | 0.71| 0.94 | 0.96
> Qwen3-cls-l2 |0.23 | 0.78 | 0.50 | 0.995 | 0.98
> Qwen3-last-l2 | 0.19 | 0.78| 0.73 | 0.94| 0.96
> gte-Qwen2-instruct|0.25 |0.82 | 0.50| 0.93 | 0.96 |
> llama-embed-nemotron| 0.190| 0.72| 0.713|0.96| 0.998|
> Baseline | 0.306 | 1.09 | 0.50 | 0.42 | 0.77 |
>
>
> We will include the full results of these ablations in the revised manuscript.

---

> ### Author Response · Authors · 2025-11-21
> **Rebuttal Part 3**
>
> **Question 3**
> > The normalization choice differs across baselines … Can you justify fairness here or add a control where text summaries are unit-standardized too?
>
> We thank the reviewer for raising this point about preprocessing differences between normalized grids and raw numerical values in the medical records. This implementation choice was made to minimize performance variance for the non-language baselines.
>
> For the text-serialization, we intentionally keep raw units and magnitudes because our goal with Record2Vec is to require minimal manual preprocessing of ICU stay information. This keeps the pipeline simple and avoids injecting additional human priors into the data distribution.
>
> By contrast, traditional ML methods operating on numeric matrices typically perform poorly without normalization and imputation. Prior benchmark work on ICU datasets (e.g., MIMIC-III/IV, HiRID, YAIB) has repeatedly shown that normalization is crucial for these models.
>
> Regarding fairness, using raw, unnormalized values for the language pipeline likely makes the comparison more conservative for Record2Vec, since baselines benefit from normalization while our summaries preserve the original data distribution. We therefore believe this design is reasonable. We will clarify this rationale and its implications in the revised manuscript.
>
>
> **Question 4**
> > Figures 3–4 rely on “rank distributions.” Please report the exact metric used to compute ranks in the captions and include a small table of corresponding absolute numbers per method/prompt.
>
> We thank the reviewer for this helpful clarification request.
>
> The metrics used to compute the ranks in Figures 3 and 4 are exactly those reported in Tables 1 and 2: mean squared error (MSE) for forecasting, mean absolute error (MAE) for length-of-stay, AUROC for mortality, and recall for both treatment planning (Drug) and measurement ordering (Lab). We will revise the captions for Figures 3 and 4 to explicitly state these metrics.
>
> For Figure 3, ranks are computed across the four summary methods {no-summary, llama, medgemma, gemini-2.0 flash}. For Figure 4, the ranks are computed across the four prompting variants {zero-shot, ICD, CoT, Trend}.
>
>
> For in-distribution results (left subfigure), ranks are aggregated over 15 tasks (5 tasks per dataset and with 3 datasets). For transfer learning, there are 30 tasks in total (5 tasks per dataset for 6 transfer directions). During our verification, we noticed that the right subfigure of Figure 4 had been accumulated over 15 transfer tasks (HiRID->PPICU, MIMIC->PPICU, HiRID->MIMIC); we will correct this and clarify the exact set of transfer directions in the revised figure and caption.
>
> Our intent with Figures 3 and 4 is to provide an intuitive visualization of how different summarization and prompting strategies affect relative performance, without overwhelming the reader with many numbers in the main text.
>
>
> In the revision, we will (i) clarify the ranking metric in the captions and (ii) add small tables with the corresponding absolute performance numbers per method/prompt in the appendix so that readers can directly inspect the underlying values used to construct these rank distributions.

---

> ### Author Response · Authors · 2025-11-21
> **Rebuttal Part 4**
>
> Here are the original numbers (4 tables we used) to generate Figure 3 and Figure 4.
>
>
>
> | Model | Forecast | LOS | Mort | Drug | Lab | Forecast | LOS | Mort | Drug | Lab | Forecast | LOS | Mort | Drug | Lab |
> | :--- | :--- | :--- | :--- | :--- | :--- | :--- | :--- | :--- | :--- | :--- | :--- | :--- | :--- | :--- | :--- |
> | **no-summary** | 0.021 | 0.3538 | 0.9 | 0.899 | 0.931 | 0.027 | 0.328 | 0.82 | 0.886 | 0.947 | 0.0217 | 0.376 | 0.64 | 0.9308 | 0.923 |
> | **llama 3.1** | 0.0214 | 0.381 | 0.8797 | 0.899 | 0.925 | 0.0288 | 0.406 | 0.8131 | 0.886 | 0.938 | 0.029 | 0.3752 | 0.63 | 0.937 | 0.9271 |
> | **medgemma** | 0.0237 | 0.3682 | 0.83 | 0.911 | 0.925 | 0.0299 | 0.4005 | 0.7788 | 0.8946 | 0.9432 | 0.029 | 0.3626 | 0.6346 | 0.925 | 0.936 |
> | **gemini** | 0.028 | 0.347 | 0.8302 | 0.9064 | 0.9307 | 0.03 | 0.3328 | 0.77 | 0.903 | 0.942 | 0.017 | 0.358 | 0.6377 | 0.9308 | 0.9328 |
>
>
> Four summarization methods performance comparison across 15 in-distribution tasks (Figure 3 left subfigure)
>
>
>
> | Model | Forecast | LOS | Mort | Drug | Lab | Forecast | LOS | Mort | Drug | Lab | Forecast | LOS | Mort | Drug | Lab | Forecast | LOS | Mort | Drug | Lab | Forecast | LOS | Mort | Drug | Lab | Forecast | LOS | Mort | Drug | Lab |
> | :--- | :--- | :--- | :--- | :--- | :--- | :--- | :--- | :--- | :--- | :--- | :--- | :--- | :--- | :--- | :--- | :--- | :--- | :--- | :--- | :--- | :--- | :--- | :--- | :--- | :--- | :--- | :--- | :--- | :--- | :--- |
> | **no-summary** | 0.21 | 0.98 | 0.66 | 0.92 | 0.96 | 0.263 | 0.77 | 0.7 | 0.92 | 0.92 | 0.1284 | 0.693 | 0.73 | 0.881 | 0.832 | 0.15971 | 0.518 | 0.75 | 0.913 | 0.8669 | 0.135 | 0.5805 | 0.8277 | 0.816 | 0.8857 | 0.0954 | 0.494 | 0.76 | 0.892 | 0.7895 |
> | **llama 3.1** | 0.1908 | 0.71 | 0.735 | 0.9243 | 0.9718 | 0.249 | 0.5522 | 0.7097 | 0.9415 | 0.9285 | 0.134 | 0.6464 | 0.82 | 0.878 | 0.851 | 0.141 | 0.443 | 0.752 | 0.877 | 0.8669 | 0.1741 | 0.572 | 0.83 | 0.903 | 0.9085 | 0.096 | 0.4393 | 0.7657 | 0.835 | 0.734 |
> | **medgemma** | 0.21 | 0.6981 | 0.74 | 0.97 | 0.98 | 0.239 | 0.5627 | 0.7145 | 0.9428 | 0.95 | 0.0922 | 0.537 | 0.81 | 0.891 | 0.83 | 0.1281 | 0.4493 | 0.81 | 0.88 | 0.9054 | 0.135 | 0.5805 | 0.8277 | 0.8653 | 0.877 | 0.089 | 0.4682 | 0.7725 | 0.8637 | 0.8042 |
> | **gemini** | 0.183 | 0.7017 | 0.7001 | 0.92 | 0.9737 | 0.258 | 0.49 | 0.73 | 0.95 | 0.9372 | 0.089 | 0.6661 | 0.8044 | 0.852 | 0.842 | 0.127 | 0.4886 | 0.7942 | 0.901 | 0.919 | 0.167 | 0.5864 | 0.8037 | 0.8239 | 0.916 | 0.083 | 0.429 | 0.81 | 0.8627 | 0.812 |
>
>
> For summarization methods performance comparison across 30 transfer learning tasks (Figure 3 left subfigure)
>
>
>
> | Model | Forecast | LOS | Mort | Drug | Lab | Forecast | LOS | Mort | Drug | Lab | Forecast | LOS | Mort | Drug | Lab |
> | :--- | :--- | :--- | :--- | :--- | :--- | :--- | :--- | :--- | :--- | :--- | :--- | :--- | :--- | :--- | :--- |
> | **zero-shot** | 0.0244 | 0.354 | 0.8501 | 0.911 | 0.931 | 0.0282 | 0.3951 | 0.77 | 0.8894 | 0.9431 | 0.017 | 0.376 | 0.6367 | 0.9346 | 0.923 |
> | **CoT** | 0.0268 | 0.381 | 0.8311 | 0.9078 | 0.925 | 0.0294 | 0.328 | 0.8101 | 0.886 | 0.938 | 0.029 | 0.358 | 0.6387 | 0.937 | 0.936 |
> | **ICD** | 0.021 | 0.3517 | 0.83 | 0.899 | 0.9305 | 0.027 | 0.3844 | 0.8018 | 0.903 | 0.947 | 0.0284 | 0.3645 | 0.64 | 0.925 | 0.9234 |
> | **Trend** | 0.028 | 0.347 | 0.9 | 0.9042 | 0.9263 | 0.03 | 0.406 | 0.82 | 0.8881 | 0.9469 | 0.0249 | 0.3713 | 0.63 | 0.9324 | 0.9308 |
>
> For prompting methods performance comparison across 15 in-distribution tasks (Figure 4 left subfigure)
>
> | Model | Forecast | LOS | Mort | Drug | Lab | Forecast | LOS | Mort | Drug | Lab | Forecast | LOS | Mort | Drug | Lab |
> | :--- | :--- | :--- | :--- | :--- | :--- | :--- | :--- | :--- | :--- | :--- | :--- | :--- | :--- | :--- | :--- |
> | **zero-shot** | 0.183 | 0.6994 | 0.6888 | 0.9359 | 0.9758 | 0.2536 | 0.57 | 0.7045 | 0.949 | 0.95 | 0.1082 | 0.6467 | 0.73 | 0.8878 | 0.83 |
> | **CoT** | 0.1996 | 0.69 | 0.66 | 0.92 | 0.9667 | 0.2461 | 0.49 | 0.7106 | 0.92 | 0.9375 | 0.089 | 0.693 | 0.7328 | 0.868 | 0.832 |
> | **ICD** | 0.21 | 0.6989 | 0.6916 | 0.933 | 0.98 | 0.239 | 0.5199 | 0.73 | 0.926 | 0.9498 | 0.134 | 0.6564 | 0.82 | 0.891 | 0.851 |
> | **Trend** | 0.1927 | 0.71 | 0.74 | 0.97 | 0.96 | 0.258 | 0.5595 | 0.7 | 0.95 | 0.92 | 0.1248 | 0.537 | 0.7629 | 0.852 | 0.847 |
>
> For prompting methods performance comparison across 15 transfer learning tasks (Figure 4 right subfigure)

---

> ### Author Response · Authors · 2025-11-21
> **Rebuttal Part 5**
>
> **Question 5**
> > Few-shot: what was the selection protocol for the 16 examples (random vs stratified), and how variable are results across seeds?
>
> We thank the reviewer for asking for more detail on the few-shot fine-tuning setup.
>
> The 16 samples are selected uniformly at random (not stratified) from the target dataset. We chose this protocol to better reflect a realistic deployment scenario, where we do not control which specific labeled cases a new hospital may have. The same 16 examples are shared across all methods to ensure a fair comparison.
>
> We now provide the mean and standard deviation across multiple random seeds we had in the experiment and Record2Vec remains the most stable method across tasks under this evaluation. We will add these details, along with the corresponding mean/SD results, to the revised manuscript.
>
> **HiRID→PPICU**
>
> | **Method** | **Forecast**   | **LoS**        | **Mortality**   |
> |-------------|---------------:|---------------:|----------------:|
> | mean        | 0.133±0.001    | 2.678±0.229    | 0.600±0.007     |
> | right       | 0.136±0.002    | 1.803±0.041    | 0.621±0.036     |
> | interp      | 0.139±0.001    | 1.999±0.123    | 0.605±0.033     |
> | TFM         | 0.144±0.005    | 0.896±0.042    | 0.686±0.003     |
> | TSDE        | 0.302±0.002    | 0.795±0.036    | 0.507±0.017     |
> | Record2Vec  | **0.022±0.000**| **0.500±0.005**| **0.792±0.002** |
>
>
> **MIMIC→PPICU**
>
> | **Method** | **Forecast**   | **LoS**        | **Mortality**   |
> |-------------|---------------:|---------------:|----------------:|
> | mean        | 0.183±0.003    | 1.171±0.284    | 0.565±0.050     |
> | right       | 0.181±0.003    | 0.662±0.056    | 0.606±0.036     |
> | interp      | 0.183±0.003    | 0.972±0.090    | 0.563±0.028     |
> | TFM         | 0.163±0.001    | 1.067±0.028    | 0.683±0.002     |
> | TSDE        | 0.392±0.016    | 0.643±0.170    | 0.586±0.025     |
> | Record2Vec  | **0.0215±0.000**| **0.448±0.003**| **0.750±0.008** |

---

### Official Review · Reviewer_t4Cv · 2025-11-01

**Soundness:** 3
**Presentation:** 2
**Contribution:** 3
**Rating:** 4
**Confidence:** 5

**Summary:**

This paper proposes a method for transfer learning clinical machine learning models to different hospitals without a harmonization process, addressing the problem of data format differences that arise during deployment. The authors designed the 'Record2Vec', which utilizes LLMs to generate 'portable' patient representations. This methodology transforms irregular ICU time-series data into a natural language summary (a clinical handoff-style summary) using LLM, and then extracts text embeddings from this summary to be used for training predictive models.

The authors claim that this representation vector overcomes the heterogeneity of data formats between hospitals, allowing a downstream predictive model trained at one hospital to operate at another with minimal-to-no retraining. Experiments conducted on multiple prediction tasks across three ICU cohorts (MIMIC-IV, HIRID, PPICU) showed that the proposed method achieves in-distribution performance comparable to existing methods, while exhibiting less performance degradation in cross-site transfer and greater efficiency in few-shot learning.

**Strengths:**

(1) Well motivaed and novel approach: It is timely and important that the paper directly addresses the practical bottleneck of inter-hospital portability in clinical ML model deployment. Unlike traditional model-centric domain adaptation or schema standardization (OMOP/FHIR), the approach of using an LLM as an 'information transformation layer' to generate semantically aligned input representations is novel and interesting.

(2) Comprehensive Experimental Design: The authors attempted to validate the core claim (portability) of the proposed methodology in various scenarios, including In-distribution, cross-site transfer, and few-shot learning, using three different ICU cohorts.

(3) Practicality and Privacy Considerations: The 'Record2Vec' pipeline is conceptually simple. Furthermore, the evaluation of privacy risks—specifically, the recoverability of sensitive information (age, sex) from the embeddings—and the finding that it does not introduce additional risk is a commendable aspect for a study utilizing clinical data.

**Weaknesses:**

1. Inadequate Baseline Comparison: To demonstrate the superiority of the proposed representation, the paper compared the performance of various representation methods by feeding them into the same downstream classifier (PatchTSMixer). However, this approach omits comparisons with key baselines in the field of EHR prediction.
(a) Traditional EHR prediction models that directly utilize medical code sequences, such as BEHRT (Li et al., 2020) or MedBERT (Ramsy et al., 2021).
(b) Models like GenHPF (Hur et al., 2023), which are trained end-to-end by converting EHR time-series inputs into text.

A direct performance comparison with these end-to-end models is necessary. Merely comparing input representations while keeping the PatchTSMixer classifier head fixed is insufficient to demonstrate the practical performance advantages of the proposed method.

2. Practicality of Transfer Learning: In real clinical settings, hospitals will likely prioritize achieving the highest accuracy by training models optimized for their own in-distribution datasets, even if it requires significant cost and time. For the portability offered by the proposed transfer learning approach to be meaningful, the performance of the model transferred from a source to a target (transfer learning performance) must be superior to, or at least comparable to, the performance of a model trained directly on the target hospital's data (in-distribution performance). The current results do not provide a clear incentive for hospitals to choose portability at the expense of accuracy.

3. Overstatement on Resolving Distribution Shift: While it is true that LLM summarization standardizes heterogeneous data formats into a common text input, the claim that this resolves the 'core' issue of cross-site distribution shift is an overstatement. The primary cause of distribution shift is not merely that the same patient data is recorded in different formats, but that the patient populations themselves are fundamentally different between hospitals (e.g., differences in severity, prevalence rates). Generating summaries can alleviate input format differences to some extent, but it seems that author overclaimed it.

4. Unclear Role of LLM Summarization and Limitations of Experimental Setup: The purpose of LLM summarization in this study is ambiguous. While reducing the input length is a general advantage of converting time-series data to text, this experiment was conducted using 'pre-selected' common features (60-75) across datasets. This presupposes that the key shared variables between datasets are already aligned. It is questionable whether performing this pre-alignment step before summarization is an appropriate experimental setup to validate the purpose of LLM summarization (without aligning process EHRs).

If the role of the LLM is 'generating a standardized format,' a comparison should have been made with recent studies that use LLMs to convert EHRs to a standard data model (CDM) like OMOP and then apply models such as BEHRT (e.g., Adams et al., 2025).

**Questions:**

1. Lack of Explanation for Performance Improvement: There is an insufficient explanation of 'how' standardizing the input format into a unified summary leads to performance benefits in prediction. An analysis is needed on how clinically important information is better preserved or extracted during the summarization process—beyond simply matching formats—and how this positively impacts downstream tasks.

2. Lack of Justification for Classifier Choice: The paper does not provide a reason or justification for choosing PatchTSMixer as the downstream predictive model for comparing the different representations. An explanation is needed as to whether this model is suitable for the task, or if the same conclusions would hold if a different classifier were used.

---

> ### Author Response · Authors · 2025-11-21
> **Rebuttal Part 1**
>
> We thank the reviewer for their careful reading of our paper and constructive feedback. We are grateful for the positive assessment of our work as a “well motivated and novel approach,” the recognition of our “comprehensive experimental design,” and the acknowledgement of its “practicality and privacy considerations.”
>
> In the following, we address each of the raised concerns and questions point by point to further clarify our methodology, experimental design, and claims.
>
>
>
> **Weakness 1**
> > Inadequate Baseline Comparison
>
> We thank the reviewer for pointing out these related works. However, we would like to respectfully clarify and partially disagree with the claim of “inadequate baselines,” given the specific deployment-driven research problem we are addressing.
>
> Our goal is to identify more portable input representations for irregular ICU time-series. To this end, we compare against strong prior ICU time-series prediction methods and use downstream task performance as a proxy for representation quality.
>
> The reviewer suggests BEHRT and MedBERT, which are end-to-end pretraining methods based on transformer architectures trained on specially processed/curated EHR records. These pretraining approaches are tied to particular populations and are therefore highly sensitive to distribution shifts and input formality (and BEHRT’s weights are not publicly available). They are also task-specific (diagnostic code prediction for BEHRT and disease prediction for MedBERT) and designed for longitudinal EHR records. Such pretrained models have not been shown to transfer well across datasets, and to date, there is no publicly released ICU time-series foundation model (whereas our baselines include a general time-series foundation model). In both Hegselmann et al. and Gao et al., it has been reported that these pretrained EHR models can underperform compared to rapidly evolving generalist models, which tend to generalize better.
>
>
> We reproduced GenHPF and found that it performs substantially worse than our method, as shown below. We replicate GenHPF with minor adaptations to ICU data preprocessing, using ICU vitals and treatments as events instead of EHR codes, while keeping the rest of the pipeline and hyperparameters unchanged. Because GenHPF is limited to classification tasks, we benchmark it on mortality, drug, and lab prediction.
>
> We believe there are two main reasons for the performance gap: (i) ICU data are much more numerically intensive than typical EHR records, with many events occurring in a short time window, making it difficult for GenHPF’s embedding and aggregation framework to capture sufficient information; and (ii) under distribution shifts, GenHPF primarily addresses schema/code differences (as claimed in the original work), whereas our method, by mapping patient state to a shared basis for diagnosis and summarization in natural language, can mitigate additional distribution shifts induced by population differences.
>
>
> We will add this discussion in the revised manuscript to clarify our baseline choices and will also explicitly cite and distinguish the works the reviewer mentioned.
>
> Here is a summary of results comparing GenHPF and ours on classification tasks.
>
> |Task | GenHPF | Record2Vec
> |-------------|---------------:|---------------:|
> |Mort(Hirid) | 0.8359 | 0.90 |
> |Mort(Ppicu) | 0.7797 | 0.86|
> |Mort(Hirid -> Ppicu) | 0.5824| 0.72 |
> |Drug(Hirid) | 0.7697 | 0.911 |
> |Drug(Ppicu) | 0.835| 0.937 |
> |Drug(Hirid -> Ppicu)| 0.6996 | 0.97|
> |Lab(Hirid) | 0.7409 |0.931|
> |Lab(Ppicu) | 0.6773| 0.936 |
> |Lab(Hirid -> Ppicu) |0.3160 |0.97 |
>
>
> Stefan Hegselmann, Georg von Arnim, Tillmann Rheude, Noel Kronenberg, David Sontag, Gerhard Hindricks, Roland Eils, and Benjamin Wild. "Large Language Models are Powerful EHR Encoders". arXiv preprint arXiv:2502.17403, 2025.
>
> Yanjun Gao, Skatje Myers, Shan Chen, Dmitriy Dligach, Timothy A. Miller, Danielle Bitterman, Matthew Churpek, and Majid Afshar. "When Raw Data Prevails: Are Large Language Model Embeddings Effective in Numerical Data Representation for Medical Machine Learning Applications?". In Findings of the Association for Computational Linguistics: EMNLP 2024, pages 5414–5428, 2024.

---

> ### Author Response · Authors · 2025-11-21
> **Rebuttal Part 2**
>
> **Weakness 2**
> > Practicality of Transfer Learning
>
> We understand the reviewer’s perspective, but we respectfully do not fully agree. We believe that transfer learning and methods that generate more unified embeddings can be practically valuable. As an analogy, hospitals historically used different note-taking and EHR systems, and in the short term it was less costly and simpler to maintain local standards. Over time, however, clinicians and health systems have increasingly advocated for unified protocols and EHR standards to enable more consistent and better care.
>
> Putting computational cost aside, many local or resource-constrained hospitals may simply not have enough samples to train a robust in-distribution model. MIMIC-IV, HiRID, and PPICU are all large, complex datasets collected over many years at major centers; smaller hospitals often lack such scale. To approximate this setting, we train an in-distribution model on PPICU using 1,000 randomly selected samples and compare it to 16-sample transfer learning. In this regime, few-shot fine-tuning actually outperforms training from scratch on the small local dataset.
>
> |Task |Small PPICU | Few-shot Finetune |
> |-------------|---------------:|---------------:|
> Mortality | 0.6843 | 0.72 |
> Los | 1.03 | 0.49 |
> Forecast | 0.62005 | 0.183|
> Feature |  0.901| 0.97 |
> Lab | 0.94 | 0.97|
>
>
>
>
> **Weakness 4**
> > Unclear Role of LLM Summarization and Limitations of Experimental Setup
>
> We would like to clarify a possible misunderstanding that our current writing may have caused.
>
> For any time-series prediction task or ML pipeline, some degree of preprocessing is unavoidable. However, this does not mean that the features from each distribution are “pre-selected” in a way that targets LLM performance. In consultation with our clinical collaborators (attending ICU physicians), we retain the most clinically relevant variables for ICU stays. While the final merged feature set is 60-75 variables, the raw EHR typically contains 200-400 unique variables (where normally pipeline would only consider ~100 unique features). Beyond standard irregular time-series preprocessing and feature merging, we do not inject additional priors or hand-crafted modifications into the ICU time-series records.
>
> Our intention is that the reviewer sees this preprocessing as necessary for any ICU modeling pipeline, rather than as tuning for the LLM. Moreover, the role of the LLM in our work is not merely to standardize format, but to convert clinical time-series data into a concise natural-language summary that captures key medical features and their temporal trajectories for each unique ICU stay.
> If possible, could you please provide a more detailed reference (e.g., venue, title) for the Adams et al. 2025 paper you mentioned, so that we can properly compare and cite it in the revision?

---

> ### Author Response · Authors · 2025-11-21
> **Rebuttal Part 3**
>
> **Weakness 3 & Question 1**
> > Overstatement on Resolving Distribution Shift; Lack of Explanation for Performance Improvement
>
> Thank you for raising this point. We did not intend to claim that we “solve” distribution shift; rather, our aim is to demonstrate empirically that Record2Vec-generated representations are more robust for downstream classifiers in cross-site prediction than existing baselines. We will revise the manuscript to soften and clarify our claims around distribution shift.
>
> In our view, there are three main sources of distribution shift:
>
> 1. **Misaligned features and units.** We mitigate this by mapping all signals into natural language descriptions, rather than relying directly on raw feature names and units.
>
> 2. **Population differences.** We address this by mapping each patient’s status into a shared, higher-level representation (e.g., ICD codes, organ failure descriptors, etc.).
>
> 3. **Site-specific treatment practices.** We handle this by decomposing time series into medical events and summarizing them (e.g., via the Trend prompt), rather than treating measurements as a uniform grid of values.
>
>
> Our pipeline is designed to approximate what clinicians do when patients are transferred or handed to another team. In addition to passing along raw measurements, clinicians summarize the patient’s status as trends, provide brief natural language notes on diagnoses, vitals to be mindful for, and suggest possible interventions if things happen later-on.
>
> Regarding the reviewer’s question on “how” standardizing the input leads to performance benefits: this paper is primarily empirical, and our experimental design is motivated by real-world hand-off practice and the capabilities of LLMs. A full theoretical treatment (e.g., via a causal diagram showing conditions under which site-invariant embeddings emerge) is beyond our current scope and we view it as important future work. Intuitively, our hypothesis is that LLMs, trained on large corpora including medical text, encode clinical knowledge and tend to produce summaries in a relatively consistent style across sites, even when underlying data distributions shift. As a result, this helps downstream models learn representations that transfer better.
>
> We agree that a more detailed analysis of how clinically important information is preserved or emphasized in the summaries would be valuable. However, our primary goal in this paper is to propose and validate a pipeline that hospitals can use to generate more portable input representations. We view a detailed information-flow analysis as complementary future work.
>
> Finally, the table below reports transferability results from using a fixed template that only matches formats, without summarization. While this “format matching” yields some improvement, it does not reproduce the substantial gains we observe with full summarization, suggesting that the benefit is not solely due to standardizing the input format.
>
> | **Method ↓**           | **HiRID→PPICU Forecast (MSE)** | **HiRID→PPICU LoS (MAE)** | **HiRID→PPICU Mort (AUROC)** | **HiRID→PPICU Drug (Recall)** | **HiRID→PPICU Lab (Recall)** |
> |------------------------|-------------------------------:|---------------------------:|------------------------------:|-------------------------------:|------------------------------:|
> | Best baseline          | 0.209                          | 0.73                       | 0.49                          | 0.93                          | 0.95                          |
> | Deterministic Template | 0.195                          | 0.98                       | **0.72**                      | 0.96                          | 0.96                          |
> | Record2Vec             | **0.183**                      | **0.69**                   | **0.72**                      | **0.97**                      | **0.97**                      |
>
> | **Method ↓**           | **MIMIC→PPICU Forecast (MSE)** | **MIMIC→PPICU LoS (MAE)** | **MIMIC→PPICU Mort (AUROC)** | **MIMIC→PPICU Drug (Recall)** | **MIMIC→PPICU Lab (Recall)** |
> |------------------------|--------------------------------:|---------------------------:|------------------------------:|-------------------------------:|------------------------------:|
> | Best baseline          | 0.269                           | 0.71                      | 0.69                          | 0.90                           | 0.90                          |
> | Deterministic Template | 0.263                           | 0.77                      | 0.71                          | 0.95                           | 0.95                          |
> | Record2Vec             | **0.242**                       | **0.49**                  | **0.72**                      | **0.95**                       | **0.95**                      |

---

> ### Author Response · Authors · 2025-11-21
> **Rebuttal Part 4**
>
> **Question 2**
> > Lack of Justification for Classifier Choice
>
> We appreciate the reviewer’s question regarding our choice of downstream classifier. In fact, we evaluated our representations with multiple classical time-series models (MLP, LSTM, PatchTSMixer, TimeMixer). We report PatchTSMixer in the main text because it provides the strongest and most consistent performance across datasets and tasks, and thus serves as a stringent, representative backbone for comparing input representations. A full table of results for all tested classifiers is included in the appendix. We will revise the manuscript to more clearly document the set of downstream models considered and to justify our decision to highlight PatchTSMixer in the main results.

---

> ### Comment · Reviewer_t4Cv · 2025-11-26
> **Thnaks for your sincere response**
>
> Thank you for thinking deeply about the concerns I made and providing such a thorough response.
>
> Rather than simply following my comments as they were, you counter-argued them and supported your points with additional experiments — I truly appreciate the effort you put into this.
>
> I found it very impressive that you directly demonstrated how the GenHPF results differed from my expectations.
>
> All my concerns are resolved.
> I have raised the score.

---

> ### Author Response · Authors · 2025-11-26
>
> We sincerely thank you for the time and effort you dedicated to this review process and for raising your score; we are delighted that our additional experiments and analysis regarding GenHPF successfully addressed your concerns.

---

### Author Response · Authors · 2025-11-21
**General Comment**

We sincerely thank all reviewers for their thoughtful, detailed, and constructive feedback on our work.

Several reviewers highlighted that our paper addresses a practically important problem of cross-hospital portability in clinical ML, that the proposed summarize-then-embed Record2Vec pipeline is a simple yet novel and deployment-oriented use of frozen LLMs, and that our experimental design is comprehensive, with multi-site evaluation across three ICU cohorts, seven tasks, systematic ablations, few-shot adaptation studies, and privacy/representation diagnostics.
The reviewers also emphasized our empirical findings that summarization improves transfer while substantially reducing token and computational costs.

Overall, our contribution is to show that a frozen summarize-then-embed pipeline can already produce portable ICU time-series representations that are competitive in-distribution, markedly more robust under cross-site transfer, and effective in realistic few-shot settings, without increasing demographic leakage relative to strong non-language baselines.

In this rebuttal, we have clarified our positioning relative to recent LLM for EHR work, refined our claims around distribution shift, privacy, and added experiments on baselines, ablations, and operational cost estimates.

#### Clarifications Summary

- **Significance of transferability.** While utility is critical for clinicians, many sites lack large, long-term datasets like HiRID, MIMIC, and PPICU. Models trained only on small local cohorts may have even lower utility. To illustrate this, we replicate baseline methods using 1,000 PPICU samples and compare them with our 16-sample finetuning results.

- **Distribution shift and explanation of gains.** We explicitly identify three sources of distribution shift and detail how our method addresses each. We also show that improvements are not due to simple format matching by comparing against a no-summarization, fixed-template variant, where our method still achieves substantially larger gains.

- **Role of LLM summarization and experimental setup.** The features used for LLM summarization are not pre-selected for the LLM; they arise from standard ICU preprocessing pipelines developed with inputs from clinical specialists. These procedures are common to ICU studies, not unique to ours.

- **Classifier choice.** In addition to PatchTSMixer, we include other strong time-series classifiers (MLP, LSTM, TimeMixer). Their results, reported in the appendix, show that our conclusions are not tied to a single downstream model.

- **Finetuning sample selection.** All finetuning samples are randomly selected and shared across methods to ensure a fair and controlled comparison.

---

#### Experiments Summary

- Replicate GenHPF with minor adaptations to better fit ICU datasets and quantify the additional performance contributed by our method.
- Replace the original mortality models with strong mortality baselines, demonstrating both improved utility and portability.
- Conduct embedder ablations over pooling strategies (mean, CLS, last) and normalization schemes (L2, none), and compare against one additional SOTA and one non-SOTA embedding model.
- Finetune TSDE under our transfer setting to assess compatibility with our framework.
- Report latency and cost statistics to contextualize operational feasibility.

---

### Author Response · Authors · 2025-11-26
**Paper revision and modification**

We thank the reviewers for the valuable feedback. We have made the following changes to address the comments and improve the clarity and robustness of our work.

## Changes to the PDF (indicated in blue for readability, will be modified to black afterwards)

***Main text***

* `[t4Cv, xPpy, rGkM]` (Section 2) Major revision and better position of our work compared with recent related works.
* `[xPBC, rGkM]` (Section 3.1) Footnote on embedder ablation experiments.
* `[t4Cv, xPBC, rGkM]` (Section 3.3) Explain data pre-processing in detail to ensure comparison fairness.
* `[t4Cv, xPpy, rGkM]` (Section 3.1, Table 1, Table 2) Add a deterministic text baseline GenHPF description and result.
* `[t4Cv, xPBC]` (Section 4, Table 1) Clearly state what downstream models we experimented with. Replace mortality with results from strong mortality baselines.
* `[rGkM, xPpy]` (Section 4.2, Table 2) Add comparison between a fixed template.
* `[xPBC, xPpy, rGkM]` (Section 4.3 & 4.4, Figure 3 & 4) Replace with high-resolution figures and modify the figure captions to better explain how rankings are obtained.
* `[t4Cv, rGkM]` (Section 4.5, Figure 5) Introduced a practicality and motivation discussion of transfer learning and added the 16-sample finetuning results of TSDE baseline.
* `[rGkM]` (Section 4.6) Tune down the privacy experiment result claim and report worst-group performance analysis.
* `[rGkM]` (Section 6) Latency and cost refer to appendix.
* `[t4Cv, xPBC, xPpy, rGkM]` (Section 6) Extended and more detailed explanation of our work’s limitations and future directions.
* `[xPBC]` All minor edits suggested by reviewer `xPBC` are fixed.

---

***Appendix***

* `[t4Cv, xPpy, rGkM]` (Appendix B) Add suggested related work and discussions to the extended related work section.
* `[rGkM]` (Appendix H) Add runtime, latency, and cost details.
* `[xPBC]` (Appendix I) Added mean and standard deviation results of few-shot fine-tuning experiments.
* `[t4Cv]` (Appendix J) Add practicality of transfer learning.
* `[xPBC]` (Appendix K) Detailed strong mortality baseline experiment results.
* `[xPBC, rGkM]` (Appendix L) Detailed results for embedding model ablation study.
* `[xPBC, xPpy, rGkM]` (Appendix M) Detailed results used for generating the rank distribution in Figure 3 and 4.
* `[t4Cv, xPBC, xPpy]` (Appendix N) Detailed explanation on how we modify GenHPF to serve as a baseline.


---

We hope these revisions effectively respond to the reviewers' suggestions and improve the overall quality of the paper. Additionally, we would like to express our sincere gratitude to the reviewers for their insightful suggestions, which have strengthened our manuscript.

---

### Meta-Review · Area_Chair_dCAQ · 2026-01-06

**Summary:**

This paper offers a novel approach to mitigating the deleterious effects of distribution shifts when embedding patient time-series data: Summarize (in natural language) such data using an LLM, and then train a predictor on top of representations of these summaries. There was consensus that this paper addresses a practical problem in clinical ML with a clever, novel method. The empirical evaluation here seems compelling, and outstanding issues regarding this raised by reviewers were addressed in rebuttal with updated results.

**Reviewer Concerns:**

Some reviewers (notably t4Cv) raised concerns about baselines, which were explicitly and satisfactorily addressed in rebuttal. Other issues included things like better contextualizing the work and adding SOTA in-domain baselines for comparison: The authors have addressed all of these, as far as I can tell.

**Reviewer Scores:**

t4Cv briefly raised their score.
xPpy primarily asked for more contextualization w/r/t existing work, which the authors provided, so it seems plausible they would have raised their score.
rGkM and xPBC were already quite positive.

---

### Decision · Program_Chairs · 2026-01-26

Accept (Poster)